# How Does Adaptive Optimization Impact Local Neural Network Geometry?

**Kaiqi Jiang**
Department of Electrical and Computer Engineering
Princeton University
Princeton, NJ 08540
kaiqij@princeton.edu

**Dhruv Malik**
Machine Learning Department
Carnegie Mellon University
Pittsburgh, PA 15213
dhruvm@andrew.cmu.edu

**Yuanzhi Li**
Machine Learning Department
Carnegie Mellon University
Pittsburgh, PA 15213
yuanzhil@andrew.cmu.edu

## Abstract

Adaptive optimization methods are well known to achieve superior convergence relative to vanilla gradient methods. The traditional viewpoint in optimization, particularly in convex optimization, explains this improved performance by arguing that, unlike vanilla gradient schemes, adaptive algorithms mimic the behavior of a second-order method by adapting to the *global* geometry of the loss function. We argue that in the context of neural network optimization, this traditional viewpoint is insufficient. Instead, we advocate for a *local* trajectory analysis. For iterate trajectories produced by running a generic optimization algorithm OPT, we introduce $R_{\text{med}}^{\text{OPT}}$, a statistic that is analogous to the condition number of the loss Hessian evaluated at the iterates. Through extensive experiments on language models where adaptive algorithms converge faster than vanilla gradient methods like SGD, we show that adaptive methods such as Adam bias the trajectories towards regions where $R_{\text{med}}^{\text{Adam}}$ is small, where one might expect faster optimization. By contrast, SGD (with momentum) biases the trajectories towards regions where $R_{\text{med}}^{\text{SGD}}$ is comparatively large. We complement these empirical observations with a theoretical result that provably demonstrates this phenomenon in the simplified setting of a two-layer linear network. We view our findings as evidence for the need of a new explanation of the success of adaptive methods, one that is different than the conventional wisdom.

## 1 Introduction

The efficient minimization of a parameterized loss function is a core primitive in statistics, optimization and machine learning. Gradient descent (GD), which iteratively updates a parameter vector with a step along the gradient of the loss function evaluated at that vector, is a simple yet canonical algorithm which has been applied to efficiently solve such minimization problems with enormous success. However, in modern machine learning, and especially deep learning, one frequently encounters problems where the loss functions are high dimensional, non-convex and non-smooth. The optimization landscape of such problems is thus extremely challenging, and in these settings gradient descent often suffers from prohibitively high iteration complexity.

37th Conference on Neural Information Processing Systems (NeurIPS 2023).

To deal with these difficulties and improve optimization efficiency, practitioners in recent years have developed many variants of GD. One prominent class of these GD variants is the family of *adaptive* algorithms [DHS11, TH+12, KB15]. At a high level, adaptive methods scale the gradient with an adpatively selected preconditioning matrix, which is constructed via a moving average of past gradients. These methods are reminiscent of second order gradient descent, since they construct approximations to the Hessian of the loss functions, while remaining computationally feasible since they eschew full computation of the Hessian. A vast line of empirical work has demonstrated the superiority of adaptive methods over GD to optimize deep neural networks, especially on Natural Language Processing (NLP) tasks with transformers [VSP+17, DCLT19].

From a theoretical perspective, adaptive methods are well understood in the traditional context of convex optimization. For instance, Duchi et al. [DHS11] show that when the loss function is convex, then the Adagrad algorithm yields regret guarantees that are provably as good as those obtained by using the best (diagonal) preconditioner in hindsight. The key mechanism that underlies this improved performance, is that the loss function has some global geometric property (such as sparsity or a coordinate wise bounded Lipschitz constant), and the algorithm adapts to this global geometry by adaptively selecting learning rates for features that are more informative.

However, in non-convex optimization, and deep learning in particular, it is highly unclear whether this simple characterization is sufficient to explain the superiority of adaptive methods over GD. Indeed, for large scale neural networks, global guarantees on the geometric properties of the loss are typically vacuous. For instance, for a 20-layer feedforward neural network, if we scale up the weights in each layer by a factor of $1.5$, then the global Lipschitz constant of the network is scaled up by a factor of at least $e^{10}$. Hence it only makes sense to study convergence by looking at the local geometry of the loss along the trajectory of the optimization algorithm [ACH18].

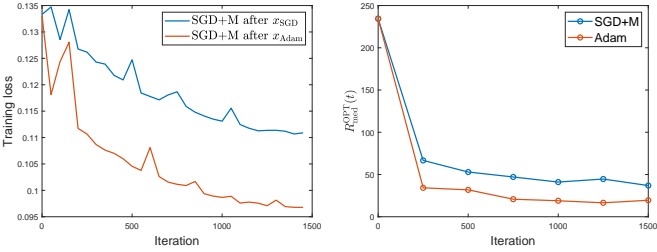
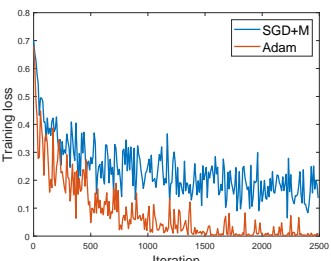

Figure 1: (left) Training losses of SGD+M starting from $x_{\mathrm{SGD}}$ and $x_{\mathrm{Adam}}$. (right) The 10th largest value over median in the diagonal of loss Hessian (which can be viewed as a variant of $R_{\mathrm{med}}^{\mathrm{OPT}}(t)$ defined in eq. (1)) for Adam and SGD+M. Since the full Hessian is too big, here we selected several layers and randomly sampled 200 coordinates per layer to compute.

Figure 2: Training losses of Adam and SGD+M on the sentence classification task described in Section 4.1.

Moreover, the interaction between an optimization algorithm and neural network geometry is highly complex — recent work has shown that geometric characteristics of iterates encountered during optimization is highly dependent on the choice of optimization algorithm and associated hyperparameters [LBD+20, CKL+21]. For instance, Cohen et al. [CKL+21] demonstrate that while training neural networks with GD, the maximum eigenvalue of the Hessian evaluated at the GD iterates first increases and then plateaus at a level that is inversely proportional to the step size. The viewpoint from convex optimization, where a loss function has some (potentially) non-uniform but fixed underlying geometry that we must adapt to, is thus insufficient for neural networks, since the choice of optimization algorithm can actually *interact* with and *influence* the observed geometry **significantly**.

To provide another example of this interactive phenomenon, we consider the following experiment. On the same network training loss function $f$, we run stochastic gradient descent with momentum (SGD+M) and Adam to obtain two different trajectories. We select an iterate $x_{\mathrm{Adam}}$ from the Adam trajectory and an iterate $x_{\mathrm{SGD}}$ from the SGD trajectory, such that $f(x_{\mathrm{Adam}}) = f(x_{\mathrm{SGD}})$. We then run SGD+M with the same configuration twice, once from $x_{\mathrm{Adam}}$ and once from $x_{\mathrm{SGD}}$. If the underlying geometry of the loss function $f$ was truly fixed, then we would not expect a significant difference in the performance of running SGD+M from either of the two iterates. However, as shown in Figure 1, there is a noticeable difference in performance, and running SGD+M from $x_{\mathrm{Adam}}$ achieves lower

loss than running SGD+M from $x_{\text{SGD}}$. This suggests that Adam may bias the optimization trajectory towards a region which is more favorable for rapid training. This motivates the following question.

*How does adaptive optimization impact the observed geometry of a neural network loss function, relative to SGD (with momentum)?*

The remainder of this paper is dedicated to answering the above question. To this end, for each iterate in a trajectory produced by running an optimization algorithm OPT, where the Hessian of the $t$th iterate is given by $H^{(t)} \in \mathbb{R}^{d \times d}$, we define the second order statistic $R_{\text{med}}^{\text{OPT}}(t)$ in the following fashion. For the $t$th iterate in the trajectory, let $R_{\text{med}}^{\text{OPT}}(t)$ be the ratio of *maximum* of the absolute entries of the diagonal of $H^{(t)}$, to the *median* of the absolute entries of the diagonal of $H^{(t)}$. Concretely, we define

$$R_{\text{med}}^{\text{OPT}}(t) = \frac{\max\{|H_{ii}^{(t)}|\}_{i=1}^d}{\text{median } \{|H_{ii}^{(t)}|\}_{i=1}^d}. \tag{1}$$

This statistic thus measures the *uniformity* of the diagonal of Hessian, where a smaller value of $R_{\text{med}}^{\text{OPT}}(t)$ implies that the Hessian has a more uniform diagonal. It can also be viewed as a stable[1] variant of the condition number. Instead of singular values, we choose diagonal entries because adaptive methods used in practice are *coordinate-wise*, which can be viewed as the diagonal scaling approaches.[2] In Appendix A.9 we discuss this intuition in detail and compare $R_{\text{med}}^{\text{OPT}}(t)$ with singular value-based metrics. As a supplementary result, in Appendix F, we demonstrate that the loss Hessian approaches diagonal during training for Adam and SGD+M. There has been prior theoretical work on overparameterized neural networks showing that a smaller condition number of Hessian, Neural Tangent Kernel [JGH18] etc. could yield to faster convergence rate for (S)GD [LZB22]. As for (diagonal) adaptive methods (*e.g.* Adagrad), they were original designed to adapt to the nonuniform diagonal geometry. Intuitively, a smaller $R_{\text{med}}^{\text{OPT}}(t)$, which implies more uniform diagonal geometry, could lead to faster convergence.

Armed with this statistic, we make the following contributions:

1. We focus on language models as on NLP tasks, adaptive algorithms show significantly faster convergence than SGD (with momentum). On a wide variety of neural network transformer architectures and language modeling datasets, we conduct experiments to compare how $R_{\text{med}}^{\text{Adam}}(t)$ and $R_{\text{med}}^{\text{SGDM}}(t)$ evolve over time, when Adam and SGD+M are run from the same initialization and with their optimal (initial) learning rates respectively. In each case, we demonstrate that the Adam trajectory attains $R_{\text{med}}^{\text{Adam}}(t)$ values that are significantly smaller than the $R_{\text{med}}^{\text{SGDM}}(t)$ values found by SGD+M. We show a simple example of this phenomenon in Figure 1(right). This suggests that relative to SGD+M, Adam biases the optimization trajectory to a region where the Hessian diagonal is more uniform. We call this phenomenon *the uniformity of diagonal geometry* for adaptive methods. Moreover, we demonstrate that a more uniform Hessian diagonal, characterized by smaller $R_{\text{med}}^{\text{OPT}}(t)$, is a contributing factor to faster optimization (see Section 4.3 for discussion). This suggests that a region where the Hessian diagonal is more uniform is also a region that is more amenable to rapid optimization.

2. We complement our empirical results with a theoretical analysis of this phenomenon in the simplified setting of large batch Adam and SGD+M, on a two-layer linear network with $d$-dimensional input and hidden layer, and one dimensional output. We show that for a wide range of $t$, $R_{\text{med}}^{\text{Adam}}(t) = 1 \pm o(1)$ but $R_{\text{med}}^{\text{SGDM}}(t) = \Omega(\log d)$. Our proof reveals that Adam induces the weight matrices to have low rank whose leading singular vectors have certain type of uniformity (see Section 6 for discussion), a fact that we also observe empirically in large scale neural networks, suggesting that this may be a mechanism by which adaptive methods bias trajectories to have uniformity of diagonal geometry.

## 2 Related work

**Existing analyses of adaptive methods.** The vast majority of prior theoretical work on adaptive methods has focused on the blackbox setting [DHS11, KB15, CZT+20, RKK18, WWB20, DBBU20,

---

[1]Consider the case where one parameter has little impact on the loss, then the second derivative w.r.t. this parameter is almost zero, making $\frac{\max\{|H_{ii}^{(t)}|\}_{i=1}^d}{\min\{|H_{ii}^{(t)}|\}_{i=1}^d}$ infinity. So we consider *median* which is more stable.

[2]Recall that the main theoretical bound in the original Adagrad paper [DHS11] is in terms of the diagonal scaling.

ENV21]. These works make minimal assumptions about the structure of the loss function, beyond (possibly) some global properties such as convexity or smoothness. These global properties (governed by parameters such as the smoothness parameter) are assumed to hold over the entire domain. Hence this style of analysis is worst case, since the resulting convergence bounds depend on polynomially on these global parameters. However, as we show in Section 3.1, in neural networks these parameters are prohibitively large. This worst case analysis is hence unlikely to explain the success of adaptive methods on neural networks. By contrast, our focus is on analyzing the local trajectory that is induced by running the optimization method.

**Existing analyses of (S)GD on neural networks.** There is an extensive literature on the analysis of GD/SGD in the non-blackbox setting, *e.g.* overparameterized neural networks, [DZPS18, JT20, AZLL19, AZLS19, ADH$^+$19, LZB22]. However, it is unclear how to translate these analyses of GD/SGD, to an analysis that explains the gap between GD/SGD and adaptive methods.

**Influence of algorithms on the loss geometry.** In many simple convex settings, *e.g.* linear or logistic regression and the Neural Tangent Kernel [JGH18], the loss geometry is usually fixed and not influenced by learning algorithms. However, in neural networks the interaction between algorithms and loss landscapes is more complicated. Lewkowycz et al. [LBD$^+$20] find a so-called catapult effect of initial learning rate on the training trajectory of SGD and related loss curvature. Cohen et al. [CKL$^+$21] demonstrate that while training neural networks with GD, the maximum eigenvalue of the Hessian evaluated at the GD iterates first increases and then plateaus at a level that is inversely proportional to the step size. However, Cohen et al. [CKL$^+$21] leave open the problem of whether similar interactive phenomena occur in algorithms that are not GD, including adaptive methods.

## 3 Overview of main results

### 3.1 Issues of prior analyses on adaptive methods

As is mentioned in Section 2, existing work on adaptive algorithms has mainly focused on black-box analysis assuming some global worst-case parameters. However, these global bounds can be extremely bad in complicated deep learning models, as is discussed in Section 1. To see this, we initialized a transformer model[3] with default initialization in Pytorch but chose a large gain[4], and computed the smoothness parameter (denoted as $l$) and the condition number (denoted as $\kappa$) of loss Hessian on one layer. We observed that setting the gain as a large constant (*e.g.* 800) results in extremely large $l$ and $\kappa$ ($l \geq 10^7$ and $\kappa \geq 10^{10}$), which makes the convergence rates in prior black-box analysis vacuous.

The failure of global worst-case analysis implies that we need to focus on the local trajectory of algorithms. However, it is unclear whether when two optimization algorithms are used, they will have the same geometry in local trajectory or not. In particular, although in theory, adaptive algorithms can yield to a convergence rate with better dependency on certain local geometry of the function comparing to SGD (with momentum), it could still be the case that the local geometry along the trajectory of adaptive algorithm can be much worse than that of SGD (with momentum).

That motivates us to study the local geometry, especially that obtained by adaptive methods comparing to SGD (with momentum) in the paper. Motivated by the diagonal scaling of Adagrad and Adam for neural network training, we ask the follow **main question** in our paper:

> How does the local diagonal geometry (diagonal of the loss Hessian) along the local trajectory of adaptive algorithms compare to that of SGD (with momentum)?

### 3.2 Overview of the experiments

As is discussed in Section 1, we consider $R_{\text{med}}^{\text{OPT}}(t)$ defined in eq. (1) as a measurement of the uniformity of the loss Hessian diagonal and conduct experiments on different NLP tasks where adaptive methods converge faster. The detailed experimental setup will be stated in Section 4. To explore potential different patterns of different layers, we do the computation layer by layer. On a

---

[3]`https://pytorch.org/tutorials/beginner/transformer_tutorial.html`
[4]This refers to the gain parameter in some commonly used initialization functions of Pytorch, *e.g.* torch.nn.init.xavier_uniform_().

wide variety of transformer architectures and language modeling datasets from the same initialization, we observe that for the vast majority of layers:

**When we train the neural network using Adam, the uniformity of diagonal geometry, measured by $R_{\text{med}}^{\text{OPT}}(t)$ is smaller than that when we train using SGD+M from the same initialization.**

Table 1 shows a typical example of $R_{\text{med}}^{\text{Adam}}(t)$ compared to $R_{\text{med}}^{\text{SGDM}}(t)$ on a sentence classification task using BERT-small [TCLT19, BDR21] (See Section 4.1 for details). We repeated the experiments for 12 times starting from the same initialization. Table 1 shows the averaged $R_{\text{med}}^{\text{Adam}}(t)$ and $R_{\text{med}}^{\text{SGDM}}(t)$ in some randomly selected layers. We also report the averaged $\frac{R_{\text{med}}^{\text{SGDM}}(t)}{R_{\text{med}}^{\text{Adam}}(t)}$ and their standard deviations in the brackets.[5] Figure 2 shows the corresponding training losses of one in these 12 experiments.

Table 1: $R_{\text{med}}^{\text{Adam}}(t)$ and $R_{\text{med}}^{\text{SGDM}}(t)$ in some layers, on the sentence classification task (see Section 4.1).

| Layer# | Iteration 0 | | Iteration 750 | | | Iteration 1250 | | |
|---|---|---|---|---|---|---|---|---|
| | $R_{\text{med}}^{\text{SGDM}}(t)$ | $R_{\text{med}}^{\text{Adam}}(t)$ | $R_{\text{med}}^{\text{SGDM}}(t)$ | $R_{\text{med}}^{\text{Adam}}(t)$ | $\frac{R_{\text{med}}^{\text{SGDM}}(t)}{R_{\text{med}}^{\text{Adam}}(t)}$ | $R_{\text{med}}^{\text{SGDM}}(t)$ | $R_{\text{med}}^{\text{Adam}}(t)$ | $\frac{R_{\text{med}}^{\text{SGDM}}(t)}{R_{\text{med}}^{\text{Adam}}(t)}$ |
| 9 | 15.7 | 15.7 | 12.76 | 9.65 | 1.45 (0.65) | 11.43 | 14.24 | 0.94 (0.40) |
| 12 | 22.63 | 22.63 | 13.17 | 7.41 | 1.92 (0.67) | 10.62 | 9.67 | 1.33 (0.75) |
| 15 | 9.35 | 9.35 | 80.57 | 53.52 | 1.65 (0.65) | 100.65 | 61.80 | 2.01 (1.00) |
| 17 | 82.37 | 82.37 | 405.02 | 223.56 | 1.91 (0.53) | 423.28 | 337.32 | 1.43 (0.63) |
| 18 | 31.32 | 31.32 | 17.07 | 13.24 | 1.43 (0.58) | 18.15 | 15.63 | 1.21 (0.36) |
| 22 | 47.13 | 47.13 | 233.72 | 72.67 | 3.54 (1.21) | 158.38 | 93.13 | 2.28 (1.18) |
| 24 | 31.17 | 31.17 | 17.52 | 17.34 | 1.13 (0.40) | 13.51 | 14.23 | 1.05 (0.36) |

To understand this phenomenon in a more principled point of view, in Section 5 we provide a formal proof of the statement in a simplified setting: large batch Adam and SGD+M on a 2-layer linear network with 1-dimensional output.

# 4 The uniformity of diagonal geometry

As is mentioned in Section 3.2, we computed $R_{\text{med}}^{\text{OPT}}(t)$ defined in eq. (1) on different language models. In this section, we present the results of SGD+M and Adam on different architectures and datasets. In Appendix A, we present the results of other adaptive algorithms.

During training we started from the same initial weights and used the same learning rate schedule (constant or decreasing) for SGD+M and Adam. We tuned and chose the best (initial) learning rate of SGD+M. The (initial) learning rate of Adam was set as a value under which Adam converged faster than SGD+M with its best learning rate. The concrete values will be stated in later parts of this section. We used large batch sizes to make the training procedure stable. When computing Hessian, we also used large batch sizes. Due to the extremely large dimension, we did the computation on some uniformly selected coordinates, more precisely, 200 coordinates per layer.

## 4.1 Experiments on real datasets

**Sentence classification task on BERT-small.** We fine-tuned BERT-small [TCLT19, BDR21] on the IMDB dataset [MDP+11]: the task is to classify whether movie reviews are positive or negative.[6] The momentum parameter $\beta$ in SGD was set as 0.9. The two momentum parameters $(\beta_1, \beta_2)$ of Adam were set as (0.9, 0.999). We trained the model using linearly decreasing learning rates for 10 epochs (2500 iterations). The initial learning rates of SGD+M and Adam were 0.001 and 5e-5, respectively. As mentioned in Section 3.2, Figure 2 and Table 1 show the training losses and the comparison between $R_{\text{med}}^{\text{Adam}}(t)$ and $R_{\text{med}}^{\text{SGDM}}(t)$, respectively.

**Translation task.** We trained a Seq2Seq network that uses Transformer to solve a machine translation task on Multi30k [EFSS16]: this task is to train a German to English translation model.[7] The

---

[5] $R_{\text{med}}^{\text{SGDM}}(t)$ values in Table 1 for most layers are roughly 1.4 to 2 times $R_{\text{med}}^{\text{Adam}}(t)$ in corresponding layers. In practice, it can be considered significant because it might imply 1.4 to 2 times faster convergence.

[6] https://huggingface.co/docs/transformers/v4.16.2/en/training

[7] https://pytorch.org/tutorials/beginner/translation_transformer.html

momentum parameter $\beta$ in SGD was set as 0.9. The two momentum parameters $(\beta_1, \beta_2)$ of Adam were set as (0.9, 0.98). We trained the model using constant learning rates (0.03 for SGD+M and 1e-4 for Adam) for 60 epochs (1800 iterations). The experiments were repeated for 8 times starting from the same initialization. Figure 3(left) shows the training losses for one among them. Table 2a shows the averaged $R_{\mathrm{med}}^{\mathrm{Adam}}(t)$, $R_{\mathrm{med}}^{\mathrm{SGDM}}(t)$ and $\frac{R_{\mathrm{med}}^{\mathrm{SGDM}}(t)}{R_{\mathrm{med}}^{\mathrm{Adam}}(t)}$ (with standard deviation in the brackets) in some randomly selected layers.

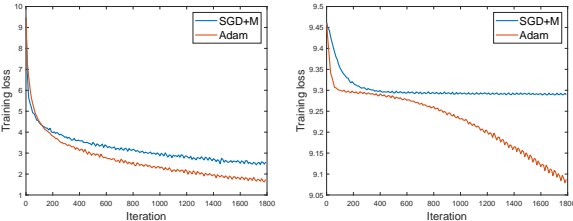

Figure 3: Training losses of Adam and SGD+M for the translation task on (left) Multi30k (see Section 4.1) and (right) Multi30k but with randomly generated targets (see Section 4.2).

Figure 4: Singular values and $R_u$ of the weight matrix in the 27-th layer on the translation task described in Section 4.1.

Table 2: $R_{\mathrm{med}}^{\mathrm{Adam}}(t)$ and $R_{\mathrm{med}}^{\mathrm{SGDM}}(t)$ in some layers for the translation task. (a) on Multi30k (see Section 4.1) and (b) on Multi30k but with randomly generated targets (see Section 4.2).

(a)

| Layer# | Epoch 0 | | Epoch 30 | | | Epoch 55 | | |
|---|---|---|---|---|---|---|---|---|
| | $R_{\mathrm{med}}^{\mathrm{SGDM}}(t)$ | $R_{\mathrm{med}}^{\mathrm{Adam}}(t)$ | $R_{\mathrm{med}}^{\mathrm{SGDM}}(t)$ | $R_{\mathrm{med}}^{\mathrm{Adam}}(t)$ | $\frac{R_{\mathrm{med}}^{\mathrm{SGDM}}(t)}{R_{\mathrm{med}}^{\mathrm{Adam}}(t)}$ | $R_{\mathrm{med}}^{\mathrm{SGDM}}(t)$ | $R_{\mathrm{med}}^{\mathrm{Adam}}(t)$ | $\frac{R_{\mathrm{med}}^{\mathrm{SGDM}}(t)}{R_{\mathrm{med}}^{\mathrm{Adam}}(t)}$ |
| 3 | 4.27 | 4.27 | 5.14 | 2.41 | 2.16 (0.75) | 3.14 | 2 | 1.58 (0.41) |
| 5 | 7.09 | 7.09 | 36.11 | 18.33 | 2.00 (0.42) | 52.12 | 16.59 | 3.16 (0.64) |
| 7 | 5.79 | 5.79 | 5.91 | 3.87 | 1.55 (0.32) | 7.52 | 3.08 | 2.45 (0.56) |
| 9 | 18.11 | 18.11 | 28.93 | 20.74 | 1.43 (0.28) | 36.67 | 18 | 2.05 (0.18) |
| 12 | 11.1 | 11.1 | 6.64 | 7.25 | 0.95 (0.21) | 9.27 | 5.06 | 1.88 (0.54) |
| 15 | 83.15 | 83.15 | 52.41 | 7.5 | 7.15 (1.63) | 46.27 | 5.69 | 8.6 (3.06) |
| 18 | 14.99 | 14.99 | 4.19 | 4.22 | 1.17 (0.45) | 3.09 | 2.72 | 1.2 (0.46) |
| 21 | 93.5 | 93.5 | 30.29 | 5.36 | 5.72 (1.05) | 19.27 | 4.8 | 4.09 (0.86) |
| 24 | 36.63 | 36.63 | 6.14 | 4.66 | 1.35 (0.31) | 5.02 | 3.2 | 1.6 (0.36) |
| 28 | 18.47 | 18.47 | 3.07 | 1.95 | 1.58 (0.16) | 2.9 | 1.59 | 1.83 (0.14) |

(b)

| Layer# | Epoch 0 | | Epoch 30 | | | Epoch 55 | | |
|---|---|---|---|---|---|---|---|---|
| | $R_{\mathrm{med}}^{\mathrm{SGDM}}(t)$ | $R_{\mathrm{med}}^{\mathrm{Adam}}(t)$ | $R_{\mathrm{med}}^{\mathrm{SGDM}}(t)$ | $R_{\mathrm{med}}^{\mathrm{Adam}}(t)$ | $\frac{R_{\mathrm{med}}^{\mathrm{SGDM}}(t)}{R_{\mathrm{med}}^{\mathrm{Adam}}(t)}$ | $R_{\mathrm{med}}^{\mathrm{SGDM}}(t)$ | $R_{\mathrm{med}}^{\mathrm{Adam}}(t)$ | $\frac{R_{\mathrm{med}}^{\mathrm{SGDM}}(t)}{R_{\mathrm{med}}^{\mathrm{Adam}}(t)}$ |
| 3 | 4.82 | 4.82 | 3.98 | 1.8 | 2.23 (0.36) | 3.79 | 1.61 | 2.36 (0.32) |
| 5 | 8.04 | 8.04 | 46.06 | 45.84 | 1.01 (0.17) | 47.83 | 34.18 | 1.41 (0.30) |
| 7 | 5.69 | 5.69 | 44.77 | 3.92 | 11.79 (2.37) | 46.5 | 2.74 | 17.4 (2.99) |
| 9 | 11.89 | 11.89 | 317.34 | 55.61 | 5.81 (0.70) | 351.85 | 46.54 | 7.61 (0.87) |
| 12 | 19.73 | 19.73 | 133.39 | 3.91 | 34.17 (4.51) | 145.09 | 2.97 | 49.49 (13.40) |
| 15 | 32.12 | 32.12 | 462.74 | 51.53 | 9.03 (0.91) | 492.73 | 50.57 | 9.84 (1.03) |
| 18 | 19.79 | 19.79 | 74.6 | 6.59 | 11.8 (3.33) | 79.02 | 3.58 | 22.75 (6.01) |
| 21 | 26.94 | 26.94 | 767.31 | 48.89 | 16.4 (3.38) | 797.49 | 36.88 | 21.98 (3.40) |
| 24 | 34.72 | 34.72 | 467.75 | 9.15 | 52.57 (11.16) | 602.03 | 3.51 | 172.65 (18.85) |
| 28 | 13.13 | 13.13 | 19.8 | 2.22 | 8.99 (1.74) | 19 | 1.63 | 11.7 (1.48) |

## 4.2 Experiments on random datasets

We used the same model and momentum parameters as in the translation task described in Section 4.1 but generated random integers as targets. Similar to the setting on real targets, the model was trained using constant learning rates (0.015 for SGD+M and 5e-5 for Adam) for 60 epochs (1800 iterations), and we repeated the experiments for 8 times starting from the same initialization. Figure 3(right) shows the training losses for one among them. Table 2b shows the averaged $R_{\mathrm{med}}^{\mathrm{Adam}}(t)$, $R_{\mathrm{med}}^{\mathrm{SGDM}}(t)$ and $\frac{R_{\mathrm{med}}^{\mathrm{SGDM}}(t)}{R_{\mathrm{med}}^{\mathrm{Adam}}(t)}$ (with standard deviation in the brackets) of the same 10 layers as in Table 2a.[8]

## 4.3 Summarization of the empirical results and discussion

Overall, through extensive experiments on language models, we demonstrate that **starting from the same initialization, for the vast majority of layers, the $R_{\mathrm{med}}^{\mathrm{OPT}}(t)$ values found by Adam are smaller than those found by SGD+M.** This suggests that Adam biases the trajectory towards a region with more uniform Hessian diagonal than SGD+M. In Appendix A.10 we also validate this observation on the in-distribution test data.

**Contribution of uniform Hessian diagonal to fast convergence.** We observe that on dataset with random targets, SGD+M plateaus after about 400 steps and thus converges much slower when compared to Adam than on real dataset (see Figure 3). On the other hand, the gaps of $R_{\mathrm{med}}^{\mathrm{SGDM}}(t)$ and

---

[8]To prevent $R_{\mathrm{med}}^{\mathrm{OPT}}(t)$ from getting too large due to tiny median, we added an additional term $0.001 \max\{|H_{ii}^{(t)}|\}_{i=1}^{d}$ to the denominator of eq. (1) when computing.

$R_{\text{med}}^{\text{Adam}}(t)$ are more significant on random data than on real data (see Table 2a and Table 2b) as well. In Appendix A.4, we conduct another experiment where we switch from SGD to Adam in the middle and compare it with the model trained by Adam from the beginning. The observation is that both the loss gap and the gap of $R_{\text{med}}^{\text{OPT}}(t)$ are gradually closed after switching (see Figure 7 and Table 8). Hence we find a positive correlation between fast convergence and uniform Hessian diagonal (small $R_{\text{med}}^{\text{OPT}}(t)$). In Appendix A we **study other adaptive algorithms (Adagrad, RMSprop and AMSGrad) and get similar observation**: all these adaptive methods converge faster than SGD or SGD+M and also bias the trajectory to regions with smaller $R_{\text{med}}^{\text{OPT}}(t)$, suggesting the universality of our observation.

Despite the above positive correlation, it is reasonable to ask whether small $R_{\text{med}}^{\text{OPT}}(t)$ indeed contributes to fast convergence or is just a byproduct of adaptive methods. To address this concern, in Appendix B.1, we add a supplementary experiment similar to that in Figure 1. We select two iterates $x_1$ and $x_2$ from two trajectories that both come from SGD+M (instead of one from Adam and one from SGD+M in Figure 1), such that the loss $f(x_1) = f(x_2)$ but $x_2$ has a smaller $R_{\text{med}}^{\text{OPT}}(t)$ than $x_1$. We then run SGD+M with the same configuration twice, once from $x_1$ and once from $x_2$. Under this setting, we get similar observation as before: running SGD+M from $x_2$ (with smaller $R_{\text{med}}^{\text{OPT}}(t)$) achieves faster convergence than from $x_1$. This suggests that the uniformity of the diagonal of loss Hessian (measured by $R_{\text{med}}^{\text{OPT}}(t)$) reveals some intrinsic trajectory property beyond the algorithm choice and is indeed a contributing factor to fast optimization. In Appendix B.2 we theoretically prove the contribution of small $R_{\text{med}}^{\text{OPT}}(t)$ to fast optimization in a simplified setting.

**More discussions on the trajectory difference.** Considering the fact that our comparison between $R_{\text{med}}^{\text{Adam}}(t)$ and $R_{\text{med}}^{\text{SGDM}}(t)$ is conditioned on the same iteration when SGD+M has larger training loss than Adam, there is a potential alternative explanation of the Hessian diagonal uniformity. That is, the global minimum has uniform Hessian, and Adam simply converges faster to it than SGD+M. To rule out this possibility, in Appendix A.3 we add a comparison of our measurements $R_{\text{med}}^{\text{Adam}}(t)$ and $R_{\text{med}}^{\text{SGDM}}(t')$, where $t, t'$ are picked such that $t$th Adam iterate and $t'$th SGD+M iterate have the *same training loss*. The results (in Table 7) show that $R_{\text{med}}^{\text{Adam}}(t) < R_{\text{med}}^{\text{SGDM}}(t')$ for most layers, thus demonstrating that the trajectories of Adam and SGD+M are truly different and that the difference is because Adam biases the local geometry (as opposed to faster convergence).

**Adding regularization.** People in practice usually add weight decay (equivalent to $l_2$ regularization) to encourage better generalization ability. In Appendix A.7 we compare SGD+M and Adam when both using small weight decay values (0.001). The results in Figure 13a and Table 9 suggest that in this case, our observation still holds: Adam converges faster than SGD+M and in the vast majority of layers, $R_{\text{med}}^{\text{Adam}}(t)$ values are smaller than $R_{\text{med}}^{\text{SGDM}}(t)$. This reveals the robustness of our observation under weak regularization. However, under large weight decay parameters, we observed cases where Adam still converged faster but $R_{\text{med}}^{\text{Adam}}(t)$ values were larger rather than smaller. In the case of strong regularization, the adaptivity of Adam requires further exploration and we hope to find new mechanisms in the future.

# 5 Theoretical analysis

In Section 4, we empirically demonstrate the uniformity of diagonal geometry. In this section, we theoretically analyze this property for large batch Adam and SGD+M on a two-layer linear network with 1-dimensional output. Although simple, the choice of 2-layer linear networks to understand learning dynamics is common in prior works (*e.g.* [TCG21]). Moreover, in language transformer models when the weights are small, the softmax in the key-value-query structure is near the linear regime. Then this structure might be approximated by the product of 3 linear operators, similar to a three-layer linear network. Hence the theoretical analysis of this phenomenon on linear networks would be a good starting point for further understanding of more complicated language models.

## 5.1 Problem setup

**Notation** Let $[d] = \{1, 2, ..., d\}$. We use $\| \cdot \|_2$ to denote the $l_2$ norm of a vector, and $\| \cdot \|_F$ to denote the Frobenius norm of a matrix. Let $\langle \cdot, \cdot \rangle$ be the Euclidean inner product between vectors or matrices. Let $\mathcal{N}(\mu, \sigma^2)$ be the one-dimensional Gaussian distribution with mean $\mu$ and variance $\sigma^2$. For a scalar (vector, matrix) $A$ which evolves over time, we use $A^{(t)}$ to denote its value at time $t$.

Let there be $m$ data points. The data matrix is $X \in \mathbb{R}^{d_x \times m}$ and the label matrix is $Y \in \mathbb{R}^{d_y \times m}$. We assume that the input dataset is whitened, i.e. $\Lambda_{xx} := \frac{1}{m} X X^T \in \mathbb{R}^{d_x \times d_x}$ is an identity matrix.

The parameters of a 2-layer linear network are given by $W := (W_2, W_1)$. Assume $W_i \in \mathbb{R}^{d_i \times d_{i-1}}$ for $i = 1, 2$. We have $d_2 = d_y, d_0 = d_x$. We consider the square loss $L(W) := \frac{1}{2m} \|W_2 W_1 X - Y\|_F^2$.

Denote $A := \frac{1}{m} Y X^T \in \mathbb{R}^{d_y \times d_x}$. [AGCH19] show that with whitened dataset,

$$L(W) := \frac{1}{2m} \|W_2 W_1 X - Y\|_F^2 = \bar{L}(W) + c, \quad \bar{L}(W) := \frac{1}{2} \|W_2 W_1 - A\|_F^2. \quad (2)$$

where $c$ does not depend on $W$. We consider the following model with small Gaussian initialization.

**Assumption 1** (Setup). *The input covariance $\Lambda_{xx} := \frac{1}{m} X X^T \in \mathbb{R}^{d_x \times d_x}$ is an identity matrix. The input and hidden layers are both of dimension $d$, i.e. $d_1 = d_0 = d$. Without loss of generality, we can assume that $A$ is a row vector (i.e. $d_2 = 1$) whose coordinates are positive[9] and $\Theta(1)$ in terms of $d$.*

**Assumption 2** (Gaussian Initialization). *$\forall i, j : w_{2i}^{(0)} \sim \mathcal{N}(0, \frac{1}{d^{2\alpha}}), W_1^{(0)}[i, j] \sim \mathcal{N}(0, \frac{1}{d^{4\alpha}})$ are independently initialized with sufficiently large $\alpha > 0$.*

Denote $\tilde{A}$ and $\tilde{\Lambda}_{xx}$ as the batch versions of $A$ and $\Lambda_{xx}$. We make the following large-batch assumption. We emphasize that large batches are commonly used in NLP tasks (*e.g.* [BMR+20]).

**Assumption 3** (Large Batch). *For the randomly selected batches, assume $\mathbb{E}[\tilde{A}] = A$, $\mathbb{E}[\tilde{\Lambda}_{xx}] = \Lambda_{xx}$. $\forall i, j \in [d] : \mathbb{E}\left[(\tilde{A}_i - A_i)^2\right] \leq \sigma^2, \mathbb{E}\left[(\tilde{\Lambda}_{xx}[i, j] - \Lambda_{xx}[i, j])^2\right] \leq \sigma^2$, and $\sigma^2 = \mathcal{O}\left(\frac{1}{poly(d)}\right)$.*

Denote $\tilde{g}^{(t)}$ as the batch gradient at time $t$. The update rules of SGD+M and Adam are given by

$$\text{SGD+M:} \quad u^{(t+1)} = \beta u^{(t)} + \tilde{g}^{(t)}, \quad W^{(t+1)} = W^{(t)} - \eta u^{(t)},$$

$$\text{Adam:} \quad \eta_t = \eta \cdot \sqrt{1 - \beta_2^{t+1}} / (1 - \beta_1^{t+1}), \quad m^{(t+1)} = \beta_1 m^{(t)} + (1 - \beta_1) \tilde{g}^{(t)},$$

$$v^{(t+1)} = \beta_2 v^{(t)} + (1 - \beta_2) \tilde{g}^{(t)} \odot \tilde{g}^{(t)}, \quad W^{(t+1)} = W^{(t)} - \eta_t \frac{m^{(t)}}{\sqrt{v^{(t)}} + \xi}, \quad (3)$$

where $\eta$ is the learning rate, $\beta, \beta_1, \beta_2$ are momentum parameters, and $\xi$ is for numerical stability. All operations on vectors are element-wise. Here and throughout, the notation $f(x) = \mathcal{O}(g(x))$ (resp. $f(x) = \Omega(g(x)), f(x) = \Theta(g(x))$) means that $\exists C_1, C_2 > 0$ such that $f(x) \leq C_2 g(x)$ (resp. $f(x) \geq C_1 g(x), C_1 g(x) \leq f(x) \leq C_2 g(x)$). Here $C_1, C_2$ may depend on $\beta, \beta_1, \beta_2$. We will also use the notation with $\sim$, i.e. $\tilde{\mathcal{O}}(\cdot), \tilde{\Omega}(\cdot), \tilde{\Theta}(\cdot)$ to hide factors of order $\log d$. In our theoretical analysis, "with high probability", or "w.h.p." for short, means that with probability at least $1 - \frac{1}{poly(d)}$.

Since the weights and Hessians in different layers may have different magnitudes, we compute the $R_{\text{med}}^{\text{OPT}}(t)$ layer by layer. Denote $R_{\text{med},k}^{\text{SGDM}}(t)$ (resp. $R_{\text{med},k}^{\text{Adam}}(t)$) as the $R_{\text{med}}^{\text{OPT}}(t)$ found by SGD+M (resp. Adam) w.r.t. $W_k$ at time $t$ where $k = 1, 2$.

## 5.2 Main results

**Theorem 1.** *Under Assumption 1, 2 and 3, consider the weights $\left\{W_{SGD}^{(t)}\right\}_{t \geq 0}$ (resp. $\left\{W_{Adam}^{(t)}\right\}_{t \geq 0}$) obtained by SGD+M (resp. Adam) defined in* (3).

*1. For any $p > 0$, pick $0 < \epsilon < \frac{1}{d^p}, \eta \leq \mathcal{O}\left(\frac{\epsilon}{d^{7\alpha/4+4}}\right)$ and $\alpha \geq 4(p + 2)$. Suppose $\sigma \leq \frac{\eta^{3/2}}{d^{\alpha/2+1}}$, then there exists $T_{SGD,1}, T_{SGD,2}$ such that w.h.p., $\bar{L}\left(W_{SGD}^{(T_{SGD,1})}\right) = \Theta(d), \bar{L}\left(W_{SGD}^{(T_{SGD,2})}\right) \leq \tilde{\mathcal{O}}\left(\frac{1}{d^p}\right)$, and*

$$\forall t \in [T_{SGD,1}, T_{SGD,2}]: \quad R_{med,k}^{SGDM}(t) = \Omega(\log d), \quad k = 1, 2.$$

---

[9] In Assumption 2 we assume Gaussian initialization. Due to the rotational invariance of Gaussian distribution, we can assume that all coordinates of $A$ are positive without loss of generality.

2. *For any $p > 0$, pick $\eta \leq \mathcal{O}\left(\frac{1}{d^{3\alpha}}\right), \xi \leq \sqrt{\frac{\eta}{d^{3\alpha-1}}}, \alpha \geq \frac{p+4}{3}$ and $\beta_2 = \beta_1^2$.[10] Suppose $\sigma \leq \frac{\eta^{3/2}\xi^2}{d^{13/4}}$,*
*Then $\exists T_{Adam,1}, T_{Adam,2}$ such that w.h.p., $\bar{L}\left(W_{Adam}^{(T_{Adam,1})}\right) = \Theta(d), \bar{L}\left(W_{Adam}^{(T_{Adam,2})}\right) \leq \tilde{\mathcal{O}}\left(\frac{1}{d^p}\right)$, and*

$$\forall t \in [T_{Adam,1}, T_{Adam,2}]: \quad R_{med,k}^{Adam}(t) = 1 + \tilde{\mathcal{O}}\left(\eta^{\frac{1}{4}} + \frac{1}{d^{\frac{\alpha}{2}-\frac{1}{4}}}\right), \quad k = 1, 2.$$

**Remark** Theorem 1 holds for all values of hyperparameters (such as $\alpha, \sigma$) in certain ranges instead of just particular values. The ranges of SGD+M and Adam overlap with each other. That means we can choose the same hyperparameters for SGD+M and Adam in the overlapped region to make a fair comparison, for example, same $\alpha, \sigma$ such that $\alpha \geq 4(p+2)$ and $\sigma \leq \min\{\frac{\eta^{3/2}}{d^{\alpha/2+1}}, \frac{\eta^{3/2}\xi^2}{d^{13/4}}\}$.

An immediate corollary of this theorem below gives the difference between iterates of Adam and SGD+M that have the same loss.

**Corollary 1.** *Under the setup in Theorem 1, w.h.p., for any $t \in [T_{SGD,1}, T_{SGD,2}]$ and $t' \in [T_{Adam,1}, T_{Adam,2}]$ such that $\bar{L}\left(W_{SGD}^{(t)}\right) = \bar{L}\left(W_{Adam}^{(t')}\right) \in \left[\tilde{\Omega}\left(\frac{1}{d^p}\right), \Theta(d)\right]$, we have*

$$R_{med,k}^{SGDM}(t) = \Omega(\log d), \quad R_{med,k}^{Adam}(t') = 1 + \tilde{\mathcal{O}}\left(\eta^{\frac{1}{4}} + \frac{1}{d^{\frac{\alpha}{2}-\frac{1}{4}}}\right), \quad k = 1, 2.$$

Theorem 1 and Corollary 1 tell us that during a long training period when the loss decreases from $\Theta(d)$ to $\tilde{\mathcal{O}}\left(\frac{1}{d^p}\right)$, the diagonal of loss Hessian for Adam keeps nice uniformity: $R_{med,k}^{Adam}(t) = 1 \pm o(1), k = 1, 2$. On the other hand, the diagonal of loss Hessian for SGD+M is less uniform. Appendix C gives a proof sketch of Theorem 1. The detailed proof can be found in Appendix D and E.

## 6 Low-rank weight matrices and uniformity of leading singular vectors

The proof sketch in Appendix C highlights one crucial intuition of Theorem 1: After $T_{\text{SGD},1}$ (resp. $T_{\text{Adam},1}$) steps, $W_1$ of SGD+M (resp. Adam) becomes an approximately rank-1 matrix. Consider the left singular vector $\boldsymbol{u} := [u_1, u_2, ..., u_d]^T$ which corresponds to the leading singular value $\sigma_1$. We can show that the distribution of $u_1^2, u_2^2, ..., u_d^2$ for Adam is more uniform than that of SGD+M. This property, we call *the uniformity of the leading singular vector*, is related to the uniformity of the diagonal of loss Hessian, see Appendix G for more details.

Similar low rank bias after training has been studied in prior works (*e.g.* [GWB$^+$17, LMZ18, CGMR20]). For more complicated models, we want to check whether the weight matrices also have low rank structures and if so, whether we can still observe *the uniformity of leading singular vectors*. More formally, consider the weight matrix in some layer $W \in \mathbb{R}^{m \times n}$, we want to check

(A) Whether $W \in \mathbb{R}^{m \times n}$ is approximately a rank $k$ matrix with $k \ll \min\{m, n\}$, and if true,

(B) Consider the top $k$ singular values $\sigma_1, ..., \sigma_k$ and left singular vectors $\boldsymbol{u}_1, ..., \boldsymbol{u}_k$. Define $\tilde{\boldsymbol{u}} := \sum_{i=1}^k \sigma_i^2 \boldsymbol{u}_i \odot \boldsymbol{u}_i := [\tilde{u}_1, ..., \tilde{u}_d]^T$ and compute $R_u := \frac{\max_i \tilde{u}_i}{\text{median } \tilde{u}_i}$, a generalized version of $\frac{\max_i u_i^2}{\text{median } u_i^2}$ in the rank-1 case. We want to see whether $R_u$ obtained by Adam is smaller than that of SGD+M.

After reviewing the weight matrices we got in different settings, we observed that (A) and (B) hold for many layers in those models. For example, on the translation task mentioned in Section 4.1, we found 12 layers which had approximately low rank structures and for 10 of them, $R_u$ values (defined in (B)) obtained by Adam were smaller than those found by SGD+M. Figure 4 shows the result on one typical layer. Results of more layers can be found in Appendix A.5.

**Remark** The definition of $R_u$ is based on the connection between diagonal of loss Hessian and weight matrices. Appendix G shows that for a 2-layer linear network, $R_{\text{med},2}^{\text{OPT}}(t) = \frac{\max_i \|W_1^{(t)}[i,:]\|_2^2}{\text{median}\|W_1^{(t)}[i,:]\|_2^2}$. When $W_1 \in \mathbb{R}^{m \times n}$ is approximately rank $k$, i.e. $W_1 \approx \sum_{i=1}^k \sigma_i \boldsymbol{u}_i \boldsymbol{v}_i^T$, denote $\boldsymbol{u}_i = [u_{i1}, u_{i2}, ..., u_{im}]^T$ and $\boldsymbol{v}_i = [v_{i1}, v_{i2}, ..., v_{in}]^T$, we have that for the $j$-th row,

---

[10]The assumption $\beta_2 = \beta_1^2$ is only for convenience to make the proof easier but is not essential. Our result can also generalize to cases with other $\beta_1$ and $\beta_2$. See Appendix C for more discussions.

$\|W_1[j,:]\|_2^2 \approx \left\|\sum_{i=1}^k \sigma_i u_{ij} \boldsymbol{v}_i^T\right\|_2^2 = \sum_{i=1}^k \sigma_i^2 u_{ij}^2$. By defining $\tilde{\boldsymbol{u}} = [\tilde{u}_1, \tilde{u}_2, ..., \tilde{u}_d]^T :=$ $\sum_{i=1}^k \sigma_i^2 \boldsymbol{u}_i \odot \boldsymbol{u}_i$, we have that $\|W_1[j,:]\|_2^2 \approx \tilde{u}_j$. Although in multi-layer nonlinear neural networks, the connection between diagonal of loss Hessian and the weight matrices is more complicated and $R_{\text{med},2}^{\text{OPT}}(t)$ may depend on the product of many weight matrices rather than one single matrix, we still believe that this definition of $R_u$ is a reasonable ratio to consider.

## 7  Conclusion and future work

We demonstrate that adaptive optimization methods bias the training trajectory towards a region where the diagonal of loss Hessian is more uniform, through extensive experiments on language models and theoretical analysis in a simplified setting of two-layer linear networks. Although our findings may not directly lead to an improved algorithm for practical use, they provide a new way of thinking when designing new algorithms: in contrast with the traditional view which tries to design a method that performs better *in* the bad loss geometry, our findings suggest that we can design algorithms which *implicitly avoid* regions with bad geometry. There are a lot of future directions along this line. For example, our theoretical results on the two-layer linear networks may be able to generalize to multi-layer networks. As is discussed in Section 5, the key-value-query structure in the transformer models might be approximated by a three-layer linear network and the analysis of multi-layer networks might provide more connection to real deep models and could be an interesting and challenging future direction. Moreover, it is also possible to relax our large-batch assumption (Assumption 3) and prove similar results in the general stochastic setting.

## Acknowledgments and Disclosure of Funding

This project was supported by NSF CCF 2145703.

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

# A More experiments of the uniformity of diagonal geometry

## A.1 Vanilla SGD vs. Adagrad

In this section, we present the $R_{\mathrm{med}}^{\mathrm{OPT}}(t)$ values defined in eq. (1) obtained by vanilla SGD and Adagrad on a language modeling task[11]. The task is to assign a probability for the likelihood of a given word (or a sequence of words) to follow a sequence of words. We trained a transformer model to solve this problem on Wikitext-2 [MXBS17](CC BY-SA 3.0) with real targets and randomly generated targets. This model has roughly 8 layers (not counting normalization and dropout layers)

The setup is the same as in Section 3.2. We used the same learning rate schedule (constant or decreasing) for SGD and Adagrad. We tuned and chose the best (initial) learning rate of SGD. The (initial) learning rate of Adagrad was set as a value under which Adagrad converged faster than SGD with its best (initial) learning rate. We used large batch sizes to make the training procedure more stable. When computing Hessian, we also used large batch sizes. Due to the extremely large dimension, we did the computation on some uniformly selected coordinates, more precisely, 200 coordinates per layer.

We tried different initialization (normal and uniform) by using different gains of the Pytorch initialization schedule.

### A.1.1 Experiments on real datasets

Figure 5a shows the training losses on Wikitext-2. Table 3 (resp. Table 4) shows the $R_{\mathrm{med}}^{\mathrm{OPT}}(t)$ for Adagrad and SGD under uniform (resp. normal) initialization with different gains.

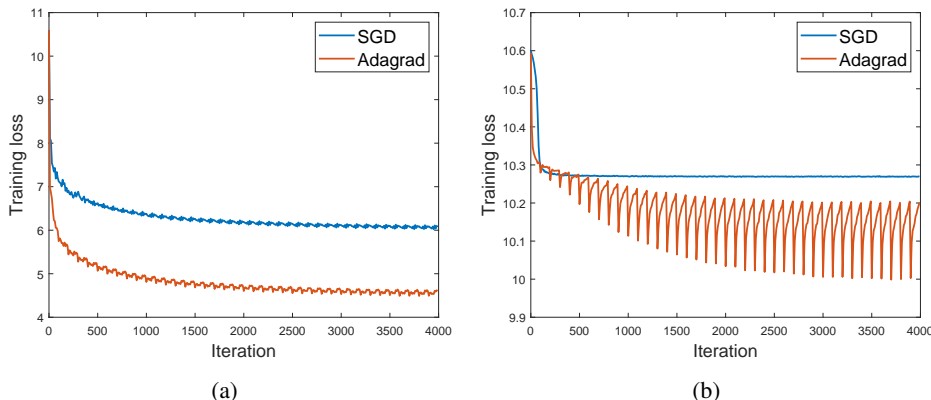

(a)  (b)

Figure 5: Training losses of Adagrad and SGD. (a) on Wikitext-2 and (b) on Wikitext-2 but with randomly generated targets.

Table 3: $R_{\mathrm{med}}^{\mathrm{SGD}}(t)$ and $R_{\mathrm{med}}^{\mathrm{Adagrad}}(t)$ under uniform initialization with different gains

(a) Gain = 2

| Layer# | Epoch 1 | | Epoch 20 | | Epoch 40 | |
|---|---|---|---|---|---|---|
| | $R_{\mathrm{med}}^{\mathrm{SGD}}(t)$ | $R_{\mathrm{med}}^{\mathrm{Adagrad}}(t)$ | $R_{\mathrm{med}}^{\mathrm{SGD}}(t)$ | $R_{\mathrm{med}}^{\mathrm{Adagrad}}(t)$ | $R_{\mathrm{med}}^{\mathrm{SGD}}(t)$ | $R_{\mathrm{med}}^{\mathrm{Adagrad}}(t)$ |
| 1 | 6.07 | 6.77 | 5.91 | 9.77 | 5.16 | 10.37 |
| 2 | 4.60 | 6.26 | 3.43 | 1.66 | 3.44 | 1.88 |
| 3 | 5.15 | 6.84 | 4.35 | 4.34 | 4.84 | 3.60 |
| 4 | 9.47 | 10.78 | 9.76 | 3.54 | 8.67 | 3.14 |
| 5 | 12.54 | 13.96 | 10.31 | 6.59 | 9.79 | 6.98 |
| 6 | 4.92 | 5.25 | 7.21 | 2.33 | 7.94 | 2.28 |
| 7 | 5.73 | 5.45 | 40.56 | 4.57 | 21.24 | 4.76 |
| 8 | 9.39 | 8.87 | 37.95 | 4.50 | 46.03 | 3.19 |

(b) Gain = 0.5

| Layer# | Epoch 1 | | Epoch 20 | | Epoch 40 | |
|---|---|---|---|---|---|---|
| | $R_{\mathrm{med}}^{\mathrm{SGD}}(t)$ | $R_{\mathrm{med}}^{\mathrm{Adagrad}}(t)$ | $R_{\mathrm{med}}^{\mathrm{SGD}}(t)$ | $R_{\mathrm{med}}^{\mathrm{Adagrad}}(t)$ | $R_{\mathrm{med}}^{\mathrm{SGD}}(t)$ | $R_{\mathrm{med}}^{\mathrm{Adagrad}}(t)$ |
| 1 | 69.36 | 78.60 | 15.26 | 7.74 | 18.22 | 7.23 |
| 2 | 24.12 | 24.36 | 4.05 | 2.30 | 3.70 | 2.04 |
| 3 | 2.83 | 2.85 | 3.78 | 4.98 | 3.56 | 4.40 |
| 4 | 5.25 | 4.74 | 3.83 | 5.68 | 3.11 | 4.81 |
| 5 | 66.49 | 67.83 | 88.75 | 19.31 | 63.01 | 15.64 |
| 6 | 6.54 | 6.91 | 3.57 | 2.08 | 3.50 | 1.97 |
| 7 | 3.22 | 3.73 | 13.03 | 3.97 | 9.55 | 4.07 |
| 8 | 6.12 | 5.99 | 6.73 | 7.82 | 5.43 | 6.98 |

---

[11]https://pytorch.org/tutorials/beginner/transformer_tutorial.html

Table 4: $R_{\mathrm{med}}^{\mathrm{SGD}}(t)$ and $R_{\mathrm{med}}^{\mathrm{Adagrad}}(t)$ under normal initialization with different gains

(a) Gain = 1

| Layer# | Epoch 1 | | Epoch 20 | | Epoch 40 | |
|---|---|---|---|---|---|---|
| | $R_{\mathrm{med}}^{\mathrm{SGD}}(t)$ | $R_{\mathrm{med}}^{\mathrm{Adagrad}}(t)$ | $R_{\mathrm{med}}^{\mathrm{SGD}}(t)$ | $R_{\mathrm{med}}^{\mathrm{Adagrad}}(t)$ | $R_{\mathrm{med}}^{\mathrm{SGD}}(t)$ | $R_{\mathrm{med}}^{\mathrm{Adagrad}}(t)$ |
| 1 | 6.76 | 6.06 | 8.27 | 12.28 | 9.69 | 11.17 |
| 2 | 9.51 | 6.61 | 3.19 | 1.87 | 3.21 | 1.73 |
| 3 | 7.38 | 7.35 | 8.61 | 3.38 | 9.25 | 3.94 |
| 4 | 18.02 | 15.63 | 6.45 | 4.86 | 7.49 | 4.44 |
| 5 | 12.70 | 9.35 | 11.69 | 11.23 | 15.07 | 12.18 |
| 6 | 12.76 | 11.86 | 3.84 | 2.32 | 3.20 | 2.09 |
| 7 | 11.79 | 8.58 | 17.95 | 4.32 | 14.99 | 4.50 |
| 8 | 17.09 | 12.73 | 26.70 | 5.16 | 26.91 | 6.73 |

(b) Gain = 0.5

| Layer# | Epoch 1 | | Epoch 20 | | Epoch 40 | |
|---|---|---|---|---|---|---|
| | $R_{\mathrm{med}}^{\mathrm{SGD}}(t)$ | $R_{\mathrm{med}}^{\mathrm{Adagrad}}(t)$ | $R_{\mathrm{med}}^{\mathrm{SGD}}(t)$ | $R_{\mathrm{med}}^{\mathrm{Adagrad}}(t)$ | $R_{\mathrm{med}}^{\mathrm{SGD}}(t)$ | $R_{\mathrm{med}}^{\mathrm{Adagrad}}(t)$ |
| 1 | 9.12 | 14.46 | 10.90 | 8.00 | 10.19 | 8.55 |
| 2 | 10.70 | 15.42 | 8.52 | 2.12 | 8.88 | 2.04 |
| 3 | 5.73 | 5.94 | 10.16 | 2.80 | 6.05 | 2.99 |
| 4 | 16.62 | 12.94 | 8.90 | 3.91 | 8.12 | 4.14 |
| 5 | 15.98 | 16.98 | 42.57 | 10.76 | 18.45 | 10.16 |
| 6 | 4.84 | 6.46 | 7.92 | 2.66 | 5.30 | 2.46 |
| 7 | 6.52 | 6.55 | 107.51 | 3.14 | 136.38 | 2.73 |
| 8 | 8.39 | 8.20 | 337.34 | 5.18 | 315.21 | 4.48 |

### A.1.2 Experiments on random datasets

Figure 5b shows the training losses on Wikitext-2 but with randomly generated targets and Table 5 shows the corresponding $R_{\mathrm{med}}^{\mathrm{SGD}}(t)$ and $R_{\mathrm{med}}^{\mathrm{Adagrad}}(t)$ in different layers.

Table 5: $R_{\mathrm{med}}^{\mathrm{SGD}}(t)$ and $R_{\mathrm{med}}^{\mathrm{Adagrad}}(t)$ on Wikitext-2 but with randomly generated targets

| Layer# | Epoch 1 | | Epoch 20 | | Epoch 40 | |
|---|---|---|---|---|---|---|
| | $R_{\mathrm{med}}^{\mathrm{SGD}}(t)$ | $R_{\mathrm{med}}^{\mathrm{Adagrad}}(t)$ | $R_{\mathrm{med}}^{\mathrm{SGD}}(t)$ | $R_{\mathrm{med}}^{\mathrm{Adagrad}}(t)$ | $R_{\mathrm{med}}^{\mathrm{SGD}}(t)$ | $R_{\mathrm{med}}^{\mathrm{Adagrad}}(t)$ |
| 1 | 10.88 | 10.98 | 9.99 | 18.66 | 9.67 | 22.37 |
| 2 | 9.47 | 12.15 | 14.98 | 4.43 | 13.01 | 3.99 |
| 3 | 7.45 | 8.52 | 459.71 | 6.09 | 451.16 | 5.11 |
| 4 | 9.84 | 10.42 | 135.37 | 7.22 | 126.91 | 6.04 |
| 5 | 7.09 | 7.88 | 103.60 | 353.89 | 184.61 | 190.17 |
| 6 | 7.68 | 8.58 | 18.38 | 4.08 | 18.69 | 2.73 |
| 7 | 7.81 | 5.40 | 294.68 | 62.72 | 229.25 | 29.76 |
| 8 | 13.51 | 9.16 | 329.12 | 20.59 | 203.70 | 9.57 |

### A.2 RMSprop and AMSGrad

In this section, we present the results of RMSprop and AMSGrad and compare them with SGD+M. The experiments were conducted on the translation task described in Section 4.1. We used learning rate 2.5e-5 for RMSprop, 5e-4 for AMSGrad and 0.03 for SGD+M. Both RMSprop and SGD+M used momentum parameter 0.9. The two momentum parameters $(\beta_1, \beta_2)$ of AMSGrad were $(0.9, 0.98)$. Figure 6 shows the training losses and Table 6 shows the corresponding $R_{\mathrm{med}}^{\mathrm{OPT}}(t)$.

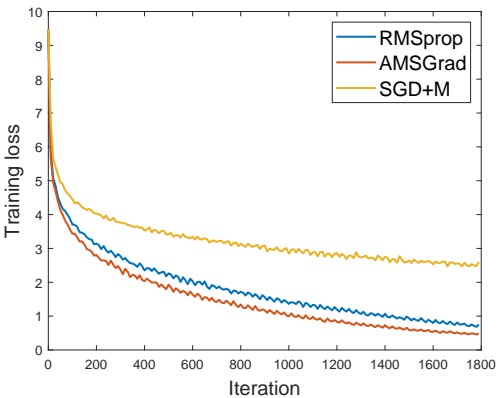

Figure 6: Training losses of RMSprop, AMSGrad and SGD+M on the translation task described in Section 4.1.

Table 6: $R_{\text{med}}^{\text{RMSprop}}(t)$, $R_{\text{med}}^{\text{AMSGrad}}(t)$ and $R_{\text{med}}^{\text{SGDM}}(t)$ in some layers, on the translation task described in Section 4.1
.

| Layer# | Epoch 10 | | | Epoch 20 | | | Epoch 40 | | |
|---|---|---|---|---|---|---|---|---|---|
| | $R_{\text{med}}^{\text{SGDM}}(t)$ | $R_{\text{med}}^{\text{RMSprop}}(t)$ | $R_{\text{med}}^{\text{AMSGrad}}(t)$ | $R_{\text{med}}^{\text{SGDM}}(t)$ | $R_{\text{med}}^{\text{RMSprop}}(t)$ | $R_{\text{med}}^{\text{AMSGrad}}(t)$ | $R_{\text{med}}^{\text{SGDM}}(t)$ | $R_{\text{med}}^{\text{RMSprop}}(t)$ | $R_{\text{med}}^{\text{AMSGrad}}(t)$ |
| 3 | 3.97 | 2.69 | 2.56 | 2.33 | 1.89 | 1.68 | 2.83 | 1.62 | 1.56 |
| 5 | 26.17 | 21.19 | 11.36 | 37.11 | 17.83 | 10.85 | 51.94 | 10.22 | 12.31 |
| 7 | 4.10 | 6.98 | 6.12 | 3.94 | 4.95 | 2.92 | 7.58 | 2.29 | 2.58 |
| 9 | 29.41 | 35.72 | 25.86 | 37.81 | 19.89 | 16.90 | 30.68 | 16.24 | 9.97 |
| 12 | 4.93 | 6.20 | 12.67 | 4.63 | 6.61 | 4.64 | 6.44 | 5.13 | 4.06 |
| 15 | 85.06 | 33.63 | 19.51 | 140.99 | 12.22 | 6.72 | 44.07 | 6.98 | 5.37 |
| 18 | 8.71 | 2.99 | 9.48 | 3.86 | 2.44 | 4.16 | 3.51 | 2.10 | 2.35 |
| 21 | 95.34 | 11.68 | 6.62 | 47.20 | 6.37 | 4.74 | 22.20 | 4.58 | 3.58 |
| 24 | 8.70 | 5.67 | 6.95 | 8.13 | 3.59 | 5.13 | 6.46 | 2.30 | 2.83 |
| 28 | 4.44 | 2.42 | 2.64 | 4.67 | 1.85 | 1.81 | 2.63 | 1.46 | 2.13 |

## A.3 Comparison conditioned on the same loss

In this section, we compare $R_{\text{med}}^{\text{SGDM}}(t)$ and $R_{\text{med}}^{\text{Adam}}(t)$ conditioned on the same training loss. More precisely, we make comparison of $R_{\text{med}}^{\text{Adam}}(t)$ and $R_{\text{med}}^{\text{SGDM}}(t')$, where $t, t'$ are picked such that $t$th Adam iterate and $t'$th SGD+M iterate have the same training loss. The details of the tasks are described in Section 4.1. Table 7 shows the results of $R_{\text{med}}^{\text{Adam}}(t)$ and $R_{\text{med}}^{\text{SGDM}}(t')$ in some layers.

Table 7: $R_{\text{med}}^{\text{Adam}}(t)$ and $R_{\text{med}}^{\text{SGDM}}(t')$ in some layers. Dataset and task: (a) sentence classification task on BERT-small, (b) translation task on Multi30k. See Section 4.1 for detailed setup.

(a)

| Layer# | Loss 0.251 | | Loss 0.170 | | Loss 0.133 | |
|---|---|---|---|---|---|---|
| | $R_{\text{med}}^{\text{SGDM}}(t')$ | $R_{\text{med}}^{\text{Adam}}(t)$ | $R_{\text{med}}^{\text{SGDM}}(t')$ | $R_{\text{med}}^{\text{Adam}}(t)$ | $R_{\text{med}}^{\text{SGDM}}(t')$ | $R_{\text{med}}^{\text{Adam}}(t)$ |
| 9 | 16.77 | 13.69 | 14.14 | 12.71 | 15.17 | 9.86 |
| 12 | 16.68 | 8.29 | 9.98 | 8.31 | 8.90 | 5.42 |
| 15 | 18.64 | 7.79 | 51.39 | 46.43 | 80.82 | 40.97 |
| 17 | 208.29 | 381.05 | 464.37 | 315.58 | 498.26 | 313.99 |
| 18 | 14.43 | 23.56 | 19.17 | 19.26 | 15.76 | 12.99 |
| 22 | 257.32 | 88.47 | 188.55 | 110.87 | 197.79 | 139.48 |
| 24 | 34.22 | 16.34 | 16.42 | 18.08 | 14.04 | 15.97 |

(b)

| Layer# | Loss 3.72 | | Loss 2.78 | | Loss 1.90 | |
|---|---|---|---|---|---|---|
| | $R_{\text{med}}^{\text{SGDM}}(t')$ | $R_{\text{med}}^{\text{Adam}}(t)$ | $R_{\text{med}}^{\text{SGDM}}(t')$ | $R_{\text{med}}^{\text{Adam}}(t)$ | $R_{\text{med}}^{\text{SGDM}}(t')$ | $R_{\text{med}}^{\text{Adam}}(t)$ |
| 3 | 4.01 | 4.45 | 5.80 | 3.02 | 2.44 | 2.28 |
| 5 | 31.19 | 27.50 | 44.29 | 21.46 | 57.83 | 19.52 |
| 7 | 5.80 | 4.38 | 7.51 | 3.71 | 5.25 | 2.87 |
| 9 | 21.23 | 53.65 | 28.99 | 20.92 | 44.26 | 28.13 |
| 13 | 53.18 | 17.77 | 51.17 | 20.64 | 35.80 | 35.49 |
| 15 | 82.30 | 186.41 | 34.17 | 13.76 | 33.87 | 5.31 |
| 21 | 100.43 | 23.66 | 23.45 | 5.12 | 12.96 | 5.35 |
| 26 | 7.45 | 3.48 | 4.69 | 3.10 | 3.33 | 2.83 |
| 30 | 19.14 | 9.54 | 10.46 | 5.48 | 9.56 | 5.33 |

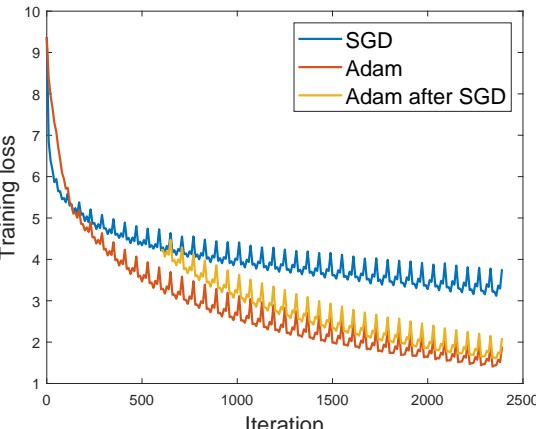

Figure 7: Training losses of SGD, Adam after SGD and Adam for the translation task

Table 8: $R_{\text{med}}^{\text{OPT}}(t)$ of SGD, Adam after SGD and Adam in some layers after roughly 2160 iterations

| Layer# | SGD | Adam | Adam after SGD | Adam |
|--------|--------|--------|----------------|--------|
| 13 | 294.76 | 150.02 | 332.96 | 150.02 |
| 14 | 14.34 | 5.84 | 5.33 | 5.84 |
| 15 | 36.38 | 16.66 | 11.86 | 16.66 |
| 16 | 6.47 | 7.05 | 3.76 | 7.05 |
| 17 | 17.17 | 6.05 | 4.76 | 6.05 |
| 26 | 5.68 | 3.53 | 2.30 | 3.53 |
| 27 | 14.33 | 15.93 | 21.76 | 15.93 |
| 28 | 9.10 | 1.71 | 1.71 | 1.71 |
| 29 | 8.22 | 3.04 | 2.82 | 3.04 |
| 30 | 11.39 | 5.12 | 5.29 | 5.12 |

## A.4 Experiments of switching from SGD to Adam

In this section we describe another learning schedule: the "Adam after SGD" schedule, where we switch from SGD to Adam in the middle to see whether the loss and $R_{\text{med}}^{\text{OPT}}(t)$ can catch up with the model trained by Adam from the very beginning. Again, we used the same model as in the translation task in Section 4.1. In this section, we did not add momentum term to SGD in order to get a larger gap between SGD and Adam than the case using momentum. We want to see whether this larger gap can be closed after switching to Adam in the middle.

As is shown in Figure 7 and Table 8, both the loss gap and the gap of $R_{\text{med}}^{\text{OPT}}(t)$ were closed after a period of training after switching algorithms, which provides evidence of the connection between convergence speed and uniformity of diagonal of loss Hessian.

## A.5 The low rank structure

In this section, we present more results for the experiments in Section 6.

We examined the weights of the model trained for the translation task in Section 4.1. Among roughly 30 layers, we observed that for 12 layers, at least the weight matrices obtained by Adam after training have approximately low rank structures.

Figure 8 shows the examples of layers with or without the low rank structure.

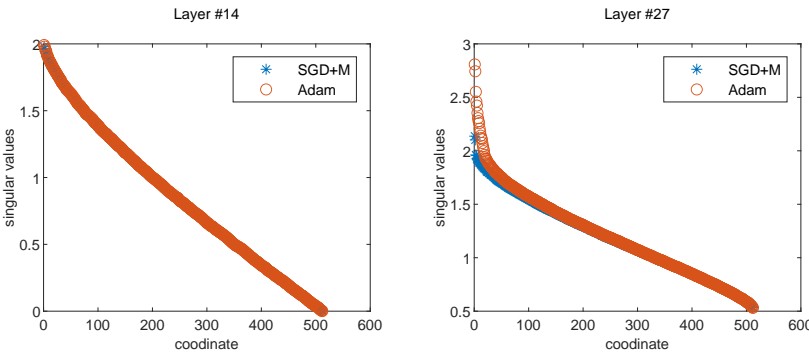

Figure 8: Examples of singular values on (left) layers without low rank structure and (right) layers with approximately low rank structure.

We then studied the uniformity of leading singular vectors of these 12 layers, i.e. computed $R_u$ defined in (B) of Section 6. The observation was that for 10 out of these 12 layers, $R_u$ values of Adam were smaller those of SGD, which implies the uniformity of leading left singular vectors of Adam. Here we may also want to consider the right singular vectors $\boldsymbol{v}_1, \boldsymbol{v}_2, ... \boldsymbol{v}_k$ and corresponding $\tilde{\boldsymbol{v}} = [\tilde{v}_1, \tilde{v}_2, ..., \tilde{v}_d]^T := \sum_{i=1}^{k} \sigma_i^2 \boldsymbol{v}_i \odot \boldsymbol{v}_i$ and compute $R_v := \frac{\max_i \tilde{v}_i}{\text{median } \tilde{v}_i}$ for Adam and SGD+M. However, on this translation task, among the 12 layers which were approximately low rank, for only 6 of them, $R_v$ of Adam were smaller, i.e. we did not observe uniformity of the leading right singular

vector for Adam. One possible reason is that for a weight matrix, its right singular vectors are closer to the input data than left singular vectors and more easily influenced by the data, therefore may not show uniformity. Figure 9 shows how $R_u$ and $R_v$ changed over time in some layers.

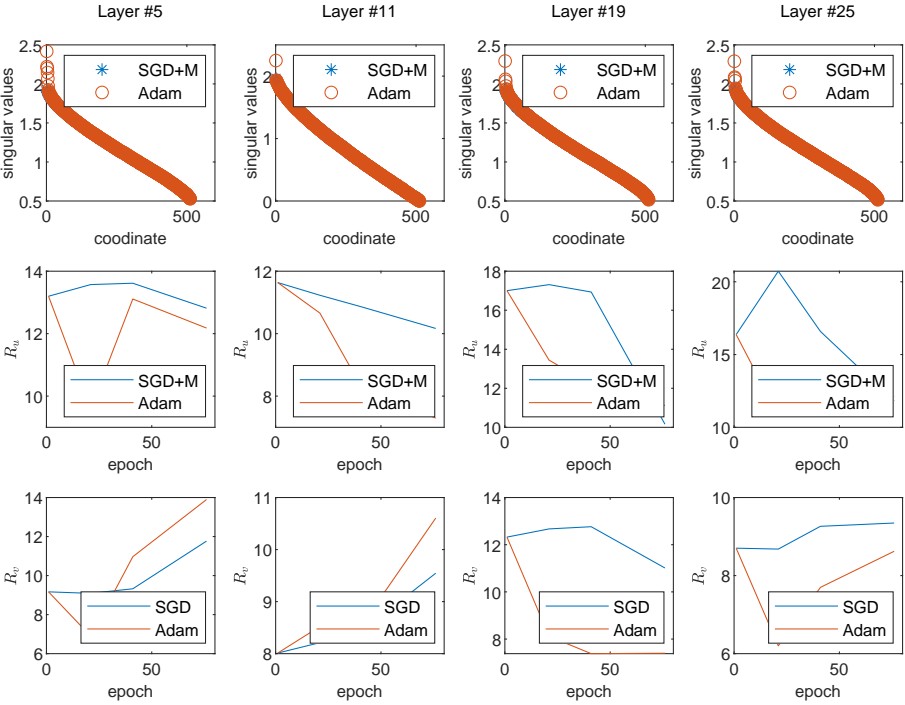

Figure 9: $R_u$ and $R_v$ for Adam and SGD+M in some layers

## A.6 How does the (adaptive) gradient align with diagonal of loss Hessian

In this section we present the uniformity of diagonal geometry of adaptive methods from another perspective. Denote $H_{ii}$ as the $(i, i)$-th element of the loss Hessian $H$ and $g_i$ as the $i$-th element of the gradient. It is conjectured that when $|H_{ii}|$ is large, the corresponding $|g_i|$ is usually large as well. For adaptive methods, we can regard the update per step as the learning rate times the "adaptive gradient". Let's use $g_{\text{adapt},i}$ to represent the $i$-th component of the adaptive gradient. Through experiments on language models, we found that $|g_{\text{adapt},i}|$ for different $i$ are quite uniform and do not align with $|H_{ii}|$ as the true gradient $|g_i|$ does.

In the experiments, we first sorted $|H_{ii}|$ in the ascent order: $|H_{i_1,i_1}| \leq |H_{i_2,i_2}| \leq ... \leq |H_{i_d,i_d}|$ (suppose $H \in \mathbb{R}^{d \times d}$), and then plotted the corresponding $|g_{i_k}|$ and $|g_{\text{adapt},i_k}|$ for $k \in [d]$.

### A.6.1 SGD vs. Adagrad

Here we compare SGD and Adagrad on the language modeling task on wikitext-2 described in Section A.1. We observed that the figures of all layers are quite similar so we select one layer as an example, as is shown in Figure 10.

### A.6.2 SGD with momentum vs. Adam

Here we compare Adam and SGD+M on the tasks described in Section 4.1. Again, we select one layer as an example for each task. Figure 11 shows the results on the sentence classification task and Figure 12 shows the results on the translation task.

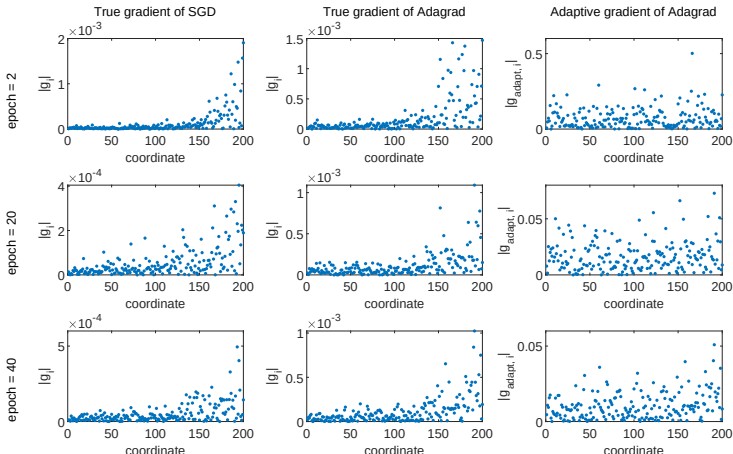

Figure 10: How the true gradient ($\{|g_{i_k}|\}_{k=1}^d$) and "adaptive gradient" ($\{|g_{\text{adapt},i_k}|\}_{k=1}^d$) align with diagonal of Hessian ($\{|H_{i_k,i_k}|\}_{k=1}^d$). Here coordinates are sorted such that $|H_{i_1,i_1}| \leq |H_{i_2,i_2}| \leq \ldots \leq |H_{i_d,i_d}|$ (suppose $H \in \mathbb{R}^{d \times d}$). Experiments were conducted on the model described in Section A.1. This figure shows the results on the 12-th layer.

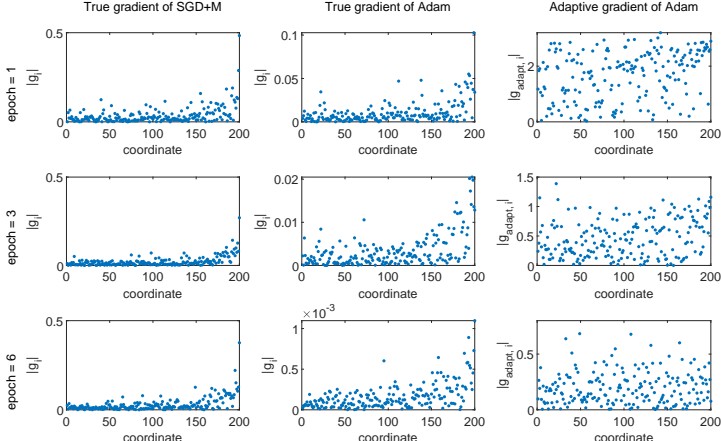

Figure 11: How the true gradient ($\{|g_{i_k}|\}_{k=1}^d$) and "adaptive gradient" ($\{|g_{\text{adapt},i_k}|\}_{k=1}^d$) align with diagonal of Hessian ($\{|H_{i_k,i_k}|\}_{k=1}^d$). Here coordinates are sorted such that $|H_{i_1,i_1}| \leq |H_{i_2,i_2}| \leq \ldots \leq |H_{i_d,i_d}|$ (suppose $H \in \mathbb{R}^{d \times d}$). Experiments were conducted on the sentence classification task described in Section 4.1. This figure shows the results on the 12-th layer.

## A.7  Adding regularization and other tricks

In this section, we add weight decay to both Adam and SGD+M on the translation task described in Section 4. The momentum parameter $\beta$ in SGD was set as 0.9. The two momentum parameters $(\beta_1, \beta_2)$ of Adam were set as (0.9, 0.98). For both algorithms, we set the weight decay parameter as 0.001. We trained the model using constant learning rates for 60 epochs (1800 iterations). We tuned and chose the best learning rate 0.03 for SGD+M. The learning rate of Adam was set as 0.0001, under which Adam converged faster than SGD+M with its best learning rate 0.03. Figure 13a shows the training losses and Table 9 shows the values of $R_{\text{med}}^{\text{Adam}}(t)$, $R_{\text{med}}^{\text{SGDM}}(t)$ and $\frac{R_{\text{med}}^{\text{SGDM}}(t)}{R_{\text{med}}^{\text{Adam}}(t)}$ in some randomly selected layers.

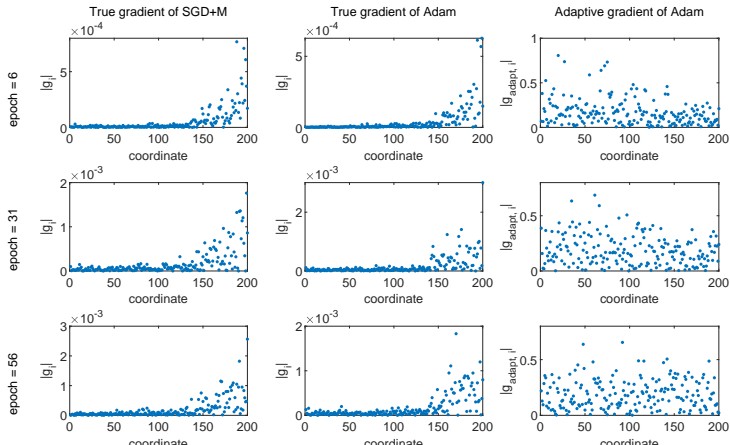

Figure 12: How the true gradient ($\{|g_{i_k}|\}_{k=1}^d$) and "adaptive gradient" ($\{|g_{\text{adapt},i_k}|\}_{k=1}^d$) align with diagonal of Hessian ($\{|H_{i_k,i_k}|\}_{k=1}^d$). Here coordinates are sorted such that $|H_{i_1,i_1}| \leq |H_{i_2,i_2}| \leq ... \leq |H_{i_d,i_d}|$ (suppose $H \in \mathbb{R}^{d \times d}$). Experiments were conducted on the translation task described in Section 4.1. This figure shows the results on the 5-th layer.

Table 9: $R_{\text{med}}^{\text{Adam}}(t)$ and $R_{\text{med}}^{\text{SGDM}}(t)$ (both using weight decay) in some layers for the translation task.

| Layer# | Epoch 0 | | Epoch 30 | | | Epoch 55 | | |
|---|---|---|---|---|---|---|---|---|
| | $R_{\text{med}}^{\text{SGDM}}(t)$ | $R_{\text{med}}^{\text{Adam}}(t)$ | $R_{\text{med}}^{\text{SGDM}}(t)$ | $R_{\text{med}}^{\text{Adam}}(t)$ | $\frac{R_{\text{med}}^{\text{SGDM}}(t)}{R_{\text{med}}^{\text{Adam}}(t)}$ | $R_{\text{med}}^{\text{SGDM}}(t)$ | $R_{\text{med}}^{\text{Adam}}(t)$ | $\frac{R_{\text{med}}^{\text{SGDM}}(t)}{R_{\text{med}}^{\text{Adam}}(t)}$ |
| 3 | 73.09 | 73.09 | 17.65 | 13.11 | 1.35 | 13.38 | 6.28 | 2.13 |
| 5 | 469.88 | 469.88 | 293.48 | 310.85 | 0.94 | 601.68 | 588.12 | 1.02 |
| 7 | 80.78 | 80.78 | 8.22 | 39.65 | 0.21 | 13.65 | 4.85 | 2.81 |
| 9 | 494.27 | 494.27 | 150.14 | 123.79 | 1.21 | 301.89 | 119.53 | 2.53 |
| 15 | 632.10 | 632.10 | 277.18 | 175.34 | 1.58 | 334.48 | 282.88 | 1.18 |
| 18 | 55.08 | 55.08 | 6.56 | 4.45 | 1.47 | 23.88 | 4.52 | 5.29 |
| 21 | 549.62 | 549.62 | 257.89 | 44.78 | 5.76 | 515.99 | 53.79 | 9.59 |
| 24 | 107.51 | 107.51 | 8.54 | 3.64 | 2.34 | 53.79 | 3.32 | 16.20 |
| 28 | 13.77 | 13.77 | 4.74 | 2.37 | 2.00 | 15.60 | 2.15 | 7.24 |
| 30 | 491.62 | 491.62 | 6.91 | 2.66 | 2.60 | 9.60 | 2.02 | 4.77 |

## A.8 Results on image tasks

Although in this paper we focus on language models where Adam shows significant fast convergence, we also add results in this section on image tasks where SGD+M performs better. We trained a ResNet[12] on CIFAR-10 dataset and compared the convergence speed and $R_{\text{med}}^{\text{OPT}}(t)$ of SGD+M and Adam. The momentum parameter $\beta$ in SGD was set as 0.9. The two momentum parameters $(\beta_1, \beta_2)$ of Adam were set as $(0.9, 0.98)$. The model was trained using constant learning rates for 41 epochs (2050 iterations). We tuned and chose the best learning rates for both algorithms: 0.5 for SGD+M and 0.005 for Adam. Figure 13b shows the training losses and Table 10 shows the values of $R_{\text{med}}^{\text{Adam}}(t)$, $R_{\text{med}}^{\text{SGDM}}(t)$ and $\frac{R_{\text{med}}^{\text{SGDM}}(t)}{R_{\text{med}}^{\text{Adam}}(t)}$.

As we can see, on this image task, Adam does not converge faster than SGD+M and in the meantime, $R_{\text{med}}^{\text{Adam}}(t)$ values were no longer smaller than $R_{\text{med}}^{\text{SGDM}}(t)$ during training. This reveals the connection between the local diagonal geometry and the convergence speed from another perspective. That is, when the diagonal of Hessian of Adam is not more uniform than SGD+M, its convergence speed is not better, either.

---

[12]We borrowed the implementation here `https://pytorch-tutorial.readthedocs.io/en/latest/tutorial/chapter03_intermediate/3_2_2_cnn_resnet_cifar10/` and replace the "layers" array [2,2,2] with [1,1,1].

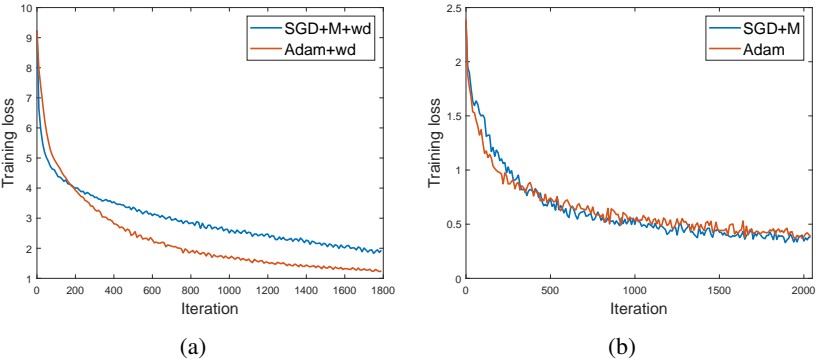

(a)  (b)

Figure 13: (a) Training losses of Adam and SGD+M for the translation task, both with weight decay. (b) Training losses of Adam and SGD+M for a ResNet trained on CIFAR-10.

Table 10: $R_{\text{med}}^{\text{Adam}}(t)$ and $R_{\text{med}}^{\text{SGDM}}(t)$ for ResNet on CIFAR-10.

| Layer# | Epoch 10 | | | Epoch 20 | | | Epoch 40 | | |
|---|---|---|---|---|---|---|---|---|---|
| | $R_{\text{med}}^{\text{SGDM}}(t)$ | $R_{\text{med}}^{\text{Adam}}(t)$ | $\frac{R_{\text{med}}^{\text{SGDM}}(t)}{R_{\text{med}}^{\text{Adam}}(t)}$ | $R_{\text{med}}^{\text{SGDM}}(t)$ | $R_{\text{med}}^{\text{Adam}}(t)$ | $\frac{R_{\text{med}}^{\text{SGDM}}(t)}{R_{\text{med}}^{\text{Adam}}(t)}$ | $R_{\text{med}}^{\text{SGDM}}(t)$ | $R_{\text{med}}^{\text{Adam}}(t)$ | $\frac{R_{\text{med}}^{\text{SGDM}}(t)}{R_{\text{med}}^{\text{Adam}}(t)}$ |
| 1 | 6.88 | 25.34 | 0.27 | 3.74 | 39.35 | 0.09 | 4.39 | 15.80 | 0.28 |
| 2 | 110.19 | 35.93 | 3.07 | 32.97 | 36.27 | 0.91 | 60.69 | 28.06 | 2.16 |
| 3 | 40.89 | 16.92 | 2.42 | 13.98 | 15.92 | 0.88 | 11.70 | 37.01 | 0.32 |
| 4 | 28.56 | 23.66 | 1.21 | 11.48 | 13.04 | 0.88 | 7.99 | 14.51 | 0.55 |
| 5 | 13.47 | 23.78 | 0.57 | 8.64 | 12.07 | 0.72 | 6.52 | 14.23 | 0.46 |
| 6 | 18.72 | 12.49 | 1.50 | 12.19 | 8.80 | 1.38 | 8.96 | 21.69 | 0.41 |
| 7 | 18.85 | 39.25 | 0.48 | 9.00 | 12.81 | 0.70 | 13.87 | 11.42 | 1.22 |
| 8 | 13.79 | 19.91 | 0.69 | 8.87 | 11.72 | 0.76 | 7.48 | 9.34 | 0.80 |
| 9 | 12.50 | 14.85 | 0.84 | 9.62 | 8.06 | 1.19 | 11.35 | 8.08 | 1.41 |
| 10 | 14.89 | 14.53 | 1.02 | 8.15 | 5.80 | 1.41 | 6.21 | 8.89 | 0.70 |

## A.9   Comparison between $R_{\text{med}}^{\text{OPT}}(t)$ and singular value-based metrics

In Section 4, through extensive experiments on language models, we demonstrate that when we train the neural network using Adam, the uniformity of diagonal geometry, measured by $R_{\text{med}}^{\text{OPT}}(t)$ is smaller than that when we train using SGD+M from the same initialization, for the vast majority of layers. We are aware that people also usually consider Hessian singular values instead of diagonal entries to measure the loss geometry. Hence in this section we make a comparison between our diagonal-based metric and singular value-based metrics.

First, we believe that our metric has a natural connection to the mechanism that underlies adaptive methods. Adaptive methods in practice choose coordinate-wise adaptive learning rates. From a high-level perspective, this procedure can be viewed as adapting to the loss smoothness with respect to each coordinate. The smoothness of certain coordinate is measured by the second derivative with respect to this coordinate and therefore corresponds to the diagonal entries of loss Hessian. Our metric, which measures these diagonal entries, is thus fundamentally intertwined with the mechanism that underlies adaptive methods.

Next, we empirically demonstrate that our metric $R_{\text{med}}^{\text{OPT}}(t)$ is a reasonable proxy of singular value-based metrics. Define a singular value-based metric $S_{\text{med}}^{\text{OPT}}(t) := \frac{\max\{\sigma_i(t)\}_{i=1}^d}{\text{median }\{\sigma_i(t)\}_{i=1}^d}$ as an analogy of our diagonal-based metric $R_{\text{med}}^{\text{OPT}}(t)$, where $\{\sigma_i(t)\}_{i=1}^d$ denotes the singular values of loss Hessian $H(t) \in \mathbb{R}^{d\times d}$ at the $t$th iterate. We compare $S_{\text{med}}^{\text{OPT}}(t)$ along the trajectories of Adam and SGD+M in the translation task described in Section 4.1. Table 11 suggests that if measured by singular values, Adam is also biased to a region with smaller $S_{\text{med}}^{\text{OPT}}(t)$ than SGD+M, similar to the observation for $R_{\text{med}}^{\text{OPT}}(t)$. This is expected because in Appendix F, we demonstrate that the loss Hessian approaches diagonal during training. The fact that our diagonal-based metric and singular value-based metric give the same result also reveals the robustness of our observation to the choice of metric, demonstrating that there does exist some geometry bias of Adam towards more uniform regions even when measured by different metrics.

Finally, there is strong reason why our metric is often easier to compute empirically and analyze theoretically than singular value-based metrics such as $S_{\text{med}}^{\text{OPT}}(t)$.

1. From the empirical computation perspective, suppose the loss Hessian is $d \times d$. Then computing its singular values, in general, requires computing the whole matrix with $d^2$ elements. However, our metric only requires computing the $d$ diagonal entries.

2. From the theoretical analysis perspective, in Appendix G, we show that the diagonal of loss Hessian in linear networks can be connected to weight matrices by simple formulas. These straightforward formulas simplify the analysis and allow us to connect our metric to the low-rank structure of weight matrices and the uniformity of their leading singular vectors (see Section 6 for more discussions). However, all these nice connections fail to hold for singular value-based metrics. The formulas of singular values are very complicated even in linear networks, making it almost impossible to theoretically analyze any singular value-based metrics.

Table 11: $S_{\text{med}}^{\text{Adam}}(t)$ and $S_{\text{med}}^{\text{SGDM}}(t)$ in some layers for the translation task.

| Layer# | Epoch 0 | | Epoch 30 | | | Epoch 55 | | |
|---|---|---|---|---|---|---|---|---|
| | $S_{\text{med}}^{\text{SGDM}}(t)$ | $S_{\text{med}}^{\text{Adam}}(t)$ | $S_{\text{med}}^{\text{SGDM}}(t)$ | $S_{\text{med}}^{\text{Adam}}(t)$ | $\frac{S_{\text{med}}^{\text{SGDM}}(t)}{S_{\text{med}}^{\text{Adam}}(t)}$ | $S_{\text{med}}^{\text{SGDM}}(t)$ | $S_{\text{med}}^{\text{Adam}}(t)$ | $\frac{S_{\text{med}}^{\text{SGDM}}(t)}{S_{\text{med}}^{\text{Adam}}(t)}$ |
| 3 | 4.53 | 4.53 | 6.08 | 2.87 | 2.12 | 5.92 | 2.58 | 2.30 |
| 5 | 14.64 | 14.64 | 40.01 | 15.38 | 2.60 | 52.28 | 15.05 | 3.47 |
| 7 | 6.91 | 6.91 | 9.84 | 5.06 | 1.94 | 12.25 | 4.22 | 2.90 |
| 9 | 24.12 | 24.12 | 42.02 | 30.89 | 1.36 | 33.20 | 21.54 | 1.54 |
| 12 | 19.07 | 19.07 | 32.41 | 24.84 | 1.30 | 28.83 | 14.23 | 2.03 |
| 15 | 47.03 | 47.03 | 69.97 | 11.54 | 6.06 | 42.71 | 7.19 | 5.94 |
| 18 | 15.96 | 15.96 | 26.03 | 29.73 | 0.88 | 18.46 | 17.94 | 1.03 |
| 21 | 31.03 | 31.03 | 25.84 | 7.92 | 3.26 | 19.71 | 7.06 | 2.79 |
| 24 | 35.42 | 35.42 | 21.31 | 18.08 | 1.18 | 14.62 | 10.33 | 1.41 |
| 28 | 55.38 | 55.38 | 6.18 | 2.77 | 2.23 | 4.84 | 2.01 | 2.41 |

### A.10 The uniformity of diagonal geometry on in-distribution test data

In this section we compare $R_{\text{med}}^{\text{Adam}}(t)$ and $R_{\text{med}}^{\text{SGDM}}(t)$ on the in-distribution test data. The task is the translation task described in Section 4.1. Table 12 validates that on in-distribution test data, Adam is also biased to a region with smaller $R_{\text{med}}^{\text{OPT}}(t)$ than SGD+M, similar to what happens on the training data shown in Table 2a. This is expected because of the same distribution. One other thing we want to emphasize is that, in real language tasks, the dataset is typically very large and the model see each training example only once. Hence the training behavior usually implies similar in-distribution test behavior.

Table 12: $R_{\text{med}}^{\text{Adam}}(t)$ and $R_{\text{med}}^{\text{SGDM}}(t)$ in some layers for the translation task on in-distribution test data.

| Layer# | Epoch 0 | | Epoch 30 | | | Epoch 55 | | |
|---|---|---|---|---|---|---|---|---|
| | $R_{\text{med}}^{\text{SGDM}}(t)$ | $R_{\text{med}}^{\text{Adam}}(t)$ | $R_{\text{med}}^{\text{SGDM}}(t)$ | $R_{\text{med}}^{\text{Adam}}(t)$ | $\frac{R_{\text{med}}^{\text{SGDM}}(t)}{R_{\text{med}}^{\text{Adam}}(t)}$ | $R_{\text{med}}^{\text{SGDM}}(t)$ | $R_{\text{med}}^{\text{Adam}}(t)$ | $\frac{R_{\text{med}}^{\text{SGDM}}(t)}{R_{\text{med}}^{\text{Adam}}(t)}$ |
| 3 | 4.39 | 4.39 | 5.80 | 3.06 | 1.89 | 6.79 | 2.81 | 2.42 |
| 5 | 7.90 | 7.90 | 38.01 | 11.71 | 3.24 | 41.40 | 10.21 | 4.06 |
| 7 | 5.77 | 5.77 | 6.00 | 4.61 | 1.30 | 5.53 | 3.20 | 1.73 |
| 9 | 25.09 | 25.09 | 28.81 | 17.17 | 1.68 | 16.67 | 14.85 | 1.12 |
| 12 | 10.24 | 10.24 | 9.13 | 8.63 | 1.06 | 13.78 | 9.09 | 1.52 |
| 15 | 79.71 | 79.71 | 77.18 | 13.56 | 5.69 | 37.93 | 9.91 | 3.83 |
| 18 | 14.78 | 14.78 | 3.94 | 7.15 | 0.55 | 5.42 | 6.04 | 0.90 |
| 21 | 83.25 | 83.25 | 26.04 | 5.44 | 4.79 | 13.11 | 5.57 | 2.36 |
| 24 | 29.91 | 29.91 | 6.89 | 5.42 | 1.27 | 6.51 | 7.16 | 0.91 |
| 28 | 22.57 | 22.57 | 5.39 | 3.94 | 1.37 | 6.13 | 2.14 | 2.87 |

# B    Supplementary result on the contribution of small $R_{\mathbf{med}}^{\mathbf{OPT}}(t)$ to fast optimization

## B.1    Supplementary empirical result

As explained in Section 1, Figure 1 demonstrates that Adam biases the trajectory towards a region which benefits fast optimization, and in the meantime, it has more uniform diagonal Hessian (characterized by smaller $R_{\text{med}}^{\text{OPT}}(t)$). However, since the two trajectories in Figure 1 come from two different algorithms (SGD+M and Adam), it is reasonable to ask whether the benefit of Adam's trajectory is indeed due to smaller $R_{\text{med}}^{\text{OPT}}(t)$ or it is due to other properties of Adam and small $R_{\text{med}}^{\text{OPT}}(t)$ is just a byproduct. To address this concern, we add the following experiment where we select two trajectories that both come from SGD+M and have different $R_{\text{med}}^{\text{OPT}}(t)$.

We ran SGD+M twice on the translation task described in Section 4.1 with learning rates 0.01 and 0.03 and got two trajectories. Then we selected an iterate $x_1$ on the trajectory with learning rate 0.01 and $x_2$ on the trajectory with learning rate 0.03 such that $x_1$ and $x_2$ have the same training loss and different $R_{\text{med}}^{\text{OPT}}(t)$. We then ran SGD+M with the same configuration twice, once starting from $x_1$ and once from $x_2$. The observation is that $x_2$ has a more uniform diagonal Hessian (characterized by smaller $R_{\text{med}}^{\text{OPT}}(t)$, see Table 13) than $x_1$, and in the meantime, SGD+M starting from $x_2$ also converges faster (see Figure 14). Hence we demonstrate that a small value of $R_{\text{med}}^{\text{OPT}}(t)$ is not just a byproduct of adaptive algorithms (since SGD+M with different learning rates can also yield trajectories with different $R_{\text{med}}^{\text{OPT}}(t)$), and it is indeed a contributing factor of fast optimization.

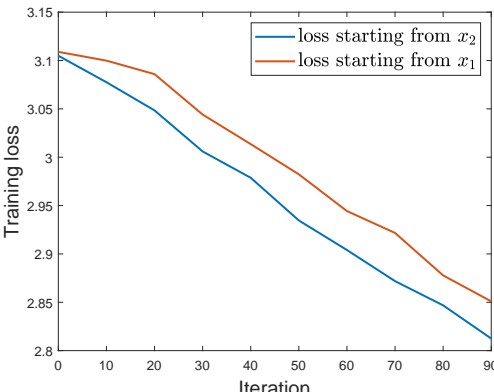

Figure 14: Training losses of SGD+M starting from $x_1$ and $x_2$

Table 13: $R_{\text{med}}^{\text{OPT}}(t)$ values of $x_1$ and $x_2$ in some layers

| Layer# | $R_{\text{med}}^{\text{OPT1}}$ (on the iterate $x_1$) | $R_{\text{med}}^{\text{OPT2}}$ (on the iterate $x_2$) | $\frac{R_{\text{med}}^{\text{OPT1}}}{R_{\text{med}}^{\text{OPT2}}}$ |
|---|---|---|---|
| 3 | 2.89 | 2.73 | 1.06 |
| 5 | 69.73 | 51.44 | 1.36 |
| 7 | 4.39 | 4.81 | 0.91 |
| 9 | 42.93 | 35.89 | 1.20 |
| 12 | 7.09 | 6.18 | 1.15 |
| 15 | 97.25 | 68.92 | 1.41 |
| 18 | 3.50 | 3.74 | 0.94 |
| 21 | 71.69 | 45.07 | 1.59 |
| 24 | 7.49 | 5.42 | 1.38 |
| 28 | 4.85 | 4.00 | 1.21 |

## B.2 Theoretical analysis on a simple setting

In this section, we add the following lemma on a simplified setting where smaller $R_{\text{med}}^{\text{OPT}}(t)$ provably contributes to faster optimization.

**Lemma 1.** *Consider a diagonal quadratic loss $f(x) = \sum_{i=1}^{d} \frac{\lambda_i}{2} x_i^2$ where $\lambda_1 \geq \lambda_2 \geq ... \geq \lambda_d \geq 0$. Run GD with learning rate $\eta = \frac{1}{\lambda_1}$ and produce $\{x_i^{(t)}\}_{t \geq 0}, 1 \leq i \leq d$. We initialize the parameters as $x_i^{(0)} = \pm 1, \forall i \in [d]$. Without loss of generality, suppose $d$ is even. Then we have that for any $p > 0$, after $T = \frac{p}{2} \cdot R_{med}^{OPT} \cdot \log d = \frac{p}{2} \cdot \frac{\lambda_1}{median\{\lambda_i\}_{i=1}^d} \log d$ steps, the loss $f(x)$ will shrink by a half. More precisely,*

$$f\left(x^{(T)}\right) \leq \left(\frac{1}{2} + d^{-p}\right) f\left(x^{(0)}\right).$$

Lemma 1 demonstrates that under this simplified setting, even when we just use the vanilla GD with no adaptive methods, a more uniform diagonal geometry (characterized by smaller $R_{\text{med}}^{\text{OPT}}$) can still lead to faster optimization.

*Proof.* By the update of GD with learning rate $\eta = \frac{1}{\lambda_1}$, we have that

$$\forall i \in [d]: \quad x_i^{(t+1)} = \left(1 - \frac{\lambda_i}{\lambda_1}\right) x_i^{(t)},$$

$$\Rightarrow \quad \left(x_i^{(t)}\right)^2 = \left(1 - \frac{\lambda_i}{\lambda_1}\right)^{2t} \left(x_i^{(0)}\right)^2 \leq \exp\left(-\frac{2\lambda_i t}{\lambda_1}\right) \left(x_i^{(0)}\right)^2.$$

Then after $T = \frac{p}{2} \cdot \frac{\lambda_1}{\text{median}\{\lambda_i\}_{i=1}^d} \log d$ steps, we have that

$$\forall i \leq \frac{d}{2}: \quad \left(x_i^{(T)}\right)^2 \leq d^{-p} \left(x_i^{(0)}\right)^2, \Rightarrow \sum_{i=1}^{d/2} \lambda_i \left(x_i^{(T)}\right)^2 \leq d^{-p} \sum_{i=1}^{d/2} \lambda_i \left(x_i^{(0)}\right)^2 \leq d^{-p} f\left(x^{(0)}\right).$$

On the other hand,

$$\sum_{i=d/2+1}^{d} \lambda_i \left(x_i^{(T)}\right)^2 \leq \sum_{i=d/2+1}^{d} \lambda_i \left(x_i^{(0)}\right)^2 \overset{(i)}{\leq} \sum_{i=1}^{d/2} \lambda_i \left(x_i^{(0)}\right)^2,$$

where $(i)$ is by definition of median and the initialization $\forall i \in [d]: x_i^{(0)} = \pm 1$. Combining with the fact $\sum_{i=d/2+1}^{d} \lambda_i \left(x_i^{(0)}\right)^2 + \sum_{i=1}^{d/2} \lambda_i \left(x_i^{(0)}\right)^2 = f\left(x^{(0)}\right)$ yields that $\sum_{i=d/2+1}^{d} \lambda_i \left(x_i^{(T)}\right)^2 \leq \frac{1}{2} f\left(x^{(0)}\right)$. Hence we have

$$f\left(x^{(T)}\right) = \sum_{i=1}^{d/2} \lambda_i \left(x_i^{(T)}\right)^2 + \sum_{i=d/2+1}^{d} \lambda_i \left(x_i^{(T)}\right)^2 \leq \left(\frac{1}{2} + d^{-p}\right) f\left(x^{(0)}\right).$$

$\square$

## C Proof sketch of Theorem 1

Now we give a proof sketch of Theorem 1, which contains three major steps. The detailed proof can be found in Appendix G, D and E.

First we relate the diagonal of Hessian to weight matrices $W_1, W_2$. Under Assumption 1, denote $W_1[i,:]$ as the $i$-th row of $W_1$ and $W_2 := [w_{2i}, w_{22}, ..., w_{2d}]$. Since the input dataset is whitened, we can show that

$$R_{\text{med},1}^{\text{OPT}}(t) = \frac{\max_i \left(w_{2i}^{(t)}\right)^2}{\text{median} \left(w_{2i}^{(t)}\right)^2}, \quad R_{\text{med},2}^{\text{OPT}}(t) = \frac{\max_i \left\|W_1^{(t)}[i,:]\right\|_2^2}{\text{median} \left\|W_1^{(t)}[i,:]\right\|_2^2}.$$

Next, due to the one-dimensional output, we can prove that $W_1$ converges to an approximately rank-1 matrix. More precisely, we have

$$W_1^{(t)} = \boldsymbol{u}^{(t)} \boldsymbol{v}^{(t)T} + R_1^{(t)},$$

$$W_2^{(t)} = c^{(t)} \boldsymbol{u}^{(t)T} + R_2^{(t)T}.$$

where $c^{(t)}$ is a scalar, $\boldsymbol{u}^{(t)}, \boldsymbol{v}^{(t)}, R_2^{(t)} \in \mathbb{R}^d$ and $R_1^{(t)} \in \mathbb{R}^{d \times d}$. Denote the $i$-th coordinate of $\boldsymbol{u}^{(t)}, \boldsymbol{v}^{(t)}, R_2^{(t)}$ as $u_i^{(t)}, v_i^{(t)}, R_{2i}^{(t)}$, respectively. Denote the $(i,j)$-th element of $R_1^{(t)}$ as $R_1^{(t)}[i,j]$. We have that $\forall i, j \in [d] : \left| R_{2i}^{(t)} \right| \ll c^{(t)} \left| u_i^{(t)} \right|$ and $\left| R_1^{(t)}[i,j] \right| \ll \left| u_i^{(t)} v_i^{(t)} \right|$.

Using the rank 1 structure, we can further simplify $R_{\text{med},1}^{\text{OPT}}(t)$ and $R_{\text{med},2}^{\text{OPT}}(t)$ by

$$R_{\text{med},k}^{\text{OPT}}(t) \approx \frac{\max_i \left( u_i^{(t)} \right)^2}{\text{median} \left( u_i^{(t)} \right)^2}, k = 1, 2. \tag{4}$$

The final step is the detailed analysis of $\boldsymbol{u}^{(t)}$.

For SGD+M, we can prove that $\boldsymbol{u}^{(t)} \approx C(t)[X_1, X_2, ..., X_d]^T$ where $C(t) \in \mathbb{R}$ and $X_i, i \in [d]$ are i.i.d. Gaussian variables. Then we have with high probability, $\frac{\max_i \left( u_i^{(t)} \right)^2}{\text{median} \left( u_i^{(t)} \right)^2} = \Omega(\log d)$.

For Adam, we can try to convert it to signed descent (*e.g.* in eq. (29) and Lemma 24). The main intuition is that as long as the learning rate is small enough, the movement of the gradient will be slow and Adam will be similar to signed descent. Since the update of signed descent per step is $\pm\eta$, we can prove that $\forall i \in [d] : u_i^{(t)} \in \{\pm 1\}$, which gives us $\frac{\max_i \left( u_i^{(t)} \right)^2}{\text{median} \left( u_i^{(t)} \right)^2} = 1$. Substituting into eq. (4) completes the proof.

Finally, we want to emphasize that the assumption $\beta_2 = \beta_1^2$ is not essential. As discussed above, as long as the learning rate is sufficiently small, Adam can be approximated to signed descent regardless of the ratio between $\beta_1, \beta_2$. Hence our result can generalize to cases with other $\beta_1, \beta_2$, for example, $\beta_2 > \beta_1^2$ where Zhang et al. [ZCS+22] proved the convergence of Adam to stationary points.

## D   Analysis of SGD+M

Note that $A = \frac{1}{m} Y X^T$, $\Lambda_{xx} := \frac{1}{m} X X^T$. Denote $g_k^{(t)} := \nabla_{W_k} L(W^{(t)}), k = 1, 2$. We have that

$$g_1^{(t)} = W_2^{(t)T} \left( W_2^{(t)} W_1^{(t)} - A \right), \quad g_2^{(t)} = \left( W_2^{(t)} W_1^{(t)} - A \right) W_1^{(t)T}.$$

Let $\tilde{A}^{(t)}, \tilde{\Lambda}_{xx}^{(t)}$ and $\tilde{g}_k^{(t)}, k = 1, 2$ be the corresponding batch versions at time $t$. Let $E^{(t)} := W_2^{(t)} W_1^{(t)} - A$, and use $E_i^{(t)}, A_i$ and $\left( W_2^{(t)} W_1^{(t)} \right)_i$ to represent the $i$-th coordinates of $E^{(t)}, A$ and $W_2^{(t)} W_1^{(t)}$, respectively. By eq. (2), the update rules of $W_1$ and $W_2$ for SGD+M are given by:

$$W_1^{(t+1)} = W_1^{(t)} - \eta \sum_{\tau=0}^{t} \beta^{t-\tau} W_2^{(\tau)T} \left( W_2^{(\tau)} W_1^{(\tau)} - A \right) - \eta \sum_{\tau=0}^{t} \beta^{t-\tau} Dg_1^{(\tau)},$$

$$W_2^{(t+1)} = W_2^{(t)} - \eta \sum_{\tau=0}^{t} \beta^{t-\tau} \left( W_2^{(\tau)} W_1^{(\tau)} - A \right) W_1^{(\tau)T} - \eta \sum_{\tau=0}^{t} \beta^{t-\tau} Dg_2^{(\tau)},$$

where

$$Dg_1^{(t)} := \tilde{g}_1^{(t)} - g_1^{(t)} = W_2^{(t)T} \left( W_2^{(t)} W_1^{(t)} \left( \tilde{\Lambda}_{xx}^{(t)} - \Lambda_{xx} \right) - \left( \tilde{A}^{(t)} - A \right) \right),$$

$$Dg_2^{(t)} := \tilde{g}_2^{(t)} - g_2^{(t)} = \left( W_2^{(t)} W_1^{(t)} \left( \tilde{\Lambda}_{xx}^{(t)} - \Lambda_{xx} \right) - \left( \tilde{A}^{(t)} - A \right) \right) W_1^{(t)T}.$$

Based on the magnitude of $W_2$ and $W_1$, we can intuitively divide the training procedure into 2 phases.

1. **First phase**: the first several iterations when $W_1$ and $W_2$ are "small" so that $W_2W_1 - A \approx -A$.

2. **Second phase**: later iterations when $W_2W_1$ cannot be ignored.

More formally, the boundary between the first and second phase is defined below.

**Definition 1** (End of the first phase). *The end of the first phase (denoted as $T_1$) is defined as*
$$T_1 := \inf\left\{t \geq 0 : \exists i,j \in [d] : \left|w_{2i}^{(t)}\right| \geq \frac{1}{d^{\frac{\alpha}{2}}} \text{ or } \left|W_1^{(t)}[i,j]\right| \geq \frac{1}{d^{\frac{\alpha}{2}}}\right\}.$$

By Assumption 2 and the assumption that $\forall j \in [d] : A_j > 0, A_j = \Theta(1)$, at the beginning, w.h.p., $\forall j \in [d] : (W_2W_1)_j - A_j < 0$. During the training, each $(W_2W_1)_j$ increases and approaches $A_j$. We hope that by choosing a small learning rate, when $(W_2W_1)_j$ overshoots for some coordinate $j$, i.e. $(W_2W_1)_j > A_j$, it will be close to convergence. To analyze this overshooting issue more carefully, let's first define the following "almost overshooting time".

**Definition 2** (Almost overshooting time). *For $\epsilon > 0$, denote $\epsilon_0 := \frac{1}{d^{\frac{1}{4}\alpha-1}} + \epsilon \log \sqrt{\frac{d}{\epsilon}}$. Define*
$$T_2 := \inf\left\{t \geq 0 : \exists j \in [d] : \left(W_2^{(t)}W_1^{(t)}\right)_j - A_j \geq -\sqrt{\epsilon_0}\right\}.$$

**Definition 3** (Convergence time). *For $\epsilon > 0$, we define the "convergence time"*
$$T_3 := \inf\left\{t \geq 0 : \left\|E^{(t)}\right\|_2^2 \leq \epsilon\right\}.$$

We can first show that after the first phase, i.e. when $t = T_1$, $W_1$ will become an approximately rank-1 matrix, as described in the following lemma.

**Lemma 2.** *Under Assumption 1, 2 and 3, suppose $\sigma \leq \frac{\eta^{3/2}}{d^{\alpha/2+1}}$. By picking $\eta \leq \mathcal{O}\left(\frac{1}{d^\alpha}\right)$, we have that when $t = T_1$, $\bar{L}\left(W^{(T_1)}\right) = \Theta(d)$, and that*
$$W_1^{(T_1)} = R_1^{(T_1)} + \boldsymbol{u}^{(T_1)}\boldsymbol{v}^{(T_1)T},$$
$$W_2^{(T_1)} = R_2^{(T_1)T} + c^{(T_1)}\boldsymbol{u}^{(T_1)T},$$

*where $c^{(T_1)} \in \mathbb{R}$, $\boldsymbol{u}^{(T_1)}, \boldsymbol{v}^{(T_1)}, R_2^{(T_1)} \in \mathbb{R}^d$ and $R_1^{(T_1)} \in \mathbb{R}^{d \times d}$. Denote the $i$-th coordinate of $\boldsymbol{u}^{(T_1)}, \boldsymbol{v}^{(T_1)}, R_2^{(T_1)}$ as $u_i^{(T_1)}, v_i^{(T_1)}, R_{2i}^{(T_1)}$, respectively, and the $(i,j)$-th element of $R_1^{(T_1)}$ as $R_1^{(T_1)}[i,j]$. Then w.h.p.,*
$$\forall 1 \leq i,j \leq d : \quad \frac{\left|R_1^{(T_1)}[i,j]\right|}{\left|u_i^{(T_1)}v_j^{(T_1)}\right|} \leq \tilde{\mathcal{O}}\left(\frac{1}{d^{\frac{1}{4}\alpha-1}}\right), \quad \frac{\left|R_{2i}^{(T_1)}\right|}{\left|c^{(T_1)}u_i^{(T_1)}\right|} \leq \tilde{\mathcal{O}}\left(\frac{1}{d^{\frac{1}{4}\alpha-1}}\right).$$

The following lemma tells us that this approximate rank-1 structure is preserved when $T_1 \leq t \leq \min\{T_2, T_3\}$.

**Lemma 3.** *Under Assumption 1, 2 and 3, suppose $\sigma \leq \frac{\eta^{3/2}}{d^{\alpha/2+1}}$. By picking $\eta \leq \mathcal{O}\left(\frac{\epsilon}{d^{\frac{7\alpha}{4}+4}}\right)$, we have that w.h.p. for $T_1 \leq t \leq \min\{T_2, T_3\}$,*
$$W_1^{(t)} = \boldsymbol{u}^{(T_1)}\boldsymbol{v}^{(t)T} + R_1^{(t)},$$
$$W_2^{(t)} = c^{(t)}\boldsymbol{u}^{(T_1)T} + R_2^{(t)T}.$$

*where*
$$\forall 1 \leq i,j \leq d : \quad \frac{\left|R_1^{(t)}[i,j]\right|}{\left|u_i^{(T_1)}v_j^{(t)}\right|} \leq \tilde{\mathcal{O}}(\epsilon_0), \quad \frac{\left|R_{2i}^{(t)}\right|}{\left|c^{(t)}u_i^{(T_1)}\right|} \leq \tilde{\mathcal{O}}(\epsilon_0),$$

*and $\epsilon_0$ is defined in Definition 2. Moreover, when $t = \min\{T_2, T_3\}$, $\bar{L}\left(W^{(t)}\right) = \mathcal{O}(\epsilon_0 d)$.*

The following lemma gives us a more detailed description of $\boldsymbol{u}^{(T_1)}$.

**Lemma 4.** *The $\boldsymbol{u}^{(T_1)}$ in Lemma 2 and 3 can be written as $\boldsymbol{u}^{(T_1)} = X + Y$ where $X_i, i \in [d]$ are i.i.d Gaussian random variables and that w.h.p. $\forall i \in [d] : \frac{|Y_i|}{|X_i|} \leq \tilde{\mathcal{O}}\left(\frac{1}{d^{\frac{1}{4}\alpha-\frac{1}{2}}}\right).$*

Now we are ready to prove the SGD+M part of Theorem 1.

### D.1 Proof of the SGD+M part of Theorem 1

Define $T_{\text{SGD},1} = T_1, T_{\text{SGD},2} = \min\{T_2, T_3\}$. By picking $\eta \le \mathcal{O}\left(\frac{\epsilon}{d^{\frac{7\alpha}{4}+4}}\right)$, we can apply Lemma 2 and 3 to conclude that $\bar{L}\left(W^{(T_{\text{SGD},1})}\right) = \Theta(d)$ and $\bar{L}\left(W^{(T_{\text{SGD},2})}\right) = \mathcal{O}(\epsilon_0 d)$. For any $p > 0$, by picking $0 < \epsilon < \frac{1}{d^p}$ and $\alpha \ge 4(p+2)$, we have $\bar{L}\left(W^{(T_{\text{SGD},2})}\right) = \mathcal{O}(\epsilon_0 d) \le \tilde{\mathcal{O}}\left(\frac{1}{d^p}\right)$.

Moreover, when $t \in [T_{\text{SGD},1}, T_{\text{SGD},2}]$, the conditions in Lemma 31 are satisfied with $\delta = \tilde{\mathcal{O}}(\epsilon_0)$. Then we can apply Lemma 31 and get that

$$R_{\text{med},1}^{\text{SGDM}}(t), R_{\text{med},2}^{\text{SGDM}}(t) \ge \left(\frac{1 - \tilde{\mathcal{O}}(\epsilon_0)}{1 + \tilde{\mathcal{O}}(\epsilon_0)}\right)^2 \cdot \frac{\max_i \left(u_i^{(T_1)}\right)^2}{\text{median}\left(u_i^{(T_1)}\right)^2}.$$

By Lemma 4, $\boldsymbol{u}^{(T_1)} = X + Y$ where w.h.p. $\forall i \in [d] : \frac{|Y_i|}{|X_i|} \le \tilde{\mathcal{O}}\left(\frac{1}{d^{\frac{1}{4}\alpha - \frac{1}{2}}}\right)$. This fact yields

$$\forall i \in [d] : \frac{\max_i \left(u_i^{(T_1)}\right)^2}{\text{median}\left(u_i^{(T_1)}\right)^2} \ge \left(\frac{1 - \tilde{\mathcal{O}}\left(\frac{1}{d^{\frac{1}{4}\alpha - \frac{1}{2}}}\right)}{1 + \tilde{\mathcal{O}}\left(\frac{1}{d^{\frac{1}{4}\alpha - \frac{1}{2}}}\right)}\right)^2 \frac{\max_i X_i^2}{\text{median } X_i^2}.$$

Here $X_i, i \in [d]$ are i.i.d Gaussian random variables by Lemma 4. To prove the concentration of median $X_i^2$, we borrow the Proposition 12 in Chapter 2.3 of [Ler19]. By setting $K = N = d$ in this proposition, we have

$$\mathbb{P}\left(\left|\text{median } X_i^2 - \mathbb{E}[X_1^2]\right| > 2\sqrt{\text{Var}(X_1^2)}\right) \le e^{-\frac{d}{8}}.$$

Denote $\sigma^2$ as the variance of $X_i, i \in [d]$. Then $\mathbb{E}[X_i^2] = \sigma^2$ and $\text{Var}(X_i^2) = 2\sigma^4$. Hence

$$\mathbb{P}\left(\left|\text{median } X_i^2 - \sigma^2\right| > 2\sqrt{2}\sigma^2\right) \le e^{-\frac{d}{8}}.$$

That means with high probability, median $X_i^2 \le C\sigma^2$ for some $C > 0$. By Lemma 35 in Appendix H, we know that w.h.p.

$$\max_{1 \le i \le d} X_i^2 = \sigma^2 \Omega(\log d),$$

which gives us w.h.p.

$$\frac{\max_{1 \le i \le d} X_i^2}{\text{median } X_i^2} = \Omega(\log d).$$

Hence we have proved that $R_{\text{med},1}^{\text{SGDM}}(t), R_{\text{med},2}^{\text{SGDM}}(t) \ge \Omega(\log d)$.

### D.2 Proof of Lemma 2

In the first phase, $W_2 W_1$ is "small", and we write the update equations in the following way

$$\begin{aligned}
W_1^{(t+1)} &= W_1^{(t)} - \eta \sum_{\tau=0}^{t} \beta^{t-\tau} W_2^{(\tau)T}\left(W_2^{(\tau)} W_1^{(\tau)} - A\right) - \eta \sum_{\tau=0}^{t} \beta^{t-\tau} D g_1^{(\tau)} \\
&= W_1^{(t)} + \eta \sum_{\tau=0}^{t} \beta^{t-\tau} W_2^{(\tau)T} A - \eta \sum_{\tau=0}^{t} \beta^{t-\tau} W_2^{(\tau)T} W_2^{(\tau)} W_1^{(\tau)} - \eta \sum_{\tau=0}^{t} \beta^{t-\tau} D g_1^{(\tau)} \\
&= W_1^{(t)} + \eta W_2^{(t)T} A \sum_{\tau=0}^{t} \beta^{t-\tau} + \eta \sum_{\tau=0}^{t} \beta^{t-\tau}\left(W_2^{(\tau)T} - W_2^{(t)T}\right) A \qquad (5) \\
&\quad - \eta \sum_{\tau=0}^{t} \beta^{t-\tau} W_2^{(\tau)T} W_2^{(\tau)} W_1^{(\tau)} - \eta \sum_{\tau=0}^{t} \beta^{t-\tau} D g_1^{(\tau)} \\
&= W_1^{(t)} + \frac{\eta}{1-\beta} W_2^{(t)T} A + \frac{\eta}{1-\beta} r_1^{(t)},
\end{aligned}$$

where

$$r_1^{(t)} = -\beta^{t+1} W_2^{(t)T} A + (1-\beta) \sum_{\tau=0}^{t} \beta^{t-\tau} \left( W_2^{(\tau)T} - W_2^{(t)T} \right) A$$

$$- (1-\beta) \sum_{\tau=0}^{t} \beta^{t-\tau} W_2^{(\tau)T} W_2^{(\tau)} W_1^{(\tau)} - (1-\beta) \sum_{\tau=0}^{t} \beta^{t-\tau} D g_1^{(\tau)}.$$

Similarly, we have

$$W_2^{(t+1)} = W_2^{(t)} - \eta \sum_{\tau=0}^{t} \beta^{t-\tau} \left( W_2^{(\tau)} W_1^{(\tau)} - A \right) W_1^{(\tau)T} = W_2^{(t)} + \frac{\eta}{1-\beta} A W_1^{(t)T} + \frac{\eta}{1-\beta} r_2^{(t)}, \quad (6)$$

where

$$r_2^{(t)} = -\beta^{t+1} A W_1^{(t)T} + (1-\beta) \sum_{\tau=0}^{t} \beta^{t-\tau} A \left( W_1^{(\tau)T} - W_1^{(t)T} \right)$$

$$- (1-\beta) \sum_{\tau=0}^{t} \beta^{t-\tau} W_2^{(\tau)} W_1^{(\tau)} W_1^{(\tau)T} - (1-\beta) \sum_{\tau=0}^{t} \beta^{t-\tau} D g_2^{(\tau)}.$$

The following lemma gives us an explicit formula of $W_2^{(t)}$.

**Lemma 5.** *Let $\lambda_1 < \lambda_2$ be the two roots of the quadratic equation $x^2 - 2x + 1 - \frac{\eta^2}{(1-\beta)^2} \|A\|_2^2 = 0$. Pick $\eta < \frac{1-\beta}{\|A\|_2}$, then we have that*

$$W_2^{(t)} = C_1 \lambda_1^t + \left( C_2 + r_5^{(t)} \right) \lambda_2^t,$$

*where $C_1 = -\frac{W_2^{(1)} - \lambda_2 W_2^{(0)}}{\lambda_2 - \lambda_1}$, $C_2 = \frac{W_2^{(1)} - \lambda_1 W_2^{(0)}}{\lambda_2 - \lambda_1}$. $r_5^{(t)}$ will be specified in the proof.*

We can prove that in the first phase, $r_5^{(t)}$ is "small". More specifically, denote its $i$-th coordinate as $r_{5i}^{(t)}$, and the $i$-th coordinate of $C_2$ as $C_{2i}$. Then the following lemmas tell us that $\forall i \in [d], \left| r_{5i}^{(t)} \right| \leq \mathcal{O}\left( \frac{1}{d^{p(\alpha)}} \right)$, where w.h.p. $\mathcal{O}\left( \frac{1}{d^{p(\alpha)}} \right) \ll \min_{i \in [d]} |C_{2i}|$.

We first have the following bounds of $\left| r_{1i}^{(t)} \right|$, $\left| r_{2i}^{(t)} \right|$ and $\left| r_{5i}^{(t)} \right|$ for $i \in [d]$.

**Lemma 6.** *Under Assumption 1, 2 and 3, suppose $\sigma \leq \frac{\eta^{3/2}}{d^{\alpha/2+1}}$ and pick $\eta \leq \mathcal{O}\left( \frac{1}{d^\alpha} \right)$. We have w.h.p. for all $t \leq T_1$, $\forall i \in [d] : \left| r_1^{(t)}[i,j] \right| \leq \tilde{\mathcal{O}}\left( \frac{1}{d^{\frac{3}{2}\alpha - 1}} \right), \left| r_{2i}^{(t)} \right| \leq \tilde{\mathcal{O}}\left( \frac{1}{d^{\frac{3}{2}\alpha - 2}} \right).$*

**Lemma 7.** *Under conditions of Lemma 6, we have that w.h.p. for all $t \leq T_1$, $\forall i \in [d] : \left| r_{5i}^{(t)} \right| \leq \tilde{\mathcal{O}}\left( \frac{1}{d^{\frac{3}{2}\alpha - 1}} \right).$*

Next we prove upper and lower bounds of $|C_{1i}|$ and $|C_{2i}|$ for $i \in [d]$.

**Lemma 8.** *Under Assumption 1, 2 and 3, suppose $\sigma \leq \frac{\eta^{3/2}}{d^{\alpha/2+1}}$. Pick $\eta < \frac{1-\beta}{\|A\|_2}$, we have that*
*i) w.h.p., $\forall i \in [d] : |C_{1i}| \leq \tilde{\mathcal{O}}\left( \frac{1}{d^\alpha} \right), |C_{2i}| \leq \tilde{\mathcal{O}}\left( \frac{1}{d^\alpha} \right)$;*
*ii) $C_2$ can be written as $C_2 := \frac{1}{2}(C_3 + C_4)$ where $C_{3i}, i \in [d]$ are i.i.d Gaussian random variables and that w.h.p. $\forall i \in [d] : \frac{|C_{4i}|}{|C_{3i}|} \leq \tilde{\mathcal{O}}\left( \frac{1}{d^{\frac{1}{4}\alpha - \frac{1}{2}}} \right)$;*
*iii) w.h.p., $\forall i \in [d], |C_{1i}| \geq \tilde{\Omega}\left( \frac{1}{d^{\frac{5}{4}\alpha}} \right), |C_{2i}| \geq \tilde{\Omega}\left( \frac{1}{d^{\frac{5}{4}\alpha}} \right).$*

Now we are ready to prove Lemma 2. Lemma 5 tells us that

$$W_2^{(t)} = C_1 \lambda_1^t + \left( C_2 + r_5^{(t)} \right) \lambda_2^t,$$

where $\lambda_1 = 1 - \frac{\eta}{1-\beta} \|A\|_2$ and $\lambda_2 = 1 + \frac{\eta}{1-\beta} \|A\|_2$.

Under the conditions of Theorem 1 and pick $\eta \leq \mathcal{O}\left(\frac{1}{d^\alpha}\right)$, by Lemma 7 and 8, we know that w.h.p. $\forall t \leq T_1, \forall 1 \leq i \leq d$,

$$\left|r_{5i}^{(t)}\right| \leq \tilde{\mathcal{O}}\left(\frac{1}{d^{\frac{3}{2}\alpha-1}}\right), \quad |C_{2i}| \geq \tilde{\Omega}\left(\frac{1}{d^{\frac{5}{4}\alpha}}\right), \quad |C_{2i}| \leq \tilde{\mathcal{O}}\left(\frac{1}{d^\alpha}\right), \quad \frac{\left|r_{5i}^{(t)}\right|}{|C_{2i}|} \leq \tilde{\mathcal{O}}\left(\frac{1}{d^{\frac{1}{4}\alpha-1}}\right).$$
(7)

We first prove that $\left|w_{2i}^{(t)}\right|$ reaches $\frac{1}{d^{\alpha/2}}$ for some coordinate $i$ before $\left|W_1^{(t)}[k,j]\right|$ for $\forall k, j \in [d]$. To see this, first note that

$$W_1^{(t)} = W_1^{(t-1)} + \frac{\eta}{1-\beta}W_2^{(t-1)T}A + \frac{\eta}{1-\beta}r_1^{(t-1)}$$

$$= W_1^{(0)} + \frac{\eta}{1-\beta}\sum_{\tau=0}^{t-1}W_2^{(\tau)T}A + \frac{\eta}{1-\beta}\sum_{\tau=0}^{t-1}r_1^{(\tau)}$$

$$= W_1^{(0)} + \frac{\eta}{1-\beta}\left(C_1\sum_{\tau=0}^{t-1}\lambda_1^\tau + C_2\sum_{\tau=0}^{t-1}\lambda_2^\tau + \sum_{\tau=0}^{t-1}\lambda_2^\tau r_5^{(\tau)}\right)^T A + \frac{\eta}{1-\beta}\sum_{\tau=0}^{t-1}r_1^{(\tau)}$$

$$= W_1^{(0)} + \frac{\eta}{1-\beta}\sum_{\tau=0}^{t-1}r_1^{(\tau)} + \frac{\eta}{1-\beta}\left(C_1\sum_{\tau=0}^{t-1}\lambda_1^\tau + \sum_{\tau=0}^{t-1}\lambda_2^\tau r_5^{(\tau)}\right)^T A + \frac{\eta}{1-\beta}\sum_{\tau=0}^{t-1}\lambda_2^\tau C_2^T A$$

$$:= W_1^{(0)} + \frac{\eta}{1-\beta}\sum_{\tau=0}^{t-1}r_1^{(\tau)} + \left(C_1\sum_{\tau=0}^{t-1}\lambda_1^\tau + \sum_{\tau=0}^{t-1}\lambda_2^\tau r_5^{(\tau)}\right)^T v^{(t)T} + u^{(t)}v^{(t)T},$$

where $v^{(t)T} = \frac{\eta}{1-\beta}A$ and

$$u^{(t)} = \sum_{\tau=0}^{t-1}\lambda_2^\tau C_2^T.$$
(8)

Moreover, we have that

$$W_2^{(t)} = C_1\lambda_1^t + \left(C_2 + r_5^{(t)}\right)\lambda_2^t := C_1\lambda_1^t + r_5^{(t)}\lambda_2^t + c^{(t)}u^{(t)T}, \quad c^{(t)} = \frac{\lambda_2^t}{\sum_{\tau=0}^{t-1}\lambda_2^\tau}.$$

For $t \leq T_1$, by eq. (7), we get that w.h.p.,

$$\forall 1 \leq i, j \leq d: \quad \frac{\left|\sum_{\tau=0}^{t-1}\lambda_2^\tau r_{5i}^{(\tau)}v_j^{(t)}\right|}{\left|u_i^{(t)}v_j^{(t)}\right|} \leq \frac{\tilde{\mathcal{O}}\left(\frac{1}{d^{\frac{3}{2}\alpha-1}}\right)\sum_{\tau=0}^{t-1}\lambda_2^\tau}{\tilde{\Omega}\left(\frac{1}{d^{\frac{5}{4}\alpha}}\right)\sum_{\tau=0}^{t-1}\lambda_2^\tau} \leq \tilde{\mathcal{O}}\left(\frac{1}{d^{\frac{1}{4}\alpha-1}}\right),$$

$$\frac{\left|\lambda_2^t r_{5i}^{(t)}\right|}{\left|c^{(t)}u_i^{(t)}\right|} = \frac{\left|r_{5i}^{(t)}\right|}{|C_{2i}|} \leq \tilde{\mathcal{O}}\left(\frac{1}{d^{\frac{1}{4}\alpha-1}}\right).$$

For $t \leq T_1$, by Lemma 6, $\forall 1 \leq i, j \leq d: \left|r_1^{(t)}[i,j]\right| \leq \tilde{\mathcal{O}}\left(\frac{1}{d^{\frac{3}{2}\alpha-1}}\right)$. Then we have that w.h.p.

$$\frac{\left|\frac{\eta}{1-\beta}\sum_{\tau=0}^{t-1}r_1^{(\tau)}[i,j]\right|}{\left|u_i^{(t)}v_j^{(t)}\right|} = \frac{\left|\sum_{\tau=0}^{t-1}r_1^{(\tau)}[i,j]\right|}{\left|\sum_{\tau=0}^{t-1}\lambda_2^\tau C_{2i}A_j\right|} \leq \frac{\left|\sum_{\tau=0}^{t-1}r_1^{(\tau)}[i,j]\right|}{\left|\sum_{\tau=0}^{t-1}C_{2i}A_j\right|} \leq \frac{\sum_{\tau=0}^{t-1}\tilde{\mathcal{O}}\left(\frac{1}{d^{\frac{3}{2}\alpha-1}}\right)}{\sum_{\tau=0}^{t-1}\tilde{\Omega}\left(\frac{1}{d^{\frac{5}{4}\alpha}}\right)}$$

$$= \tilde{\mathcal{O}}\left(\frac{1}{d^{\frac{1}{4}\alpha-1}}\right).$$

Here we used $\forall i \in [d]: A_i = \Theta(1)$ by Assumption 1.

Since $\lambda_1 = 1 - \frac{\eta}{1-\beta}\|A\|_2$, we have that $|C_{1i}\lambda_1^t| \leq |C_{1i}| \leq \tilde{\mathcal{O}}\left(\frac{1}{d^\alpha}\right)$ and that

$$\left|C_{1i}\sum_{\tau=0}^{t-1}\lambda_1^\tau v_j^{(t)}\right| = \frac{\eta A_j}{1-\beta}\left|C_{1i}\sum_{\tau=0}^{t-1}\lambda_1^\tau\right| \leq \frac{\eta A_j|C_{1i}|}{(1-\beta)(1-\lambda_1)} \leq \frac{A_j|C_{1i}|}{\|A\|_2} \leq \tilde{\mathcal{O}}\left(\frac{1}{d^{\alpha+\frac{1}{2}}}\right).$$

Using the Gaussian tail bound and union bound, we have w.h.p. $\forall 1 \leq i, j \leq d :$ $\left|W_1^{(0)}[i,j]\right| = \tilde{\mathcal{O}}\left(\frac{1}{d^{2\alpha}}\right)$. Combining the above bounds together yields that for $t \leq T_1$ and $\forall i,j \in [d]$,

$$
\begin{aligned}
W_1^{(t)}[i,j] &= R_{11}^{(t)}[i,j] + u_i^{(t)} v_j^{(t)} (1 + e_1^{(t)}[i,j]), \\
w_{2i}^{(t)} &= R_{21,i}^{(t)} + c^{(t)} u_i^{(t)} (1 + e_{2i}^{(t)}).
\end{aligned}
\tag{9}
$$

where for $\forall i,j \in [d]$. $\left|R_{11}^{(t)}[i,j]\right| \leq \tilde{\mathcal{O}}\left(\frac{1}{d^{\alpha+\frac{1}{2}}}\right)$, $\left|R_{21,i}^{(t)}\right| \leq \tilde{\mathcal{O}}\left(\frac{1}{d^\alpha}\right)$ and $\left|e_1^{(t)}[i,j]\right|, \left|e_{2i}^{(t)}\right| \leq \tilde{\mathcal{O}}\left(\frac{1}{d^{\frac{1}{4}\alpha-1}}\right)$.

Further we notice that for $t \leq T_1$, we have $\forall j \in [d]$,

$$
\frac{\left|v_j^{(t)}\right|}{\left|c^{(t)}\right|} = \frac{\eta A_j}{1-\beta} \cdot \frac{\sum_{\tau=0}^{t-1} \lambda_2^\tau}{\lambda_2^t} = \frac{\eta A_j}{1-\beta} \frac{\lambda_2^t - 1}{\lambda_2^t(\lambda_2 - 1)} = \frac{A_j(\lambda_2^t - 1)}{\lambda_2^t \|A\|_2} \leq \frac{A_j}{\|A\|_2} = \mathcal{O}\left(\frac{1}{\sqrt{d}}\right).
$$

which yields that $\left|u_i^{(t)} v_j^{(t)}\right| \leq \mathcal{O}\left(\frac{1}{\sqrt{d}}\right) \left|c^{(t)} u_i^{(t)}\right|$. Together with eq. (9) gives us that $\left|w_{2i}^{(t)}\right|$ reaches $\frac{1}{d^{\alpha/2}}$ for some $i \in [d]$ before $\left|W_1^{(t)}[k,j]\right|$ for $\forall k,j \in [d]$, i.e. $T_1 = \inf\left\{t \geq 0 : \exists i \in [d] : \left|w_{2i}^{(t)}\right| \geq \frac{1}{d^{\frac{\alpha}{2}}}\right\}$.

Further, we know that at time $T_1$, $\left|c^{(T_1)} u_{i_0}^{(T_1)}\right| = |C_{2i_0}| \lambda_2^{T_1} = \Theta\left(\frac{1}{d^{\alpha/2}}\right)$ for some $i_0 \in [d]$, which means w.h.p.

$$
\frac{\Theta\left(\frac{1}{d^{\frac{\alpha}{2}}}\right)}{\tilde{\mathcal{O}}\left(\frac{1}{d^\alpha}\right)} \leq \lambda_2^{T_1} \leq \frac{\Theta\left(\frac{1}{d^{\frac{\alpha}{2}}}\right)}{\tilde{\Omega}\left(\frac{1}{d^{\frac{5}{4}\alpha}}\right)}, \quad \Rightarrow \quad \tilde{\Omega}\left(d^{\frac{\alpha}{2}}\right) \leq \lambda_2^{T_1} = \left(1 + \frac{\eta}{1-\beta}\|A\|_2\right)^{T_1} \leq \tilde{\mathcal{O}}\left(d^{\frac{3}{4}\alpha}\right),
$$

$$
\Rightarrow \quad T_1 = \Theta\left(\frac{\log d}{\eta\|A\|_2}\right).
\tag{10}
$$

This is the length of the first phase. As for $c^{(T_1)} u_i^{(T_1)}$ and $u_i^{(T_1)} v_j^{(T_1)}$ for other coordinates, we have that w.h.p. $\forall 1 \leq i,j \leq d$,

$$
\begin{aligned}
\left|u_i^{(T_1)} v_j^{(T_1)}\right| &= \frac{\eta}{1-\beta} \sum_{\tau=0}^{T_1-1} \lambda_2^\tau |C_{2i} A_j| = \frac{\eta}{1-\beta} \cdot \frac{\lambda_2^{T_1} - 1}{\lambda_2 - 1} |C_{2i} A_j| \stackrel{(i)}{=} \frac{\lambda_2^{T_1} - 1}{\|A\|_2} |C_{2i} A_j| \\
&\geq \frac{\tilde{\Omega}\left(d^{\alpha/2}\right)}{\Theta\left(\sqrt{d}\right)} \tilde{\Omega}\left(\frac{1}{d^{\frac{5}{4}\alpha}}\right) = \tilde{\Omega}\left(\frac{1}{d^{\frac{3}{4}\alpha+\frac{1}{2}}}\right), \\
\left|c^{(T_1)} u_i^{(T_1)}\right| &= |C_{2i}| \lambda_2^{T_1} \geq \tilde{\Omega}\left(d^{\alpha/2}\right) \tilde{\Omega}\left(\frac{1}{d^{\frac{5}{4}\alpha}}\right) = \tilde{\Omega}\left(\frac{1}{d^{\frac{3}{4}\alpha}}\right).
\end{aligned}
$$

Here in $(i)$ we used $\lambda_2 = 1 + \frac{\eta}{1-\beta}\|A\|_2$. Then we have at time $T_1$, $\forall i,j \in [d]$, $\frac{\left|R_{11}^{(T_1)}[i,j]\right|}{\left|u_i^{(T_1)} v_j^{(T_1)}\right|} \leq \tilde{\mathcal{O}}\left(\frac{1}{d^{\frac{1}{4}\alpha}}\right)$ and that $\frac{\left|R_{21,i}^{(T_1)}\right|}{\left|c^{(T_1)} u_i^{(T_1)}\right|} \leq \tilde{\mathcal{O}}\left(\frac{1}{d^{\frac{1}{4}\alpha}}\right)$. Together with eq. (9), we have the following weight structure:

$$
\begin{aligned}
W_1^{(T_1)} &= R_1^{(T_1)} + \boldsymbol{u}^{(T_1)} \boldsymbol{v}^{(T_1)T}, \\
W_2^{(T_1)} &= R_2^{(T_1)T} + c^{(T_1)} \boldsymbol{u}^{(T_1)T},
\end{aligned}
$$

where w.h.p.,

$$
\forall 1 \leq i,j \leq d : \quad \frac{\left|R_1^{(T_1)}[i,j]\right|}{\left|u_i^{(T_1)} v_j^{(T_1)}\right|} \leq \tilde{\mathcal{O}}\left(\frac{1}{d^{\frac{1}{4}\alpha-1}}\right), \quad \frac{\left|R_{2i}^{(T_1)}\right|}{\left|c^{(T_1)} u_i^{(T_1)}\right|} \leq \tilde{\mathcal{O}}\left(\frac{1}{d^{\frac{1}{4}\alpha-1}}\right).
$$

Finally, we consider the loss. Since $\forall j \in [d] : \left(W_2^{(T_1)} W_1^{(T_1)}\right)_j - A_j = -\Theta(1)$, we know that $\bar{L}\left(W^{(T_1)}\right) = \Theta(d)$.

### D.3 Proof of Lemma 4

Eq. (8) tells us that $\boldsymbol{u}^{(T_1)} = \sum_{\tau=0}^{T_1-1} \lambda_2^\tau C_2^T$. Lemma 8 tells us that $C_2$ can be written as $C_2 := \frac{1}{2}(C_3 + C_4)$ where $C_{3i}, i \in [d]$ are i.i.d Gaussian random variables and that w.h.p. $\forall i \in [d] : \frac{|C_{4i}|}{|C_{3i}|} \leq \tilde{\mathcal{O}}\left(\frac{1}{d^{\frac{1}{4}\alpha - \frac{1}{2}}}\right)$. Combining these two facts together finishes the proof.

### D.4 Proof of Lemma 5

Replacing $t$ by $t-1$ in eq. (6), we get

$$W_2^{(t)} = W_2^{(t-1)} + \frac{\eta}{1-\beta} A W_1^{(t-1)T} + \frac{\eta}{1-\beta} r_2^{(t-1)}. \tag{11}$$

Eq. (6)-(11) and substituting eq. (5) yield

$$W_2^{(t+1)} - W_2^{(t)} = W_2^{(t)} - W_2^{(t-1)} + \frac{\eta^2}{(1-\beta)^2}\|A\|_2^2 W_2^{(t-1)} + \frac{\eta^2}{(1-\beta)^2} A r_1^{(t-1)T}$$
$$+ \frac{\eta}{1-\beta}\left(r_2^{(t)} - r_2^{(t-1)}\right),$$
$$\Rightarrow \quad W_2^{(t+1)} = 2W_2^{(t)} - \left(1 - \frac{\eta^2}{(1-\beta)^2}\|A\|_2^2\right) W_2^{(t-1)} + r_3^{(t)},$$

where $r_3(t) := \frac{\eta^2}{(1-\beta)^2} A r_1^{(t-1)T} + \frac{\eta}{1-\beta}\left(r_2^{(t)} - r_2^{(t-1)}\right)$.

For the equation $x^2 - 2x + 1 - \frac{\eta^2}{(1-\beta)^2}\|A\|_2^2 = 0$, the roots are $\lambda_1 = 1 - \frac{\eta}{1-\beta}\|A\|_2$ and $\lambda_2 = 1 + \frac{\eta}{1-\beta}\|A\|_2$. We have that

$$W_2^{(t+1)} - \lambda_2 W_2^{(t)} = \lambda_1\left(W_2^{(t)} - \lambda_2 W_2^{(t-1)}\right) + r_3^{(t)}$$
$$\Rightarrow \quad W_2^{(t)} - \lambda_2 W_2^{(t-1)} = \lambda_1^{t-1}\left(W_2^{(1)} - \lambda_2 W_2^{(0)}\right) + \sum_{\tau=1}^{t-1} \lambda_1^{t-1-\tau} r_3^{(\tau)}$$
$$:= \lambda_1^{t-1}\left(W_2^{(1)} - \lambda_2 W_2^{(0)}\right) + r_4^{(t)}.$$

We further have

$$W_2^{(t)} = \lambda_2^t W_2^{(0)} + \sum_{\tau=0}^{t-1} \lambda_2^{t-1-\tau} \lambda_1^\tau\left(W_2^{(1)} - \lambda_2 W_2^{(0)}\right) + \sum_{\tau=1}^t \lambda_2^{t-\tau} r_4^{(\tau)}$$
$$= \lambda_2^t W_2^{(0)} + \frac{\lambda_2^t - \lambda_1^t}{\lambda_2 - \lambda_1}\left(W_2^{(1)} - \lambda_2 W_2^{(0)}\right) + \sum_{\tau=1}^t \lambda_2^{t-\tau} r_4^{(\tau)}$$
$$= C_1 \lambda_1^t + C_2 \lambda_2^t + \sum_{\tau=1}^t \lambda_2^{t-\tau} r_4^{(\tau)}$$
$$= C_1 \lambda_1^t + \left(C_2 + r_5^{(t)}\right) \lambda_2^t,$$

where $r_5^{(t)} = \sum_{\tau=1}^t \lambda_2^{-\tau} r_4^{(\tau)}$, $C_1 = -\frac{W_2^{(1)} - \lambda_2 W_2^{(0)}}{\lambda_2 - \lambda_1}$ and $C_2 = \frac{W_2^{(1)} - \lambda_1 W_2^{(0)}}{\lambda_2 - \lambda_1}$.

### D.5 Proof of Lemma 6

Write $r_1^{(t)} = -\beta^{t+1} W_2^{(t)T} A + q_{12}^{(t)} + q_{13}^{(t)} + q_{14}^{(t)}$ where $q_{12}^{(t)} = (1 - \beta)\sum_{\tau=0}^t \beta^{t-\tau}\left(W_2^{(\tau)T} - W_2^{(t)T}\right) A$, $q_{13}^{(t)} = -(1 - \beta)\sum_{\tau=0}^t \beta^{t-\tau} W_2^{(\tau)T} W_2^{(\tau)} W_1^{(\tau)}$ and $q_{14}^{(t)} = -(1 - \beta)\sum_{\tau=0}^t \beta^{t-\tau} D g_1^{(\tau)}$. And write $r_2^{(t)} = -\beta^{t+1} A W_1^{(t)T} + q_{22}^{(t)} + q_{23}^{(t)} + q_{24}^{(t)}$, where $q_{22}^{(t)} = (1 - \beta)\sum_{\tau=0}^t \beta^{t-\tau} A\left(W_1^{(\tau)T} - W_1^{(t)T}\right)$, $q_{23}^{(t)} = -(1 - \beta)\sum_{\tau=0}^t \beta^{t-\tau} W_2^{(\tau)} W_1^{(\tau)} W_1^{(\tau)T}$ and $q_{24}^{(t)} = -(1 - \beta)\sum_{\tau=0}^t \beta^{t-\tau} D g_2^{(\tau)}$.

Let's first try to bound $\left|q_{12}^{(t)}[i,j]\right|$ and $\left|q_{22,i}^{(t)}\right|$. For any $\tau \leq T_1$, we have that

$$\forall i \in [d]: \quad \left|\left(W_2^{(\tau)} W_1^{(\tau)}\right)_i\right| = \left|\sum_{j=1}^{d} w_{2j}^{(\tau)} W_1^{(\tau)}[j,i]\right| \leq \sum_{j=1}^{d} \left|w_{2j}^{(\tau)}\right| \left|W_1^{(\tau)}[j,i]\right| \leq \sum_{i=1}^{d} \frac{1}{d^\alpha} = \frac{1}{d^{\alpha-1}},$$

and thus $\forall i \in [d]: \left|E_i^{(\tau)}\right| = \mathcal{O}(1)$. Then we have for all $i,j \in [d]$,

$$\left|W_1^{(\tau+1)}[i,j] - W_1^{(\tau)}[i,j]\right| \leq \eta \sum_{k=0}^{\tau} \beta^{\tau-k} \left|w_{2i}^{(k)} E_j^{(k)}\right| \leq \eta \sum_{k=0}^{\tau} \beta^{\tau-k} \mathcal{O}\left(\frac{1}{d^{\alpha/2}}\right)$$

$$= \eta \mathcal{O}\left(\frac{1}{d^{\alpha/2}}\right),$$

$$\left|w_{2i}^{(\tau+1)} - w_{2i}^{(\tau)}\right| \leq \eta \sum_{k=0}^{\tau} \beta^{\tau-k} \sum_{j=1}^{d} \left|E_j^{(k)} W_1^{(k)}[i,j]\right| \leq \eta \sum_{k=0}^{\tau} \beta^{\tau-k} \mathcal{O}\left(\frac{1}{d^{\alpha/2-1}}\right)$$

$$= \eta \mathcal{O}\left(\frac{1}{d^{\alpha/2-1}}\right).$$

That gives us $\forall i,j \in [d]$,

$$\left|q_{12}^{(t)}[i,j]\right| \leq (1-\beta) \sum_{\tau=0}^{t} \beta^{t-\tau} \left|\left(w_{2i}^{(\tau)} - w_{2i}^{(t)}\right) A_j\right| \leq \eta(1-\beta) \sum_{\tau=0}^{t} \mathcal{O}\left(\frac{\beta^{t-\tau}(t-\tau)}{d^{\alpha/2-1}}\right)$$

$$= \mathcal{O}\left(\frac{\eta}{d^{\alpha/2-1}}\right),$$

$$\left|q_{22,i}^{(t)}\right| \leq (1-\beta) \sum_{\tau=0}^{t} \beta^{t-\tau} \sum_{j=1}^{d} \left|A_j \left(W_1^{(\tau)}[i,j] - W_1^{(t)}[i,j]\right)\right|$$

$$\leq \eta(1-\beta) \sum_{\tau=0}^{t} \mathcal{O}\left(\frac{\beta^{t-\tau}(t-\tau)}{d^{\alpha/2-1}}\right) = \mathcal{O}\left(\frac{\eta}{d^{\alpha/2-1}}\right).$$

Then we bound $\left|q_{13}^{(t)}[i,j]\right|$ and $\left|q_{23,i}^{(t)}\right|$. We have for $\forall i,j \in [d]$,

$$\left|q_{13}^{(t)}[i,j]\right| \leq (1-\beta) \sum_{\tau=0}^{t} \beta^{t-\tau} \left|w_{2i}^{(\tau)} \left(W_2^{(\tau)} W_1^{(\tau)}\right)_j\right| \leq (1-\beta) \sum_{\tau=0}^{t} \beta^{t-\tau} \frac{1}{d^{\frac{\alpha}{2}}} \cdot \frac{1}{d^{\alpha-1}}$$

$$= \mathcal{O}\left(\frac{1}{d^{\frac{3}{2}\alpha-1}}\right),$$

$$\left|q_{23,i}^{(t)}\right| \leq (1-\beta) \sum_{\tau=0}^{t} \beta^{t-\tau} \sum_{j=1}^{d} \left|\left(W_2^{(t)} W_1^{(t)}\right)_j W_1^{(t)}[i,j]\right|$$

$$\leq (1-\beta) \sum_{\tau=0}^{t} \beta^{t-\tau} \sum_{i=1}^{d} \frac{1}{d^{\alpha-1+\frac{\alpha}{2}}} = \mathcal{O}\left(\frac{1}{d^{\frac{3}{2}\alpha-2}}\right).$$

Finally we use Lemma 32 to bound $\left|q_{14}^{(t)}[i,j]\right|$ and $\left|q_{24,i}^{(t)}\right|$. For $t \leq T_1$, the $M_1^{(t)}, M_2^{(t)}$ in Lemma 32 are upper bounded by $\frac{1}{d^{\frac{\alpha}{2}}}$. In the theorem we consider the training period before $T_{\text{SGD},2}$ so the time $T$ in Lemma 32 is set as $T_{\text{SGD},2}$. In the following sections, we will prove that $T_{\text{SGD},2} \leq \mathcal{O}\left(\frac{d^\alpha \log(\sqrt{d/\epsilon})}{\eta}\right)$. Then by Lemma 32, we have with probability at least $1 - \frac{1}{d}$, for $\forall t \leq T_1$ and

$\forall i, j \in [d]$,

$$\left| Dg_1^{(t)}[i,j] \right| = \left| \tilde{g}_1^{(t)}[i,j] - g_1^{(t)}[i,j] \right| \leq \mathcal{O}\left( \frac{1}{d^{\frac{3\alpha}{2}-3}} \sigma \sqrt{\frac{d^{\alpha+1}}{\eta} \log \frac{d}{\epsilon}} \right) + \mathcal{O}\left( \frac{1}{d^{\frac{\alpha}{2}}} \sigma \sqrt{\frac{d^{\alpha+2}}{\eta} \log \frac{d}{\epsilon}} \right)$$

$$\leq \tilde{\mathcal{O}}\left( \frac{1}{d^{\frac{\alpha}{2}}} \sigma \sqrt{\frac{d^{\alpha+2}}{\eta}} \right),$$

$$\left| Dg_{2i}^{(t)} \right| = \left| \tilde{g}_{2i}^{(t)} - g_{2i}^{(t)} \right| \leq \mathcal{O}\left( \frac{1}{d^{\frac{3\alpha}{2}-4}} \sigma \sqrt{\frac{d^{\alpha+1}}{\eta} \log \frac{d}{\epsilon}} \right) + \mathcal{O}\left( \frac{1}{d^{\frac{\alpha}{2}-1}} \sigma \sqrt{\frac{d^{\alpha+2}}{\eta} \log \frac{d}{\epsilon}} \right)$$

$$\leq \tilde{\mathcal{O}}\left( \frac{1}{d^{\frac{\alpha}{2}-1}} \sigma \sqrt{\frac{d^{\alpha+2}}{\eta}} \right).$$

By picking $\sigma \leq \frac{\eta^{3/2}}{d^{\alpha/2+1}}$, we have w.h.p. for $\forall t \leq T_1$ and $\forall i, j \in [d]$, $\left| Dg_1^{(t)}[i,j] \right| \leq \eta \tilde{\mathcal{O}}\left( \frac{1}{d^{\frac{\alpha}{2}}} \right)$ and $\left| Dg_{2i}^{(t)} \right| \leq \eta \tilde{\mathcal{O}}\left( \frac{1}{d^{\frac{\alpha}{2}-1}} \right)$, which yields

$$\left| q_{14}^{(t)}[i,j] \right| \leq (1-\beta) \sum_{\tau=0}^{t} \beta^{t-\tau} \left| Dg_1^{(\tau)}[i,j] \right| \leq (1-\beta) \sum_{\tau=0}^{t} \beta^{t-\tau} \eta \tilde{\mathcal{O}}\left( \frac{1}{d^{\frac{\alpha}{2}}} \right) = \eta \tilde{\mathcal{O}}\left( \frac{1}{d^{\frac{\alpha}{2}}} \right),$$

$$\left| q_{24,i}^{(t)} \right| \leq (1-\beta) \sum_{\tau=0}^{t} \beta^{t-\tau} \left| Dg_{2i}^{(\tau)} \right| \leq (1-\beta) \sum_{\tau=0}^{t} \beta^{t-\tau} \eta \tilde{\mathcal{O}}\left( \frac{1}{d^{\frac{\alpha}{2}-1}} \right) = \eta \tilde{\mathcal{O}}\left( \frac{1}{d^{\frac{\alpha}{2}-1}} \right).$$

Combining all the above bounds and substituting $\eta \leq \mathcal{O}\left( \frac{1}{d^\alpha} \right)$ gives us for $\forall t \leq T_1$ and $\forall i, j \in [d]$,

$$\left| r_1^{(t)}[i,j] \right| \leq \beta^{t+1} \left| w_{2i}^{(t)} A_j \right| + \tilde{\mathcal{O}}\left( \frac{1}{d^{\frac{3}{2}\alpha-1}} \right), \quad \left| r_{2i}^{(t)} \right| \leq \beta^{t+1} \left| \sum_{j=1}^{d} A_j W_1^{(t)}[i,j] \right| + \tilde{\mathcal{O}}\left( \frac{1}{d^{\frac{3}{2}\alpha-2}} \right).$$

$$(12)$$

For $t \leq T_1$, we have $\forall i, j \in [d]$, $\left| w_{2i}^{(t)} A_j \right| \leq \mathcal{O}\left( \frac{1}{d^{\alpha/2}} \right)$ and $\left| \sum_{j=1}^{d} A_j W_1^{(t)}[i,j] \right| \leq \mathcal{O}\left( \frac{1}{d^{\alpha/2-1}} \right)$, which gives us $\left| r_1^{(t)}[i,j] \right| \leq \mathcal{O}\left( \frac{1}{d^{\alpha/2}} \right)$ and $\left| r_{2i}^{(t)} \right| \leq \mathcal{O}\left( \frac{1}{d^{\alpha/2-1}} \right)$. Substituting into eq. (5) and eq. (6) yields that for $t \leq T_1$ and $\forall i, j \in [d]$,

$$\left| W_1^{(t+1)}[i,j] - W_1^{(t)}[i,j] \right| \leq \mathcal{O}\left( \frac{\eta}{d^{\alpha/2}} \right), \quad \left| w_{2i}^{(t+1)} - w_{2i}^{(t)} \right| \leq \mathcal{O}\left( \frac{\eta}{d^{\alpha/2-1}} \right).$$

Hence for $t \leq \min\left\{ \frac{\alpha \log d}{\log(1/\beta)}, T_1 \right\}$, we have $\forall i, j \in [d]$,

$$\left| W_1^{(t)}[i,j] \right| \leq \left| W_1^{(0)}[i,j] \right| + \frac{\alpha \log d}{\log(1/\beta)} \mathcal{O}\left( \frac{\eta}{d^{\alpha/2}} \right) \leq \tilde{\mathcal{O}}\left( \frac{1}{d^{\frac{3\alpha}{2}}} \right),$$

$$\left| w_{2i}^{(t)} \right| \leq \left| w_{2i}^{(0)} \right| + \frac{\alpha \log d}{\log(1/\beta)} \mathcal{O}\left( \frac{\eta}{d^{\alpha/2-1}} \right) \leq \tilde{\mathcal{O}}\left( \frac{1}{d^{\frac{3\alpha}{2}-1}} \right).$$

Then we know that $T_1 > \frac{\alpha \log d}{\log(1/\beta)}$ and also get tighter bounds of $\left| W_1^{(t)}[i,j] \right|, \left| w_{2i}^{(t)} \right|$ for $t \leq \frac{\alpha \log d}{\log(1/\beta)}$. Now we use these new bounds to analyze $\left| r_1^{(t)}[i,j] \right|$ and $\left| r_{2i}^{(t)} \right|$ again.

When $t \leq \frac{\alpha \log d}{\log(1/\beta)}$, we have for all $i, j \in [d]$, $\beta^{t+1} \left| w_{2i}^{(t)} A_j \right| \leq \left| w_{2i}^{(t)} A_j \right| \leq \tilde{\mathcal{O}}\left( \frac{1}{d^{\frac{3\alpha}{2}-1}} \right)$ and $\beta^{t+1} \left| \sum_{j=1}^{d} A_j W_1^{(t)}[i,j] \right| \leq \left| \sum_{j=1}^{d} A_j W_1^{(t)}[i,j] \right| \leq \tilde{\mathcal{O}}\left( \frac{1}{d^{\frac{3\alpha}{2}-1}} \right)$. When $\frac{\alpha \log d}{\log(1/\beta)} < t \leq T_1$, we have $\beta^{t+1} \leq \frac{1}{d^\alpha}$, suggesting that $\forall i, j \in [d]$, $\beta^{t+1} \left| w_{2i}^{(t)} A_j \right| \leq \frac{1}{d^\alpha} \tilde{\mathcal{O}}\left( \frac{1}{d^{\frac{\alpha}{2}}} \right) \leq \tilde{\mathcal{O}}\left( \frac{1}{d^{\frac{3\alpha}{2}}} \right)$ and $\beta^{t+1} \left| \sum_{j=1}^{d} A_j W_1^{(t)}[i,j] \right| \leq \frac{1}{d^\alpha} \tilde{\mathcal{O}}\left( \frac{1}{d^{\frac{\alpha}{2}-1}} \right) \leq \tilde{\mathcal{O}}\left( \frac{1}{d^{\frac{3\alpha}{2}-1}} \right)$. Substituting into (12) completes the proof.

### D.6 Proof of Lemma 7

Based on the bound in Lemma 6, we have

$$\left| r_{3i}^{(t)} \right| = \left| \frac{\eta^2}{(1-\beta)^2} \sum_{j=1}^{d} A_j r_1^{(t-1)}[i,j] + \frac{\eta}{1-\beta} \left( r_{2i}^{(t)} - r_{2i}^{(t-1)} \right) \right|$$

$$\leq \frac{\eta^2}{(1-\beta)^2} \sum_{j=1}^{d} \left| A_j r_1^{(t-1)}[i,j] \right| + \frac{\eta}{1-\beta} \left| r_{2i}^{(t)} \right| + \frac{\eta}{1-\beta} \left| r_{2i}^{(t-1)} \right|$$

$$\leq \eta^2 \tilde{\mathcal{O}} \left( \frac{1}{d^{\frac{3}{2}\alpha-2}} \right) + 2\eta \tilde{\mathcal{O}} \left( \frac{1}{d^{\frac{3}{2}\alpha-2}} \right) = \eta \tilde{\mathcal{O}} \left( \frac{1}{d^{\frac{3}{2}\alpha-2}} \right).$$

Since $\lambda_1 = 1 - \frac{\eta}{1-\beta} \|A\|_2$, $\lambda_2 = 1 + \frac{\eta}{1-\beta} \|A\|_2$, and note that $\|A\|_2 = \Theta\left(\sqrt{d}\right)$, we have that

$$\left| r_{4i}^{(t)} \right| = \left| \sum_{\tau=1}^{t-1} \lambda_1^{t-1-\tau} r_{3i}^{(\tau)} \right| \leq \sum_{\tau=1}^{t-1} \lambda_1^{t-1-\tau} \tilde{\mathcal{O}} \left( \frac{\eta}{d^{\frac{3}{2}\alpha-2}} \right) \leq \frac{\eta}{1-\lambda_1} \tilde{\mathcal{O}} \left( \frac{1}{d^{\frac{3}{2}\alpha-2}} \right) = \tilde{\mathcal{O}} \left( \frac{1}{d^{\frac{3}{2}(\alpha-1)}} \right),$$

$$\left| r_{5i}^{(t)} \right| = \left| \sum_{\tau=1}^{t} \lambda_2^{-\tau} r_{4i}^{(\tau)} \right| \leq \eta \sum_{\tau=1}^{t} \lambda_2^{-\tau} \tilde{\mathcal{O}} \left( \frac{1}{d^{\frac{3}{2}(\alpha-1)}} \right) \leq \frac{\eta}{\lambda_2-1} \tilde{\mathcal{O}} \left( \frac{1}{d^{\frac{3}{2}(\alpha-1)}} \right) = \tilde{\mathcal{O}} \left( \frac{1}{d^{\frac{3}{2}\alpha-1}} \right).$$

### D.7 Proof of Lemma 8

For the equation $x^2 - 2x + 1 - \frac{\eta^2}{(1-\beta)^2} \|A\|_2^2 = 0$, the roots are $\lambda_1 = 1 - \frac{\eta}{1-\beta} \|A\|_2$ and $\lambda_2 = 1 + \frac{\eta}{1-\beta} \|A\|_2$, which gives us

$$\begin{aligned} C_2 &= \frac{W_2^{(1)} - \lambda_1 W_2^{(0)}}{\lambda_2 - \lambda_1} \\ &= \frac{W_2^{(0)} + \eta A W_1^{(0)T} + \eta \tilde{r}_2^{(0)} - W_2^{(0)} + \frac{\eta}{1-\beta} \|A\|_2 W_2^{(0)}}{\frac{2\eta}{1-\beta} \|A\|_2} \\ &= \frac{1}{2} W_2^{(0)} + \frac{1-\beta}{2\|A\|_2} A W_1^{(0)T} + \frac{1-\beta}{2\|A\|_2} \tilde{r}_2^{(0)}, \end{aligned} \tag{13}$$

where $\tilde{r}_2^{(0)} = -W_2^{(0)} W_1^{(0)} W_1^{(0)T} - Dg_2^{(0)}$. Note that this is slightly different from the definition of $r_2^{(0)}$ in eq. (6). Now let's bound the $i$-th coordinate of $\tilde{r}_2^{(0)}$.

In Section D.5 we have shown that w.h.p. for $\forall t \leq T_1$ and $\forall i,j \in [d]$, $\left| Dg_{2i}^{(t)} \right| \leq \eta \tilde{\mathcal{O}} \left( \frac{1}{d^{\frac{\alpha}{2}-1}} \right) = \tilde{\mathcal{O}} \left( \frac{1}{d^{\frac{3\alpha}{2}-1}} \right)$, which also applies to $t = 0$. Using the Gaussian tail bound and union bound, w.p. at least $1 - \delta$, for ever $1 \leq i,j \leq d$, we have that

$$\left| w_{2i}^{(0)} \right| \leq \sqrt{\frac{2}{d^{2\alpha}} \log \frac{2d}{\delta}}, \quad \left| W_1^{(0)}[i,j] \right| \leq \sqrt{\frac{2}{d^{4\alpha}} \log \frac{2d^2}{\delta}}.$$

Then we have that w.p. at least $1 - \delta$, $\forall 1 \leq i,j \leq d$ :,

$$\left| \left( W_2^{(0)} W_1^{(0)} \right)_i \right| = \left| \sum_{j=1}^{d} w_{2j}^{(0)} W_1^{(0)}[j,i] \right| \leq \sum_{j=1}^{d} \left| w_{2j}^{(0)} \right| \left| W_1^{(0)}[j,i] \right|$$

$$\leq \sum_{i=1}^{d} \sqrt{\frac{2}{d^{2\alpha}} \log \frac{2d}{\delta}} \sqrt{\frac{2}{d^{4\alpha}} \log \frac{2d^2}{\delta}} \leq \frac{2}{d^{3\alpha-1}} \log \frac{2d^2}{\delta},$$

$$\Rightarrow \quad \left| \tilde{r}_{2i}^{(0)} \right| \leq \sum_{j=1}^{d} \left| \left( W_2^{(0)} W_1^{(0)} \right)_j \right| \left| W_1^{(0)}[i,j] \right| + \left| Dg_{2i}^{(0)} \right|$$

$$\leq \sum_{i=1}^{d} \frac{2}{d^{3\alpha-1}} \log \frac{2d^2}{\delta} \sqrt{\frac{2}{d^{4\alpha}} \log \frac{2d^2}{\delta}} + \tilde{\mathcal{O}} \left( \frac{1}{d^{\frac{3\alpha}{2}-1}} \right) = \tilde{\mathcal{O}} \left( \frac{1}{d^{\frac{3\alpha}{2}-1}} \right). \tag{14}$$

Next, we bound the $i$-th coordinate of $W_2^{(0)} + \frac{1-\beta}{\|A\|_2} A W_1^{(0)T}$, i.e. $w_{2i}^{(0)} + \frac{1-\beta}{\|A\|_2} A \left( W_1^{(0)}[i,:] \right)^T$.

By independence under Assumption 2, we have that

$$\text{Var}\left( w_{2i}^{(0)} + \frac{1-\beta}{\|A\|_2} A \left( W_1^{(0)}[i,:] \right)^T \right) = \text{Var}\left( w_{2i}^{(0)} \right) + \frac{(1-\beta)^2}{\|A\|_2^2} \sum_{j=1}^d A_j^2 \text{Var}\left( W_1^{(0)}[i,j] \right)$$

$$= \frac{1}{d^{2\alpha}} + \frac{(1-\beta)^2}{\|A\|_2^2} \sum_{i=1}^d A_j^2 \frac{1}{d^{4\alpha}} = \mathcal{O}\left( \frac{1}{d^{2\alpha}} \right).$$

Using the Gaussian tail bound and union bound, w.p. at least $1-\delta$, for ever $1 \le i \le d$, we have that

$$\left| w_{2i}^{(0)} + \frac{1-\beta}{\|A\|_2} A \left( W_1^{(0)}[i,:] \right)^T \right| \le \mathcal{O}\left( \sqrt{\frac{1}{d^{2\alpha}} \log \frac{d}{\delta}} \right) = \tilde{\mathcal{O}}\left( \frac{1}{d^\alpha} \right).$$

Since for $X \sim \mathcal{N}(0,\sigma^2)$, we have that $P(|X| \le t) \le \frac{2t}{\sqrt{2\pi}\sigma}$, then for a fixed $i$,

$$P\left( \left| w_{2i}^{(0)} + \frac{1-\beta}{\|A\|_2} A \left( W_1^{(0)}[i,:] \right)^T \right| \le \frac{1}{d^{\frac{5}{4}\alpha}} \right) \le \mathcal{O}\left( \frac{2/d^{\frac{5}{4}\alpha}}{\sqrt{2\pi} \cdot \sqrt{1/d^{2\alpha}}} \right) = \Theta\left( \frac{1}{d^{\frac{\alpha}{4}}} \right).$$

Then by union bound, we have that w.p. at least $1 - \frac{1}{d^{\frac{\alpha}{4}-1}}$, for every $1 \le i \le d$,

$$\left| w_{2i}^{(0)} + \frac{1-\beta}{\|A\|_2} A \left( W_1^{(0)}[i,:] \right)^T \right| \ge \Theta\left( \frac{1}{d^{\frac{5}{4}\alpha}} \right).$$

Now define $C_3 := W_2^{(0)} + \frac{1-\beta}{\|A\|_2} A W_1^{(0)T}$ and $C_4 := \frac{1-\beta}{2\|A\|_2} \tilde{r}_{2i}^{(0)}$. We get that $C_{3i}, i \in [d]$ are i.i.d Gaussian random variables and that $C_2 = \frac{1}{2}(C_3 + C_4)$, where w.h.p. for all $i \in [d]$,

$$|C_{3i}| \le \tilde{\mathcal{O}}\left( \frac{1}{d^\alpha} \right), \quad |C_{3i}| \ge \Theta\left( \frac{1}{d^{\frac{5}{4}\alpha}} \right), \quad |C_{4i}| \stackrel{(i)}{\le} \tilde{\mathcal{O}}\left( \frac{1}{d^{\frac{3\alpha}{2}-\frac{1}{2}}} \right), \tag{15}$$

where $(i)$ follows from eq. (14) and the fact that $\|A\|_2 = \sqrt{d}$. Then we get that w.h.p.

$$\forall i \in [d]: \frac{|C_{4i}|}{|C_{3i}|} \le \frac{\tilde{\mathcal{O}}\left( \frac{1}{d^{\frac{3\alpha}{2}-\frac{1}{2}}} \right)}{\Omega\left( \frac{1}{d^{\frac{5}{4}\alpha}} \right)} = \tilde{\mathcal{O}}\left( \frac{1}{d^{\frac{1}{4}\alpha-\frac{1}{2}}} \right).$$

Substituting eq. (15) into eq. (13), we get that w.h.p.,

$$|C_{2i}| = \Theta\left( \left| w_{2i}^{(0)} + \frac{1-\beta}{\|A\|_2} A \left( W_1^{(0)}[i,:] \right)^T \right| \right) \in \left[ \tilde{\Omega}\left( \frac{1}{d^{\frac{5}{4}\alpha}} \right), \tilde{\mathcal{O}}\left( \frac{1}{d^\alpha} \right) \right].$$

Similarly, note that

$$C_1 = -\frac{W_2^{(1)} - \lambda_2 W_2^{(0)}}{\lambda_2 - \lambda_1}$$

$$= -\frac{W_2^{(0)} + \eta A W_1^{(0)T} + \eta \tilde{r}_2^{(0)} - W_2^{(0)} - \frac{\eta}{1-\beta} \|A\|_2 W_2^{(0)}}{\frac{2\eta}{1-\beta} \|A\|_2}$$

$$= \frac{1}{2} W_2^{(0)} - \frac{1-\beta}{2\|A\|_2} A W_1^{(0)T} - \frac{1-\beta}{2\|A\|_2} \tilde{r}_2^{(0)},$$

we can use the same techniques to get that i) w.p. at least $1-\delta$, $\forall i \in [d]: |C_{1i}| \le \tilde{\mathcal{O}}\left( \frac{1}{d^\alpha} \right)$, ii) w.p. at least $1 - \delta - \frac{1}{d^{\frac{\alpha}{4}-1}}$, $\forall i \in [d], |C_{1i}| \ge \tilde{\Omega}\left( \frac{1}{d^{\frac{5}{4}\alpha}} \right)$.

## D.8 Proof of Lemma 3

The proof in Section D.2 tells us that at the end of the first phase (when $t = T_1$),

$$W_1^{(T_1)} = \boldsymbol{u}^{(T_1)}\boldsymbol{v}^{(T_1)T} + R_1^{(T_1)},$$
$$W_2^{(T_1)} = c^{(T_1)}\boldsymbol{u}^{(T_1)T} + R_2^{(T_1)T}, \tag{16}$$
$$\text{where } \boldsymbol{v}^{(T_1)T} = \frac{\eta A}{1-\beta}, \quad c^{(T_1)} = \frac{\lambda_2^{T_1}}{\sum_{\tau=0}^{T_1-1} \lambda_2^\tau}.$$

Denote the $i$-th coordinate of $\boldsymbol{u}^{(t)}, \boldsymbol{v}^{(t)}, R_2^{(t)}$ as $u_i^{(t)}, v_i^{(t)}, R_{2i}^{(t)}$, respectively. Denote the $(i, j)$-th element of $R_1^{(t)}$ as $R_1^{(t)}[i, j]$. For $t \geq T_1$, we prove by induction that,

$$W_1^{(t)} = \boldsymbol{u}^{(T_1)}\boldsymbol{v}^{(t)T} + R_1^{(t)},$$
$$W_2^{(t)} = c^{(t)}\boldsymbol{u}^{(T_1)T} + R_2^{(t)T}, \tag{17}$$

where

$$\boldsymbol{v}^{(t+1)T} = \boldsymbol{v}^{(t)T} - \eta_t c^{(t)} E^{(t)},$$
$$R_1^{(t+1)} = R_1^{(t)} - \eta_t R_2^{(t)} E^{(t)} + r_1^{(t)},$$
$$c^{(t+1)} = c^{(t)} - \eta_t E^{(t)} \boldsymbol{v}^{(t)},$$
$$R_2^{(t+1)T} = R_2^{(t)T} - \eta_t E^{(t)} R_1^{(t)T} + r_2^{(t)},$$

with $r_1^{(t)} := \eta \sum_{\tau=0}^t \beta^{t-\tau} \left(W_2^{(t)T} E^{(t)} - W_2^{(\tau)T} E^{(\tau)}\right) - \eta \sum_{\tau=0}^t \beta^{t-\tau} Dg_1^{(\tau)}$, $E^{(t)} := W_2^{(t)} W_1^{(t)} - A$, $\eta_t = \eta \sum_{\tau=0}^t \beta^{t-\tau}$ and $r_2^{(t)} = \eta \sum_{\tau=0}^t \beta^{t-\tau} \left(E^{(t)} W_1^{(t)T} - E^{(\tau)} W_1^{(\tau)T}\right) - \eta \sum_{\tau=0}^t \beta^{t-\tau} Dg_2^{(\tau)}$.

Note that the $r_1^{(t)}$ and $r_2^{(t)}$ here are different from those defined in Section D.2, but we abuse the notation and still use $r_1^{(t)}$ and $r_2^{(t)}$ to represent the error terms.

The base case is already given by eq. (16).

Suppose our lemma holds for $t$, then for $t + 1$, using the same techniques as in eq. (5) and eq. (6), we have that

$$W_1^{(t+1)} = W_1^{(t)} - \eta \sum_{\tau=0}^t \beta^{t-\tau} W_2^{(\tau)T} E^{(\tau)} - \eta \sum_{\tau=0}^t \beta^{t-\tau} Dg_1^{(\tau)}$$
$$= W_1^{(t)} - \eta_t W_2^{(t)T} E^{(t)} + r_1^{(t)},$$
$$W_2^{(t+1)} = W_2^{(t)} - \eta \sum_{\tau=0}^t \beta^{t-\tau} E^{(\tau)} W_1^{(\tau)T} - \eta \sum_{\tau=0}^t \beta^{t-\tau} Dg_2^{(\tau)}$$
$$= W_2^{(t)} - \eta_t E^{(t)} W_1^{(t)T} + r_2^{(t)},$$

Plugging in the inductive hypothesis yields

$$W_1^{(t+1)} = W_1^{(t)} - \eta_t W_2^{(t)T} E^{(t)} + r_1^{(t)}$$
$$= \boldsymbol{u}^{(T_1)}\boldsymbol{v}^{(t)T} + R_1^{(t)} - \eta_t \left(c^{(t)}\boldsymbol{u}^{(T_1)} + R_2^{(t)}\right) E^{(t)} + r_1^{(t)}$$
$$= \boldsymbol{u}^{(T_1)} \left(\boldsymbol{v}^{(t)T} - \eta_t c^{(t)} E^{(t)}\right) + R_1^{(t)} - \eta_t R_2^{(t)} E^{(t)} + r_1^{(t)},$$

$$W_2^{(t+1)} = W_2^{(t)} - \eta_t E^{(t)} W_1^{(t)T} + r_2^{(t)}$$
$$= c^{(t)}\boldsymbol{u}^{(T_1)T} + R_2^{(t)T} - \eta_t E^{(t)} \left(\boldsymbol{v}^{(t)}\boldsymbol{u}^{(T_1)T} + R_1^{(t)T}\right) + r_2^{(t)}$$
$$= \left(c^{(t)} - \eta_t E^{(t)}\boldsymbol{v}^{(t)}\right) \boldsymbol{u}^{(T_1)T} + R_2^{(t)T} - \eta_t E^{(t)} R_1^{(t)T} + r_2^{(t)}.$$

It implies that our lemma holds for $t + 1$, which completes the proof.

Now we analyze the error terms $\left|R_1^{(t)}[i,j]\right|$ and $\left|R_{2i}^{(t)}\right|$. Eq. (16) tells us that $c^{(T_1)}$ and $\forall i \in [d], v_i^{(T_1)}$ are all positive. We first prove by induction that for all $T_1 \leq t \leq T_2$, $c^{(t)} > 0, \forall i \in [d], v_i^{(t)} > 0$.

The above discussion already proves the base case. Suppose at time $t$, we have $c^{(t)} > 0, \forall i \in [d], v_i^{(t)} > 0$. Note that when $T_1 \leq t < T_2, \forall i \in [d] : E_i^{(t)} \leq 0$, then for $t+1$,

$$v_i^{(t+1)} = v_i^{(t)} - \eta_t c^{(t)} E_i^{(t)} > 0,$$

$$c^{(t+1)} = c^{(t)} - \eta_t \sum_{i=1}^{d} E_i^{(t)} v_i^{(t)} > 0.$$

Therefore by induction, we have proved that for all $T_1 \leq t \leq T_2$, $c^{(t)} > 0, \forall i \in [d], v_i^{(t)} > 0$.

Now we prove that for all $T_1 \leq t \leq T_2$,

$$\forall 1 \leq i,j \leq d : \quad 0 \leq \frac{\left|R_1^{(t)}[i,j]\right|}{\left|u_i^{(T_1)}\right|\left|v_j^{(t)}\right|} \leq \delta_i + \sum_{\tau=T_1}^{t-1} \epsilon_i^{(\tau)}, \quad 0 \leq \frac{\left|R_{2i}^{(t)}\right|}{c^{(t)}\left|u_i^{(T_1)}\right|} \leq \delta_i + \sum_{\tau=T_1}^{t-1} \epsilon_i^{(\tau)}, \quad (18)$$

where

$$\delta_i := \max\left\{\max_j \frac{\left|R_1^{(T_1)}[i,j]\right|}{\left|u_i^{(T_1)}\right|\left|v_j^{(T_1)}\right|}, \frac{\left|R_{2i}^{(T_1)}\right|}{c^{(T_1)}\left|u_i^{(T_1)}\right|}\right\}, \quad \epsilon_i^{(t)} := \max\left\{\max_j \frac{\left|r_1^{(t)}[i,j]\right|}{\left|u_i^{(T_1)}\right|\left|v_j^{(t)}\right|}, \frac{\left|r_{2i}^{(t)}\right|}{c^{(t)}\left|u_i^{(T_1)}\right|}\right\}.$$

The left hand sides of the inequalities are trivial since we have proved that $c^{(t)} > 0, \forall i \in [d], v_i^{(t)} > 0$ for all $T_1 \leq t \leq T_2$. Now we prove the right hand sides by induction.

The base case is already verified by the definition of $\delta_i$. Suppose eq.(18) holds for $T_1 \leq t < T_2$. Then for $t+1$, using $\forall i \in [d] : E_i^{(t)} \leq 0$ and $v^{(t+1)} \geq v^{(t)}, c^{(t+1)} \geq c^{(t)}$, we can get that $\forall 1 \leq i,j \leq d$

$$\frac{\left|R_1^{(t+1)}[i,j]\right|}{\left|u_i^{(T_1)}\right|\left|v_j^{(t+1)}\right|} \leq \frac{\left|R_1^{(t)}[i,j]\right|\frac{1}{\left|u_i^{(T_1)}\right|} + \eta_t\left|R_{2i}^{(t)}\right|\frac{1}{\left|u_i^{(T_1)}\right|}\left(-E_j^{(t)}\right)}{v_j^{(t)} + \eta_t c^{(t)}\left(-E_j^{(t)}\right)} + \frac{\left|r_1^{(t)}[i,j]\right|}{\left|u_i^{(T_1)}\right|\left|v_j^{(t)}\right|}$$

$$\leq \frac{\left(\delta_i + \sum_{\tau=T_1}^{t-1} \epsilon_i^{(\tau)}\right)v_j^{(t)} + \eta_t\left(\delta_i + \sum_{\tau=T_1}^{t-1} \epsilon_i^{(\tau)}\right)c^{(t)}\left(-E_j^{(t)}\right)}{v_j^{(t)} + \eta_t c^{(t)}\left(-E_j^{(t)}\right)} + \epsilon_i^{(t)}$$

$$= \delta_i + \sum_{\tau=T_1}^{t} \epsilon_i^{(\tau)}.$$

Similarly, we have that $\forall 1 \leq i \leq d$

$$\frac{\left|R_{2i}^{(t+1)}\right|}{c^{(t+1)}\left|u_i^{(T_1)}\right|} \leq \frac{\left|R_{2i}^{(t)}\right|\frac{1}{\left|u_i^{(T_1)}\right|} + \eta_t\sum_{j=1}^{d}\left(-E_j^{(t)}\right)\left|R_1^{(t)}[i,j]\right|\frac{1}{\left|u_i^{(T_1)}\right|}}{c^{(t)} + \eta_t\sum_{j=1}^{d}\left(-E_j^{(t)}\right)v_j^{(t)}} + \frac{\left|r_{2i}^{(t)}\right|}{\left|u_i^{(T_1)}\right|c^{(t)}}$$

$$\leq \frac{\left(\delta_i + \sum_{\tau=T_1}^{t-1} \epsilon_i^{(\tau)}\right)c^{(t)} + \eta_t\left(\delta_i + \sum_{\tau=T_1}^{t-1} \epsilon_i^{(\tau)}\right)\sum_{j=1}^{d}\left(-E_j^{(t)}\right)v_j^{(t)}}{c^{(t)} + \eta_t\sum_{j=1}^{d}\left(-E_j^{(t)}\right)v_j^{(t)}} + \epsilon_i^{(t)}$$

$$= \delta_i + \sum_{\tau=T_1}^{t} \epsilon_i^{(\tau)}.$$

Therefore by induction, eq. (18) holds for all $t$ in the second phase.

So far we have proved the rank 1 structure stated in Lemma 3. The remaining part of the proof is given by the following lemma, whose proof is deferred to Section D.9.

**Lemma 9.** *Under Assumption 1, 2 and 3, suppose $\sigma \leq \frac{\eta^{3/2}}{d^{\alpha/2+1}}$. By picking $\eta \leq \mathcal{O}\left(\frac{\epsilon}{d^{\frac{7\alpha}{4}+4}}\right)$, we have that w.h.p. for $T_1 \leq t \leq \min\{T_2, T_3\}$,*

$$\forall 1 \leq i, j \leq d: \quad 0 \leq \frac{\left|R_1^{(t)}[i,j]\right|}{\left|u_i^{(T_1)}\right|\left|v_j^{(t)}\right|} \leq \tilde{\mathcal{O}}(\epsilon_0), \quad 0 \leq \frac{\left|R_{2i}^{(t)}\right|}{c^{(t)}\left|u_i^{(T_1)}\right|} \leq \tilde{\mathcal{O}}(\epsilon_0), \tag{19}$$

*and that when $t = \min\{T_2, T_3\}$, we have $\left\|E^{(t)}\right\|_2^2 = \mathcal{O}(\epsilon_0 d)$.*

### D.9 Proof of Lemma 9

We first have the following lemma which describes the structure of $\boldsymbol{v}^{(t)}$ for $t \geq T_1$.

**Lemma 10.** *Under Assumption 1, 2 and 3, for $t \geq T_1$, we can write $\boldsymbol{v}^{(t)T}$ as $\boldsymbol{v}^{(t)T} = a^{(t)}A + R_v^{(t)T}$, with $a^{(T_1)} = \frac{\eta}{1-\beta}$, $R_v^{(T_1)T} = [0, 0, ..., 0]$, and*

$$a^{(t+1)} = \left(1 - \eta_t c^{(t)} d^{(t)}\right) a^{(t)} + \eta_t c^{(t)},$$

$$R_v^{(t+1)} = \left(1 - \eta_t c^{(t)} d^{(t)}\right) R_v^{(t)} - \eta_t c^{(t)} R_3^{(t)},$$

*where $d^{(t)} := c^{(t)}\left\|\boldsymbol{u}^{(T_1)}\right\|^2 + R_2^{(t)T}\boldsymbol{u}^{(T_1)}$, $R_3^{(t)T} := c^{(t)}\boldsymbol{u}^{(T_1)T}R_1^{(t)} + R_2^{(t)T}R_1^{(t)}$.*

*Moreover, we have that*

$$W_2^{(t)}W_1^{(t)} = d^{(t)}\boldsymbol{v}^{(t)T} + R_3^{(t)T} = d^{(t)}a^{(t)}A + d^{(t)}R_v^{(t)T} + R_3^{(t)T}. \tag{20}$$

We prove Lemma 9 by induction. Denote the $i$-th coordinate of $R_3^{(t)}$ and $R_v^{(t)}$ as $R_{3i}^{(t)}$ and $R_{vi}^{(t)}$, respectively. The following lemmas constitute the inductive part.

**Lemma 11.** *Under Assumption 1, 2 and 3, suppose $\sigma \leq \frac{\eta^{3/2}}{d^{\alpha/2+1}}$ and pick $\eta \leq \mathcal{O}\left(\frac{1}{d^\alpha}\right)$. Consider any $t$ such that $T_1 \leq t < \min\{T_2, T_3\}$. Suppose for all $T_1 \leq \tau \leq t$, we have $\forall i, j \in [d]: \left|w_{2i}^{(\tau)}\right| \leq \mathcal{O}\left(d^{1/4}\right), \left|W_1^{(\tau)}[i,j]\right| \leq \mathcal{O}\left(\frac{1}{d^{1/4}}\right)$, then we have that $\forall i, j \in [d]: \left|r_1^{(t)}[i,j]\right| = \tilde{\mathcal{O}}\left(\eta^2 d^{11/4}\right), \left|r_{2i}^{(t)}\right| = \tilde{\mathcal{O}}\left(\eta^2 d^{13/4}\right)$. Moreover, we can get that $\forall i \in [d]: \epsilon_i^{(t)} = \tilde{\mathcal{O}}\left(\eta^2 d^{\frac{3}{4}\alpha+\frac{13}{4}}\right)$, where $\epsilon_i^{(t)}$ is defined in eq. (18).*

**Lemma 12.** *Under the conditions of Lemma 11 and pick $\eta \leq \mathcal{O}\left(\frac{\epsilon}{d^{\frac{7\alpha}{4}+4}}\right)$, we have that at time $t+1$,*

$$\forall 1 \leq i, j \leq d: \quad 0 \leq \frac{\left|R_1^{(t+1)}[i,j]\right|}{\left|u_i^{(T_1)}\right|\left|v_j^{(t+1)}\right|} \leq \tilde{\mathcal{O}}(\epsilon_0), \quad 0 \leq \frac{\left|R_{2i}^{(t+1)}\right|}{c^{(t+1)}\left|u_i^{(T_1)}\right|} \leq \tilde{\mathcal{O}}(\epsilon_0).$$

*where $\epsilon_0$ is defined in Definition 2.*

**Lemma 13.** *Under the conditions of Lemma 11 and pick $\eta \leq \mathcal{O}\left(\frac{\epsilon}{d^{\frac{7\alpha}{4}+4}}\right)$, we have that at time $t+1$,*

$$0 \leq \frac{\left|R_2^{(t+1)T}\boldsymbol{u}^{(T_1)}\right|}{c^{(t+1)}\left\|\boldsymbol{u}^{(T_1)}\right\|^2} \leq \tilde{\mathcal{O}}(\epsilon_0), \quad \forall j \in [d]: \quad 0 \leq \frac{\left|R_{3j}^{(t+1)}\right|}{c^{(t+1)}\left\|\boldsymbol{u}^{(T_1)}\right\|^2 v_j^{(t)}} \leq \tilde{\mathcal{O}}(\epsilon_0).$$

*Moreover,*

$$\forall j \in [d]: \frac{\left|R_{3j}^{(t+1)}\right|}{A_j} \leq \tilde{\mathcal{O}}(\epsilon_0). \tag{21}$$

**Lemma 14.** *Under the conditions of Lemma 11 and pick $\eta \leq \mathcal{O}\left(\frac{\epsilon}{d^{\frac{7\alpha}{4}+4}}\right)$, if we further suppose that $\forall j \in [d]: \frac{v_j^{(t)}}{c^{(t)}} = \Theta\left(\frac{1}{\sqrt{d}}\right), \frac{\left|R_{3j}^{(t)}\right|}{A_j}$ and $\frac{\left|R_{vj}^{(t)}\right|}{a^{(t)}A_j}$ are of order $\tilde{\mathcal{O}}(\epsilon_0)$, then we have that at time $t+1$,*

*(A)* $\forall i, j \in [d] : \frac{E_j^{(t)}}{E_i^{(t)}} = \Theta(1)$,

*(B)* $\forall j \in [d] : \frac{v_j^{(t+1)}}{c^{(t+1)}} = \Theta\left(\frac{1}{\sqrt{d}}\right)$,

*(C)* $\forall i, j \in [d] : \left|w_{2i}^{(t+1)}\right| \le \mathcal{O}\left(d^{1/4}\right), \left|W_1^{(t+1)}[i,j]\right| \le \mathcal{O}\left(\frac{1}{d^{1/4}}\right)$,

*(D)* $\forall j \in [d]$, $\frac{\left|R_{3j}^{(t+1)}\right|}{A_j}$ and $\frac{\left|R_{vj}^{(t+1)}\right|}{a^{(t+1)}A_j}$ are of order $\tilde{\mathcal{O}}(\epsilon_0)$.

By combining Lemma 11, 12 and 14, we can prove by induction that for all $T_1 \le t \le \min\{T_2, T_3\}$, eq. (19) holds (which follows from Lemma 12), and

$$\forall i, j \in [d] : \frac{E_j^{(t)}}{E_i^{(t)}} = \Theta(1), \tag{22}$$

which follows from the part (A) of Lemma 14. Now the only thing to verify is the base case, i.e. when $t = T_1$. More specifically, we want to prove that 1) $\forall i, j \in [d] : \left|w_{2i}^{(T_1)}\right| \le \mathcal{O}\left(d^{1/4}\right), \left|W_1^{(T_1)}[i,j]\right| \le \mathcal{O}\left(\frac{1}{d^{1/4}}\right)$ and that 2) $\forall j \in [d] : \frac{v_j^{(T_1)}}{c^{(T_1)}} = \Theta\left(\frac{1}{\sqrt{d}}\right)$, and that 3) $\frac{\left|R_{3j}^{(T_1)}\right|}{A_j}$ and $\frac{\left|R_{vj}^{(T_1)}\right|}{a^{(T_1)}A_j}$ are of order $\tilde{\mathcal{O}}(\epsilon_0)$. All of them can be verified by the proof in Section D.2 and the definition of $R_v^{(t)}, R_3^{(t)}$.

So far we have proved eq. (19) in Lemma 9. Now let's prove when $t = \min\{T_2, T_3\}$, we have that $\left\|E^{(t)}\right\|_2^2 = \mathcal{O}(\epsilon_0 d)$.

If $\min\{T_2, T_3\} = T_3$, by Definition 3, we have $\left\|E^{(t)}\right\|_2^2 \le \epsilon$. If $\min\{T_2, T_3\} = T_2$, by Definition 2, there exists $j \in [d]$ such that $E_j^{(t)} = -\Theta\left(\sqrt{\epsilon_0}\right)$. Combining with eq. (22) gives us $\forall i \in [d] :$ $E_i^{(t)} = -\Theta\left(\sqrt{\epsilon_0}\right)$. Combining these two cases, we get that when $t = \min\{T_2, T_3\}$, $\left\|E^{(t)}\right\|_2^2 \le \max\{\epsilon, \Theta(\epsilon_0 d)\} = \mathcal{O}(\epsilon_0 d)$.

### D.10 Proof of Lemma 10

We prove this lemma by induction. The base case ($t = T_1$) of $v^{(t)}$ is verified by eq. (16).

Suppose at time $t$, $v^{(t)T} = a^{(t)}A + R_v^{(t)T}$, then by eq. 17, we have that

$$\begin{aligned}
W_2^{(t)}W_1^{(t)} &= \left(c^{(t)}u^{(T_1)T} + R_2^{(t)T}\right)\left(u^{(T_1)}v^{(t)T} + R_1^{(t)}\right) \\
&= \left(c^{(t)}\left\|u^{(T_1)}\right\|^2 + R_2^{(t)T}u^{(T_1)}\right)v^{(t)} + c^{(t)}u^{(T_1)T}R_1^{(t)} + R_2^{(t)T}R_1^{(t)} \\
&= d^{(t)}v^{(t)T} + R_3^{(t)T} \\
&= d^{(t)}a^{(t)}A + d^{(t)}R_v^{(t)T} + R_3^{(t)T},
\end{aligned}$$

where $d^{(t)} := c^{(t)}\left\|u^{(T_1)}\right\|^2 + R_2^{(t)T}u^{(T_1)}$, $R_3^{(t)T} := c^{(t)}u^{(T_1)T}R_1^{(t)} + R_2^{(t)T}R_1^{(t)}$. That gives us

$$\begin{aligned}
v^{(t+1)T} &= v^{(t)T} - \eta_t c^{(t)}E^{(t)} \\
&= a^{(t)}A + R_v^{(t)T} - \eta_t c^{(t)}\left(d^{(t)}a^{(t)}A + d^{(t)}R_v^{(t)T} + R_3^{(t)T} - A\right) \\
&= \left(\left(1 - \eta_t c^{(t)}d^{(t)}\right)a^{(t)} + \eta_t c^{(t)}\right)A + \left(1 - \eta_t c^{(t)}d^{(t)}\right)R_v^{(t)T} - \eta_t c^{(t)}R_3^{(t)T} \\
&:= a^{(t+1)}A + R_v^{(t+1)T}.
\end{aligned}$$

Therefore we have proved by induction that for $t$ in the second phase, $v^{(t)} = a^{(t)}A + R_v^{(t)T}$. The above steps also proved eq. (20).

## D.11 Proof of Lemma 11

Write $r_1^{(t)} = q_{11}^{(t)} + q_{12}^{(t)}$ where we have $q_{11}^{(t)} = \eta \sum_{\tau=0}^{t} \beta^{t-\tau} \left( W_2^{(t)T} E^{(t)} - W_2^{(\tau)T} E^{(\tau)} \right)$, $q_{12}^{(t)} = -\eta \sum_{\tau=0}^{t} \beta^{t-\tau} Dg_1^{(\tau)}$. Write $r_2^{(t)} = q_{21}^{(t)} + q_{22}^{(t)}$ where $q_{22}^{(t)} = -\eta \sum_{\tau=0}^{t} \beta^{t-\tau} Dg_2^{(\tau)}$, $q_{21}^{(t)} = \eta \sum_{\tau=0}^{t} \beta^{t-\tau} \left( E^{(t)} W_1^{(t)T} - E^{(\tau)} W_1^{(\tau)T} \right)$.

Let's first bound $\left| q_{11}^{(t)}[i,j] \right|$ and $\left| q_{21,i}^{(t)} \right|$. By definition of $T_2$, we know that for $T_1 \leq \tau \leq t$, $\forall i \in [d]$ : $\left| E_i^{(\tau)} \right| = \mathcal{O}(1)$. Then we have for all $i, j \in [d]$,

$$\left| W_1^{(\tau+1)}[i,j] - W_1^{(\tau)}[i,j] \right| \leq \eta \sum_{k=0}^{\tau} \beta^{\tau-k} \left| w_{2i}^{(k)} E_j^{(k)} \right| \leq \eta \sum_{k=0}^{\tau} \beta^{\tau-k} \mathcal{O}\left( d^{1/4} \right) = \eta \mathcal{O}\left( d^{1/4} \right),$$

$$\left| w_{2i}^{(\tau+1)} - w_{2i}^{(\tau)} \right| \leq \eta \sum_{k=0}^{\tau} \beta^{\tau-k} \sum_{j=1}^{d} \left| E_j^{(k)} W_1^{(k)}[j,i] \right| \leq \eta \sum_{k=0}^{\tau} \beta^{\tau-k} \mathcal{O}\left( d^{3/4} \right) = \eta \mathcal{O}\left( d^{3/4} \right).$$
(23)

Note that

$$\left| E_j^{(\tau+1)} - E_j^{(\tau)} \right| = \sum_{i=1}^{d} \left( \left( w_{2i}^{(\tau+1)} - w_{2i}^{(\tau)} \right) W_1^{(\tau)}[i,j] + w_{2i}^{(\tau)} \left( W_1^{(\tau+1)}[i,j] - W_1^{(\tau)}[i,j] \right) \right)$$

$$+ \sum_{i=1}^{d} \left( \left( w_{2i}^{(\tau+1)} - w_{2i}^{(\tau)} \right) \left( W_1^{(\tau+1)}[i,j] - W_1^{(\tau)}[i,j] \right) \right).$$

We can further get that for $\forall j \in [d]$,

$$\left| E_j^{(\tau+1)} - E_j^{(\tau)} \right| \leq \eta d \mathcal{O}\left( d^{3/4} \right) \mathcal{O}\left( d^{-1/4} \right) + \eta d \mathcal{O}\left( d^{1/4} \right) \mathcal{O}\left( d^{1/4} \right) + \eta^2 d \mathcal{O}\left( d^{3/4} \right) \mathcal{O}\left( d^{1/4} \right)$$

$$= \mathcal{O}\left( \eta d^{3/2} + \eta^2 d^2 \right) = \mathcal{O}\left( \eta d^{3/2} \right).$$

Combining the above inequalities gives us $\forall i, j \in [d]$,

$$\left| q_{11}^{(t)}[i,j] \right| = \eta \left| \sum_{\tau=0}^{t} \beta^{t-\tau} \left( w_{2i}^{(t)} E_j^{(t)} - w_{2i}^{(\tau)} E_j^{(\tau)} \right) \right|$$

$$\leq \eta \sum_{\tau=0}^{t} \beta^{t-\tau} \left( \left| w_{2i}^{(t)} - w_{2i}^{(\tau)} \right| \left| E_j^{(t)} \right| + \left| w_{2i}^{(\tau)} \right| \left| E_j^{(t)} - E_j^{(\tau)} \right| \right)$$

$$\leq \eta^2 \sum_{\tau=0}^{t} \beta^{t-\tau}(t-\tau) \left( \mathcal{O}\left( d^{3/4} \right) \mathcal{O}(1) + \mathcal{O}\left( d^{1/4} \right) \mathcal{O}\left( d^{3/2} \right) \right) = \mathcal{O}\left( \eta^2 d^{7/4} \right),$$

$$\left| q_{12,i}^{(t)} \right| = \eta \left| \sum_{\tau=0}^{t} \beta^{t-\tau} \sum_{j=1}^{d} \left( E_j^{(t)} W_1^{(t)}[i,j] - E_j^{(\tau)} W_1^{(\tau)}[i,j] \right) \right|$$

$$\leq \eta \sum_{\tau=0}^{t} \beta^{t-\tau} \sum_{j=1}^{d} \left( \left| E_j^{(t)} \right| \left| W_1^{(t)}[i,j] - W_1^{(\tau)}[i,j] \right| + \left| E_j^{(t)} - E_j^{(\tau)} \right| \left| W_1^{(\tau)}[i,j] \right| \right)$$

$$\leq \eta^2 d \sum_{\tau=0}^{t} \beta^{t-\tau}(t-\tau) \left( \mathcal{O}(1) \mathcal{O}\left( d^{1/4} \right) + \mathcal{O}\left( d^{3/2} \right) \mathcal{O}\left( d^{-1/4} \right) \right) = \mathcal{O}\left( \eta^2 d^{9/4} \right).$$

Next let's bound $\left| q_{12}^{(t)}[i,j] \right|$ and $\left| q_{22,i}^{(t)} \right|$. By the assumption of this lemma and the analysis before $T_1$, we know that for all $\tau \leq t$, the $M_1^{(\tau)}, M_2^{(\tau)}$ in Lemma 32 are upper bounded by $\mathcal{O}\left( \frac{1}{d^{1/4}} \right)$ and $\mathcal{O}\left( d^{1/4} \right)$, respectively. In the theorem we consider the training period before $T_{\text{SGD},2}$ so the time $T$ in

Lemma 32 is set as $T_{\text{SGD},2}$. In the following sections, we will prove that $T_{\text{SGD},2} \le \mathcal{O}\left(\frac{d^\alpha \log(\sqrt{d/\epsilon})}{\eta}\right)$.

Then by Lemma 32, we have with probability at least $1 - \frac{1}{d}$, for $\forall \tau \le t$ and $\forall i, j \in [d]$,

$$\left|Dg_1^{(\tau)}[i,j]\right| = \left|\tilde{g}_1^{(\tau)}[i,j] - g_1^{(\tau)}[i,j]\right| \le \mathcal{O}\left(d^{\frac{13}{4}}\sigma\sqrt{\frac{d^{\alpha+1}}{\eta}\log\frac{d}{\epsilon}}\right) + \mathcal{O}\left(d^{\frac{1}{4}}\sigma\sqrt{\frac{d^{\alpha+2}}{\eta}\log\frac{d}{\epsilon}}\right)$$

$$\le \tilde{\mathcal{O}}\left(d^{\frac{13}{4}}\sigma\sqrt{\frac{d^{\alpha+1}}{\eta}}\right),$$

$$\left|Dg_{2i}^{(\tau)}\right| = \left|\tilde{g}_{2i}^{(\tau)} - g_{2i}^{(\tau)}\right| \le \mathcal{O}\left(d^{\frac{15}{4}}\sigma\sqrt{\frac{d^{\alpha+1}}{\eta}\log\frac{d}{\epsilon}}\right) + \mathcal{O}\left(d^{\frac{3}{4}}\sigma\sqrt{\frac{d^{\alpha+2}}{\eta}\log\frac{d}{\epsilon}}\right)$$

$$= \tilde{\mathcal{O}}\left(d^{\frac{15}{4}}\sigma\sqrt{\frac{d^{\alpha+1}}{\eta}}\right).$$

By picking $\sigma \le \frac{\eta^{3/2}}{d^{\alpha/2+1}}$, we have $\left|Dg_1^{(\tau)}[i,j]\right| \le \eta\tilde{\mathcal{O}}\left(d^{\frac{11}{4}}\right)$ and $\left|Dg_{2i}^{(\tau)}\right| \le \eta\tilde{\mathcal{O}}\left(d^{\frac{13}{4}}\right)$, which yields

$$\left|q_{12}^{(t)}[i,j]\right| \le \eta\sum_{\tau=0}^{t}\beta^{t-\tau}\left|Dg_1^{(\tau)}[i,j]\right| \le \tilde{\mathcal{O}}\left(\eta^2 d^{\frac{11}{4}}\right),$$

$$\left|q_{22,i}^{(t)}\right| \le \eta\sum_{\tau=0}^{t}\beta^{t-\tau}\left|Dg_{2i}^{(\tau)}\right| \le \tilde{\mathcal{O}}\left(\eta^2 d^{\frac{13}{4}}\right).$$

Combining the above bounds, we get that $\forall i, j \in [d]$,

$$\left|r_1^{(t)}[i,j]\right| \le \tilde{\mathcal{O}}\left(\eta^2 d^{\frac{11}{4}}\right), \quad \left|r_{2i}^{(t)}\right| \le \tilde{\mathcal{O}}\left(\eta^2 d^{\frac{13}{4}}\right).$$

By the analysis in Section D.2, we know that at time $T_1$, for some $i_0 \in [d]$, $c^{(T_1)}\left|u_{i_0}^{(T_1)}\right| = \Theta\left(\frac{1}{d^{\frac{\alpha}{2}}}\right)$, and for $\forall i, j \in [d]$, we have $c^{(T_1)}\left|u_i^{(T_1)}\right| = \tilde{\Omega}\left(\frac{1}{d^{\frac{3\alpha}{4}}}\right)$ and $\left|u_i^{(T_1)}\right|\left|v_j^{(T_1)}\right| = \tilde{\Omega}\left(\frac{1}{d^{\frac{3\alpha}{4}+\frac{1}{2}}}\right)$, which gives us $\forall i, j \in [d]$,

$$\frac{\left|r_1^{(t)}[i,j]\right|}{\left|u_i^{(T_1)}\right|\left|v_j^{(t)}\right|} \le \frac{\left|r_1^{(t)}[i,j]\right|}{\left|u_i^{(T_1)}\right|\left|v_j^{(T_1)}\right|} = \tilde{\mathcal{O}}\left(\eta^2 d^{\frac{3}{4}\alpha+\frac{13}{4}}\right), \frac{\left|r_{2i}^{(t)}\right|}{c^{(t)}\left|u_i^{(T_1)}\right|} \le \frac{\left|r_{2i}^{(t)}\right|}{c^{(T_1)}\left|u_i^{(T_1)}\right|} = \tilde{\mathcal{O}}\left(\eta^2 d^{\frac{3}{4}\alpha+\frac{13}{4}}\right).$$

Hence we get the bound $\forall i \in [d] : \epsilon_i^{(t)} \le \tilde{\mathcal{O}}\left(\eta^2 d^{\frac{3}{4}\alpha+\frac{13}{4}}\right)$.

### D.12 Proof of Lemma 12

Let's first try to bound the length of $\min\{T_2, T_3\}$. More formally, we prove that under the conditions of Lemma 11 and pick $\eta \le \mathcal{O}\left(\frac{\epsilon}{d^{\frac{7\alpha}{4}+4}}\right)$, we have that $\min\{T_2, T_3\} \le \mathcal{O}\left(\frac{d^\alpha \log(\sqrt{d/\epsilon})}{\eta}\right)$.

Under the conditions of Lemma 11, we know that

$$\forall j \in [d] : \quad \left|\left(r_2^{(t)}W_1^{(t)}\right)_j\right| \le \sum_{i=1}^{d}\left|r_{2i}^{(t)}W_1^{(t)}[i,j]\right| = \mathcal{O}\left(\eta^2 d^4\right),$$

$$\left|\left(W_2^{(t)}r_1^{(t)}\right)_j\right| \le \sum_{i=1}^{d}\left|w_{2i}^{(t)}r_1^{(t)}[i,j]\right| = \mathcal{O}\left(\eta^2 d^4\right).$$

Combining with eq. (23), we get

$$E^{(t+1)}$$
$$=E^{(t)} + \left(W_2^{(t+1)} - W_2^{(t)}\right)W_1^{(t)} + W_2^{(t)}\left(W_1^{(t+1)} - W_1^{(t)}\right) + \left(W_2^{(t+1)} - W_2^{(t)}\right)\left(W_1^{(t+1)} - W_1^{(t)}\right)$$
$$=E^{(t)} - \eta_t E^{(t)} W_1^{(t)T} W_1^{(t)} + r_2^{(t)} W_1^{(t)} - \eta_t W_2^{(t)} W_2^{(t)T} E^{(t)} + W_2^{(t)} r_1^{(t)} + \mathcal{O}\left(\eta^2 d\right)$$
$$=E^{(t)}\left(I - \eta_t W_1^{(t)T} W_1^{(t)} - \eta_t \left\|W_2^{(t)}\right\|_2^2 I\right) + \mathcal{O}\left(\eta^2 d^4\right) + \mathcal{O}\left(\eta^2 d^4\right) + \mathcal{O}\left(\eta^2 d\right).$$

Then we have

$$\left\|E^{(t+1)}\right\|_2 \le \left\|E^{(t)}\right\|_2 \left\|I - \eta_t W_1^{(t)T} W_1^{(t)} - \eta_t \left\|W_2^{(t)}\right\|_2^2 I\right\|_2 + \mathcal{O}\left(\eta^2 d^4\right)$$
$$\le \left(1 - \eta_t \left\|W_2^{(t)}\right\|_2^2\right)\left\|E^{(t)}\right\|_2 + \mathcal{O}\left(\eta^2 d^4\right).$$

When $T_1 \le t < T_2$, we have proved that $c^{(t)}$ is increasing over time in Section D.8, which implies that $\left\|W_2^{(t)}\right\|_2^2 \ge C\left\|W_2^{(T_1)}\right\|_2^2$ since $c^{(t)} u^{(T_1)T}$ is the leading term of $W_2^{(t)}$. Combining with $\eta_t \ge \eta$ gives us

$$\left\|E^{(t+1)}\right\|_2 \le \left(1 - \eta C \left\|W_2^{(T_1)}\right\|_2^2\right)\left\|E^{(t)}\right\|_2 + \mathcal{O}\left(\eta^2 d^4\right),$$
$$\Rightarrow \quad \left\|E^{(t)}\right\|_2 \le \frac{\mathcal{O}\left(\eta^2 d^4\right)}{\eta C \left\|W_2^{(T_1)}\right\|_2^2} + \left(1 - \eta C \left\|W_2^{(T_1)}\right\|_2^2\right)^{t-T_1}\left(\left\|E^{(T_1)}\right\|_2 - \frac{\mathcal{O}\left(\eta^2 d^4\right)}{\eta C \left\|W_2^{(T_1)}\right\|_2^2}\right)$$
$$\overset{(i)}{\le} \mathcal{O}\left(\frac{\eta d^4}{\left\|W_2^{(T_1)}\right\|_2^2}\right) + \exp\left(-\eta C \left\|W_2^{(T_1)}\right\|_2^2 (t - T_1)\right)\mathcal{O}\left(\sqrt{d}\right),$$

where $(i)$ uses $\left\|E^{(T_1)}\right\|_2 = \mathcal{O}(\sqrt{d})$. By picking $\eta \le \mathcal{O}\left(\frac{\epsilon}{d^{\frac{7\alpha}{4}+4}}\right)$ and noticing that $\left\|W_2^{(T_1)}\right\|_2^2 \ge \Omega\left(\frac{1}{d^\alpha}\right)$, we have $\frac{\eta d^4}{\left\|W_2^{(T_1)}\right\|_2^2} < \frac{\sqrt{\epsilon}}{2}$. Hence when $t - T_1 \ge \Theta\left(\frac{\log\left(\sqrt{d/\epsilon}\right)}{\eta \left\|W_2^{(T_1)}\right\|_2^2}\right)$, we have that $\left\|E^{(t)}\right\|_2 \le \sqrt{\epsilon}$, i.e. $\left\|E^{(t)}\right\|_2^2 \le \epsilon$.

That means after at most $\mathcal{O}\left(\frac{\log\left(\sqrt{d/\epsilon}\right)}{\eta \left\|W_2^{(T_1)}\right\|_2^2}\right)$ steps from $T_1$, either $t \ge T_2$, or we have $\left\|E^{(t)}\right\|_2^2 \le \epsilon$. In other words, $\min\{T_2, T_3\} \le T_1 + \mathcal{O}\left(\frac{\log\left(\sqrt{d/\epsilon}\right)}{\eta \left\|W_2^{(T_1)}\right\|_2^2}\right) \le \mathcal{O}\left(\frac{d^\alpha \log\left(\sqrt{d/\epsilon}\right)}{\eta}\right).$

Now we are ready to bound eq. 18.

Combining $\min\{T_2, T_3\} \le \mathcal{O}\left(\frac{d^\alpha \log(\sqrt{d/\epsilon})}{\eta}\right)$ and Lemma 11 yields that for $t + 1 \le \min\{T_2, T_3\}$, $\forall i \in [d]$,

$$\sum_{\tau=T_1}^{t+1} \epsilon_i^{(\tau)} \le (t + 1 - T_1)\tilde{\mathcal{O}}\left(\eta^2 d^{\frac{3}{4}\alpha + \frac{13}{4}}\right) \le \tilde{\mathcal{O}}\left(\eta d^{\frac{7}{4}\alpha + \frac{13}{4}}\log\sqrt{\frac{d}{\epsilon}}\right) = \tilde{\mathcal{O}}\left(\epsilon \log\sqrt{\frac{d}{\epsilon}}\right).$$

Lemma 2 tells us that $\delta_i = \tilde{\mathcal{O}}\left(\frac{1}{d^{\frac{1}{4}\alpha - 1}}\right)$. Substituting these bounds into eq. (18) completes the proof.

## D.13 Proof of Lemma 13

The proof in Section D.8 tells us that for $T_1 \leq \tau \leq T_2$, $c^{(\tau)} > 0, \forall j \in [d] : v_j^{(\tau)} > 0$, which gives us
$0 \leq \frac{\left| R_2^{(t+1)T} \boldsymbol{u}^{(T_1)} \right|}{c^{(t+1)} \left\| \boldsymbol{u}^{(T_1)} \right\|^2}$ and $0 \leq \frac{\left| R_{3j}^{(t+1)} \right|}{c^{(t+1)} \left\| \boldsymbol{u}^{(T_1)} \right\|^2 v_j^{(t+1)}}$. By Lemma 12, we have that

$$\forall 1 \leq i, j \leq d : \quad 0 \leq \frac{\left| R_1^{(t+1)}[i,j] \right|}{\left| u_i^{(T_1)} \right| v_j^{(t+1)}} \leq \tilde{\mathcal{O}}(\epsilon_0), \quad 0 \leq \frac{\left| R_{2i}^{(t+1)} \right|}{c^{(t+1)} \left| u_i^{(T_1)} \right|} \leq \tilde{\mathcal{O}}(\epsilon_0),$$

which gives us

$$\frac{\left| R_2^{(t+1)T} \boldsymbol{u}^{(T_1)} \right|}{c^{(t+1)} \left\| \boldsymbol{u}^{(T_1)} \right\|^2} \leq \frac{\sum_{i=1}^d \left| u_i^{(T_1)} \right| \left| R_{2i}^{(t+1)} \right|}{c^{(t+1)} \sum_{i=1}^d \left| u_i^{(T_1)} \right|^2} \leq \frac{\tilde{\mathcal{O}}(\epsilon_0) c^{(t+1)} \sum_{i=1}^d \left| u_i^{(T_1)} \right|^2}{c^{(t+1)} \sum_{i=1}^d \left| u_i^{(T_1)} \right|^2} = \tilde{\mathcal{O}}(\epsilon_0).$$

Lemma 10 tells us that

$$R_3^{(t+1)T} = c^{(t+1)} \boldsymbol{u}^{(T_1)T} R_1^{(t+1)} + R_2^{(t+1)T} R_1^{(t+1)}.$$

And we have that

$$\frac{\left| \left( c^{(t+1)} \boldsymbol{u}^{(T_1)T} R_1^{(t+1)} \right)_j \right|}{c^{(t+1)} \left\| \boldsymbol{u}^{(T_1)} \right\|^2 v_j^{(t+1)}} \leq \frac{c^{(t+1)} \sum_{i=1}^d \left| u_i^{(T_1)} \right| \left| R_1^{(t+1)}[i,j] \right|}{c^{(t+1)} \sum_{i=1}^d \left| u_i^{(T_1)} \right|^2 v_j^{(t+1)}}$$

$$\leq \frac{\tilde{\mathcal{O}}(\epsilon_0) c^{(t+1)} \sum_{i=1}^d \left| u_i^{(T_1)} \right|^2 v_j^{(t+1)}}{c^{(t+1)} \sum_{i=1}^d \left| u_i^{(T_1)} \right|^2 v_j^{(t+1)}} = \tilde{\mathcal{O}}(\epsilon_0),$$

$$\frac{\left| \left( R_2^{(t+1)T} R_1^{(t+1)} \right)_j \right|}{c^{(t+1)} \left\| \boldsymbol{u}^{(T_1)} \right\|^2 v_j^{(t+1)}} \leq \frac{\sum_{i=1}^d \left| R_{2i}^{(t+1)} \right| \left| R_1^{(t+1)}[i,j] \right|}{c^{(t+1)} \sum_{i=1}^d \left| u_i^{(T_1)} \right|^2 v_j^{(t+1)}} \leq \frac{\tilde{\mathcal{O}}\left(\epsilon_0^2\right) c^{(t+1)} \sum_{i=1}^d \left| u_i^{(T_1)} \right|^2 v_j^{(t+1)}}{c^{(t+1)} \sum_{i=1}^d \left| u_i^{(T_1)} \right|^2 v_j^{(t+1)}}$$

$$= \tilde{\mathcal{O}}\left(\epsilon_0^2\right).$$

Therefore

$$\frac{\left| R_{3j}^{(t+1)} \right|}{c^{(t+1)} \left\| \boldsymbol{u}^{(T_1)} \right\|^2 v_j^{(t+1)}} \leq \frac{\left| \left( c^{(t+1)} \boldsymbol{u}^{(T_1)T} R_1^{(t+1)} \right)_j \right|}{c^{(t+1)} \left\| \boldsymbol{u}^{(T_1)} \right\|^2 v_j^{(t+1)}} + \frac{\left| \left( R_2^{(t+1)T} R_1^{(t+1)} \right)_j \right|}{c^{(t+1)} \left\| \boldsymbol{u}^{(T_1)} \right\|^2 v_j^{(t+1)}} \leq \tilde{\mathcal{O}}(\epsilon_0).$$

By Lemma 10,

$$W_2^{(t+1)} W_1^{(t+1)} = c^{(t+1)} \left\| \boldsymbol{u}^{(T_1)} \right\|^2 \boldsymbol{v}^{(t+1)T} + R_2^{(t+1)T} \boldsymbol{u}^{(T_1)} \boldsymbol{v}^{(t+1)T} + R_3^{(t+1)T}.$$

Then we have that $\forall j \in [d]$,

$$\left( W_2^{(t+1)} W_1^{(t+1)} \right)_j = c^{(t+1)} \left\| \boldsymbol{u}^{(T_1)} \right\|^2 v_j^{(t+1)} \left( 1 + e_j^{(t+1)} \right), \quad \text{where } \left| e_j^{(t+1)} \right| \leq \tilde{\mathcal{O}}(\epsilon_0). \quad (24)$$

Since $t < T_2$, we have $\forall j \in [d] : \frac{\left( W_2^{(t+1)} W_1^{(t+1)} \right)_j}{A_j} = \mathcal{O}(1)$, which yields

$$0 \leq \frac{c^{(t+1)} \left\| \boldsymbol{u}^{(T_1)} \right\|^2 v_j^{(t+1)}}{A_i} \leq \mathcal{O}(1), \quad (25)$$

which proves eq. (21), since $\forall j \in [d] : \quad 0 \leq \frac{\left| R_{3j}^{(t+1)} \right|}{c^{(t+1)} \left\| \boldsymbol{u}^{(T_1)} \right\|^2 v_j^{(t+1)}} \leq \tilde{\mathcal{O}}(\epsilon_0).$

## D.14   Proof of Lemma 14

(A) Under the conditions of Lemma 11 and pick $\eta \leq \mathcal{O}\left(\frac{\epsilon}{d^{\frac{7\alpha}{4}+4}}\right)$, we can apply the technique when proving eq. (24) to show that eq. (24) also holds at time $t$. Since $\frac{\left|R_{vj}^{(t)}\right|}{a^{(t)}A_j} \leq \tilde{\mathcal{O}}(\epsilon_0)$, we get that

$$v_j^{(t)} = a^{(t)}A_j + R_{vj}^{(t)} = a^{(t)}A_j\left(1 + e_{vj}^{(t)}\right), \quad \text{where } \left|e_{vj}^{(t)}\right| \leq \tilde{\mathcal{O}}(\epsilon_0).$$

Substituting into the time $t$ version of eq.(24) yields

$$\forall j \in [d]: \quad \left(W_2^{(t)}W_1^{(t)}\right)_j = a^{(t)}c^{(t)}\left\|\boldsymbol{u}^{(T_1)}\right\|^2 A_j\left(1 + \tilde{e}_j^{(t)}\right), \quad \text{where } \left|\tilde{e}_j^{(t)}\right| \leq \tilde{\mathcal{O}}(\epsilon_0),$$

That gives us

$$\forall j \in [d]: E_j^{(t)} = A_j\left(a^{(t)}c^{(t)}\left\|\boldsymbol{u}^{(T_1)}\right\|^2 - 1 + a^{(t)}c^{(t)}\left\|\boldsymbol{u}^{(T_1)}\right\|^2 \tilde{e}_j^{(t)}\right).$$

Since $t < T_2$, we have $E_j^{(t)} < -\sqrt{\epsilon_0}$. Combining with $A_j = \Theta(1)$, gives us $a^{(t)}c^{(t)}\left\|\boldsymbol{u}^{(T_1)}\right\|^2 - 1 = -\Omega\left(\sqrt{\epsilon_0}\right)$. Then we can rewrite $E_j^{(t)}$ as $\forall j \in [d]$,

$$E_j^{(t)} = A_j\left(a^{(t)}c^{(t)}\left\|\boldsymbol{u}^{(T_1)}\right\|^2 - 1\right)\left(1 + \frac{a^{(t)}c^{(t)}\left\|\boldsymbol{u}^{(T_1)}\right\|^2}{a^{(t)}c^{(t)}\left\|\boldsymbol{u}^{(T_1)}\right\|^2 - 1}\tilde{e}_j^{(t)}\right)$$

$$:= A_j\left(a^{(t)}c^{(t)}\left\|\boldsymbol{u}^{(T_1)}\right\|^2 - 1\right)\left(1 + e_{Ej}^{(t)}\right),$$

where $\left|e_{Ej}^{(t)}\right| = \tilde{\mathcal{O}}(\sqrt{\epsilon_0})$. Hence $\forall i,j \in [d]: \frac{E_j^{(t)}}{E_i^{(t)}} = \Theta(1)$.

(B) Note that we assume $\forall j \in [d]: \frac{v_j^{(t)}}{c^{(t)}} = \Theta\left(\frac{1}{\sqrt{d}}\right)$, then we have for $j \in [d]$,

$$\frac{-E^{(t)}\boldsymbol{v}^{(t)}}{c^{(t)}\left(-E_j^{(t)}\right)} = \frac{\sum_{i=1}^d \left(-E_i^{(t)}\right)v_i^{(t)}}{c^{(t)}\left(-E_j^{(t)}\right)} = \sum_{i=1}^d \frac{E_i^{(t)}}{E_j^{(t)}}\cdot\frac{v_i^{(t)}}{c^{(t)}} = \sum_{i=1}^d \Theta\left(\frac{1}{\sqrt{d}}\right) = \Theta\left(\sqrt{d}\right),$$

$$\Rightarrow \quad \frac{c^{(t)}\left(-E_j^{(t)}\right)}{-E^{(t)}\boldsymbol{v}^{(t)}} = \Theta\left(\frac{1}{\sqrt{d}}\right).$$

Then for $t+1$, we have that for $j \in [d]$,

$$\frac{v_j^{(t+1)}}{c^{(t+1)}} = \frac{v_j^{(t)} + \eta_t c^{(t)}\left(-E_j^{(t)}\right)}{c^{(t)} + \eta_t\left(-E^{(t)}\right)\boldsymbol{v}^{(t)}} = \Theta\left(\frac{1}{\sqrt{d}}\right).$$

(C) Combining eq. (25) and $\forall j \in [d]: A_j = \Theta(1)$, we know that

$$c^{(t+1)}\left\|\boldsymbol{u}^{(T_1)}\right\|^2 v_j^{(t+1)} \leq \mathcal{O}(1),$$

which yields $\forall j \in [d]$,

$$\left\|\boldsymbol{u}^{(T_1)}\right\|^2\left(v_j^{(t+1)}\right)^2 \leq \frac{v_j^{(t+1)}}{c^{(t+1)}}\mathcal{O}(1) = \mathcal{O}\left(\frac{1}{\sqrt{d}}\right), \quad \left(c^{(t+1)}\right)^2\left\|\boldsymbol{u}^{(T_1)}\right\|^2 \leq \frac{c^{(t+1)}}{v_j^{(t+1)}}\mathcal{O}(1) = \mathcal{O}\left(\sqrt{d}\right). \tag{26}$$

Hence $\forall i,j \in [d]$,

$$c^{(t+1)}\left|u_i^{(T_1)}\right| = \mathcal{O}\left(d^{1/4}\right) \quad \Rightarrow \left|w_{2i}^{(t+1)}\right| \leq c^{(t+1)}\left|u_i^{(T_1)}\right| + \left|R_{2i}^{(t+1)}\right| \stackrel{(i)}{=} \mathcal{O}\left(d^{1/4}\right),$$

$$\left|u_i^{(T_1)}\right|v_j^{(t+1)} = \mathcal{O}\left(\frac{1}{d^{1/4}}\right) \quad \Rightarrow \left|W_1^{(t+1)}[i,j]\right| \leq \left|u_i^{(T_1)}\right|v_j^{(t+1)} + \left|R_1^{(t+1)}[i,j]\right| \stackrel{(ii)}{=} \mathcal{O}\left(\frac{1}{d^{1/4}}\right),$$

where $(i)$ and $(ii)$ use Lemma 12.

(D) The fact that $\forall j \in [d]: \dfrac{\left|R_{3j}^{(t+1)}\right|}{A_j} \leq \tilde{\mathcal{O}}(\epsilon_0)$ was already proved in Lemma 13 in eq.(21). To analyze $\dfrac{\left|R_{vi}^{(t+1)}\right|}{a^{(t+1)}A_i}$, we first prove that $1 - \eta_t c^{(t)} d^{(t)} > 0$.

It is not hard to prove that eq.(26) also holds for time $t$. Recall that $d^{(t)} = c^{(t)} \left\|\boldsymbol{u}^{(T_1)}\right\|^2 + R_2^{(t)T} \boldsymbol{u}^{(T_1)}$ and Lemma 13 tells us that $0 \leq \dfrac{\left|R_2^{(t)T}\boldsymbol{u}^{(T_1)}\right|}{c^{(t)}\left\|\boldsymbol{u}^{(T_1)}\right\|^2} \leq \tilde{\mathcal{O}}(\epsilon_0)$, then we have

$$c^{(t)}d^{(t)} = \left(c^{(t)}\right)^2 \left\|\boldsymbol{u}^{(T_1)}\right\|^2 + c^{(t)}R_2^{(t)T}\boldsymbol{u}^{(T_1)} \leq \mathcal{O}\left(\sqrt{d}\right).$$

Under the conditions of Lemma 11, and pick $\eta \leq \mathcal{O}\left(\dfrac{\epsilon}{d^{\frac{7\alpha}{4}+4}}\right)$, we have that $1 - \eta_t c^{(t)} d^{(t)} \geq 1 - \eta c^{(t)} d^{(t)} > 0$.

The assumption $\forall j \in [d]: \dfrac{\left|R_{3j}^{(t)}\right|}{A_j} \leq \tilde{\mathcal{O}}(\epsilon_0)$ together with $c^{(t)} > 0$ gives us

$$\frac{\eta_t c^{(t)} \left|R_{3j}^{(t)}\right|}{\eta_t c^{(t)} A_j} \leq \tilde{\mathcal{O}}(\epsilon_0).$$

Combining with the assumption $\dfrac{\left|R_{vi}^{(t)}\right|}{a^{(t)}A_i} \leq \tilde{\mathcal{O}}(\epsilon_0)$ yields

$$\forall i \in [d]: \quad \frac{\left|R_{vi}^{(t+1)}\right|}{a^{(t+1)}A_i} \leq \frac{\left(1 - \eta_t c^{(t)} d^{(t)}\right)\left|R_v^{(t)}\right| + \eta_t c^{(t)}\left|R_{3i}^{(t)}\right|}{\left(1 - \eta_t c^{(t)} d^{(t)}\right)a^{(t)}A_i + \eta_t c^{(t)}A_i} \leq \tilde{\mathcal{O}}(\epsilon_0).$$

## E Analysis of Adam

Note that $A = \frac{1}{m}YX^T$, $\Lambda_{xx} := \frac{1}{m}XX^T$. Denote $g_k^{(t)} := \nabla_{W_k} L(W^{(t)}), k = 1,2$. We have that
$$g_1^{(t)} = W_2^{(t)T}\left(W_2^{(t)}W_1^{(t)} - A\right), \quad g_2^{(t)} = \left(W_2^{(t)}W_1^{(t)} - A\right)W_1^{(t)T}.$$

Let $\tilde{A}^{(t)}$, $\tilde{\Lambda}_{xx}^{(t)}$ and $\tilde{g}_k^{(t)}, k = 1,2$ be the corresponding batch versions at time $t$. Let $E^{(t)} := W_2^{(t)}W_1^{(t)} - A$, and denote $E_j^{(t)}$ as the $j$-th component of $E^{(t)}$. We also denote $\Delta w_{2i}^{(t)} := w_{2i}^{(t+1)} - w_{2i}^{(t)}$, $\Delta W_1^{(t)}[i,j] := W_1^{(t+1)}[i,j] - W_1^{(t)}[i,j]$. By eq. (2), the update equations of Adam are given by

$$\eta_t = \eta \cdot \frac{\sqrt{1-\beta_2^{t+1}}}{1-\beta_1^{t+1}}, \quad g_1^{(t)}[i,j] = w_{2i}^{(t)}E_j^{(t)}, \quad g_{2i}^{(t)} = \left\langle E^{(t)}, W_1^{(t)}[i,:]\right\rangle,$$

$$W_1^{(t+1)}[i,j] - W_1^{(t)}[i,j] = -\eta_t \frac{m_1^{(t)}[i,j]}{\sqrt{v_1^{(t)}[i,j]}} = -\eta_t \frac{(1-\beta_1)\sum_{\tau=0}^t \beta_1^{t-\tau}\tilde{g}_1^{(\tau)}[i,j]}{\sqrt{(1-\beta_2)\sum_{\tau=0}^t \beta_2^{t-\tau}\left(\tilde{g}_1^{(\tau)}[i,j]\right)^2 + \xi}}$$

$$= -\eta_t \frac{(1-\beta_1)\sum_{\tau=0}^t \beta_1^{t-\tau}g_1^{(\tau)}[i,j] + r_{1n}^{(t)}[i,j]}{\sqrt{(1-\beta_2)\sum_{\tau=0}^t \beta_2^{t-\tau}\left(g_1^{(\tau)}[i,j]\right)^2 + r_{1d}^{(t)}[i,j] + \xi}},$$

$$w_{2i}^{(t+1)} - w_{2i}^{(t)} = -\eta_t \frac{m_{2i}^{(t)}}{\sqrt{v_{2i}^{(t)}}} = -\eta_t \frac{(1-\beta_1)\sum_{\tau=0}^t \beta_1^{t-\tau}\tilde{g}_{2i}^{(\tau)}}{\sqrt{(1-\beta_2)\sum_{\tau=0}^t \beta_2^{t-\tau}\left(\tilde{g}_{2i}^{(\tau)}\right)^2 + \xi}}$$

$$= -\eta_t \frac{(1-\beta_1)\sum_{\tau=0}^t \beta_1^{t-\tau}g_{2i}^{(\tau)} + r_{2n,i}^{(t)}}{\sqrt{(1-\beta_2)\sum_{\tau=0}^t \beta_2^{t-\tau}\left(g_{2i}^{(\tau)}\right)^2 + +r_{2d,i}^{(t)} + \xi}}.$$

$$(27)$$

where $Dg_1^{(t)} := \tilde{g}_1^{(t)} - g_1^{(t)}$ and $Dg_2^{(t)} := \tilde{g}_2^{(t)} - g_2^{(t)}$, and

$$r_{1n}^{(t)}[i,j] := (1 - \beta_1) \sum_{\tau=0}^{t} \beta_1^{t-\tau} Dg_1^{(\tau)}[i,j],$$

$$r_{1d}^{(t)}[i,j] = (1 - \beta_2) \sum_{\tau=0}^{t} \beta_2^{t-\tau} \left( 2g_1^{(\tau)}[i,j] Dg_1^{(\tau)}[i,j] + \left( Dg_1^{(\tau)}[i,j] \right)^2 \right),$$

$$r_{2n,i}^{(t)} := (1 - \beta_1) \sum_{\tau=0}^{t} \beta_1^{t-\tau} Dg_{2i}^{(\tau)},$$

$$r_{2d,i}^{(t)} = (1 - \beta_2) \sum_{\tau=0}^{t} \beta_2^{t-\tau} \left( 2g_{2i}^{(\tau)} Dg_{2i}^{(\tau)} + \left( Dg_{2i}^{(\tau)} \right)^2 \right).$$

(28)

Denote the $i$-th coordinate of $W_2 W_1$ and $A$ as $(W_2 W_1)_i$ and $A_i$, respectively. By Assumption 2 and the assumption that $\forall i \in [d] : A_i > 0, A_i = \Omega(1)$, at the beginning, w.h.p., $\forall i \in [d] : (W_2 W_1)_i - A_i < 0$. Based on this, we divide the training procedure into two phases (note that these two phases are different from those of SGD+M).

1. **First phase**: when the error $(W_2 W_1)_i - A_i$ is negative and its absolute value is big for all $i \in [d]$.
2. **Second phase**: when $(W_2 W_1)_i - A_i$ is close to zero for some coordinate $i \in [d]$.

More formally, we define the boundary between the two phases below.

**Definition 4** (End of the first phase). *The end of the first phase (denoted as $T_1$) is defined as* $T_1 = \inf \left\{ t > 0 : \exists i \in [d] : E_i^{(t)} \geq -\sqrt{\eta d} \right\}.$

In the second phase, we define some time points.

**Definition 5.** *Define* $T_g := \inf \left\{ t > T_1 : \exists i \in [d] : \left| g_{2i}^{(t)} \right| \leq d\sqrt{\eta} \right\}.$

For $t < T_1$, we have $\forall i \in [d] : E_i^{(t)} < 0$ by Definition 4. For $t > T_1$, some $E_i^{(t)}$ may flip the sign and become positive. For certain coordinate $i$, we define the following "flip time".

**Definition 6.** *Define* $T_{f,i} := \inf \left\{ t > T_1 : E_i^{(t)} \geq -\sqrt{\eta d} \right\}$. *Define* $T_f := \max_i T_{f,i}$ *as the largest "flip time" over all $i \in [d]$, i.e. the "flip time" of the last $E_i$ which flips the sign. Moreover, denote* $\tilde{T} := \min \{ T_g, T_f \}$.

We can first show that after a few steps in the first phase, $W_1$ will become an approximately rank-1 matrix, as described in the following lemma.

**Lemma 15.** *Under Assumption 1, 2 and 3, suppose $\sigma \leq \frac{\eta^{3/2} \xi^2}{d^{13/4}}$. By picking $\eta \leq \mathcal{O}\left(\frac{1}{d^{3\alpha}}\right), \xi \leq \sqrt{\frac{\eta}{d^{3\alpha-1}}}$, and $\beta_2 = \beta_1^2$, there exists $t_{inc} > 0$ such that w.h.p. for $t_{inc} \leq t < T_1$,*

$$\forall i, j \in [d] : \quad w_{2i}^{(t)} = sign\left(w_{2i}^{(0)}\right) \eta \left(t - t_{inc}\right) + R_{2i}^{(t)},$$

$$W_1^{(t)}[i,j] = sign\left(w_{2i}^{(0)}\right) \eta \left(t - t_{inc}\right) + R_1^{(t)}[i,j],$$

*where* $\frac{\left| R_1^{(t)}[i,j] \right|}{\eta(t - t_{inc})} = \tilde{\mathcal{O}}\left( \sqrt{\eta} + \frac{1}{\eta(t-t_{inc})d^\alpha} \right), \quad \frac{\left| R_{2i}^{(t)} \right|}{\eta(t - t_{inc})} = \tilde{\mathcal{O}}\left( \sqrt{\eta} + \frac{1}{\eta(t-t_{inc})d^\alpha} \right).$

*Specially, when $t = T_1$, we have that*

$$\forall i, j \in [d] : \quad w_{2i}^{(T_1)} = sign\left(w_{2i}^{(0)}\right) \eta \left(T_1 - t_{inc}\right) + R_{2i}^{(T_1)},$$

$$W_1^{(T_1)}[i,j] = sign\left(w_{2i}^{(0)}\right) \eta (T_1 - t_{inc}) + R_1^{(T_1)}[i,j],$$

*where $\eta (T_1 - t_{inc}) = \Theta\left(\frac{1}{\sqrt{d}}\right)$ and*

$$\frac{\left| R_1^{(T_1)}[i,j] \right|}{\eta (T_1 - t_{inc})} = \tilde{\mathcal{O}}\left( \sqrt{\eta} + \frac{1}{d^{\alpha-\frac{1}{2}}} \right), \quad \frac{\left| R_{2i}^{(T_1)} \right|}{\eta (T_1 - t_{inc})} = \tilde{\mathcal{O}}\left( \sqrt{\eta} + \frac{1}{d^{\alpha-\frac{1}{2}}} \right).$$

The following lemma tells us that this approximate rank-1 structure is preserved when $T_1 \le t \le \tilde{T}$.

**Lemma 16.** *Under Assumption 1, 2 and 3, suppose $\sigma \le \frac{\eta^{3/2}\xi^2}{d^{13/4}}$. By picking $\eta \le \mathcal{O}\left(\frac{1}{d^{3\alpha}}\right), \xi \le \sqrt{\frac{\eta}{d^{3\alpha-1}}}$, and $\beta_2 = \beta_1^2$, we have w.h.p. for $T_1 \le t < \tilde{T}$,*

$$\forall i, j \in [d]: \quad w_{2i}^{(t)} = sign\left(w_{2i}^{(0)}\right) c^{(t)} + R_{2i}^{(t)},$$

$$W_1^{(t)}[i, j] = sign\left(w_{2i}^{(0)}\right) V_j^{(t)} + R_1^{(t)}[i, j],$$

$$where \quad \frac{\left|R_{2i}^{(t)}\right|}{\left|c^{(t)}\right|} = \tilde{\mathcal{O}}\left(\sqrt{\eta} + \frac{1}{d^{\alpha-1/2}}\right), \quad \frac{\left|R_1^{(t)}[i, j]\right|}{\left|V_j^{(t)}\right|} \le \tilde{\mathcal{O}}\left(\eta^{\frac{1}{4}} + \frac{1}{d^{\frac{\alpha}{2}-\frac{1}{4}}}\right),$$

*and that $L\left(W^{(\tilde{T})}\right) \le \tilde{\mathcal{O}}\left(\eta d^4\right)$.*

Now we are ready to prove the Adam part of Theorem 1.

### E.1 Proof of the Adam part of Theorem 1

Define $T_{\text{Adam},1} = t_{\text{inc}} + \frac{1}{\eta d^{\frac{\alpha}{2}}}$. Note that this choice of $T_{\text{Adam},1}$ gives $\eta\left(T_{\text{Adam},1} - t_{\text{inc}}\right) = \frac{1}{d^{\frac{\alpha}{2}}}$. By picking $\eta \le \mathcal{O}\left(\frac{1}{d^{3\alpha}}\right), \xi \le \sqrt{\frac{\eta}{d^{3\alpha-1}}}$ and $\beta_2 = \beta_1^2$, we can apply Lemma 15 to get that $\forall i, j \in [d]$ : $\left|w_{2i}^{(T_{\text{Adam},1})}\right| = \Theta\left(\frac{1}{d^{\frac{\alpha}{2}}}\right), \left|W_1^{(T_{\text{Adam},1})}[i, j]\right| = \Theta\left(\frac{1}{d^{\frac{\alpha}{2}}}\right)$, and therefore $\forall i \in [d]$ : $E_i^{(T_{\text{Adam},1})} = -\Theta(1)$ and $L\left(W^{(T_{\text{Adam},1})}\right) = \Theta(d)$. Define $T_{\text{Adam},2} = \tilde{T}$. By Lemma 16, we have $L\left(W^{(T_{\text{Adam},2})}\right) = \tilde{\mathcal{O}}\left(\eta d^4\right)$. For any $p > 0$, by picking $\alpha \ge \frac{p+4}{3}$, we have $L\left(W^{(T_{\text{Adam},2})}\right) = \tilde{\mathcal{O}}\left(\eta d^4\right) \le \tilde{\mathcal{O}}\left(\frac{1}{d^p}\right)$.

Moreover, combining Lemma 15 and 16, we get that when $t \in [T_{\text{Adam},1}, T_{\text{Adam},2}]$, the conditions in Lemma 31 are satisfied with $\delta = \tilde{\mathcal{O}}\left(\eta^{\frac{1}{4}} + \frac{1}{d^{\frac{\alpha}{2}-\frac{1}{4}}}\right)$. The $i$-th component of the $u$ vector (denoted as $u_i$) is $sign\left(w_{2i}^{(0)}\right)$. That means $\forall i \in [d]$ : $u_i^2 = 1$ and $\frac{\max_i(u_i)^2}{\text{median}(u_i)^2} = 1$. Then we can apply Lemma 31 and get that

$$R_{\text{med},1}^{\text{Adam}}(t), R_{\text{med},2}^{\text{Adam}}(t) \in \left[\left(\frac{1-\delta}{1+\delta}\right)^2 \frac{\max_i(u_i)^2}{\text{median}(u_i)^2}, \left(\frac{1+\delta}{1-\delta}\right)^2 \frac{\max_i(u_i)^2}{\text{median}(u_i)^2}\right]$$

$$= \left[\left(\frac{1-\delta}{1+\delta}\right)^2, \left(\frac{1+\delta}{1-\delta}\right)^2\right].$$

Note that by definition, $R_{\text{med},1}^{\text{Adam}}(t)$ and $R_{\text{med},2}^{\text{Adam}}(t)$ are always larger than or equal to 1, then we have

$$R_{\text{med},1}^{\text{Adam}}(t), R_{\text{med},2}^{\text{Adam}}(t) = 1 + \mathcal{O}(\delta) = 1 + \tilde{\mathcal{O}}\left(\eta^{\frac{1}{4}} + \frac{1}{d^{\frac{\alpha}{2}-\frac{1}{4}}}\right).$$

### E.2 Proof of Lemma 15

For some time $t$, we introduce two conditions.
**Condition 1.**

$$\forall \tau \in [H]: sign\left(g_1^{(t-\tau)}[i, j]\right) = s_1^{(t)}[i, j], \quad (1-\beta_1)\left|\sum_{\tau=0}^{H} \beta_1^{(\tau)} g_1^{(t-\tau)}[i, j]\right| \ge \Omega(\xi).$$

**Condition 2.**

$$\forall \tau \in [H]: sign\left(g_{2i}^{(t-\tau)}\right) = s_{2i}^{(t)}, \quad (1-\beta_1)\left|\sum_{\tau=0}^{H} \beta_1^{(\tau)} g_{2i}^{(t-\tau)}\right| \ge \Omega(\xi).$$

Next prove that, under Assumption 1 and 2, by picking $\eta \leq \mathcal{O}\left(\frac{1}{d^{3\alpha}}\right), \xi \leq \sqrt{\frac{\eta}{d^{3\alpha-1}}}$, and $\beta_2 = \beta_1^2$, there exists $t_{\text{inc}} > 0$ such that for $t_{\text{inc}} \leq t < T_1$, the update of Adam can be approximated as that of signed descent.

$$
\begin{aligned}
W_1^{(t+1)}[i,j] &= W_1^{(t)}[i,j] - \eta \left( \text{sign}\left( g_1^{(t)}[i,j] \right) + e_1^{(t)}[i,j] \right), \\
w_{2i}^{(t+1)} &= w_{2i}^{(t)} - \eta \left( \text{sign}\left( g_{2i}^{(t)} \right) + e_{2i}^{(t)} \right),
\end{aligned}
\tag{29}
$$

where $\left| e_1^{(t)}[i,j] \right| = \tilde{\mathcal{O}}\left(\sqrt{\eta}\right), \quad \left| e_{2i}^{(t)} \right| = \tilde{\mathcal{O}}\left(\sqrt{\eta}\right)$.

Before we dive into the proof, let's introduce some useful lemmas.

The following lemma reflects our key idea: converting the exponential average in Adam to a finite-step average, and trying to bound the stochastic error terms in eq. (28).

**Lemma 17.** *Under Assumption 1, 2 and 3 and pick* $\beta_2 = \beta_1^2$. *Let* $M_1^{(t)} := \max_{i,j\in[d], \tau\leq t} \left| W_1^{(\tau)}[i,j] \right|, M_2^{(t)} := \max_{i,j\in[d], \tau\leq t} \left| w_{2i}^{(\tau)} \right|, G_1^{(t)} := \max_{i,j\in[d], \tau\leq t} \left| g_1^{(\tau)}[i,j] \right|$ *and* $G_2^{(t)} := \max_{i,j\in[d], \tau\leq t} \left| g_{2i}^{(\tau)} \right|$. *We have that w.h.p., for all* $t \leq \tilde{\mathcal{O}}\left(\frac{1}{\sqrt{d}\eta}\right)$ *and* $\forall i,j \in [d]$,*

$$
\Delta W_1^{(t)}[i,j] = -\eta_t \frac{(1-\beta_1)\sum_{\tau=0}^{H} \beta_1^{\tau} g_1^{(t-\tau)}[i,j] + \epsilon_{1n}^{(t)}[i,j]}{\sqrt{(1-\beta_2)\sum_{\tau=0}^{H} \beta_2^{\tau} \left( g_1^{(t-\tau)}[i,j] \right)^2 + \epsilon_{1d}^{(t)}[i,j]} + \xi},
$$

$$
\Delta w_{2i}^{(t)} = -\eta_t \frac{(1-\beta_1)\sum_{\tau=0}^{H} \beta_1^{\tau} g_{2i}^{(t-\tau)} + \epsilon_{2n,i}^{(t)}}{\sqrt{(1-\beta_2)\sum_{\tau=0}^{H} \beta_2^{\tau} \left( g_{2i}^{(t-\tau)} \right)^2 + \epsilon_{2d,i}^{(t)}} + \xi},
$$

*where* $H \geq \frac{1}{1-\beta_1} \log \frac{\max\left\{ G_1^{(t)}, G_2^{(t)}, \left(G_1^{(t)}\right)^2, \left(G_2^{(t)}\right)^2 \right\}}{\eta\xi^2}$ *and*

$$
\left| \epsilon_{1n}^{(t)}[i,j] \right| \leq \mathcal{O}(\eta\xi^2) + \mathcal{O}\left(D_1^{(t)}\right), \quad \left| \epsilon_{1d}^{(t)}[i,j] \right| \leq \mathcal{O}(\eta\xi^2) + \mathcal{O}\left(D_1^{(t)}G_1^{(t)} + \left(D_1^{(t)}\right)^2\right),
$$

$$
\left| \epsilon_{2n,i}^{(t)} \right| \leq \mathcal{O}(\eta\xi^2) + \mathcal{O}\left(D_2^{(t)}\right), \quad \left| \epsilon_{2d,i}^{(t)} \right| \leq \mathcal{O}(\eta\xi^2) + \mathcal{O}\left(D_2^{(t)}G_2^{(t)} + \left(D_2^{(t)}\right)^2\right),
$$

*with*

$$
D_1^{(t)} \leq \tilde{\mathcal{O}}\left( d^3 M_1^{(t)} \left(M_2^{(t)}\right)^2 \sigma \sqrt{\frac{d^{1/2}}{\eta}} \right) + \tilde{\mathcal{O}}\left( M_2^{(t)} \sigma \sqrt{\frac{d^{3/2}}{\eta}} \right),
$$

$$
D_2^{(t)} \leq \tilde{\mathcal{O}}\left( d^4 \left(M_1^{(t)}\right)^2 M_2^{(t)} \sigma \sqrt{\frac{d^{1/2}}{\eta}} \right) + \tilde{\mathcal{O}}\left( d M_1^{(t)} \sigma \sqrt{\frac{d^{3/2}}{\eta}} \right).
$$

**Corollary 2.** *Under the conditions of Lemma 17 and suppose* $\sigma \leq \frac{\eta^{3/2}\xi^2}{d^{13/4}}$. *Consider any* $t \leq \tilde{\mathcal{O}}\left(\frac{1}{\sqrt{d}\eta}\right)$. *If* $M_1^{(t)}, M_2^{(t)} \leq \tilde{\mathcal{O}}\left(\frac{1}{\sqrt{d}}\right), G_1^{(t)} \leq \tilde{\mathcal{O}}\left(\frac{1}{\sqrt{d}}\right), G_2^{(t)} \leq \tilde{\mathcal{O}}\left(\sqrt{d}\right)$, *then* $H$ *in Lemma 17 can be picked as* $\frac{1}{1-\beta_1} \log \frac{d}{\eta\xi^2}$ *and we can get that* $\forall i,j \in [d], \left| \epsilon_{1n}^{(t)}[i,j] \right|, \left| \epsilon_{1d}^{(t)}[i,j] \right|, \left| \epsilon_{2n,i}^{(t)} \right|, \left| \epsilon_{2d,i}^{(t)} \right| \leq \tilde{\mathcal{O}}(\eta\xi^2)$.

The following lemma analyzes the magnitude of weights during a short period at the beginning.

**Lemma 18.** *Under Assumption 1, 2 and 3, suppose* $\sigma \leq \frac{\eta^{3/2}\xi^2}{d^{13/4}}$. *Pick* $\xi \leq \frac{1}{d^{\frac{3}{2}\alpha}}$, *then there exists some time point* $t_{inc} \in (H, T_1)$, *such that w.h.p., for* $t \leq t_{inc}$, *for every* $i,j \in [d]$,

$$
\left| \Delta W_1^{(t)}[i,j] \right| \leq \tilde{\mathcal{O}}(\eta), \left| \Delta w_{2i}^{(t)} \right| \leq \tilde{\mathcal{O}}(\eta),
$$

$$
\left| W_1^{(t)}[i,j] \right| \leq \mathcal{O}\left(\frac{1}{d^{\frac{3}{2}\alpha+1}}\right), \Omega\left(\frac{1}{d^{\frac{3}{2}\alpha}}\right) \leq \left| w_{2i}^{(t)} \right| \leq \mathcal{O}\left(\frac{1}{d^{\alpha}}\right).
$$

Specifically, when $t = t_{inc}$, we have $sign\left(w_{2i}^{(t_{inc})}\right) = sign\left(W_1^{(t_{inc})}[i,j]\right) = sign\left(w_{2i}^{(0)}\right)$, $\left|W_1^{(t_{inc})}[i,j]\right| = \Theta\left(\frac{1}{d^{\frac{3}{2}\alpha+1}}\right)$ and $\left|g_1^{(t_{inc})}[i,j]\right| \geq \Omega(\xi), \left|g_{2i}^{(t_{inc})}\right| \geq \Omega(\xi)$. Moreover, Condition 1 and 2 are satisfied for $t = t_{inc}$. The $s_1^{(t)}[i,j]$ and $s_{2i}^{(t)}$ in the conditions are both $-sign\left(w_{2i}^{(0)}\right)$.

The following lemma gives us lower bounds of $\left|g_1^{(t)}[i,j]\right|$ and $\left|g_{2i}^{(t)}\right|$.

**Lemma 19.** *Under Assumption 1, 2 and 3, suppose $\sigma \leq \frac{\eta^{3/2}\xi^2}{d^{13/4}}$. Pick $\xi \leq \sqrt{\frac{\eta}{d^{3\alpha-1}}}, \eta \leq \mathcal{O}\left(\frac{1}{d^{3\alpha}}\right)$. Consider $t_{inc}$ in Lemma 18. We have w.h.p. for any $t \in [t_{inc}, T_1)$, and for $\forall i, j \in [d]$, $sign\left(\Delta W_1^{(t)}[i,j]\right) = sign\left(\Delta w_{2i}^{(t)}\right) = sign\left(w_{2i}^{(0)}\right)$ and that $\forall i, j \in [d] : \left|g_1^{(t)}[i,j]\right| \geq \tilde{\Omega}\left(\sqrt{\eta}\right), \left|g_{2i}^{(t)}\right| \geq \tilde{\Omega}\left(\sqrt{\eta}d\right)$. Moreover, we have $\forall \tau \leq t, \forall i, j \in [d] : \left|W_1^{(\tau)}[i,j]\right| \leq \tilde{\mathcal{O}}\left(\frac{1}{\sqrt{d}}\right), \left|w_{2i}^{(\tau)}\right| \leq \tilde{\mathcal{O}}\left(\frac{1}{\sqrt{d}}\right)$ and $\left|g_1^{(\tau)}[i,j]\right| \leq \tilde{\mathcal{O}}\left(\frac{1}{\sqrt{d}}\right), \left|g_{2i}^{(\tau)}\right| \leq \tilde{\mathcal{O}}\left(\sqrt{d}\right)$.*

The following lemma shows that when $t_{inc} \leq t < T_1$, we have $\forall i, j \in [d] : \left|g_{2i}^{(t)}\right| \gg \left|g_{2i}^{(t)} - g_{2i}^{(t-1)}\right|$ and that $\left|g_1^{(t)}[i,j]\right| \gg \left|g_1^{(t)}[i,j] - g_1^{(t-1)}[i,j]\right|$.

**Lemma 20.** *Under Assumption 1, 2 and 3, suppose $\sigma \leq \frac{\eta^{3/2}\xi^2}{d^{13/4}}$. Pick $\xi \leq \sqrt{\frac{\eta}{d^{3\alpha-1}}}, \eta \leq \mathcal{O}\left(\frac{1}{d^{3\alpha}}\right)$. For $t_{inc}$ in Lemma 18, we have that w.h.p. for $t_{inc} \leq t < T_1$ and $\tau \leq t, \forall i, j \in [d]$,*

$$\frac{\left|g_1^{(t)}[i,j] - g_1^{(t-\tau)}[i,j]\right|}{\left|g_1^{(t)}[i,j]\right|} = \tilde{\mathcal{O}}\left(\sqrt{\eta}\tau\right), \quad \frac{\left|g_{2i}^{(t)} - g_{2i}^{(t-\tau)}\right|}{\left|g_{2i}^{(t)}\right|} = \tilde{\mathcal{O}}\left(\sqrt{\eta}\tau\right), \tag{30}$$

$$\frac{\left|\left(g_1^{(t)}[i,j]\right)^2 - \left(g_1^{(t-\tau)}[i,j]\right)^2\right|}{\left(g_1^{(t)}[i,j]\right)^2} = \tilde{\mathcal{O}}\left(\sqrt{\eta}\tau\right) + \tilde{\mathcal{O}}\left(\eta\tau^2\right),$$

$$\frac{\left|\left(g_{2i}^{(t)}\right)^2 - \left(g_{2i}^{(t-\tau)}\right)^2\right|}{\left(g_{2i}^{(t)}\right)^2} = \tilde{\mathcal{O}}\left(\sqrt{\eta}\tau\right) + \tilde{\mathcal{O}}\left(\eta\tau^2\right). \tag{31}$$

Equipped with these lemmas, now let's prove eq. (29).

For any $t \in [t_{inc}, T_1)$, by Lemma 19, we know that $M_1^{(t)}, M_2^{(t)} \leq \tilde{\mathcal{O}}\left(\frac{1}{\sqrt{d}}\right)$, and that $G_1^{(t)} \leq \tilde{\mathcal{O}}\left(\frac{1}{\sqrt{d}}\right), G_2^{(t)} \leq \tilde{\mathcal{O}}\left(\sqrt{d}\right)$. At the end of the proof for this lemma, we will show that $T_1 = \Theta\left(\frac{1}{\sqrt{d}\eta}\right)$. Then we can pick $H := \frac{1}{1-\beta_1}\log\frac{d}{\eta\xi^2}$ and apply Lemma 17 and Corollary 2 to get that, w.h.p., for all $t \in [t_{inc}, T_1)$ and $\forall i, j \in [d]$, eq. (27) can be written as

$$\Delta W_1^{(t)}[i,j] = -\eta_t \frac{(1-\beta_1)\sum_{\tau=0}^{H}\beta_1^\tau g_1^{(t-\tau)}[i,j] + \epsilon_{1n}^{(t)}[i,j]}{\sqrt{(1-\beta_2)\sum_{\tau=0}^{H}\beta_2^\tau\left(g_1^{(t-\tau)}[i,j]\right)^2 + \epsilon_{1d}^{(t)}[i,j]} + \xi},$$

$$\Delta w_{2i}^{(t)} = -\eta_t \frac{(1-\beta_1)\sum_{\tau=0}^{H}\beta_1^\tau g_{2i}^{(t-\tau)} + \epsilon_{2n,i}^{(t)}}{\sqrt{(1-\beta_2)\sum_{\tau=0}^{H}\beta_2^\tau\left(g_{2i}^{(t-\tau)}\right)^2 + \epsilon_{2d,i}^{(t)}} + \xi}, \tag{32}$$

where $\forall i, j \in [d], \left|\epsilon_{1n}^{(t)}[i,j]\right|, \left|\epsilon_{1d}^{(t)}[i,j]\right|, \left|\epsilon_{2n,i}^{(t)}\right|, \left|\epsilon_{2d,i}^{(t)}\right| \leq \tilde{\mathcal{O}}(\eta\xi^2)$.

Let's first look at the update of $W_1^{(t)}[i,j]$. For $t$ in the first phase, we write the RHS of eq. (32) as

$$\frac{(1-\beta_1)\sum_{\tau=0}^{H}\beta_1^\tau g_1^{(t-\tau)}[i,j]+\epsilon_{1n}^{(t)}[i,j]}{\sqrt{(1-\beta_2)\sum_{\tau=0}^{H}\beta_2^\tau\left(g_1^{(t-\tau)}[i,j]\right)^2+\epsilon_{1d}^{(t)}[i,j]}+\xi}$$

$$=\frac{(1-\beta_1)g_1^{(t)}[i,j]\sum_{\tau=0}^{H}\beta_1^\tau+(1-\beta_1)\sum_{\tau=0}^{H}\beta_1^\tau\left(g_1^{(t-\tau)}[i,j]-g_1^{(t)}[i,j]\right)+\epsilon_{1n}^{(t)}[i,j]}{\sqrt{(1-\beta_2)\left(g_1^{(t)}[i,j]\right)^2\sum_{\tau=0}^{H}\beta_2^\tau+(1-\beta_2)\sum_{\tau=0}^{H}\beta_2^\tau\left(\left(g_1^{(t-\tau)}[i,j]\right)^2-\left(g_1^{(t)}[i,j]\right)^2\right)+\epsilon_{1d}^{(t)}[i,j]}+\xi}$$

$$:=\frac{g_1^{(t)}[i,j](1-\beta_1^{H+1})+e_{1n}^{(t)}[i,j]+\epsilon_{1n}^{(t)}[i,j]}{\sqrt{\left(g_1^{(t)}[i,j]\right)^2(1-\beta_2^{H+1})+e_{1d}^{(t)}[i,j]+\epsilon_{1d}^{(t)}[i,j]}+\xi},$$

where

$$e_{1n}^{(t)}[i,j]:=(1-\beta_1)\sum_{\tau=0}^{H}\beta_1^\tau\left(g_1^{(t-\tau)}[i,j]-g_1^{(t)}[i,j]\right),$$

$$e_{1d}^{(t)}[i,j]:=(1-\beta_2)\sum_{\tau=0}^{H}\beta_2^\tau\left(\left(g_1^{(t-\tau)}[i,j]\right)^2-\left(g_1^{(t)}[i,j]\right)^2\right).$$

We have already shown that $\left|\epsilon_{1n}^{(t)}[i,j]\right|,\left|\epsilon_{1d}^{(t)}[i,j]\right|\le\tilde{\mathcal{O}}(\eta\xi^2)$. By Lemma 20, we have that $\forall i,j\in[d]$,

$$\left|e_{1n}^{(t)}[i,j]\right|\le(1-\beta_1)\sum_{\tau=0}^{H}\beta_1^\tau\left|g_1^{(t-\tau)}[i,j]-g_1^{(t)}[i,j]\right|$$

$$\le\left|g_1^{(t)}[i,j]\right|\tilde{\mathcal{O}}\left(\sqrt{\eta}\right)(1-\beta_1)\sum_{\tau=0}^{H}\beta_1^\tau\tau=\left|g_1^{(t)}[i,j]\right|\tilde{\mathcal{O}}\left(\sqrt{\eta}\right).$$

Similarly, we have $\forall i,j\in[d]$,

$$\left|e_{1d}^{(t)}[i,j]\right|\le\left(g_1^{(t)}[i,j]\right)^2\tilde{\mathcal{O}}\left(\sqrt{\eta}\right)(1-\beta_2)\sum_{\tau=0}^{H}\beta_1^\tau\tau+\left(g_1^{(t)}[i,j]\right)^2\tilde{\mathcal{O}}\left(\sqrt{\eta}\right)(1-\beta_2)\sum_{\tau=0}^{H}\beta_1^\tau\tau^2$$

$$=\left(g_1^{(t)}[i,j]\right)^2\tilde{\mathcal{O}}\left(\sqrt{\eta}\right).$$

By Lemma 19, we know that $\left|g_1^{(t)}[i,j]\right|=\Omega\left(\sqrt{\eta}\right)$. Then we have that

$$\forall i,j\in[d]:\quad\left|\epsilon_{1n}^{(t)}[i,j]\right|\le\tilde{\mathcal{O}}(\eta\xi^2)\le\tilde{\mathcal{O}}\left(\sqrt{\eta}\right)\left|g_1^{(t)}[i,j]\right|,\quad\left|\epsilon_{1d}^{(t)}[i,j]\right|\le\tilde{\mathcal{O}}\left(\sqrt{\eta}\right)\xi^2.$$

Therefore by Lemma 34 in Appendix H, we have

$$\frac{g_1^{(t)}[i,j]\left(1-\beta_1^{H+1}\right)+e_{1n}^{(t)}[i,j]+\epsilon_{1n}^{(t)}[i,j]}{\sqrt{\left(g_1^{(t)}[i,j]\right)^2\left(1-\beta_2^{H+1}\right)+e_{1d}^{(t)}[i,j]+\epsilon_{1d}^{(t)}[i,j]}+\xi}$$

$$=\frac{1-\beta_1^{H+1}}{\sqrt{1-\beta_2^{H+1}}}\left(\mathrm{sign}\left(g_1^{(t)}[i,j]\right)+\tilde{e}_1^{(t)}[i,j]\right),$$

where $\left|\tilde{e}_1^{(t)}[i,j]\right|=\tilde{\mathcal{O}}\left(\sqrt{\eta}\right)$.

Since $\beta\in(0,1)$, we know that $\log\beta\le\beta-1<0$. Then our choice of $H$ gives us $H=\frac{1}{1-\beta_1}\log\frac{d}{\eta\xi^2}\ge\frac{\log\frac{\eta\xi^2}{d}}{\log\beta_1}$ and $H>\frac{1}{1-\beta_2}\log\frac{d}{\eta\xi^2}\ge\frac{\log\frac{\eta\xi^2}{d}}{\log\beta_2}$, which implies that $\beta_1^H,\beta_2^H\le\eta\xi^2/d$. Hence for $t\ge t_{\mathrm{inc}}>H$, $\eta_t\frac{1-\beta_1^{H+1}}{\sqrt{1-\beta_2^{H+1}}}=\eta\frac{\sqrt{1-\beta_2^{t+1}}}{\sqrt{1-\beta_2^{H+1}}}\frac{1-\beta_1^{H+1}}{1-\beta_1^{t+1}}=\eta(1\pm\mathcal{O}(\eta))$.

Combining all of the above yields that

$$
\begin{aligned}
W_1^{(t+1)}[i,j] &= W_1^{(t)}[i,j] - \eta_t \frac{(1-\beta_1)\sum_{\tau=0}^t \beta_1^\tau g_1^{(t-\tau)}[i,j]}{\sqrt{(1-\beta_2)\sum_{\tau=0}^t \beta_2^\tau \left(g_1^{(t-\tau)}[i,j]\right)^2 + \xi}} \\
&= W_1^{(t)}[i,j] - \eta_t \frac{1-\beta_1^{H+1}}{\sqrt{1-\beta_2^{H+1}}} \left(\operatorname{sign}\left(g_1^{(t)}[i,j]\right) + \tilde{e}_1^{(t)}[i,j]\right) \\
&= W_1^{(t)}[i,j] - \eta \left(\operatorname{sign}\left(g_1^{(t)}[i,j]\right) + e_1^{(t)}[i,j]\right),
\end{aligned}
$$

where $\left|e_1^{(t)}[i,j]\right| = \tilde{\mathcal{O}}\left(\sqrt{\eta}\right)$. The proof for $w_{2i}^{(t)}$ is similar.

So far we have successfully proved eq. (29). By $\operatorname{sign}\left(\Delta W_1^{(t)}[i,j]\right) = \operatorname{sign}\left(\Delta w_{2i}^{(t)}\right) = \operatorname{sign}\left(w_{2i}^{(0)}\right)$ in Lemma 19, we know that $\operatorname{sign}\left(-g_1^{(t)}[i,j]\right) = \operatorname{sign}\left(-g_{2i}^{(t)}\right) = \operatorname{sign}\left(w_{2i}^{(0)}\right)$, which gives us

$$
\begin{aligned}
\forall i,j \in [d]: \quad w_{2i}^{(t)} &= \operatorname{sign}\left(w_{2i}^{(0)}\right)\eta(t-t_{\text{inc}}) + R_{2i}^{(t)}, \\
W_1^{(t)}[i,j] &= \operatorname{sign}\left(w_{2i}^{(0)}\right)\eta(t-t_{\text{inc}}) + R_1^{(t)}[i,j],
\end{aligned}
$$

where $\frac{\left|R_1^{(t)}[i,j]\right|}{\eta(t-t_{\text{inc}})} = \tilde{\mathcal{O}}\left(\sqrt{\eta} + \frac{\left|W_1^{(t_{\text{inc}})}[i,j]\right|}{\eta(t-t_{\text{inc}})}\right)$ and $\frac{\left|R_{2i}^{(t)}\right|}{\eta(t-t_{\text{inc}})} = \tilde{\mathcal{O}}\left(\sqrt{\eta} + \frac{\left|w_{2i}^{(t_{\text{inc}})}\right|}{\eta(t-t_{\text{inc}})}\right)$. Now it suffices to show that $\forall i,j \in [d]: \left|w_{2i}^{(t_{\text{inc}})}\right| \le \mathcal{O}\left(\frac{1}{d^\alpha}\right), \left|W_1^{(t_{\text{inc}})}[i,j]\right| \le \mathcal{O}\left(\frac{1}{d^\alpha}\right)$, which is implied by Lemma 18.

Finally to complete the proof, we show that $T_1 = \Theta\left(\frac{1}{\sqrt{d}\eta}\right)$. When $t = T_1$, we have $\forall j \in [d]:$ $\sum_{i=1}^d w_{2i}^{(T_1)} W_1^{(T_1)}[i,j] = \Theta(1)$. Combining with the above results, we know that $d\eta^2(T_1-t_{\text{inc}})^2 = \Theta(1)$, i.e. $\eta(T_1-t_{\text{inc}}) = \Theta\left(\frac{1}{\sqrt{d}}\right)$. In Section E.5, we will prove $t_{\text{inc}} = \Theta\left(\frac{1}{\eta d^{\frac{3}{2}\alpha+1}}\right)$. Then we have $T_1 = \Theta\left(\frac{1}{\sqrt{d}\eta}\right)$.

### E.3 Proof of Lemma 17

For certain $t$ and $H$, we write eq. (27) as

$$
\begin{aligned}
\Delta W_1^{(t)}[i,j] &= -\eta_t \frac{(1-\beta_1)\sum_{\tau=0}^H \beta_1^\tau g_1^{(t-\tau)}[i,j] + \epsilon_{1n}^{(t)}[i,j]}{\sqrt{(1-\beta_2)\sum_{\tau=0}^H \beta_2^\tau \left(g_1^{(t-\tau)}[i,j]\right)^2 + \epsilon_{1d}^{(t)}[i,j] + \xi}}, \\
\Delta w_{2i}^{(t)} &= -\eta_t \frac{(1-\beta_1)\sum_{\tau=0}^H \beta_1^\tau g_{2i}^{(t-\tau)} + \epsilon_{2n,i}^{(t)}}{\sqrt{(1-\beta_2)\sum_{\tau=0}^H \beta_2^\tau \left(g_{2i}^{(t-\tau)}\right)^2 + \epsilon_{2d,i}^{(t)} + \xi}},
\end{aligned}
$$

where

$$\epsilon_{1n}^{(t)}[i,j] := \underbrace{(1-\beta_1)\sum_{\tau=H+1}^{t}\beta_1^{\tau}g_1^{(t-\tau)}[i,j]}_{:=q_{1n}^{(t)}[i,j]} + r_{1n}^{(t)}[i,j], \epsilon_{2n,i}^{(t)} := \underbrace{(1-\beta_1)\sum_{\tau=H+1}^{t}\beta_1^{\tau}g_{2i}^{(t-\tau)}}_{:=q_{2n,i}^{(t)}} + r_{2n,i}^{(t)},$$

$$\epsilon_{1d}^{(t)}[i,j] := \underbrace{(1-\beta_2)\sum_{\tau=H+1}^{t}\beta_2^{\tau}\left(g_1^{(t-\tau)}[i,j]\right)^2}_{:=q_{1d}^{(t)}[i,j]} + r_{1d}^{(t)}[i,j],$$

$$\epsilon_{2d,i}^{(t)} := \underbrace{(1-\beta_2)\sum_{\tau=H+1}^{t}\beta_2^{\tau}\left(g_{2i}^{(t-\tau)}\right)^2}_{:=q_{2d,i}^{(t)}} + r_{2d,i}^{(t)},$$

and $r_{1n}^{(t)}[i,j], r_{1d}^{(t)}[i,j], r_{2n,i}^{(t)}, r_{2d,i}^{(t)}$ are defined in eq. (28).

Since $\beta_2 = \beta_1^2 < \beta_1$, then if we pick $H \geq \frac{1}{1-\beta_1}\log\frac{\max\left\{G_1^{(t)},G_2^{(t)},\left(G_1^{(t)}\right)^2,\left(G_2^{(t)}\right)^2\right\}}{\eta\xi^2}$, we can get that $H \geq \frac{1}{1-\beta_1}\log\frac{G_1^{(t)}}{\eta\xi^2}, H \geq \frac{1}{1-\beta_2}\log\frac{\left(G_1^{(t)}\right)^2}{\eta\xi^2}, H \geq \frac{1}{1-\beta_1}\log\frac{G_2^{(t)}}{\eta\xi^2}, H \geq \frac{1}{1-\beta_2}\log\frac{\left(G_2^{(t)}\right)^2}{\eta\xi^2}$. Hence we can apply Lemma 33 in Appendix H to get that $\left|q_{1n}^{(t)}[i,j]\right|, \left|q_{1d}^{(t)}[i,j]\right|, \left|q_{2n,i}^{(t)}\right|, \left|q_{2d,i}^{(t)}\right| \leq \eta\xi^2$.

Pick $T$ in Lemma 32 as of order $\tilde{\mathcal{O}}\left(\frac{1}{\sqrt{d}\eta}\right)$. By Lemma 32, we have with probability at least $1 - \frac{1}{d}$, for all $t \leq T, \forall \tau \leq t$ and $\forall i,j \in [d]$,

$$\left|Dg_1^{(\tau)}[i,j]\right| = \left|\tilde{g}_1^{(\tau)}[i,j] - g_1^{(\tau)}[i,j]\right| \leq \tilde{\mathcal{O}}\left(d^3 M_1^{(t)}\left(M_2^{(t)}\right)^2\sigma\sqrt{\frac{d^{1/2}}{\eta}}\right) + \tilde{\mathcal{O}}\left(M_2^{(t)}\sigma\sqrt{\frac{d^{3/2}}{\eta}}\right)$$
$$:= D_1^{(t)},$$

$$\left|Dg_{2i}^{(\tau)}\right| = \left|\tilde{g}_{2i}^{(\tau)} - g_{2i}^{(\tau)}\right| \leq \tilde{\mathcal{O}}\left(d^4\left(M_1^{(t)}\right)^2 M_2^{(t)}\sigma\sqrt{\frac{d^{1/2}}{\eta}}\right) + \tilde{\mathcal{O}}\left(dM_1^{(t)}\sigma\sqrt{\frac{d^{3/2}}{\eta}}\right) := D_2^{(t)}.$$

Plugging into eq. (28) gives us

$$\left|r_{1n}^{(t)}[i,j]\right| \leq (1-\beta_1)\sum_{\tau=0}^{t}\beta_1^{t-\tau}\left|Dg_1^{(\tau)}[i,j]\right| \leq \mathcal{O}\left(D_1^{(t)}\right),$$

$$\left|r_{1d}^{(t)}[i,j]\right| \leq (1-\beta_2)\sum_{\tau=0}^{t}\beta_2^{t-\tau}\left|2g_1^{(\tau)}[i,j]Dg_1^{(\tau)}[i,j]\right| + \left|Dg_1^{(\tau)}[i,j]\right|^2$$
$$\leq \mathcal{O}\left(D_1^{(t)}G_1^{(t)} + \left(D_1^{(t)}\right)^2\right),$$

$$\left|r_{2n,i}^{(t)}\right| \leq (1-\beta_1)\sum_{\tau=0}^{t}\beta_1^{t-\tau}\left|Dg_{2i}^{(\tau)}\right| \leq \mathcal{O}\left(D_2^{(t)}\right),$$

$$\left|r_{2d,i}^{(t)}\right| \leq (1-\beta_2)\sum_{\tau=0}^{t}\beta_2^{t-\tau}\left|2g_{2i}^{(\tau)}Dg_{2i}^{(\tau)}\right| + \left|Dg_{2i}^{(\tau)}\right|^2 \leq \mathcal{O}\left(D_2^{(t)}G_2^{(t)} + \left(D_2^{(t)}\right)^2\right).$$

### E.4  Proof of Corollary 2

Since $G_1^{(t)} \leq \tilde{\mathcal{O}}\left(\frac{1}{\sqrt{d}}\right), G_2^{(t)} \leq \tilde{\mathcal{O}}\left(\sqrt{d}\right)$, then $H := \frac{1}{1-\beta_1}\log\frac{d}{\eta\xi^2}$ is bigger than $\frac{1}{1-\beta_1}\log\frac{\max\left\{G_1^{(t)},G_2^{(t)},\left(G_1^{(t)}\right)^2,\left(G_2^{(t)}\right)^2\right\}}{\eta\xi^2}$.

By $M_1^{(t)}, M_2^{(t)} \le \tilde{\mathcal{O}}\left(\frac{1}{\sqrt{d}}\right), G_1^{(t)} \le \tilde{\mathcal{O}}\left(\frac{1}{\sqrt{d}}\right), G_2^{(t)} \le \tilde{\mathcal{O}}\left(\sqrt{d}\right)$ and the assumption $\sigma \le \frac{\eta^{3/2}\xi^2}{d^{13/4}}$, we get that $D_1^{(t)}$ and $D_2^{(t)}$ are upper bounded by $D_1^{(t)} \le \tilde{\mathcal{O}}\left(d^{7/4}\sigma\eta^{-1/2}\right)$ and $D_2^{(t)} \le \tilde{\mathcal{O}}\left(d^{11/4}\sigma\eta^{-1/2}\right)$, which yields $\forall i, j \in [d], \left|\epsilon_{1n}^{(t)}[i,j]\right|, \left|\epsilon_{1d}^{(t)}[i,j]\right|, \left|\epsilon_{2n,i}^{(t)}\right|, \left|\epsilon_{2d,i}^{(t)}\right| \le \tilde{\mathcal{O}}(\eta\xi^2)$.

## E.5 Proof of Lemma 18

The proof is based on the following two lemmas.

**Lemma 21.** *Under Assumption 1 and 2, we have that w.p. at least $1 - \frac{1}{d^{\frac{\alpha}{2}-1}}$, for every $1 \le i \le d$,*
$$\frac{\sqrt{\pi}}{d^{\frac{3}{2}\alpha}} \le \left|w_{2i}^{(0)}\right| \le \sqrt{\frac{2}{d^{2\alpha}}\log\frac{2d}{\delta}}, \text{ and that w.p. at least } 1 - \delta \text{ for any given } \delta > 0, \left|W_1^{(0)}[i,j]\right| \le \sqrt{\frac{2}{d^{4\alpha}}\log\frac{2d^2}{\delta}}.$$

**Lemma 22.** *Under Assumption 1, 2 and 3, suppose $\sigma \le \frac{\eta^{3/2}\xi^2}{d^{13/4}}$. Pick $\beta_2 = \beta_1^2, \xi \in (0,1), \eta < \frac{1}{4}$. Consider any time point $t \le \tilde{\mathcal{O}}\left(\frac{1}{\sqrt{d}\eta}\right)$. If $\forall \tau \le t, \forall i, j \in [d] : \left|W_1^{(\tau)}[i,j]\right| \le \tilde{\mathcal{O}}\left(\frac{1}{\sqrt{d}}\right), \left|w_{2i}^{(\tau)}\right| \le \tilde{\mathcal{O}}\left(\frac{1}{\sqrt{d}}\right)$ and $\left|g_1^{(\tau)}[i,j]\right| \le \tilde{\mathcal{O}}\left(\frac{1}{\sqrt{d}}\right), \left|g_{2i}^{(\tau)}\right| \le \tilde{\mathcal{O}}\left(\sqrt{d}\right)$, we will have*
$$\left|\Delta W_1^{(t)}[i,j]\right| \le \tilde{\mathcal{O}}(\eta) \quad \left|\Delta w_{2i}^{(t)}\right| \le \tilde{\mathcal{O}}(\eta),$$

*where the $\tilde{\mathcal{O}}$ notation depends on $H = \frac{1}{1-\beta_1}\log\frac{d}{\eta\xi^2}$.*

*Furthermore, if for certain $i, j \in [d]$, Condition 1 (resp. Condition 2) is satisfied, we will have*
$$sign\left(\Delta W_1^{(t)}[i,j]\right) = -s_1^{(t)}[i,j], \left|\Delta W_1^{(t)}[i,j]\right| = \tilde{\Theta}(\eta)$$
$$\left(resp. \ sign\left(\Delta w_{2i}^{(t)}\right) = -s_{2i}^{(t)}, \left|\Delta w_{2i}^{(t)}\right| = \tilde{\Theta}(\eta)\right).$$

Now we prove Lemma 18. Define $t_d := \inf\left\{t : \exists i, j : \left|W_1^{(t)}[i,j]\right| > \frac{1}{d} \text{ or } \left|w_{2i}^{(t)}\right| > \frac{1}{d}\right\}$. Now we want to find a time point $t_{\text{inc}}$ before $t_d$ for the lemma to hold. During the period $t < t_d$, we have $\forall j \in [d], E_j = -\Theta(1)$ (which means $t_d < T_1$) and therefore for all $i, j \in [d], \left|g_1^{(t)}[i,j]\right| \le \frac{1}{d}$ and $\left|g_{2i}^{(t)}\right| \le 1$. Then we can use Lemma 22 to get that for $t \le \min\left\{t_d, \frac{1}{\sqrt{d}\eta}\right\}$, we have $\left|\Delta W_1^{(t)}[i,j]\right| \le \tilde{\mathcal{O}}(\eta), \left|\Delta w_{2i}^{(t)}\right| \le \tilde{\mathcal{O}}(\eta)$. Hence $t_d \ge \tilde{\Omega}\left(\frac{1}{\eta d}\right)$.

Define $t_{\text{sign}} = \inf\left\{t < \min\left\{t_d, \frac{1}{\sqrt{d}\eta}\right\} : \exists i \in [d] : \left|w_{2i}^{(t)}\right| \le \frac{1}{d^{\frac{3}{2}\alpha}}\right\}$. By Lemma 21, w.h.p. $\forall i \in [d] : \left|w_{2i}^{(0)}\right| \ge \frac{\sqrt{\pi}}{d^{\frac{3}{2}\alpha}}$, combining with $\left|\Delta w_{2i}^{(t)}\right| \le \tilde{\mathcal{O}}(\eta)$ gives us that w.h.p., $t_{\text{sign}} \ge \frac{\sqrt{\pi}-1}{d^{\frac{3}{2}\alpha}}/\tilde{\mathcal{O}}(\eta) = \tilde{\Omega}\left(\frac{1}{\eta d^{\frac{3}{2}\alpha}}\right)$.

Now let's analyze the behavior of $W_1$ during the period $t < t_{\text{sign}}$. Consider any $i, j \in [d]$. By definition, $sign\left(w_{2i}^{(t)}\right) = sign\left(w_{2i}^{(0)}\right)$. Note that $E_j^{(t)} = -\Theta(1)$, then we have $sign\left(g_1^{(t)}[i,j]\right) = -sign\left(w_{2i}^{(0)}\right)$ and that $\left|g_1^{(t)}[i,j]\right| = \Omega\left(\frac{1}{d^{\frac{3}{2}\alpha}}\right) = \Omega(\xi)$ by our choice of $\xi$. Then we know that Condition 1 is satisfied with $s_1^{(t)}[i,j] = -sign\left(w_{2i}^{(0)}\right)$ (for all $H < t \le t_{\text{sign}}$), which by Lemma 22 yields $sign\left(\Delta W_1^{(t)}[i,j]\right) = sign\left(w_{2i}^{(0)}\right)$ and $\left|\Delta W_1^{(t)}[i,j]\right| = \tilde{\Theta}(\eta)$.

Lemma 21 tells us that w.h.p., $\forall i, j \in [d] : \left|W_1^{(0)}[i,j]\right| = \tilde{\mathcal{O}}\left(\frac{1}{d^{2\alpha}}\right)$. For any $i, j$, if initially $sign\left(W_1^{(0)}[i,j]\right) = sign\left(w_{2i}^{(0)}\right)$, then for the following steps before $t_{\text{sign}}$, we will have $sign\left(W_1^{(t)}[i,j]\right) = sign\left(w_{2i}^{(0)}\right)$. If initially $sign\left(W_1^{(0)}[i,j]\right) \ne sign\left(w_{2i}^{(0)}\right)$, then after at most $t_0 = \tilde{\mathcal{O}}\left(\frac{1}{\eta d^{2\alpha}}\right)$ steps, $W_1[i,j]$ will flip the sign. Note that $t_0 = \tilde{\mathcal{O}}\left(\frac{1}{\eta d^{2\alpha}}\right)$ is smaller than $t_{\text{sign}}$.

Hence we have shown that at some time point $t_0$, we have $\forall i, j \in [d] : \text{sign}\left(W_1^{(t)}[i,j]\right) = \text{sign}\left(w_{2i}^{(t)}\right) = \text{sign}\left(w_{2i}^{(0)}\right)$. Now we analyze the period $t \geq t_0$.

When $t_0 < t \leq t_{\text{sign}}$, we still have $\text{sign}\left(\Delta W_1^{(t)}[i,j]\right) = \text{sign}\left(w_{2i}^{(0)}\right)$ and $\left|\Delta W_1^{(t)}[i,j]\right| = \tilde{\Theta}(\eta)$. Combining these two with the fact $\text{sign}\left(W_1^{(t_0)}[i,j]\right) = \text{sign}\left(w_{2i}^{(0)}\right)$, we know that for all $t \in [t_0, t_{\text{sign}}]$, $\text{sign}\left(W_1^{(t)}[i,j]\right) = \text{sign}\left(w_{2i}^{(0)}\right)$ and that $\forall i, j \in [d] : \left|W_1^{(t+1)}[i,j]\right| = \left|W_1^{(t)}[i,j]\right| + \tilde{\Theta}(\eta)$. Then at certain step $t_{\text{inc}}$ which satisfies $t_{\text{inc}} = t_0 + \tilde{\Theta}\left(\frac{1}{\eta d^{\frac{3}{2}\alpha+1}}\right) \in (H, t_{\text{sign}})$, we will have $\forall t_{\text{inc}} - H \leq \tau \leq t_{\text{inc}}, \forall i, j \in [d] : \left|W_1^{(\tau)}[i,j]\right| = \Theta\left(\frac{1}{d^{\frac{3}{2}\alpha+1}}\right)$ and therefore $\left|g_{2i}^{(\tau)}\right| = \left|\sum_{j=1}^d W_1^{(\tau)}[i,j]E_j^{(\tau)}\right| = \sum_{j=1}^d \left|W_1^{(\tau)}[i,j]E_j^{(\tau)}\right| = \Theta\left(\frac{1}{d^{\frac{3}{2}\alpha}}\right) = \Omega(\xi)$. For $t \leq t_{\text{inc}}$, we have $\forall i, j \in [d] : \left|W_1^{(t)}[i,j]\right| = \mathcal{O}\left(\frac{1}{d^{\frac{3}{2}\alpha+1}}\right)$.

Since $t_{\text{inc}} < t_{\text{sign}}$, we have $\left|w_{2i}^{t_{\text{inc}}}\right| = \Omega\left(\frac{1}{d^{\frac{3}{2}\alpha}}\right)$. For $t \leq t_{\text{inc}}$, note that $\left|\Delta w_{2i}^{(t)}\right| \leq \tilde{\mathcal{O}}(\eta)$, $t_{\text{inc}} = t_0 + \tilde{\Theta}\left(\frac{1}{\eta d^{\frac{3}{2}\alpha+1}}\right) = \tilde{\Theta}\left(\frac{1}{\eta d^{\frac{3}{2}\alpha+1}}\right)$, combining with the upper bound in Lemma 21 yields

$$\left|w_{2i}^{(t)}\right| \leq \left|w_{2i}^{(0)}\right| + t_{\text{inc}}\tilde{\mathcal{O}}(\eta) \leq \tilde{\mathcal{O}}\left(\frac{1}{d^{\frac{3}{2}\alpha+1}}\right) \leq \mathcal{O}\left(\frac{1}{d^\alpha}\right).$$

Moreover, $\forall t_{\text{inc}} - H \leq \tau \leq t_{\text{inc}}, \forall i \in [d] : \text{sign}\left(g_{2i}^{(\tau)}\right) = -\text{sign}\left(w_{2i}^{(0)}\right)$. Then Condition 2 is satisfied with $s_{2i}^{(t)} = -\text{sign}\left(w_{2i}^{(0)}\right)$ for $t = t_{\text{inc}}$. In the analysis of $g_1^{(t)}[i,j]$, we have already shown that for all $t \leq t_{\text{sign}}$ (and thus for $t = t_{\text{inc}}$), Condition 1 is satisfied, which completes the proof.

### E.6 Proof of Lemma 21

Since for $X \sim \mathcal{N}\left(0, \sigma^2\right)$, we have that $P(|X| \leq t) \leq \frac{2t}{\sqrt{2\pi}\sigma}$, then for a fixed $i$,

$$P\left(\left|w_{2i}^{(0)}\right| \leq \frac{\sqrt{\pi}}{d^{\frac{3}{2}\alpha}}\right) \leq \frac{2\sqrt{\pi}/d^{\frac{3}{2}\alpha}}{\sqrt{2\pi} \cdot \sqrt{2/d^{2\alpha}}} = \frac{1}{d^{\frac{\alpha}{2}}}.$$

Then by union bound, we have that w.p. at least $1 - \frac{1}{d^{\frac{\alpha}{2}-1}}$, for every $1 \leq i \leq d$, $\left|w_{2i}^{(0)}\right| \geq \frac{\sqrt{\pi}}{d^{\frac{3}{2}\alpha}}$.

As for the upper bounds, using the Gaussian tail bound and union bound, we have w.p. at least $1 - \delta$,

$$\forall i, j \in [d] : \quad \left|w_{2i}^{(0)}\right| \leq \sqrt{\frac{2}{d^{2\alpha}} \log \frac{2d}{\delta}}, \quad \left|W_1^{(0)}[i,j]\right| \leq \sqrt{\frac{2}{d^{4\alpha}} \log \frac{2d^2}{\delta}}.$$

### E.7 Proof of Lemma 22

Now we analyze the magnitude order of $\Delta W_1^{(t)}[i,j]$. The analysis of $\Delta w_{2i}^{(t)}$ is similar.

For $t \leq \tilde{\mathcal{O}}\left(\frac{1}{\sqrt{d}\eta}\right)$. By assumption, $M_1^{(t)}, M_2^{(t)} \leq \tilde{\mathcal{O}}\left(\frac{1}{\sqrt{d}}\right)$, $G_1^{(t)} \leq \tilde{\mathcal{O}}\left(\frac{1}{\sqrt{d}}\right)$, $G_2^{(t)} \leq \tilde{\mathcal{O}}\left(\sqrt{d}\right)$, and $\sigma \leq \frac{\eta^{3/2}\xi^2}{d^{13/4}}$. Hence we can pick $H := \frac{1}{1-\beta_1} \log \frac{d}{\eta\xi^2}$ and apply Lemma 17 and Corollary 2 to get that, w.h.p., for all $t \leq \tilde{\mathcal{O}}\left(\frac{1}{\sqrt{d}\eta}\right)$ and $\forall i, j \in [d]$, eq. (27) can be written as

$$\Delta W_1^{(t)}[i,j] = -\eta_t \frac{(1-\beta_1)\sum_{\tau=0}^H \beta_1^\tau g_1^{(t-\tau)}[i,j] + \epsilon_{1n}^{(t)}[i,j]}{\sqrt{(1-\beta_2)\sum_{\tau=0}^H \beta_2^\tau \left(g_1^{(t-\tau)}[i,j]\right)^2 + \epsilon_{1d}^{(t)}[i,j] + \xi}},$$

$$\Delta w_{2i}^{(t)} = -\eta_t \frac{(1-\beta_1)\sum_{\tau=0}^H \beta_1^\tau g_{2i}^{(t-\tau)} + \epsilon_{2n,i}^{(t)}}{\sqrt{(1-\beta_2)\sum_{\tau=0}^H \beta_2^\tau \left(g_{2i}^{(t-\tau)}\right)^2 + \epsilon_{2d,i}^{(t)} + \xi}},$$

(33)

where $\forall i, j \in [d]$, $\left|\epsilon_{1n}^{(t)}[i,j]\right|, \left|\epsilon_{1d}^{(t)}[i,j]\right|, \left|\epsilon_{2n,i}^{(t)}\right|, \left|\epsilon_{2d,i}^{(t)}\right| \leq \tilde{\mathcal{O}}(\eta\xi^2)$.

On one hand, using $\left|\epsilon_{1n}^{(t)}[i,j]\right|, \left|\epsilon_{1d}^{(t)}[i,j]\right| \leq \tilde{\mathcal{O}}(\eta\xi^2)$ and $\beta_2 = \beta_1^2$, and $\sqrt{x+y} \geq \sqrt{x} - \sqrt{|y|}$ when $x \geq 0, x+y \geq 0$, we get from eq. (33) that

$$\left|\Delta W_1^{(t)}[i,j]\right| \leq \eta_t \frac{(1-\beta_1)\left|\sum_{\tau=0}^H \beta_1^\tau g_1^{(t-\tau)}[i,j]\right| + \tilde{\mathcal{O}}(\eta\xi^2)}{\sqrt{(1-\beta_2)\sum_{\tau=0}^H \left(\beta_1^\tau g_1^{(t-\tau)}[i,j]\right)^2 - \tilde{\mathcal{O}}(\sqrt{\eta}\xi) + \xi}}$$

$$\overset{(i)}{\leq} \eta_t \frac{(1-\beta_1)\sqrt{H+1}\sqrt{\sum_{\tau=0}^H \left(\beta_1^\tau g_1^{(t-\tau)}[i,j]\right)^2} + \tilde{\mathcal{O}}(\eta\xi^2)}{\sqrt{(1-\beta_2)\sum_{\tau=0}^H \left(\beta_1^\tau g_1^{(t-\tau)}[i,j]\right)^2 + \xi/2}} \leq \mathcal{O}\left(\sqrt{H}\eta\right) = \tilde{\mathcal{O}}(\eta),$$

where $(i)$ uses Cauchy-Schwarz inequality for the numerator.

On the other hand, when $\text{sign}\left(g_1^{(t-H)}[i,j]\right) = \text{sign}\left(g_1^{(t-H+1)}[i,j]\right) = \ldots = \text{sign}\left(g_1^{(t)}[i,j]\right) = s_1^{(t)}[i,j]$, we have

$$\text{sign}\left(\sum_{\tau=0}^H \beta_1^\tau g_1^{(t-\tau)}[i,j]\right) = s_1^{(t)}[i,j], \quad \left|\sum_{\tau=0}^H \beta_1^\tau g_1^{(t-\tau)}[i,j]\right| \geq \sqrt{\sum_{\tau=0}^H \left(\beta_1^\tau g_1^{(t-\tau)}[i,j]\right)^2}.$$

If we further have $(1-\beta_1)\left|\sum_{\tau=0}^H \beta_1^{(\tau)} g_1^{(t-\tau)}[i,j]\right| \geq \Omega(\xi)$, then combining with $\left|\epsilon_{1n}^{(t)}[i,j]\right| \leq \tilde{\mathcal{O}}(\eta\xi^2) < \xi$ we will get

$$\text{sign}\left(\Delta W_1^{(t)}[i,j]\right) = -\text{sign}\left(\sum_{\tau=0}^H \beta_1^\tau g_1^{(t-\tau)}[i,j] + \epsilon_{1n}^{(t)}[i,j]\right) = -\text{sign}\left(\sum_{\tau=0}^H \beta_1^\tau g_1^{(t-\tau)}[i,j]\right)$$

$$= -s_1^{(t)}[i,j].$$

Using $\sqrt{x+y} \leq \sqrt{|x|} + \sqrt{|y|}$, we obtain that

$$\left|\Delta W_1^{(t)}[i,j]\right| \geq \eta_t \frac{(1-\beta_1)\left|\sum_{\tau=0}^H \beta_1^{(\tau)} g_1^{(t-\tau)}[i,j]\right| - \tilde{\mathcal{O}}(\eta\xi^2)}{\sqrt{(1-\beta_2)\sum_{\tau=0}^H \left(\beta_1^\tau g_1^{(t-\tau)}[i,j]\right)^2 + \tilde{\mathcal{O}}(\sqrt{\eta}\xi) + \xi}}$$

$$\geq \eta_t \frac{\frac{1-\beta_1}{2}\left|\sum_{\tau=0}^H \beta_1^{(\tau)} g_1^{(t-\tau)}[i,j]\right|}{2\max\left\{\sqrt{(1-\beta_2)\sum_{\tau=0}^H \left(\beta_1^\tau g_1^{(t-\tau)}[i,j]\right)^2}, \frac{3}{2}\xi\right\}} = \Omega(\eta).$$

Together with the upper bound completes the proof.

## E.8 Proof of Lemma 19

The proof is based on the following lemma, which gives a coarse analysis on the magnitude of weights and their increments per step during the first phase.

**Lemma 23.** *Under Assumption 1, 2 and 3, suppose $\sigma \leq \frac{\eta^{3/2}\xi^2}{d^{13/4}}$. Pick $\xi \leq \min\left\{\sqrt{\frac{\eta}{d^{3\alpha-1}}}, \frac{1}{d^{\frac{3}{2}\alpha}}\right\}$, for $t_{inc}$ in Lemma 18, we have that w.h.p. for all $t_{inc} \leq t \leq T_1, \forall i, j \in [d]$.*

$$\text{sign}\left(\Delta W_1^{(t)}[i,j]\right) = \text{sign}\left(\Delta w_{2i}^{(t)}\right) = \text{sign}\left(w_{2i}^{(0)}\right), \quad \left|\Delta W_1^{(t)}[i,j]\right| = \tilde{\Theta}(\eta), \left|\Delta w_{2i}^{(t)}\right| = \tilde{\Theta}(\eta),$$

$$\text{sign}\left(W_1^{(t)}[i,j]\right) = \text{sign}\left(w_{2i}^{(t)}\right) = \text{sign}\left(w_{2i}^{(0)}\right), \quad \left|W_1^{(t)}[i,j]\right| = \tilde{\mathcal{O}}\left(\frac{1}{\sqrt{d}}\right), \left|w_{2i}^{(t)}\right| = \tilde{\mathcal{O}}\left(\frac{1}{\sqrt{d}}\right).$$

*Specially, at the end of the first phase ($t = T_1$), we have $\forall i, j \in [d]$, $\left|w_{2i}^{(T_1)}\right| = \tilde{\Theta}\left(\frac{1}{\sqrt{d}}\right)$ and $\left|W_1^{(T_1)}[i,j]\right| = \tilde{\Theta}\left(\frac{1}{\sqrt{d}}\right)$.*

Now we go back to the proof of Lemma 19. For $t_{\text{inc}} \leq t < T_1$, since $E_j^{(t)} = \left(W_2^{(t)} W_1^{(t)}\right)_j - A_j = \sum_{i=1}^{d} w_{2i}^{(t)} W_1^{(t)}[i,j] - A_j$, we have,

$$\Delta E_j^{(t)} := E_j^{(t+1)} - E_j^{(t)}$$
$$= \sum_{i=1}^{d} \left(w_{2i}^{(t+1)} W_1^{(t+1)}[i,j] - w_{2i}^{(t+1)} W_1^{(t)}[i,j] + w_{2i}^{(t+1)} W_1^{(t)}[i,j] - w_{2i}^{(t)} W_1^{(t)}[i,j]\right) \quad (34)$$
$$= \sum_{i=1}^{d} \left(w_{2i}^{(t+1)} \Delta W_1^{(t)}[i,j] + \Delta w_{2i}^{(t)} W_1^{(t)}[i,j]\right).$$

Combining Lemma 23 and eq. (34) gives us $\forall j \in [d]$,

$$\Delta E_j^{(t)} > 0, \quad \left|\Delta E_j^{(t)}\right| = \sum_{i=1}^{d} \left|w_{2i}^{(t+1)} \Delta W_1^{(t)}[i,j]\right| + \left|\Delta w_{2i}^{(t)} W_1^{(t)}[i,j]\right|$$
$$\leq \sum_{i=1}^{d} \tilde{\mathcal{O}}\left(\eta \frac{1}{\sqrt{d}}\right) = \tilde{\mathcal{O}}\left(\eta \sqrt{d}\right). \quad (35)$$

Let's first analyze $g_1^{(t)}[i,j]$. Note that

$$\Delta g_1^{(t)}[i,j] = w_{2i}^{(t+1)} E_j^{(t+1)} - w_{2i}^{(t+1)} E_j^{(t)} + w_{2i}^{(t+1)} E_j^{(t)} - w_{2i}^{(t)} E_j^{(t)}$$
$$= w_{2i}^{(t+1)} \Delta E_j^{(t)} + \Delta w_{2i}^{(t)} E_j^{(t)}, \quad (36)$$

where $\text{sign}\left(w_{2i}^{(t+1)} \Delta E_j^{(t)}\right) = \text{sign}\left(w_{2i}^{(0)}\right)$ while $\text{sign}\left(\Delta w_{2i}^{(t)} E_j^{(t)}\right) = -\text{sign}\left(w_{2i}^{(0)}\right)$.

Now we analyze the sign of $g_1^{(t)}[i,j]$ when $t_{\text{inc}} \leq t < T_1$. Using $\left|w_{2i}^{(t_{\text{inc}}+1)}\right| = \tilde{\mathcal{O}}\left(\frac{1}{d^\alpha}\right)$ and eq. (35), we get that $\left|w_{2i}^{(t_{\text{inc}}+1)} \Delta E_j^{(t_{\text{inc}})}\right| \leq \tilde{\mathcal{O}}\left(\frac{1}{d^\alpha} \cdot \sqrt{d}\eta\right)$. While on the other hand, $\left|\Delta w_{2i}^{(t_{\text{inc}})} E_j^{(t_{\text{inc}})}\right| = \tilde{\Theta}(\eta)$. That means $\text{sign}\left(\Delta g_1^{(t_{\text{inc}})}[i,j]\right) = -\text{sign}\left(w_{2i}^{(0)}\right)$. Note that $\text{sign}\left(g_1^{(t_{\text{inc}})}[i,j]\right) = -\text{sign}\left(w_{2i}^{(t_{\text{inc}})}\right) = -\text{sign}\left(w_{2i}^{(0)}\right)$, we know that $\left|g_1^{(t)}[i,j]\right|$ will increase when $t = t_{\text{inc}}$.

In the following steps, $\left|g_1^{(t)}[i,j]\right|$ will keep increasing as long as $\left|\Delta w_{2i}^{(t)} E_j^{(t)}\right| > \left|w_{2i}^{(t+1)} \Delta E_j^{(t)}\right|$. Since $\left|W_1^{(t)}[i,j]\right|, \left|w_{2i}^{(t)}\right|$ keep increasing while $\left|\Delta W_1^{(t)}[i,j]\right|, \left|\Delta w_{2i}^{(t)}\right|$ remain $\tilde{\Theta}(\eta)$, by eq. (35), we know that the trend of $\left|\Delta E_j^{(t)}\right|$ is to increase. On the other hand, $\left|E_j^{(t)}\right|$ keeps decreasing since $E_j^{(t)} < 0$ while $\Delta E_j^{(t)} > 0$. Then after some time point we will have $\left|\Delta w_{2i}^{(t)} E_j^{(t)}\right| < \left|w_{2i}^{(t+1)} \Delta E_j^{(t)}\right|$ and in the following steps $\left|g_1^{(t)}[i,j]\right|$ will have the trend to decrease. Specially, when $t = T_1 - 1$, we have $\left|E_j^{(t)}\right| = \Theta\left(\sqrt{\eta d}\right)$ and $\left|W_1^{(t)}\right| = \tilde{\Theta}\left(\frac{1}{\sqrt{d}}\right), \left|w_{2i}^{(t+1)}\right| = \tilde{\Theta}\left(\frac{1}{\sqrt{d}}\right)$ by Lemma 23, which gives us

$$\left|\Delta E_j^{(t)}\right| = \sum_{i=1}^{d} \left|w_{2i}^{(t+1)} \Delta W_1^{(t)}[i,j]\right| + \left|\Delta w_{2i}^{(t)} W_1^{(t)}[i,j]\right| \leq \sum_{i=1}^{d} \tilde{\Theta}\left(\eta \frac{1}{\sqrt{d}}\right) = \tilde{\Theta}\left(\eta \sqrt{d}\right).$$

Hence $\left|w_{2i}^{(t+1)} \Delta E_j^{(t)}\right| = \tilde{\Theta}(\eta) > \left|\Delta w_{2i}^{(t)} E_j^{(t)}\right| = \tilde{\Theta}\left(\eta \sqrt{\eta d}\right).$

Therefore we have proved that when $t_{\text{inc}} \leq t < T_1$, the trend of $\left|g_1^{(t)}[i,j]\right|$ is to first increase and then decrease. In order to prove $\left|g_1^{(t)}[i,j]\right| = \tilde{\Omega}\left(\sqrt{\eta}\right)$, it suffices to show that $\left|g_1^{(t_{\text{inc}})}[i,j]\right| = \tilde{\Omega}\left(\sqrt{\eta}\right)$ and $\left|g_1^{(T_1)}[i,j]\right| = \tilde{\Omega}\left(\sqrt{\eta}\right)$.

When $t = t_{\text{inc}}$,

$$\left|g_1^{(t_{\text{inc}})}[i,j]\right| = \left|w_{2i}^{(t_{\text{inc}})}\right| \cdot \left|E_j^{(t_{\text{inc}})}\right| = \Omega\left(\frac{1}{d^{\frac{3}{2}\alpha}}\right) \cdot \Theta(1) = \Omega\left(\sqrt{\eta}\right).$$

When $t = T_1$, we have

$$\left|g_1^{(T_1)}[i,j]\right| = \left|w_{2i}^{(T_1)}\right| \cdot \left|E_j^{(t_1)}\right| = \tilde{\Theta}\left(\frac{1}{\sqrt{d}} \cdot \sqrt{\eta d}\right) = \tilde{\Theta}\left(\sqrt{\eta}\right).$$

As for $g_{2i}^{(t)}$, since for $\forall i \in [d]$, $W_1^{(t)}[i,j]$ for different $j$ have the same sign. Combining with $\forall j \in [d] : E_j^{(t)} < 0$ gives us

$$\left|g_{2i}^{(t)}\right| = \left|\sum_{j=1}^d E_j^{(t)} W_1^{(t)}[i,j]\right| = \sum_{j=1}^d \left|E_j^{(t)} W_1^{(t)}[i,j]\right|.$$

Then it suffices to show that for $t_{\mathrm{inc}} \le t < T_1$, $\left|E_j^{(t)} W_1^{(t)}[i,j]\right| = \tilde{\Omega}\left(\sqrt{\eta}\right)$, which can be proven using the same technique as above.

Finally, for $\forall \tau \le t, \forall i,j \in [d]$, note that the upper bounds of $\left|W_1^{(\tau)}[i,j]\right|$ and $\left|w_{2i}^{(\tau)}\right|$ are already given in Lemma 23. As for $\left|g_1^{(\tau)}[i,j]\right|$ and $\left|g_{2i}^{(\tau)}\right|$, we have $\left|g_1^{(\tau)}[i,j]\right| = \left|w_{2i}^{(\tau)} E_j^{(\tau)}\right| = \tilde{\mathcal{O}}\left(\frac{1}{\sqrt{d}}\right), \left|g_{2i}^{(\tau)}\right| \le \sum_{j=1}^d \left|E_j^{(\tau)} W_1^{(\tau)}[i,j]\right| = \tilde{\mathcal{O}}\left(\sqrt{d}\right).$

### E.9   Proof of Lemma 23

For any $i,j \in [d]$, and any $t$ in the interval $[t_{\mathrm{inc}}, T_1]$, we prove by induction that

(A) $\left|W_1^{(t)}[i,j]\right| = \tilde{\mathcal{O}}\left(\frac{1}{\sqrt{d}}\right), \left|w_{2i}^{(t)}\right| = \tilde{\mathcal{O}}\left(\frac{1}{\sqrt{d}}\right).$

(B) $\forall \tau \in [t-H, t] : \mathrm{sign}\left(W_1^{(\tau)}[i,j]\right) = \mathrm{sign}\left(w_{2i}^{(\tau)}\right) = \mathrm{sign}\left(w_{2i}^{(0)}\right).$

(C) $\left|g_1^{(t)}[i,j]\right| \ge \Omega(\xi), \left|g_{2i}^{(t)}\right| \ge \Omega(\xi).$

The base case $t = t_{\mathrm{inc}}$ was already proven by Lemma 18.

For $t \in [t_{\mathrm{inc}}, T_1)$, suppose (B) and (C) hold for time $t$ and (A) holds for all $\tau \in [t_{\mathrm{inc}}, t]$. From (A), we get that $\forall \tau \in [t_{\mathrm{inc}}, t] : \left|g_1^{(\tau)}[i,j]\right| = \left|w_{2i}^{(\tau)} E_j^{(\tau)}\right| = \tilde{\mathcal{O}}\left(\frac{1}{\sqrt{d}}\right), \left|g_{2i}^{(\tau)}\right| \le \sum_{j=1}^d \left|E_j^{(\tau)} W_1^{(\tau)}[i,j]\right| = \tilde{\mathcal{O}}\left(\sqrt{d}\right).$ Since when $t < T_1, \forall j \in [d] : E_j^{(t)} < 0$, from (B) we know that $\forall \tau \in [t-H, t] : \mathrm{sign}\left(g_1^{(\tau)}[i,j]\right) = \mathrm{sign}\left(g_{2i}^{(\tau)}\right) = -\mathrm{sign}\left(w_{2i}^{(0)}\right).$ Combining with (C) tells us that Condition 1 and 2 are satisfied.

In Section E.2 we have shown that $T_1 = \Theta\left(\frac{1}{\sqrt{d}\eta}\right)$. Then for $t \in [t_{\mathrm{inc}}, T_1)$, we can use Lemma 22 to get that $\forall t_{\mathrm{inc}} \le \tau \le t, \forall i,j \in [d]$,

$$\mathrm{sign}\left(\Delta W_1^{(\tau)}[i,j]\right) = \mathrm{sign}\left(\Delta w_{2i}^{(\tau)}\right) = \mathrm{sign}\left(w_{2i}^{(0)}\right), \quad \left|\Delta W_1^{(\tau)}[i,j]\right| = \tilde{\Theta}(\eta), \left|\Delta w_{2i}^{(\tau)}\right| = \tilde{\Theta}(\eta).$$

Since when $t = t_{\mathrm{inc}}, \mathrm{sign}\left(W_1^{(t_{\mathrm{inc}})}[i,j]\right) = \mathrm{sign}\left(w_{2i}^{(t_{\mathrm{inc}})}\right) = \mathrm{sign}\left(w_{2i}^{(0)}\right).$ We get that for $t_{\mathrm{inc}} \le \tau \le t$,

$$\forall i,j \in [d] : \quad \left|W_1^{(\tau+1)}[i,j]\right| = \left|W_1^{(\tau)}[i,j]\right| + \tilde{\Theta}(\eta), \quad \left|w_{2i}^{(\tau+1)}\right| = \left|w_{2i}^{(\tau)}\right| + \tilde{\Theta}(\eta).$$

Now for $t + 1$, we have $\forall i,j \in [d]$,

$$\mathrm{sign}\left(W_1^{(t+1)}[i,j]\right) = \mathrm{sign}\left(w_{2i}^{(0)}\right), \quad \left|W_1^{(t+1)}[i,j]\right| = \left|W_1^{(t_{\mathrm{inc}})}[i,j]\right| + (t+1-t_{\mathrm{inc}})\,\tilde{\Theta}(\eta),$$

$$\mathrm{sign}\left(w_{2i}^{(t+1)}\right) = \mathrm{sign}\left(w_{2i}^{(0)}\right), \quad \left|w_{2i}^{(t+1)}\right| = \left|w_{2i}^{(t_{\mathrm{inc}})}\right| + (t+1-t_{\mathrm{inc}})\,\tilde{\Theta}(\eta).$$

That means $\forall \tau \in [t+1-H, t+1] : \mathrm{sign}\left(W_1^{(\tau)}[i,j]\right) = \mathrm{sign}\left(w_{2i}^{(\tau)}\right) = \mathrm{sign}\left(w_{2i}^{(0)}\right).$ This proves (B) for time $t + 1$.

On the other hand, we get that $\left|W_1^{(t+1)}[i,j]\right| \geq \left|W_1^{(t_{\text{inc}})}[i,j]\right| = \Theta\left(\frac{1}{d^{\frac{3}{2}\alpha+1}}\right)$ and $\left|w_{2i}^{(t+1)}\right| \geq \left|w_{2i}^{(t_{\text{inc}})}\right| = \Omega\left(\frac{1}{d^{\frac{3}{2}\alpha}}\right)$. Since $t+1 \leq T_1$ which means $\forall j \in [d] : \left|E_j^{(t+1)}\right| \geq \sqrt{\eta d}$. Then

$$\left|g_1^{(t+1)}[i,j]\right| = \left|w_{2i}^{(t+1)} E_j^{(t+1)}\right| \geq \Omega\left(\frac{1}{d^{\frac{3}{2}\alpha}}\right)\sqrt{\eta d} = \Omega(\xi),$$

$$\left|g_{2i}^{(t+1)}\right| = \left|\sum_{j=1}^{d} E_j^{(t+1)} W_1^{(t+1)}[i,j]\right| = \sum_{j=1}^{d}\left|E_j^{(t+1)} W_1^{(t+1)}[i,j]\right| \geq d\Theta\left(\frac{1}{d^{\frac{3}{2}\alpha+1}}\right)\sqrt{\eta d} = \Omega(\xi).$$

This proves (C) at time $t+1$.

Since $t+1 \leq T_1$ which means $\forall j \in [d] : \left(W_2^{(t+1)} W_1^{(t+1)}\right)_j \leq \mathcal{O}(1)$, we obtain that

$$\sum_{i=1}^{d} w_{2i}^{(t+1)} W_1^{(t+1)}[i,j] = \sum_{i=1}^{d}\left|w_{2i}^{(t+1)}\right|\left|W_1^{(t+1)}[i,j]\right|$$

$$= \sum_{i=1}^{d}\left(\left|w_{2i}^{(t_{\text{inc}})}\right| + (t+1-t_{\text{inc}})\tilde{\Theta}(\eta)\right)\left(\left|W_1^{(t_{\text{inc}})}[i,j]\right| + (t+1-t_{\text{inc}})\tilde{\Theta}(\eta)\right) \leq \mathcal{O}(1).$$

Note that $\left|W_1^{(t_{\text{inc}})}[i,j]\right|, \left|w_{2i}^{(t_{\text{inc}})}\right| < \frac{1}{d}$ (since $t_{\text{inc}} < t_d$), we get that $(t+1-t_{\text{inc}})\tilde{\Theta}(\eta) = \mathcal{O}\left(\frac{1}{\sqrt{d}}\right)$, which gives us $\left|w_{2i}^{(t+1)}\right| = \tilde{\mathcal{O}}\left(\frac{1}{\sqrt{d}}\right)$ and $\left|W_1^{(t+1)}[i,j]\right| = \tilde{\mathcal{O}}\left(\frac{1}{\sqrt{d}}\right)$ and hence (A) holds at time $t+1$.

Therefore by induction, we can prove that (A), (B), (C) hold for all $t_{\text{inc}} \leq t \leq T_1$. Then applying Lemma 22, we get that for all $t_{\text{inc}} \leq t \leq T_1, \forall i,j \in [d] : \left|\Delta W_1^{(t)}[i,j]\right| = \tilde{\Theta}(\eta), \left|\Delta w_{2i}^{(t)}\right| = \tilde{\Theta}(\eta)$.

Specially, at the end of the first phase, we have $\forall j \in [d] : \left(W_2^{(t+1)} W_1^{(t+1)}\right)_j = \Theta(1)$. Repeating the above proof techniques gives us $\left|w_{2i}^{(T_1)}\right| = \tilde{\Theta}\left(\frac{1}{\sqrt{d}}\right)$ and $\left|W_1^{(T_1)}[i,j]\right| = \tilde{\Theta}\left(\frac{1}{\sqrt{d}}\right)$ for $\forall i,j \in [d]$.

### E.10 Proof of Lemma 20

Let's first prove eq. (30).

By Lemma 19, for $t_{\text{inc}} \leq t < T_1$, we have $\forall i,j \in [d], \left|g_1^{(t)}[i,j]\right| = \tilde{\Omega}\left(\sqrt{\eta}\right), \left|g_{2i}^{(t)}\right| = \tilde{\Omega}\left(\sqrt{\eta d}\right)$. Then it suffices to show that for $t_{\text{inc}} \leq t < T_1$, $\left|g_1^{(t)}[i,j] - g_1^{(t-\tau)}[i,j]\right| = \tau\tilde{\mathcal{O}}(\eta)$ and $\left|g_{2i}^{(t)} - g_{2i}^{(t-\tau)}\right| = \tau\tilde{\mathcal{O}}(\eta d)$. It suffices to show that when $t < T_1, \left|g_1^{(t+1)}[i,j] - g_1^{(t)}[i,j]\right| = \tilde{\mathcal{O}}(\eta)$ and $\left|g_{2i}^{(t+1)} - g_{2i}^{(t)}\right| = \tilde{\mathcal{O}}(\eta d)$.

By Lemma 18 and 23, we know that when $t < T_1, \forall i,j \in [d], \left|\Delta W_1^{(t)}[i,j]\right| \leq \tilde{\mathcal{O}}(\eta), \left|\Delta w_{2i}^{(t)}\right| \leq \tilde{\mathcal{O}}(\eta)$ and that $\left|W_1^{(t)}[i,j]\right| \leq \tilde{\mathcal{O}}\left(\frac{1}{\sqrt{d}}\right), \left|w_{2i}^{(t)}\right| \leq \tilde{\mathcal{O}}\left(\frac{1}{\sqrt{d}}\right)$. Then the bound $\left|\Delta E_j^{(t)}\right| \leq \tilde{\mathcal{O}}\left(\eta\sqrt{d}\right)$ in eq. (35) hold for all $t < T_1$ (not only $t_{\text{inc}} \leq t < T_1$). Substituting these bounds into eq. (36) gives us $\forall t < T_1$,

$$\left|g_1^{(t+1)}[i,j] - g_1^{(t)}[i,j]\right| \leq \left|w_{2i}^{(t+1)}\right|\left|\Delta E_j^{(t)}\right| + \left|\Delta w_{2i}^{(t)}\right|\left|E_j^{(t)}\right|$$

$$= \tilde{\mathcal{O}}\left(\frac{1}{\sqrt{d}}\right)\tilde{\mathcal{O}}\left(\eta\sqrt{d}\right) + \tilde{\Theta}(\eta)\mathcal{O}(1) = \tilde{\mathcal{O}}(\eta).$$

Similarly, we have that $\left|g_{2i}^{(t+1)} - g_{2i}^{(t)}\right| = \tilde{\mathcal{O}}(\eta d)$, which proves eq. (30).

Note that for $a,b \in \mathbb{R}$:

$$\frac{|a^2-b^2|}{a^2} = \frac{|a^2-(a-b-a)^2|}{a^2} = \frac{|2a(a-b)-(a-b)^2|}{a^2} \leq 2\frac{|a-b|}{|a|} + \left(\frac{|a-b|}{|a|}\right)^2.$$

Then eq. (31) immediately follows from eq. (30).

## E.11  Proof of Lemma 16

We divide Lemma 16 into the following three lemmas. Combining them together immediately gives us the whole proof.

The first lemma below gives us the structure of $W_2$ in the second phase and that of $W_1$ under some conditions.

**Lemma 24.** *Under Assumption 1, 2 and 3, suppose* $\sigma \leq \frac{\eta^{3/2}\xi^2}{d^{13/4}}$. *By picking* $\eta \leq \mathcal{O}\left(\frac{1}{d^{3\alpha}}\right), \xi \leq \sqrt{\frac{\eta}{d^{3\alpha-1}}}$, *and* $\beta_2 = \beta_1^2$, *we have w.h.p. for* $T_1 \leq t < \tilde{T}$,

$$\forall i \in [d]: \quad w_{2i}^{(t+1)} = w_{2i}^{(t)} - \eta\left(sign\left(g_{2i}^{(t)}\right) + e_{2i}^{(t)}\right), \quad where \ \left|e_{2i}^{(t)}\right| = \tilde{\mathcal{O}}\left(\sqrt{\eta}\right),$$

*and moreover*

$$\forall i \in [d]: \quad w_{2i}^{(t)} = sign\left(w_{2i}^{(0)}\right)c^{(t)} + R_{2i}^{(t)}, \quad where \quad \frac{\left|R_{2i}^{(t)}\right|}{c^{(t)}} = \tilde{\mathcal{O}}\left(\sqrt{\eta} + \frac{1}{d^{\alpha-1/2}}\right).$$

*As for $W_1$, if for certain $i.j \in [d]$ and certain $t \in [T_1, \tilde{T})$ we have* $\left|g_1^{(t)}[i,j]\right| = \tilde{\Omega}\left(\sqrt{\eta}\right)$, *then*

$$W_1^{(t+1)}[i,j] = W_1^{(t)}[i,j] - \eta\left(sign\left(g_1^{(t)}[i,j]\right) + e_1^{(t)}[i,j]\right), \quad where \ \left|e_1^{(t)}[i,j]\right| = \tilde{\mathcal{O}}\left(\sqrt{\eta}\right).$$

The second lemma below also analyzes the structure of $W_1$ but removes the conditions in Lemma 24.

**Lemma 25.** *Under Assumption 1, 2 and 3, suppose* $\sigma \leq \frac{\eta^{3/2}\xi^2}{d^{13/4}}$. *By picking* $\eta \leq \mathcal{O}\left(\frac{1}{d^{3\alpha}}\right), \xi \leq \sqrt{\frac{\eta}{d^{3\alpha-1}}}$, *and* $\beta_2 = \beta_1^2$, *we have w.h.p. for* $T_1 \leq t < \tilde{T}$, $\forall i,j \in [d]$, $\left|W_1^{(t)}[i,j]\right| = \tilde{\Omega}\left(\frac{1}{\sqrt{d}}\right)$ *and for any $j \in [d]$,*

$$W_1^{(t)}[i,j] = sign\left(w_{2i}^{(0)}\right)V_j^{(t)} + R_1^{(t)}[i,j], \quad where \quad \frac{\left|R_1^{(t)}[i,j]\right|}{\left|V_j^{(t)}\right|} \leq \tilde{\mathcal{O}}\left(\eta^{\frac{1}{4}} + \frac{1}{d^{\frac{\alpha}{2}-\frac{1}{4}}}\right).$$

The third lemma proves the convergence of Adam at time $\tilde{T}$.

**Lemma 26.** *Under Assumption 1, 2 and 3, suppose* $\sigma \leq \frac{\eta^{3/2}\xi^2}{d^{13/4}}$. *By picking* $\eta \leq \mathcal{O}\left(\frac{1}{d^{3\alpha}}\right), \xi \leq \sqrt{\frac{\eta}{d^{3\alpha-1}}}$, *and* $\beta_2 = \beta_1^2$, *at time $\tilde{T}$, we have that w.h.p.* $\forall j \in [d]: \left|E_j^{(\tilde{T})}\right| \leq \tilde{\mathcal{O}}\left(d\sqrt{\eta d}\right)$, *which implies* $\left\|E^{(\tilde{T})}\right\|_2^2 \leq \tilde{\mathcal{O}}\left(\eta d^4\right)$.

## E.12  Proof of Lemma 24

The proof is based on the following lemma, which gives a coarse analysis on the magnitude of weights and their increments per step during the second phase.

**Lemma 27.** *Under Assumption 1, 2 and 3, suppose* $\sigma \leq \frac{\eta^{3/2}\xi^2}{d^{13/4}}$. *By picking* $\eta \leq \mathcal{O}\left(\frac{1}{d^{3\alpha}}\right), \xi \leq \sqrt{\frac{\eta}{d^{3\alpha-1}}}$, *and* $\beta_2 = \beta_1^2$, *we have w.h.p. for all* $T_1 \leq t < \tilde{T}$,

$$\forall i,j \in [d]: \quad \left|w_{2i}^{(t+1)}\right| > \left|w_{2i}^{(t)}\right|, \quad \left|\Delta w_{2i}^{(t)}\right| = \tilde{\Theta}(\eta), \quad \left|\Delta W_1^{(t)}[i,j]\right| \leq \tilde{\mathcal{O}}(\eta).$$

*Moreover, we have that* $\forall i,j \in [d]: \quad \left|w_{2i}^{(t)}\right| = \tilde{\Theta}\left(\frac{1}{\sqrt{d}}\right), \left|W_1^{(t)}[i,j]\right| = \tilde{\mathcal{O}}\left(\frac{1}{\sqrt{d}}\right).$

Equipped with Lemma 27, we are ready to prove Lemma 24. We will only prove the results of $w_{2i}^{(t)}$. The proof for $W_1^{(t)}[i,j]$ uses the same techniques.

Lemma 27 gives us upper bounds of $\left|w_{2i}^{(t)}\right|, \left|W_1^{(t)}[i,j]\right|$, as well as $\left|\Delta w_{2i}^{(t)}\right|$ and $\left|\Delta W_1^{(t)}[i,j]\right|$ for all $i,j \in [d]$. Then we know that eq.(35) still holds, which gives us $\forall j \in [d] : \left|E_j^{(t+1)} - E_j^{(t)}\right| = \tilde{\mathcal{O}}\left(\sqrt{d}\eta\right)$. Then we can use the same strategy in Lemma 20 to prove that $\left|g_{2i}^{(t+1)} - g_{2i}^{(t)}\right| = \tilde{\mathcal{O}}(\eta d)$.

By definition, for $T_1 \le t < \tilde{T}$, we know that $\left|g_{2i}^{(t)}\right| = \Omega\left(d\sqrt{\eta}\right)$. Combining with the bound $\left|g_{2i}^{(t+1)} - g_{2i}^{(t)}\right| = \tilde{\mathcal{O}}(\eta d)$, we know that the $g_{2i}^{(t)}$ parts in eq.(30) and eq.(31) still hold. Then we can use the same strategy in Section E.2 to prove that the $w_{2i}^{(t)}$ part of eq. (29) still holds, which gives us

$$\forall i \in [d]: \quad w_{2i}^{(t+1)} = w_{2i}^{(t)} - \eta\left(\operatorname{sign}\left(g_{2i}^{(t)}\right) + e_{2i}^{(t)}\right), \quad \text{where } \left|e_{2i}^{(t)}\right| = \tilde{\mathcal{O}}\left(\sqrt{\eta}\right).$$

By Lemma 15, we have that at the end of the first phase ($t = T_1$),

$$\forall i \in [d]: \quad w_{2i}^{(T_1)} = \operatorname{sign}\left(w_{2i}^{(0)}\right)c^{(T_1)} + R_{2i}^{(T_1)}, \quad \text{where} \quad \frac{\left|R_{2i}^{(T_1)}\right|}{c^{(T_1)}} = \tilde{\mathcal{O}}\left(\sqrt{\eta} + \frac{1}{d^{\alpha-1/2}}\right).$$

Combining with $\forall i \in [d], \forall t \le T_i : \operatorname{sign}\left(g_{2i}^{(t)}\right) = -\operatorname{sign}\left(w_{2i}^{(0)}\right)$ yields that during the second phase, for $t \le \tilde{T}$, we have

$$\forall i \in [d]: \quad w_{2i}^{(t)} = \operatorname{sign}\left(w_{2i}^{(0)}\right)c^{(t)} + R_{2i}^{(t)}, \quad \text{where} \quad \frac{\left|R_{2i}^{(t)}\right|}{c^{(t)}} = \tilde{\mathcal{O}}\left(\sqrt{\eta} + \frac{1}{d^{\alpha-1/2}}\right).$$

### E.13 Proof of Lemma 27

By definition of $T_f$, there exists $j_0 \in [d]$ such that $E_{j_0}^{(\tau)} < -\sqrt{\eta d}$ for $T_1 \le t \le \tilde{T}$. We prove by induction that during this period, $\forall i \in [d] : \operatorname{sign}\left(w_{2i}^{(t)}\right) = \operatorname{sign}\left(W_1^{(t)}[i,j_0]\right) = \operatorname{sign}\left(w_{2i}^{(0)}\right)$ and that $\forall i,j \in [d] : \left|w_{2i}^{(t)}\right| = \tilde{\Theta}\left(\frac{1}{\sqrt{d}}\right), \left|W_1^{(t)}[i,j]\right| = \tilde{\mathcal{O}}\left(\frac{1}{\sqrt{d}}\right)$.

The base case ($t = T_1$) was already proven by Lemma 23. Now suppose for some $t$ such that $T_1 \le t < \tilde{T}$, for all $\tau$ such that $T_1 \le \tau \le t$, we have $\forall i \in [d] : \operatorname{sign}\left(w_{2i}^{(\tau)}\right) = \operatorname{sign}\left(W_1^{(\tau)}[i,j_0]\right) = \operatorname{sign}\left(w_{2i}^{(0)}\right)$ and that $\forall i,j \in [d] : \left|w_{2i}^{(\tau)}\right| = \tilde{\Theta}\left(\frac{1}{\sqrt{d}}\right), \left|W_1^{(\tau)}[i,j]\right| = \tilde{\mathcal{O}}\left(\frac{1}{\sqrt{d}}\right)$. Using these bounds, we get that $\forall j \in [d] : \left|E_j^{(\tau)}\right| \le \sum_{i=1}^d \left|w_{2i}^{(\tau)} W_1^{(\tau)}[i,j]\right| + |A_j| = \mathcal{O}(1)$, which then yields two upper bounds $\left|g_1^{(\tau)}[i,j]\right| = \left|w_{2i}^{(\tau)} E_j^{(\tau)}\right| = \tilde{\mathcal{O}}\left(\frac{1}{\sqrt{d}}\right)$ and $\left|g_{2i}^{(\tau)}\right| \le \sum_{j=1}^d \left|E_j^{(\tau)} W_1^{(\tau)}[i,j]\right| = \tilde{\mathcal{O}}\left(\sqrt{d}\right)$.

By definition of $T_g$, we know that for all $T_1 \le \tau \le t, \forall i \in [d] : \left|g_{2i}^{(\tau)}\right| \ge d\sqrt{\eta} = \Omega(\xi)$ and that $\operatorname{sign}\left(g_{2i}^{(\tau)}\right) = -\operatorname{sign}\left(w_{2i}^{(0)}\right)$, which implies that Condition 2 is satisfied for $\forall i \in [d]$. At the end of the proof of this lemma, we will show that $\tilde{T} = \tilde{\Theta}\left(\frac{1}{\sqrt{d}\eta}\right)$. Together with the upper bound of $\left|g_{2i}^{(\tau)}\right|$, we can apply Lemma 22 to get that w.h.p. for $T_1 \le \tau \le t$, $\operatorname{sign}\left(\Delta w_{2i}^{(\tau)}\right) = \operatorname{sign}\left(w_{2i}^{(0)}\right)$ and $\left|\Delta w_{2i}^{(\tau)}\right| = \tilde{\Theta}(\eta)$. Combining with the inductive hypothesis $\operatorname{sign}\left(w_{2i}^{(\tau)}\right) = \operatorname{sign}\left(w_{2i}^{(0)}\right)$ gives us that $\left|w_{2i}^{(\tau+1)}\right| = \left|w_{2i}^{(\tau)}\right| + \tilde{\Theta}(\eta)$. Specially, when $\tau = t$, we get the lower bound $\left|w_{2i}^{(t+1)}\right| \ge \left|w_{2i}^{(t)}\right| = \tilde{\Omega}\left(\frac{1}{\sqrt{d}}\right)$ and that $\operatorname{sign}\left(w_{2i}^{(t+1)}\right) = \operatorname{sign}\left(w_{2i}^{(0)}\right)$.

Since $E_{j_0}^{(\tau)} < -\sqrt{\eta d}$, we have that $\forall i \in [d] : \left|g_1^{(\tau)}[i,j_0]\right| = \left|w_{2i}^{(\tau)}\right|\left|E_{j_0}^{(\tau)}\right| = \tilde{\Omega}\left(\sqrt{\eta}\right) = \Omega(\xi)$ and that $\operatorname{sign}\left(g_1^{(\tau)}[i,j_0]\right) = -\operatorname{sign}\left(w_{2i}^{(0)}\right)$. That means Condition 1 is satisfied for $\forall i \in [d]$ and $j_0$. Using the same technique as when we deal with $w_{2i}^{(\tau)}$, we get that for $T_1 \le \tau \le t$,

$\forall i \in [d] : \left| W_1^{(\tau+1)}[i, j_0] \right| = \left| W_1^{(\tau)}[i, j_0] \right| + \tilde{\Theta}(\eta)$, $\text{sign}\left( W_1^{(t+1)}[i, j_0] \right) = \text{sign}\left( w_{2i}^{(0)} \right)$ and that $\forall i, j \in [d], \left| \Delta W_1^{(\tau)}[i, j] \right| = \tilde{\mathcal{O}}(\eta)$.

Now we analyze the magnitude order of $\left| w_{2i}^{(t+1)} \right|, \left| W_1^{(t+1)}[i, j] \right|$. Let's first analyze $\left| w_{2i}^{(t+1)} \right|$.

By Lemma 15, when $t = T_1$, $\forall i, j \in [d]$,

$$\frac{\left| w_{2i}^{(T_1)} \right|}{\left| w_{2j}^{(T_1)} \right|} = 1 \pm \tilde{\mathcal{O}}\left( \sqrt{\eta} + \frac{1}{d^{\alpha-1/2}} \right), \quad \frac{\left| W_1^{(T_1)}[i, j_0] \right|}{\left| w_{2i}^{(T_1)} \right|} = 1 \pm \tilde{\mathcal{O}}\left( \sqrt{\eta} + \frac{1}{d^{\alpha-1/2}} \right).$$

Combining with the facts that for $T_1 \leq \tau \leq t$, $\left| W_1^{(\tau+1)}[i, j_0] \right| = \left| W_1^{(\tau)}[i, j_0] \right| + \tilde{\Theta}(\eta)$ and $\left| w_{2i}^{(\tau+1)} \right| = \left| w_{2i}^{(\tau)} \right| + \tilde{\Theta}(\eta)$ yields $\frac{\left| W_1^{(t+1)}[i, j_0] \right|}{\left| w_{2i}^{(t+1)} \right|} = \tilde{\Theta}(1)$. Since we just proved $\forall i \in [d] :$ $\text{sign}\left( w_{2i}^{(t+1)} \right) = \text{sign}\left( W_1^{(t+1)}[i, j_0] \right) = \text{sign}\left( w_{2i}^{(0)} \right)$, we get that

$$(W_2 W_1)_{j_0}^{(t+1)} = \sum_{i=1}^{d} w_{2i}^{(t+1)} W_1^{(t+1)}[i, j_0] = \sum_{i=1}^{d} \left| w_{2i}^{(t+1)} \right| \left| W_1^{(t+1)}[i, j_0] \right| = \mathcal{O}(1),$$

which gives us that $\left| w_{2i}^{(t+1)} \right| = \tilde{\mathcal{O}}\left( \frac{1}{\sqrt{d}} \right)$. Recall that we have shown $\left| w_{2i}^{(t+1)} \right| \geq \tilde{\Omega}\left( \frac{1}{\sqrt{d}} \right)$, then $\left| w_{2i}^{(t+1)} \right| = \tilde{\Theta}\left( \frac{1}{\sqrt{d}} \right)$.

Now we prove $\left| W_1^{(t+1)}[i, j] \right| = \tilde{\mathcal{O}}\left( \frac{1}{\sqrt{d}} \right)$. We have proved that $T_1 \leq \tau \leq t$, $\forall i, j \in [d]$, $\left| \Delta W_1^{(\tau)}[i, j] \right| = \tilde{\mathcal{O}}(\eta)$ and $\left| w_{2i}^{(\tau+1)} \right| - \left| w_{2i}^{(\tau)} \right| = \tilde{\Theta}(\eta)$, then $\forall i, j \in [d]$,

$$\frac{\left| W_1^{(t+1)}[i, j] \right|}{\left| w_{2i}^{(t+1)} \right|} \leq \frac{\left| W_1^{(T_1)}[i, j] \right| + \sum_{\tau=T_1}^{t} \left| W_1^{(\tau+1)}[i, j] - W_1^{(\tau)}[i, j] \right|}{\left| w_{2i}^{(T_1)} \right| + \sum_{\tau=T_1}^{t} \left| w_{2i}^{(\tau+1)} \right| - \left| w_{2i}^{(\tau)} \right|}$$

$$\leq \frac{\left| W_1^{(T_1)}[i, j] \right| + (t + 1 - T_1)\tilde{\mathcal{O}}(\eta)}{\left| w_{2i}^{(T_1)} \right| + (t + 1 - T_1)\tilde{\Theta}(\eta)} = \tilde{\mathcal{O}}(1),$$

where the last equality uses $\frac{\left| W_1^{(T_1)}[i, j] \right|}{\left| w_{2i}^{(T_1)} \right|} = 1 \pm \tilde{\mathcal{O}}\left( \sqrt{\eta} + \frac{1}{d^{\alpha-1/2}} \right)$. Since we already proved that $\left| w_{2i}^{(t+1)} \right| = \tilde{\Theta}\left( \frac{1}{\sqrt{d}} \right)$, we get $\left| W_1^{(t+1)} \right| = \tilde{\mathcal{O}}\left( \frac{1}{\sqrt{d}} \right)$.

Therefore by induction, for all $t$ in the interval $[T_1, \tilde{T})$, we have $\forall i, j \in [d] : \left| w_{2i}^{(t)} \right| = \tilde{\Theta}\left( \frac{1}{\sqrt{d}} \right)$, $\left| W_1^{(t)}[i, j] \right| = \tilde{\mathcal{O}}\left( \frac{1}{\sqrt{d}} \right)$. From the proof we also get $\forall i \in [d] : \left| w_{2i}^{(t+1)} \right| > \left| w_{2i}^{(t)} \right|$, and that $\left| \Delta w_{2i}^{(t)} \right| = \tilde{\Theta}(\eta), \left| \Delta W_1^{(t)}[i, j] \right| \leq \tilde{\mathcal{O}}(\eta)$.

Now we verify that $\tilde{T} = \tilde{\Theta}\left( \frac{1}{\sqrt{d}\eta} \right)$. Combining $\forall i, j \in [d] : \left| w_{2i}^{(\tilde{T})} \right| = \tilde{\Theta}\left( \frac{1}{\sqrt{d}} \right)$ and $\forall t \in [T_1, \tilde{T}), \left| w_{2i}^{(t+1)} \right| - \left| w_{2i}^{(t)} \right| = \tilde{\Theta}(\eta)$, we immediately get that $\tilde{T} - T_1 = \tilde{\Theta}\left( \frac{1}{\sqrt{d}\eta} \right)$. In Section E.2 we have shown that $T_1 = \Theta\left( \frac{1}{\sqrt{d}\eta} \right)$, then we get $\tilde{T} = \tilde{\Theta}\left( \frac{1}{\sqrt{d}\eta} \right)$.

### E.14 Proof of Lemma 25

We prove this lemma by induction. The base case ($t = T_1$) can be verified by Lemma 15. Now suppose for $t$ in the interval $[T_1, \tilde{T})$, we have $\forall i, j \in [d], \left| W_1^{(t)}[i, j] \right| = \tilde{\Omega}\left( \frac{1}{\sqrt{d}} \right)$.

For $t \in [T_1, \tilde{T})$, by the proof of Lemma 27 (Section E.13), we know that for $\forall \tau \leq t, \forall i, j \in [d]$ :
$\left| w_{2i}^{(\tau)} \right| = \tilde{\Theta}\left(\frac{1}{\sqrt{d}}\right)$, $\left| W_1^{(\tau)}[i,j] \right| = \tilde{\mathcal{O}}\left(\frac{1}{\sqrt{d}}\right)$ and that $\left| g_1^{(\tau)}[i,j] \right| \leq \tilde{\mathcal{O}}\left(\frac{1}{\sqrt{d}}\right)$, $\left| g_{2i}^{(\tau)} \right| \leq \tilde{\mathcal{O}}\left(\sqrt{d}\right)$, and
that $\tilde{T}_1 = \tilde{\Theta}\left(\frac{1}{\sqrt{d}\eta}\right)$. Then we can pick $H := \frac{1}{1-\beta_1} \log \frac{d}{\eta \xi^2}$ and apply Lemma 17 and Corollary 2 to
get that, w.h.p., for all $t \in [T_1, \tilde{T})$ and $\forall i, j \in [d]$, the update of $W_1$ can be written as

$$W_1^{(t+1)}[i,j] = W_1^{(t)}[i,j] - \eta_t \frac{(1-\beta_1)\sum_{\tau=0}^{H} \beta_1^\tau g_1^{(t-\tau)}[i,j] + \epsilon_{1n}^{(t)}[i,j]}{\sqrt{(1-\beta_2)\sum_{\tau=0}^{H} \beta_2^\tau \left( g_1^{(t-\tau)}[i,j] \right)^2 + \epsilon_{1d}^{(t)}[i,j] + \xi}},$$

where $\left| \epsilon_{1n}^{(t)}[i,j] \right|, \left| \epsilon_{1d}^{(t)}[i,j] \right| \leq \tilde{\mathcal{O}}(\eta \xi^2)$. By Lemma 24, we have that for $1 \leq i, j \leq d$,

$$g_1^{(t)}[i,j] = w_{2i}^{(t)} E_j^{(t)} = c^{(t)} \text{sign}\left( w_{2i}^{(0)} \right) E_j^{(t)} + R_{g,1}^{(t)}[i,j], \qquad \frac{\left| R_{g,1}^{(t)}[i,j] \right|}{c^{(t)} \left| E_j^{(t)} \right|} = \tilde{\mathcal{O}}\left( \sqrt{\eta} + \frac{1}{d^{\alpha-1/2}} \right),$$

$$\Rightarrow \quad \sum_{\tau=0}^{H} \beta_1^\tau g_1^{(t-\tau)}[i,j] = \text{sign}\left( w_{2i}^{(0)} \right) \sum_{\tau=0}^{H} \beta_1^\tau c^{(t-\tau)} E_j^{(t-\tau)} + \sum_{\tau=0}^{H} \beta_1^\tau R_{g,1}^{(t-\tau)}[i,j]. \qquad (37)$$

Using the fact that for $a, b \in \mathbb{R}, \frac{|a^2-b^2|}{a^2} \leq 2\frac{|a-b|}{|a|} + \left(\frac{|a-b|}{|a|}\right)^2$, we get that

$$\left( g_1^{(t)}[i,j] \right)^2 = \left( c^{(t)} \text{sign}\left( w_{2i}^{(0)} \right) E_j^{(t)} + R_{g,1}^{(t)}[i,j] \right)^2 := \left( c^{(t)} E_j^{(t)} \right)^2 + R_{\text{gsqr},1}^{(t)}[i,j],$$

where $\frac{\left| R_{\text{gsqr},1}^{(t)}[i,j] \right|}{\left( c^{(t)} E_j^{(t)} \right)^2} = \tilde{\mathcal{O}}\left( \sqrt{\eta} + \frac{1}{d^{\alpha-1/2}} \right)$. That yields

$$\sum_{\tau=0}^{H} \beta_2^\tau \left( g_1^{(t-\tau)}[i,j] \right)^2 = \sum_{\tau=0}^{H} \beta_2^\tau \left( c^{(t-\tau)} E_j^{(t-\tau)} \right)^2 + \sum_{\tau=0}^{H} \beta_2^\tau R_{\text{gsqr},1}^{(t-\tau)}[i,j]. \qquad (38)$$

Since $\left( c^{(t-\tau)} E_j^{(t-\tau)} \right)^2 > 0$, in eq. (38) we have that

$$\frac{\left| \sum_{\tau=0}^{H} \beta_2^\tau R_{\text{gsqr},1}^{(t-\tau)}[i,j] \right|}{\left| \sum_{\tau=0}^{H} \beta_2^\tau \left( c^{(t-\tau)} E_j^{(t-\tau)} \right)^2 \right|} = \tilde{\mathcal{O}}\left( \sqrt{\eta} + \frac{1}{d^{\alpha-1/2}} \right). \qquad (39)$$

However we cannot similarly prove that $\left| \sum_{\tau=0}^{H} \beta_1^\tau R_{g,1}^{(t-\tau)}[i,j] \right| \ll \left| \sum_{\tau=0}^{H} \beta_1^\tau c^{(t-\tau)} E_j^{(t-\tau)} \right|$ in
eq. (37) because $c^{(t-\tau)} E_j^{(t-\tau)}$ may not have the same sign for $\tau = 0, 1, ..., H$. To deal with eq.(37),
we need to consider the two cases where $\left| \sum_{\tau=0}^{H} \beta_1^\tau R_{g,1}^{(t-\tau)}[i,j] \right| \ll \left| \sum_{\tau=0}^{H} \beta_1^\tau c^{(t-\tau)} E_j^{(t-\tau)} \right|$ or
$\left| \sum_{\tau=0}^{H} \beta_1^\tau R_{g,1}^{(t-\tau)}[i,j] \right| \not\ll \left| \sum_{\tau=0}^{H} \beta_1^\tau c^{(t-\tau)} E_j^{(t-\tau)} \right|$.

**Case 1.** $\left| (1-\beta_1)\sum_{\tau=0}^{H} \beta_1^\tau R_{g,1}^{(t-\tau)}[i,j] + \epsilon_{1n}^{(t)}[i,j] \right| \leq \delta \left| (1-\beta_1)\sum_{\tau=0}^{H} \beta_1^\tau c^{(t-\tau)} E_j^{(t-\tau)} \right|$ where
$\delta = \left( \eta^{\frac{1}{4}} + \frac{1}{d^{\frac{\alpha}{2}-\frac{1}{4}}} \right)$.

Note that from eq. (39) we have

$$\left| (1-\beta_1)\sum_{\tau=0}^{H} \beta_2^\tau R_{\text{gsqr},1}^{(t-\tau)}[i,j] \right| \leq \tilde{\mathcal{O}}\left( \sqrt{\eta} + \frac{1}{d^{\alpha-1/2}} \right) \left| (1-\beta_1)\sum_{\tau=0}^{H} \beta_2^\tau \left( c^{(t-\tau)} E_j^{(t-\tau)} \right)^2 \right|.$$

Combining with $\left|\epsilon_{1d}^{(t)}[i,j]\right| \leq \tilde{\mathcal{O}}(\eta\xi^2) \leq \tilde{\mathcal{O}}\left(\eta^{\frac{1}{4}} + \frac{1}{d^{\frac{\alpha}{2}-\frac{1}{4}}}\right)^2 \xi^2$, we can apply Lemma 34 to get that

$$
\begin{aligned}
& W_1^{(t+1)}[i,j] - W_1^{(t)}[i,j] \\
&= -\eta_t \frac{(1-\beta_1)\mathrm{sign}\left(w_{2i}^{(0)}\right)\sum_{\tau=0}^{H}\beta_1^\tau c^{(t-\tau)}E_j^{(t-\tau)} + (1-\beta_1)\sum_{\tau=0}^{H}\beta_1^\tau R_{g,1}^{(t-\tau)}[i,j] + \epsilon_{1n}^{(t)}[i,j]}{\sqrt{(1-\beta_2)\sum_{\tau=0}^{H}\beta_2^\tau\left(c^{(t-\tau)}E_j^{(t-\tau)}\right)^2 + (1-\beta_2)\sum_{\tau=0}^{H}\beta_2^\tau R_{gsqr,1}^{(t-\tau)}[i,j] + \epsilon_{1d}^{(t)}[i,j]} + \xi} \\
&= -\eta_t \frac{1-\beta_1}{\sqrt{1-\beta_2}} \cdot \frac{\mathrm{sign}\left(w_{2i}^{(0)}\right)\sum_{\tau=0}^{H}\beta_1^\tau c^{(t-\tau)}E_j^{(t-\tau)}}{\sqrt{\sum_{\tau=0}^{H}\beta_2^\tau\left(c^{(t-\tau)}E_j^{(t-\tau)}\right)^2} + \xi}\left(1 + e_1^{(t)}[i,j]\right) \\
&:= -\mathrm{sign}\left(w_{2i}^{(0)}\right)v_j^{(t)}\left(1 + e_1^{(t)}[i,j]\right),
\end{aligned}
$$

where $\left|e_1^{(t)}[i,j]\right| = \tilde{\mathcal{O}}\left(\eta^{\frac{1}{4}} + \frac{1}{d^{\frac{\alpha}{2}-\frac{1}{4}}}\right)$. Since $\left|W_1^{(t+1)}[i,j] - W_1^{(t)}[i,j]\right| = \tilde{\mathcal{O}}(\eta)$, we get that $\left|v_j^{(t)}\right| = \tilde{\mathcal{O}}(\eta)$.

**Case 2.** $\left|(1-\beta_1)\sum_{\tau=0}^{H}\beta_1^\tau R_{g,1}^{(t-\tau)}[i,j] + \epsilon_{1n}^{(t)}[i,j]\right| > \delta\left|(1-\beta_1)\sum_{\tau=0}^{H}\beta_1^\tau c^{(t-\tau)}E_j^{(t-\tau)}\right|$ where $\delta = \left(\eta^{\frac{1}{4}} + \frac{1}{d^{\frac{\alpha}{2}-\frac{1}{4}}}\right)$.

Since $\frac{\left|R_{g,1}^{(t)}[i,j]\right|}{c^{(t)}\left|E_j^{(t)}\right|} = \tilde{\mathcal{O}}\left(\sqrt{\eta} + \frac{1}{d^{\alpha-1/2}}\right)$, we have that

$$
\begin{aligned}
\left|(1-\beta_1)\sum_{\tau=0}^{H}\beta_1^\tau R_{g,1}^{(t-\tau)}[i,j]\right| &\leq \tilde{\mathcal{O}}\left(\sqrt{\eta} + \frac{1}{d^{\alpha-1/2}}\right)(1-\beta_1)\sum_{\tau=0}^{H}\beta_1^\tau\left|c^{(t-\tau)}E_j^{(t-\tau)}\right| \\
&\overset{(i)}{\leq} \tilde{\mathcal{O}}\left(\sqrt{\eta} + \frac{1}{d^{\alpha-1/2}}\right)\sqrt{(H+1)(1-\beta_1)\sum_{\tau=0}^{H}\beta_2^\tau\left(c^{(t-\tau)}E_j^{(t-\tau)}\right)^2} \\
&\overset{(ii)}{=} \tilde{\mathcal{O}}\left(\sqrt{\eta} + \frac{1}{d^{\alpha-1/2}}\right)\sqrt{(1-\beta_1)\sum_{\tau=0}^{H}\beta_2^\tau\left(g_1^{(t-\tau)}[i,j]\right)^2},
\end{aligned}
$$

where $(i)$ uses Cauchy-Schwarz inequality and $\beta_2 = \beta_1^2$, $(ii)$ uses eq. (38) and (39).

Combining with $\left|\epsilon_{1n}^{(t)}[i,j]\right| \leq \tilde{\mathcal{O}}(\eta\xi^2) \leq \tilde{\mathcal{O}}\left(\sqrt{\eta} + \frac{1}{d^{\alpha-1/2}}\right)\left(\xi - \sqrt{\left|\epsilon_{1d}^{(t)}[i,j]\right|}\right)$ gives us

$$
\begin{aligned}
\left|(1-\beta_1)\sum_{\tau=0}^{H}\beta_1^\tau c^{(t-\tau)}E_j^{(t-\tau)}\right| &< \frac{\left|(1-\beta_1)\sum_{\tau=0}^{H}\beta_1^\tau R_{g,1}^{(t-\tau)}[i,j] + \epsilon_{1n}^{(t)}[i,j]\right|}{\eta^{\frac{1}{4}} + \frac{1}{d^{\frac{\alpha}{2}-\frac{1}{4}}}} \\
&\leq \frac{\tilde{\mathcal{O}}\left(\sqrt{\eta} + \frac{1}{d^{\alpha-1/2}}\right)}{\eta^{\frac{1}{4}} + \frac{1}{d^{\frac{\alpha}{2}-\frac{1}{4}}}}\left(\sqrt{\sum_{\tau=0}^{H}\beta_2^\tau\left(g_1^{(t-\tau)}[i,j]\right)^2} - \sqrt{\left|\epsilon_{1d}^{(t)}[i,j]\right|} + \xi\right) \\
&\leq \tilde{\mathcal{O}}\left(\eta^{\frac{1}{4}} + \frac{1}{d^{\frac{\alpha}{2}-\frac{1}{4}}}\right)\left(\sqrt{\sum_{\tau=0}^{H}\beta_2^\tau\left(g_1^{(t-\tau)}[i,j]\right)^2 + \epsilon_{1d}^{(t)}[i,j]} + \xi\right),
\end{aligned}
$$

which implies

$$
\left|W_1^{(t+1)}[i,j] - W_1^{(t)}[i,j]\right| \leq \eta\tilde{\mathcal{O}}\left(\eta^{\frac{1}{4}} + \frac{1}{d^{\frac{\alpha}{2}-\frac{1}{4}}}\right).
$$

Consider certain $i.j \in [d]$ and the period from $T_1$ to $t$. Denote $\mathcal{T}$ as the set of time points when Case 1 is satisfied. By Lemma 27, we know that $\eta(t - T_1) = \tilde{\mathcal{O}}\left(\frac{1}{\sqrt{d}}\right)$, which gives us

$$\sum_{\tau \notin \mathcal{T}} \Delta W_1^{(\tau)}[i, j] \leq (t - T_1)\eta\tilde{\mathcal{O}}\left(\eta^{\frac{1}{4}} + \frac{1}{d^{\frac{\alpha}{2} - \frac{1}{4}}}\right) = \tilde{\mathcal{O}}\left(\frac{\eta^{\frac{1}{4}}}{d^{\frac{1}{2}}} + \frac{1}{d^{\frac{\alpha}{2} + \frac{1}{4}}}\right).$$

By the first phase analysis, we have that

$$W_1^{(T_1)}[i, j] = \text{sign}\left(w_{2i}^{(0)}\right) V_j^{(T_1)} + R_1^{(T_1)}[i, j],$$

where $V_j^{(T_1)} = \mathcal{O}\left(\frac{1}{\sqrt{d}}\right), \left|R_1^{(T_1)}[i, j]\right| = \tilde{\mathcal{O}}\left(\sqrt{\frac{\eta}{d}} + \frac{1}{d^\alpha}\right)$. Combining with the analysis of Case 1, we have that

$$W_1^{(T_1)}[i, j] + \sum_{\tau \in \mathcal{T}} \Delta W_1^{(\tau)}[i, j] = \text{sign}\left(w_{2i}^{(0)}\right)\left(V_j^{(T_1)} - \sum_{\tau \in \mathcal{T}} v_j^{(\tau)}\right) + R_{\mathcal{T}}[i, j],$$

where $|R_{\mathcal{T}}[i, j]| \leq \mathcal{O}\left(\eta^{\frac{1}{4}} + \frac{1}{d^{\frac{\alpha}{2} - \frac{1}{4}}}\right)\left(\left|V_j^{(T_1)}\right| + \sum_{\tau \in \mathcal{T}}\left|v_j^{(\tau)}\right|\right)$.

Since for $\tau \in \mathcal{T}, \left|v_j^{(\tau)}\right| = \tilde{\mathcal{O}}(\eta), V_j^{(T_1)} = \mathcal{O}\left(\frac{1}{\sqrt{d}}\right)$ and $\eta(t - T_1) = \tilde{\mathcal{O}}\left(\frac{1}{\sqrt{d}}\right)$, we can bound $|R_{\mathcal{T}}[i, j]|$ by

$$|R_{\mathcal{T}}[i, j]| \leq \tilde{\mathcal{O}}\left(\eta^{\frac{1}{4}} + \frac{1}{d^{\frac{\alpha}{2} - \frac{1}{4}}}\right)\left(\mathcal{O}\left(\frac{1}{\sqrt{d}}\right) + (t - T_1)\tilde{\mathcal{O}}(\eta)\right) \leq \tilde{\mathcal{O}}\left(\frac{\eta^{\frac{1}{4}}}{d^{\frac{1}{2}}} + \frac{1}{d^{\frac{\alpha}{2} + \frac{1}{4}}}\right).$$

Combining the above results together yields

$$W_1^{(t)}[i, j] = W_1^{(T_1)} + \sum_{\tau = T_1}^{t-1} \Delta W_1^{(\tau)}[i, j] = W_1^{(T_1)} + \sum_{\tau \in \mathcal{T}} \Delta W_1^{(\tau)}[i, j] + \sum_{\tau \notin \mathcal{T}} \Delta W_1^{(\tau)}[i, j]$$

$$= \text{sign}\left(w_{2i}^{(0)}\right)\left(V_j^{(T_1)} - \sum_{\tau \in \mathcal{T}} v_j^{(\tau)}\right) + \tilde{\mathcal{O}}\left(\frac{\eta^{\frac{1}{4}}}{d^{\frac{1}{2}}} + \frac{1}{d^{\frac{\alpha}{2} + \frac{1}{4}}}\right)$$

$$:= \text{sign}\left(w_{2i}^{(0)}\right) V_j^{(t)} + R_1^{(t)}[i, j], \text{ where } \left|R_1^{(t)}[i, j]\right| \leq \tilde{\mathcal{O}}\left(\frac{\eta^{\frac{1}{4}}}{d^{\frac{1}{2}}} + \frac{1}{d^{\frac{\alpha}{2} + \frac{1}{4}}}\right).$$

By the inductive hypothesis $\left|W_1^{(t)}[i, j]\right| = \tilde{\Omega}\left(\frac{1}{\sqrt{d}}\right)$, we get that $\left|V_j^{(t)}\right| = \tilde{\Omega}\left(\frac{1}{\sqrt{d}}\right)$, which gives us $\frac{\left|R_1^{(t)}[i,j]\right|}{\left|V_j^{(t)}\right|} \leq \tilde{\mathcal{O}}\left(\eta^{\frac{1}{4}} + \frac{1}{d^{\frac{\alpha}{2} - \frac{1}{4}}}\right).$

Therefore, we have that for any $j \in [d]$ and any $i_1, i_2 \in [d]$,

$$\frac{\left|W_1^{(t)}[i_1, j]\right|}{\left|W_1^{(t)}[i_2, j]\right|} = \frac{\left|\text{sign}\left(w_{2i_1}^{(0)}\right) V_j^{(t)}\left(1 \pm \tilde{\mathcal{O}}\left(\eta^{\frac{1}{4}} + \frac{1}{d^{\frac{\alpha}{2} - \frac{1}{4}}}\right)\right)\right|}{\left|\text{sign}\left(w_{2i_2}^{(0)}\right) V_j^{(t)}\left(1 \pm \tilde{\mathcal{O}}\left(\eta^{\frac{1}{4}} + \frac{1}{d^{\frac{\alpha}{2} - \frac{1}{4}}}\right)\right)\right|} = 1 \pm \tilde{\mathcal{O}}\left(\eta^{\frac{1}{4}} + \frac{1}{d^{\frac{\alpha}{2} - \frac{1}{4}}}\right).$$

By Lemma 24, we know that $\left|w_{2i}^{(t)}\right|$ with different $i$ are also roughly equal, i.e. $\frac{\left|w_{2i_1}^{(t)}\right|}{\left|w_{2i_2}^{(t)}\right|} = 1 \pm \tilde{\mathcal{O}}\left(\sqrt{\eta} + \frac{1}{d^{\alpha - 1/2}}\right)$. Then we have for any $j \in [d]$,

$$(W_2 W_1)_j^{(t)} = \sum_{i=1}^d w_{2i}^{(t)} W_1^{(t)}[i, j] = \sum_{i=1}^d \left|w_{2i}^{(t)}\right|\left|W_1^{(t)}[i, j]\right| = \Theta\left(d\left|w_{2k}^{(t)}\right|\left|W_1^{(t)}[k, j]\right|\right)$$

$$= \tilde{\Theta}\left(\sqrt{d}\left|W_1^{(t)}[k, j]\right|\right).$$

where $k$ can be any index in $\{1, 2, ..., d\}$ and the last equality uses $\forall i \in [d] : \left|w_{2i}^{(t)}\right| = \tilde{\Theta}\left(\frac{1}{\sqrt{d}}\right)$.

Now we analyze the lower bound of $\left|W_1^{(t+1)}[k,j]\right|$. Although it may decrease during some period, we observe that once $\left|W_1^{(t)}[k,j]\right|$ decreases to some value of order $\tilde{\Theta}\left(\frac{1}{\sqrt{d}}\right)$ such that $(W_2 W_1)_j^{(t)} < A_j - \sqrt{\eta d}$, i.e. $E_j^{(t)} < -\sqrt{\eta d}$, we can apply the technique in Section E.13 when analyzing $W_1^{(t)}[i,j_0]$ to get that $\left|W_1^{(t)}[k,j]\right|$ will increase in the next step. This mechanism ensures a $\tilde{\Omega}\left(\frac{1}{\sqrt{d}}\right)$ lower bound of $\left|W_1^{(t+1)}[k,j]\right|$. Since $k,j$ are arbitrary, we have proved that at time $t+1$, $\forall i,j \in [d]: \left|W_1^{(t+1)}[i,j]\right| \geq \tilde{\Omega}\left(\frac{1}{\sqrt{d}}\right)$.

Therefore by induction, we conclude that when $T_1 \leq t < \tilde{T}$, for $\forall i,j \in [d]$, $\left|W_1^{(t)}[i,j]\right| = \tilde{\Omega}\left(\frac{1}{\sqrt{d}}\right)$. The remaining part of this lemma has also been proved by the analysis above.

### E.15  Proof of Lemma 26

Lemma 27 tells us that for any $i \in [d]$, $\left|w_{2i}^{(t)}\right|$ keeps increasing when $t < \tilde{T}$. However, the behavior of $W_1^{(t)}[i,j]$ is more complicated. The following lemma tells us that $\left|W_1^{(t)}[i,j]\right|$ will increase until $T_{f,j}$. After that $\left|W_1^{(t)}[i,j]\right|$ and $E_j^{(t)}$ may zigzag, but $E_j^{(t)}$ will not fluctuate dramatically and will be trapped in a small interval around zero.

**Lemma 28.** *Under Assumption 1, 2 and 3, suppose $\sigma \leq \frac{\eta^{3/2}\xi^2}{d^{13/4}}$. Pick $\eta \leq \mathcal{O}\left(\frac{1}{d^{3\alpha}}\right), \xi \leq \sqrt{\frac{\eta}{d^{3\alpha-1}}}$, and $\beta_2 = \beta_1^2$. Consider certain coordinate $j$. For $T_1 \leq t < \min\left\{\tilde{T}, T_{f,j}\right\}$, we have $\forall i \in [d]:$ $\left|W_1^{(t)}[i,j]\right|$ keeps increasing. If $T_{f,j} < \tilde{T}$, then for $T_{f,j} \leq t < \tilde{T}$, we will have $-\tilde{\mathcal{O}}\left(\sqrt{\eta d}\right) \leq E_j^{(t)} \leq \tilde{\mathcal{O}}\left(\sqrt{\eta d}\right)$.*

Now we start proving Lemma 26. At time $\tilde{T}$, denote $S := \left\{j : T_{f,j} < \tilde{T}\right\}$, i.e. the set of coordinates whose $E_j$ have passed its "flip time". By Lemma 28, we know that $\forall j \in S$, $\left|E_j^{(\tilde{T})}\right| \leq \tilde{\mathcal{O}}\left(\sqrt{\eta d}\right)$. If $S^c = \phi$, which means $\forall j \in [d] : \left|E_j^{(\tilde{T})}\right| \leq \tilde{\mathcal{O}}\left(\sqrt{\eta d}\right)$, then our lemma will immediately follow. If $S^c \neq \phi$, we have $\tilde{T} = \min\{T_g, T_f\} = T_g$ and that $\forall j \in S^c : E_j^{(\tilde{T})} < 0$. By the definition of $T_g$, we know that $\exists i_0 \in [d] : \left|g_{2i_0}^{(\tilde{T})}\right| \leq \mathcal{O}\left(d\sqrt{\eta}\right)$. Then

$$\left|\sum_{j\in S^c}^{d} E_j^{(\tilde{T})} W_1^{(\tilde{T})}[i_0,j]\right| = \left|\sum_{j=1}^{d} E_j^{(\tilde{T})} W_1^{(\tilde{T})}[i_0,j] - \sum_{j\in S} E_j^{(\tilde{T})} W_1^{(\tilde{T})}[i_0,j]\right|$$

$$\leq \left|g_{2i_0}^{(\tilde{T})}\right| + \left|\sum_{j\in S} E_j^{(\tilde{T})} W_1^{(\tilde{T})}[i_0,j]\right| \leq \mathcal{O}\left(d\sqrt{\eta}\right) + d\tilde{\mathcal{O}}\left(\sqrt{\eta d}\right)\tilde{\mathcal{O}}\left(\frac{1}{\sqrt{d}}\right) = \tilde{\mathcal{O}}\left(d\sqrt{\eta}\right).$$

By Lemma 25, we know that when $T_1 \leq t < \tilde{T}$, for $\forall i,j \in [d]$, $\left|W_1^{(t)}[i,j]\right| = \tilde{\Omega}\left(\frac{1}{\sqrt{d}}\right)$. Since the update per step $\left|\Delta W_1^{(t)}[i,j]\right| \leq \tilde{\mathcal{O}}(\eta)$, we know that $\text{sign}\left(W_1^{(t)}[i,j]\right)$ remains unchanged during this period and $\text{sign}\left(W_1^{(t)}[i,j]\right) = \text{sign}\left(W_1^{(T_1)}[i,j]\right) = \text{sign}\left(w_{2i}^{(0)}\right)$ independent of $j$. Combining with $\forall j \in S^c : E_j^{(\tilde{T})} < 0$ gives us that $E_j^{(\tilde{T})} W_1^{(\tilde{T})}[i_0,j]$ for different $j$ have the same sign. Therefore for any $j_0 \in S^c$,

$$\tilde{\mathcal{O}}\left(d\sqrt{\eta}\right) \geq \left|\sum_{j\in S^c}^{d} E_j^{(\tilde{T})} W_1^{(\tilde{T})}[i_0,j]\right| = \sum_{j\in S^c}^{d} \left|E_j^{(\tilde{T})} W_1^{(\tilde{T})}[i_0,j]\right| \geq \left|E_{j_0}^{(\tilde{T})} W_1^{(\tilde{T})}[i_0,j_0]\right|$$

$$\geq \left|E_{j_0}^{(\tilde{T})}\right| \tilde{\Omega}\left(\frac{1}{\sqrt{d}}\right) \quad \Rightarrow \left|E_{j_0}^{(\tilde{T})}\right| \leq \tilde{\mathcal{O}}\left(d\sqrt{\eta d}\right).$$

Note that the above inequality holds for any $j_0 \in S^c$, which means $\forall j \in S^c : \left| E_j^{(\tilde{T})} \right| \leq \tilde{\mathcal{O}} \left( d\sqrt{\eta d} \right)$. Combining with the fact that $\forall j \in S : \left| E_j^{(\tilde{T})} \right| \leq \tilde{\mathcal{O}} \left( d\sqrt{\eta d} \right)$ completes the proof.

### E.16 Proof of Lemma 28

Consider certain $j \in [d]$, when $t < \min \left\{ \tilde{T}, T_{f,j} \right\}$, we have that $E_j^{(t)} < -\sqrt{\eta d}$. Therefore we can use the same argument as in Section E.13 to prove that $\left| W_1^{(t)}[i, j] \right|$ keeps increasing, and $\text{sign} \left( W_1^{(t)}[i, j] \right) = \text{sign} \left( w_{2i}^{(0)} \right)$ for all $i \in [d]$.

At time the "flip time" $t = T_{f,j}$, by definition, $E_j^{(t)} \geq -\sqrt{\eta d}$. After that $E_j^{(t)}$ may oscillate. Now we prove that once $E_j^{(t)} \geq \sqrt{\eta d}$ (or $E_j^{(t)} \leq -\sqrt{\eta d}$), after a short period $E_j^{(t)}$ will decrease (or increase) until $E_j^{(t)} \leq \tilde{\mathcal{O}} \left( \sqrt{\eta d} \right)$ (or $E_j^{(t)} \geq -\tilde{\mathcal{O}} \left( \sqrt{\eta d} \right)$). Moreover, during this period, $E_j^{(t)}$ won't change too much.

We first recall that when $T_1 \leq t < \tilde{T}$, Lemma 27 gives us for all $i \in [d]$, $\left| w_{2i}^{(t)} \right| = \tilde{\Theta} \left( \frac{1}{\sqrt{d}} \right)$ and $\left| W_1^{(t)}[i, j] \right| = \tilde{\mathcal{O}} \left( \frac{1}{\sqrt{d}} \right)$. Then eq.(35) we obtained in the first phase analysis still holds, which tells us that the change of $E_j^{(t)}$ per step satisfies $\left| E_j^{(t+1)} - E_j^{(t)} \right| = \tilde{\mathcal{O}} \left( \eta\sqrt{d} \right)$ for all $T_1 \leq t < \tilde{T}$.

We divide the analysis into two cases, based on whether these $E_j^{(t)} \geq \sqrt{\eta d}$ or $E_j^{(t)} \leq -\sqrt{\eta d}$. By Lemma 25, we know that when $T_1 \leq t < \tilde{T}, \forall i \in [d], \left| W_1^{(t)}[i, j] \right| = \tilde{\Omega} \left( \frac{1}{\sqrt{d}} \right)$. Since the update per step $\left| \Delta W_1^{(t)}[i, j] \right| \leq \tilde{\mathcal{O}}(\eta)$, we know that $\text{sign} \left( W_1^{(t)}[i, j] \right)$ remains unchanged during this period and $\text{sign} \left( W_1^{(t)}[i, j] \right) = \text{sign} \left( W_1^{(T_1)}[i, j] \right) = \text{sign} \left( w_{2i}^{(0)} \right)$.

By the analysis of $w_{2i}^{(t)}$ in Lemma 24, we have for all $i \in [d]$, $w_{2i}^{(t+1)} = w_{2i}^{(t)} + \text{sign} \left( w_{2i}^{(0)} \right) \Delta_{2i}^{(t)}$, where $\Delta_{2i}^{(t)} = \eta \left( 1 \pm \tilde{\mathcal{O}} \left( \sqrt{\eta} \right) \right)$.

**Case 1.** Consider some time point $t$ such that $E_j^{(t)} \leq -\sqrt{\eta d}$. Note that for all $i \in [d]$, $\left| g_1^{(t)}[i, j] \right| = \left| w_{2i}^{(t)} E_j^{(t)} \right| = \tilde{\Omega} \left( \sqrt{\eta} \right)$ and that $\text{sign} \left( g_1^{(t)}[i, j] \right) = -\text{sign} \left( w_{2i}^{(t)} \right) = -\text{sign} \left( w_{2i}^{(0)} \right)$. By Lemma 24, for all $i \in [d]$ we have $W_1^{(t+1)}[i, j] = W_1^{(t)}[i, j] + \text{sign} \left( w_{2i}^{(0)} \right) \Delta_1^{(t)}[i, j]$ with $\Delta_1^{(t)}[i, j] = \eta \left( 1 \pm \tilde{\mathcal{O}} \left( \sqrt{\eta} \right) \right)$. That gives us

$$
\begin{aligned}
E_j^{(t+1)} &= \sum_{i=1}^{d} w_{2i}^{(t+1)} W_1^{(t+1)}[i, j] - A_j \\
&= \sum_{i=1}^{d} \left( w_{2i}^{(t)} + \text{sign} \left( w_{2i}^{(0)} \right) \Delta_{2i}^{(t)} \right) \left( W_1^{(t)}[i, j] + \text{sign} \left( w_{2i}^{(0)} \right) \Delta_1^{(t)}[i, j] \right) - A_j \\
&= \sum_{i=1}^{d} \left( w_{2i}^{(t)} W_1^{(t)}[i, j] + \text{sign} \left( w_{2i}^{(0)} \right) \left( w_{2i}^{(t)} \Delta_1^{(t)}[i, j] + \Delta_{2i}^{(t)} W_1^{(t)}[i, j] \right) + \Delta_{2i}^{(t)} \Delta_1^{(t)}[i, j] \right) - A_j \\
&\overset{(i)}{=} E_j^{(t)} + \sum_{i=1}^{d} \left( \left| w_{2i}^{(t)} \right| \Delta_1^{(t)}[i, j] + \Delta_{2i}^{(t)} \left| W_1^{(t)}[i, j] \right| + \Delta_{2i}^{(t)} \Delta_1^{(t)}[i, j] \right), \\
\Rightarrow \quad E_j^{(t+1)} &> E_j^{(t)},
\end{aligned}
$$

where $(i)$ is because $\text{sign}\left(w_{2i}^{(t)}\right) = \text{sign}\left(W_1^{(t)}[i,j]\right) = \text{sign}\left(w_{2i}^{(0)}\right)$. Therefore we have proved that $E_j^{(t)}$ will increase in the next step. After that for $\tau \geq t+1$, as long as $E_j^{(\tau)} \leq -\sqrt{\eta d}$, the above analysis will hold and $E_j^{(\tau)}$ will keep increasing until $E_j^{(\tau)} > -\sqrt{\eta d}$ or we reach $\tilde{T}$.

**Case 2.** Consider some time point $t$ such that $E_j^{(t)} \geq \sqrt{\eta d}$. We will prove that $E_j^{(t)}$ will decrease after a short period, and during this period, the change of it is at most $\tilde{\mathcal{O}}\left(\sqrt{\eta d}\right)$.

By similar arguments as in Case 1, we can get that $W_1^{(t+1)}[i,j] = W_1^{(t)}[i,j] - \text{sign}\left(w_{2i}^{(0)}\right)\Delta_1^{(t)}[i,j]$, where $\Delta_1^{(t)}[i,j] = \eta\left(1 \pm \tilde{\mathcal{O}}\left(\sqrt{\eta}\right)\right)$, Then

$$
\begin{aligned}
E_j^{(t+1)} &= \sum_{i=1}^{d} w_{2i}^{(t+1)} W_1^{(t+1)}[i,j] - A_j \\
&= \sum_{i=1}^{d} \left(w_{2i}^{(t)} + \text{sign}\left(w_{2i}^{(0)}\right)\Delta_{2i}^{(t)}\right)\left(W_1^{(t)}[i,j] - \text{sign}\left(w_{2i}^{(0)}\right)\Delta_1^{(t)}[i,j]\right) - A_j \\
&= \sum_{i=1}^{d} \left(w_{2i}^{(t)} W_1^{(t)}[i,j] - \text{sign}\left(w_{2i}^{(0)}\right)\left(w_{2i}^{(t)}\Delta_1^{(t)}[i,j] - \Delta_{2i}^{(t)} W_1^{(t)}[i,j]\right) - \Delta_{2i}^{(t)}\Delta_1^{(t)}[i,j]\right) - A_j \\
&\overset{(i)}{=} E_j^{(t)} - \sum_{i=1}^{d}\left(\left|w_{2i}^{(t)}\right|\Delta_1^{(t)}[i,j] - \Delta_{2i}^{(t)}\left|W_1^{(t)}[i,j]\right| + \Delta_{2i}^{(t)}\Delta_1^{(t)}[i,j]\right),
\end{aligned}
$$

where $(i)$ is because $\text{sign}\left(w_{2i}^{(t)}\right) = \text{sign}\left(W_1^{(t)}[i,j]\right) = \text{sign}\left(w_{2i}^{(0)}\right)$. $E_j^{(t+1)}$ may not be smaller than $E_j^{(t)}$, but we will show that after at most $t_s$ steps for some $t_s$, we will have $E_j^{(t+t_s+1)} < E_j^{(t+t_s)}$.

To see this, first note that by the bounds of $\Delta_1^{(t)}[i,j]$ and $\Delta_{2i}^{(t)}$, we get $\Delta_1^{(t)}[i,j] \geq \Delta_{2i}^{(t)} - \eta\tilde{\mathcal{O}}\left(\sqrt{\eta}\right)$. Since $\left|w_{2i}^{(t)}\right|$ increases by $\tilde{\Theta}(\eta)$ per step, and $\left|W_1^{(t)}[i,j]\right|$ keeps decreasing, then we have either i) after $t_s$ steps for some $t_s$, $\forall i \in [d]: \left|w_{2i}^{(t+t_s)}\right| \geq \left|W_1^{(t+t_s)}[i,j]\right| + \sqrt{\eta}$ or ii) we reach $\tilde{T}$.

For i), if $E_j^{(t+t_s)} < \sqrt{\eta d}$, then it's already what we want. Otherwise we will have $\Delta_1^{(t+t_s)}[i,j] = \eta\left(1 \pm \tilde{\mathcal{O}}\left(\sqrt{\eta}\right)\right)$. Hence

$$
\begin{aligned}
&E_j^{(t+t_s)} - E_j^{(t+t_s+1)} \\
&= \sum_{i=1}^{d}\left(\left|w_{2i}^{(t+t_s)}\right|\Delta_1^{(t+t_s)}[i,j] - \Delta_{2i}^{(t+t_s)}\left|W_1^{(t+t_s)}[i,j]\right| + \Delta_{2i}^{(t+t_s)}\Delta_1^{(t+t_s)}[i,j]\right) \\
&\geq \sum_{i=1}^{d}\left(\left|W_1^{(t+t_s)}[i,j]\right|\left(\Delta_1^{(t+t_s)}[i,j] - \Delta_{2i}^{(t+t_s)}\right) + \sqrt{\eta}\Delta_1^{(t+t_s)}[i,j] + \Delta_{2i}^{(t+t_s)}\Delta_1^{(t+t_s)}[i,j]\right) \\
&\geq \sum_{i=1}^{d}\left(-\eta\tilde{\mathcal{O}}\left(\sqrt{\eta}\right)\left|W_1^{(t+t_s)}[i,j]\right| + \eta\sqrt{\eta} + \Delta_{2i}^{(t+t_s)}\Delta_1^{(t+t_s)}[i,j]\right) > 0,
\end{aligned}
$$

where the last inequality uses $\forall i,j \in [d]: \left|W_1^{(t+t_s)}[i,j]\right| = \tilde{\mathcal{O}}\left(\frac{1}{\sqrt{d}}\right)$. Therefore $E_j^{(t+t_s+1)} < E_j^{(t+t_s)}$. After that for $\tau \geq t+t_s+1$, as long as $E_j^{(\tau)} \geq \sqrt{\eta d}$, the above analysis will hold and $E_j^{(\tau)}$ will keep decreasing until $E_j^{(\tau)} < \sqrt{\eta d}$ or we reach $\tilde{T}$.

Now we prove that during these $t_s$ steps, the change of $E_j$ is $\tilde{\mathcal{O}}\left(\sqrt{\eta d}\right)$. Since at each step the difference $|w_{2i}| - |W_1[i,j]|$ will be enlarged by $\tilde{\Omega}(\eta)$, then we know that $t_s = \sqrt{\eta}/\tilde{\Omega}(\eta) = \tilde{\mathcal{O}}\left(\frac{1}{\sqrt{\eta}}\right)$.

Combining with the fact that for all $T_1 \le \tau \le \tilde{T}$, $\left| E_j^{(\tau+1)} - E_j^{(\tau)} \right| = \tilde{\mathcal{O}}\left(\eta\sqrt{d}\right)$ gives us

$$E_j^{(t+t_s)} - E_j^{(t)} \le \tilde{\mathcal{O}}\left(\eta t_s \sqrt{d}\right) = \tilde{\mathcal{O}}\left(\sqrt{\eta d}\right).$$

For ii), we reach $\tilde{T}$ before $\forall i \in [d]: \left| w_{2i}^{(t+t_s)} \right| \ge \left| W_1^{(t+t_s)}[i,j] \right| + \sqrt{\eta}$. Then we have $\tilde{T} - t \le \sqrt{\eta}/\tilde{\Omega}(\eta) = \tilde{\mathcal{O}}\left(\frac{1}{\sqrt{\eta}}\right)$, which yields $E_j^{(\tilde{T})} - E_j^{(t)} \le \tilde{\mathcal{O}}\left(\eta(\tilde{T} - t)\sqrt{d}\right) \le \tilde{\mathcal{O}}\left(\sqrt{\eta d}\right)$.

Combining the above two cases, we find that if for some $t$, $E_j^{(t)} \ge \sqrt{\eta d}$, then after at most $t_s$ steps $E_j$ will decrease and keeps decreasing until $E_j < \sqrt{\eta d}$ or we reach $\tilde{T}$. During these steps, $E_j$ can increase at most $\tilde{\mathcal{O}}\left(\sqrt{\eta d}\right)$. If for some $t$, $E_j^{(t)} \le -\sqrt{\eta d}$, then after one step it will increase and keeps increasing until $E_j > \sqrt{\eta d}$ or we reach $\tilde{T}$. That means once for some coordinate $j$, $E_j$ overshoots, it will zigzag in a small region around zero, which is $\left[ -\tilde{\mathcal{O}}\left(\sqrt{\eta d}\right), \tilde{\mathcal{O}}\left(\sqrt{\eta d}\right) \right]$.

# F   Hessian tends to become more and more diagonal during training

In this section, we empirically demonstrate that the trend of loss Hessian in practice is to become more and more diagonal during training. We also give a rigorous theoretical analysis on a two-layer network under Assumption 1 and 2.

## F.1   Empirical Results

Let's first define the diagonal domination of the $i$-th coordinate at time $t$.

$$r_{\text{diag},i}^{\text{OPT}}(t) := \frac{\sqrt{\sum_{j \ne i} \left( H^{(t)}[i,j] \right)^2}}{\left| H^{(t)}[i,i] \right|}.$$

To measure the diagonal domination of the whole Hessian, we need to consider the distribution of $r_{\text{diag},i}^{\text{OPT}}(t)$ for different $i$. Figure 15 shows the mean and median of $r_{\text{diag},i}^{\text{SGDM}}(t)$ and $r_{\text{diag},i}^{\text{Adam}}(t)$ on the sentence classification task (see Section 4.1). Here we chose 4 layers (Layer #6, 12, 17 and 22) and computed the Hessians across these 4 layers. Since the number of parameters is very large, we did the computation by random sampling. As we can see, for both $r_{\text{diag},i}^{\text{SGDM}}(t)$ and $r_{\text{diag},i}^{\text{Adam}}(t)$, the trend of their mean or median is to decrease over time, although there might be some oscillation.

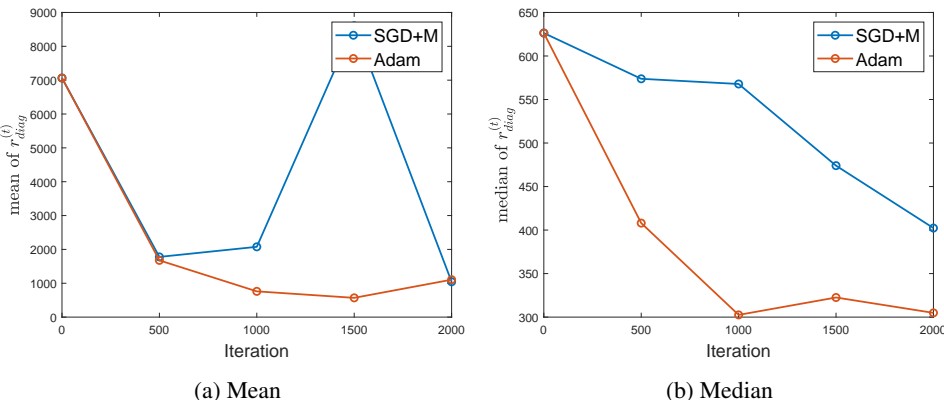

(a) Mean                                        (b) Median

Figure 15: Mean and median of $r_{\text{diag},i}^{\text{SGDM}}(t)$ and $r_{\text{diag},i}^{\text{Adam}}(t)$ for the full hessian across the four layers (#6,12,17,22)

## F.2   Theoretical Analysis

To simplify the theoretical analysis, we consider the mean of $r_{\text{diag},i}^{\text{OPT}}(t)$ over all coordinate and define

$$R_{\text{diag}}^{\text{OPT}}(t) := \text{mean}\left( r_{\text{diag},i}^{\text{OPT}}(t) \right). \tag{40}$$

We consider a 2-layer network under Assumption 1 and 2, and have two goals in our proof:

1. To show that $R_{\text{diag}}^{\text{OPT}}(t)$ after training is smaller than that before training ($t = 0$).

2. Note that in our setting (see in Assumption 1), the Hessian is a $(d^2+d) \times (d^2+d)$ matrix. For a completely "uniform" matrix with the same size, we have that $R_{\text{diag}}^{\text{OPT}}(t) = \Theta\left(\sqrt{d^2 + d}\right) = \Theta(d)$. Hence our second goal is to show that the $R_{\text{diag}}^{\text{OPT}}(t)$ after training is on lower order than $\Theta(d)$.

**Theorem 2.** *Consider the ratio $R_{diag}^{OPT}(t)$ defined in eq. (40). Under Assumption 1 and 2, we have that before training ($t = 0$), with high probability,*

$$R_{diag}^{OPT}(0) \geq \tilde{\Omega}\left(d^{4\alpha - \frac{3}{2}}\right). \tag{41}$$

*For SGD+M defined in eq. (3). For any $p > 0$, by picking the same hyperparameters as in Theorem 1, for $T_{SGD,1}, T_{SGD,2}$ mentioned in Theorem 1, we have with constant probability, for any $t \in [T_{SGD,1}, T_{SGD,2}]$,*

$$R_{diag}^{SGDM}(t) \leq \tilde{\mathcal{O}}\left(\sqrt{d}\right) + q^{(t)}, \tag{42}$$

*where the trend of $q^{(t)}$ is to decrease over time and $q^{(T_{SGD,2})} \leq \tilde{\mathcal{O}}\left(\frac{1}{d^{p/2-1}}\right) = o(d)$.*

*For Adam defined in eq. (3). For any $p > 0$, by picking the same hyperparameters as in Theorem 1, for $T_{Adam,1}, T_{Adam,2}$ mentioned in Theorem 1, we have with high probability, for any $t \in [T_{Adam,1}, T_{Adam,2}]$,*

$$R_{diag}^{Adam}(t) \leq \tilde{\mathcal{O}}\left(\sqrt{d}\right) + r^{(t)}, \tag{43}$$

*where the trend of $r^{(t)}$ is to decrease over time and $r^{(T_{Adam,2})} \leq \tilde{\mathcal{O}}\left(\frac{1}{d^{\frac{p-1}{2}}}\right) = o\left(\sqrt{d}\right)$.*

### F.3 Proof of Theorem 2

Lemma 4.3 of [Kaw16] gives us the following forms of Hessian.

For any $k \in \{1, 2, ..., H + 1\}$, we know that $\nabla_{vec(W_k)}(\nabla_{vec(W_k)}L(W))$ equals

$$((W_{H+1}\ldots W_{k+1})^T(W_{H+1}\ldots W_{k+1}) \otimes (W_{k-1}\ldots W_1)(W_{k-1}\ldots W_1)^T,$$

and for $k \in \{2, 3, ..., H + 1\}$,

$$\nabla_{vec(W_k)}(\nabla_{vec(W_1)}L(W))$$
$$=(C^T(W_{H+1}\ldots W_{k+1}) \otimes (W_{k-1}\ldots W_1)^T)$$
$$+[(W_{k-1}\ldots W_2)^T \otimes I][I_{d_{k-1}} \otimes (r(W_{H+1}\ldots W_{k+1}))_{.,1} \cdots I_{d_{k-1}} \otimes (r(W_{H+1}\ldots W_{k+1}))_{.,d_k}],$$

where $r = (W_{H+1}\ldots W_1 - A)^T, C = W_{H+1}W_H \cdots W_2$.

For the 2-layer linear network, write the Hessian as

$$H := \left[\begin{array}{cc} H_{22} & H_{21}^T \\ H_{21} & H_{11} \end{array}\right],$$

then we have that

$$H_{11} = (W_2^T W_2) \otimes I_d \in \mathbb{R}^{d^2 \times d^2},$$
$$H_{22} = W_1 W_1^T \in \mathbb{R}^{d \times d},$$
$$H_{21} = W_2^T \otimes W_1^T + I_d \otimes (W_2 W_1 - A)^T \in \mathbb{R}^{d^2 \times d}.$$

Intuitively, before training the elements of $W_1$ and $W_2$ are very close to zero, and $W_2 W_1 - A \approx -A$. Since the elements of $A$ are $\Theta(1)$, we know that the magnitudes of elements of $H_{21}$ are much bigger than those of $H_{11}$ and $H_{22}$.

After training, for both SGD+M and Adam, $W_2 W_1 - A \approx 0$. Then $H_{21} \approx (W_2)^T \otimes (W_1)^T$ and the magnitudes of its elements are no longer much larger than those of $H_{11}$ and $H_{22}$. From the formula of $H_{11}$, we know that all the diagonal entries are nonzero, and among the $d^4 - d^2$ off-diagonal entries, there are only $d^3 - d^2$ nonzero entries, which helps us to bound $R_{\text{diag}}^{\text{OPT}}(t)$.

### F.3.1 Proof of eq. (41)

Let's first analyze the weights and Hessian before training ($t = 0$). For ease of notation, we omit the superscript $(t)$.

For the $i$-th row where $1 \le i \le d$, i.e. the $i$-th row of the submatrix $[H_{22} \quad H_{21}^T]$, we have

$$\sum_{j \neq i} H^2[i,j] = \sum_{j \neq i} H_{22}^2[i,j] + \sum_{j=1}^{d^2} H_{21}^2[j,i]$$

$$\ge \sum_{j=(i-1)d}^{id} H_{21}^2[j,i] = \sum_{j=1}^{d} \left(w_{2i} W_1[i,j] + (W_2 W_1 - A)_j\right)^2 = \Theta(d).$$

On the other hand, for the diagonal elements, we have w.h.p.

$$|H[i,i]| = |H_{22}[i,i]| = \|W_1[i,:]\|_2^2 = \sum_{j=1}^{d} W_1^2[i,j] \le \tilde{\mathcal{O}}\left(\frac{1}{d^{4\alpha-1}}\right).$$

Then we have that for $1 \le i \le d$,

$$\frac{\sqrt{\sum_{j \neq i} H^2[i,j]}}{|H[i,i]|} \ge \frac{\sqrt{\Omega(d)}}{\tilde{\mathcal{O}}\left(\frac{1}{d^{4\alpha-1}}\right)} = \tilde{\Omega}\left(d^{4\alpha-\frac{1}{2}}\right).$$

For the $(id + k)$-th row where $1 \le i \le d, 1 \le k \le d$, i.e. the $((i-1)d + k)$-th row of the submatrix $[H_{21} \quad H_{11}]$, we have

$$\sum_{j \neq id+k} H^2[i,j] = \sum_{j \neq (i-1)d+k} H_{11}^2[(i-1)d+k,j] + \sum_{j=1}^{d} H_{21}^2[(i-1)d+k,j]$$

$$\ge H_{21}^2[(i-1)d+k,i] = \left(w_{2i} W_1[i,k] + (W_2 W_1 - A)_k\right)^2 = \Theta(1).$$

On the other hand, for the diagonal elements, we have w.h.p.

$$|H[id+k,id+k]| = |H_{11}[(i-1)d+k,(i-1)d+k]| = w_{2i}^2 \le \tilde{\mathcal{O}}\left(\frac{1}{d^{2\alpha}}\right).$$

Then we have that for $1 \le i \le d, 1 \le k \le d$,

$$\frac{\sqrt{\sum_{j \neq id+k} H^2[i,j]}}{|H[id+k,id+k]|} \ge \frac{\sqrt{\Omega(1)}}{\tilde{\mathcal{O}}\left(\frac{1}{d^{2\alpha}}\right)} = \tilde{\Omega}\left(d^{2\alpha}\right).$$

Taking the average, we obtain that before training, i.e. when $t = 0$,

$$R_{\text{diag}}^{\text{OPT}}(0) \ge \frac{d\tilde{\Omega}\left(d^{4\alpha-\frac{1}{2}}\right) + d^2\tilde{\Omega}\left(d^{2\alpha}\right)}{d^2 + d} = \tilde{\Omega}\left(d^{4\alpha-\frac{3}{2}}\right).$$

### F.3.2 Proof of eq. (42)

The proof is based on the lemma below.

**Lemma 29.** *Suppose the weight matrices have the following structure:*

$$W_1 = \boldsymbol{u}\boldsymbol{v}^T + R_1,$$
$$W_2 = c\boldsymbol{u}^T + R_2^T,$$

*where $\forall 1 \le i, j \le d : \quad \frac{|R_1[i,j]|}{|u_i v_j|} \le \delta, \quad \frac{|R_{2i}|}{|cu_i|} \le \delta, \quad \delta \in (0,1)$.*

*Then we have for $1 \le i \le d$,*

$$\frac{\sqrt{\sum_{j \neq i} H^2[i,j]}}{|H[i,i]|} \le \frac{1+\delta}{1-\delta}\left(1 + \frac{|c|}{\|\boldsymbol{v}\|_2}\right)\sqrt{\frac{\sum_{j=1}^{d} u_j^2}{u_i^2}} + \frac{\|E\|_2}{(1-\delta)^2 u_i^2 \|v\|_2^2},$$

*and for $1 \le i \le d, 1 \le k \le d$,*

$$\frac{\sqrt{\sum_{j \neq id+k} H^2[i,j]}}{|H[id+k,id+k]|} \le \frac{1+\delta}{1-\delta}\left(1 + \frac{|v_k|}{|c|}\right)\sqrt{\frac{\sum_{j=1}^{d} u_j^2}{u_i^2}} + \frac{|E_k|}{(1-\delta)^2 c^2 u_i^2}.$$

Now we are ready to prove eq. (42).

By the analyses in Section D.1, we know that for $t \in [T_{\text{SGD},1}, T_{\text{SGD},2}]$, the weights obtained by GD with momentum satisfy

$$W_1^{(t)} = \boldsymbol{u}^{(T_1)} \boldsymbol{v}^{(t)T} + R_1^{(t)},$$
$$W_2^{(t)} = c^{(t)} \boldsymbol{u}^{(T_1)T} + R_2^{(t)T},$$

where $T_{\text{SGD},1} = T_1$ and

$$\forall 1 \leq i, j \leq d : \quad \frac{\left| R_1^{(t)}[i,j] \right|}{\left| u_i^{(T_1)} v_j^{(t)} \right|} \leq \tilde{\mathcal{O}}(\epsilon_0), \quad \frac{\left| R_{2i}^{(t)} \right|}{\left| c^{(t)} u_i^{(T_1)} \right|} \leq \tilde{\mathcal{O}}(\epsilon_0).$$

Here $\epsilon_0$ is defined in Definition 2. Since $\boldsymbol{u}^{(T_1)}$ doesn't depend on time $t$ in the period $(T_{\text{SGD},1}, T_{\text{SGD},2}]$, we write $\boldsymbol{u}^{(T_1)}$ as $\boldsymbol{u}$ for ease of notation.

Hence by Lemma 29, when $t \in [T_{\text{SGD},1}, T_{\text{SGD},2}]$, we have for $1 \leq i \leq d$,

$$\frac{\sqrt{\sum_{j \neq i} \left( H^{(t)}[i,j] \right)^2}}{\left| H^{(t)}[i,i] \right|} \leq \frac{1 + \tilde{\mathcal{O}}(\epsilon_0)}{1 - \tilde{\mathcal{O}}(\epsilon_0)} \left( 1 + \frac{\left| c^{(t)} \right|}{\left\| \boldsymbol{v}^{(t)} \right\|_2} \right) \sqrt{\frac{\sum_{j=1}^{d} u_j^2}{u_i^2}} + \frac{\left\| E^{(t)} \right\|_2}{\left( 1 - \tilde{\mathcal{O}}(\epsilon_0) \right)^2 u_i^2 \left\| \boldsymbol{v}^{(t)} \right\|_2^2}$$

$$= \mathcal{O} \left( 1 + \frac{\left| c^{(t)} \right|}{\left\| \boldsymbol{v}^{(t)} \right\|_2} \right) \sqrt{\frac{\sum_{j=1}^{d} u_j^2}{u_i^2}} + \mathcal{O} \left( \frac{\left\| E^{(t)} \right\|_2}{u_i^2 \left\| \boldsymbol{v}^{(t)} \right\|_2^2} \right),$$

(44)

and for $1 \leq i \leq d, 1 \leq k \leq d$,

$$\frac{\sqrt{\sum_{j \neq id+k} \left( H^{(t)}[i,j] \right)^2}}{\left| H^{(t)}[id+k, id+k] \right|} \leq \frac{1 + \tilde{\mathcal{O}}(\epsilon_0)}{1 - \tilde{\mathcal{O}}(\epsilon_0)} \left( 1 + \frac{\left| v_k^{(t)} \right|}{\left| c^{(t)} \right|} \right) \sqrt{\frac{\sum_{j=1}^{d} u_j^2}{u_i^2}} + \frac{\left| E_k^{(t)} \right|}{\left( 1 - \tilde{\mathcal{O}}(\epsilon_0) \right)^2 \left( c^{(t)} \right)^2 u_i^2}$$

$$= \mathcal{O} \left( 1 + \frac{\left| v_k^{(t)} \right|}{\left| c^{(t)} \right|} \right) \sqrt{\frac{\sum_{j=1}^{d} u_j^2}{u_i^2}} + \mathcal{O} \left( \frac{\left| E_k^{(t)} \right|}{\left( c^{(t)} \right)^2 u_i^2} \right).$$

(45)

By Lemma 4, we have $\boldsymbol{u} = X + Y$ where $X_i, i \in [d]$ are i.i.d Gaussian random variables and w.h.p.,

$$\forall i \in [d] : \frac{|Y_i|}{|X_i|} \leq \tilde{\mathcal{O}} \left( \frac{1}{d^{\frac{1}{4}\alpha - \frac{1}{2}}} \right) := \delta_{xy},$$

(46)

which yields that

$$\forall i \in [d] : \frac{\sqrt{\sum_{j=1}^{d} u_j^2}}{|u_i|} \leq \left( \frac{1 + \delta_{xy}}{1 - \delta_{xy}} \right) \frac{\sqrt{\sum_{j=1}^{d} X_j^2}}{|X_i|}, \quad \frac{1}{u_i^2} \leq \left( \frac{1}{1 - \delta_{xy}} \right)^2 \frac{1}{X_i^2}.$$

(47)

By the proof in Section D.8, we know that for $t \in [T_{\text{SGD},1}, T_{\text{SGD},2}], \forall i \in [d] : v_i^{(t)}, c^{(t)}$ are positive. The induction in Section D.9 further gives us that for $t \in [T_{\text{SGD},1}, T_{\text{SGD},2}]$, w.h.p. $\forall k \in [d] : \frac{v_k^{(t)}}{c^{(t)}} = \Theta \left( \frac{1}{\sqrt{d}} \right)$, which yields $\frac{c^{(t)}}{\left\| \boldsymbol{v}^{(t)} \right\|_2} = \Theta(1)$. Combining with eq. (47), we obtain

$$\left( 1 + \frac{\left| c^{(t)} \right|}{\left\| \boldsymbol{v}^{(t)} \right\|_2} \right) \sqrt{\frac{\sum_{j=1}^{d} u_j^2}{u_i^2}} \leq \mathcal{O} \left( \frac{\sqrt{\sum_{j=1}^{d} X_j^2}}{|X_i|} \right),$$

$$\left( 1 + \frac{\left| v_k^{(t)} \right|}{\left| c^{(t)} \right|} \right) \sqrt{\frac{\sum_{j=1}^{d} u_j^2}{u_i^2}} \leq \mathcal{O} \left( \frac{\sqrt{\sum_{j=1}^{d} X_j^2}}{|X_i|} \right).$$

(48)

By the proof in Section D.8, we know that for $t \in [T_{\text{SGD},1}, T_{\text{SGD},2}], \forall i \in [d] : v_i^{(t)}, c^{(t)}$ are positive and monotonically increasing. On the other hand, the proof in Section D.2 and D.9 tells us that w.h.p. $\left\|E^{(t)}\right\|_2$ (resp. $\forall k \in [d], \left|E_k^{(t)}\right|$) decreases from $\Theta(\sqrt{d})$ (resp. $\Theta(1)$) when $t = T_{\text{SGD},1}$ to $\mathcal{O}(\sqrt{\epsilon_0 d})$ (resp. $\mathcal{O}(\sqrt{\epsilon_0})$) when $t = T_{\text{SGD},2}$. Therefore, the trend of $\frac{\left\|E^{(t)}\right\|_2}{u_i^2 \left\|\boldsymbol{v}^{(t)}\right\|_2^2}$ and $\frac{\left|E_k^{(t)}\right|}{\left(c^{(t)}\right)^2 u_i^2}$ is to decrease over time, and when $t = T_{\text{SGD},2}$, we have w.h.p.

$$\forall k \in [d] : \left|E_k^{(t)}\right| = \mathcal{O}\left(\sqrt{\epsilon_0}\right), \quad \left\|E^{(t)}\right\|_2 = \mathcal{O}\left(\sqrt{\epsilon_0 d}\right). \tag{49}$$

Moreover, when $t = T_{\text{SGD},2}$, the inequality in eq. (26) becomes equality, i.e. $c^2 \|\boldsymbol{u}\|_2^2 = \Theta\left(\sqrt{d}\right)$ and $\forall j \in [d] : \|\boldsymbol{u}\|_2^2 v_j^2 = \Theta\left(\frac{1}{\sqrt{d}}\right)$.

Using $\boldsymbol{u} = X + Y$ and eq. (46), we have

$$c^2 \|X\|_2^2 = \Theta\left(\sqrt{d}\right), \quad \forall j \in [d] : \|X\|_2^2 v_j^2 \Theta\left(\frac{1}{\sqrt{d}}\right), \quad \Rightarrow \quad \|X\|_2^2 \|\boldsymbol{v}\|_2^2 = \Theta\left(\sqrt{d}\right),$$

which together with the second inequality in eq. (47) yields

$$\frac{1}{u_i^2 \|\boldsymbol{v}\|_2^2} \leq \left(\frac{1}{1 - \delta_{xy}}\right)^2 \frac{1}{X_i^2 \|\boldsymbol{v}\|_2^2} = \Theta\left(\frac{\sum_{j=1}^d X_j^2}{X_i^2 \sqrt{d}}\right),$$

$$\frac{1}{c^2 u_i^2} \leq \left(\frac{1}{1 - \delta_{xy}}\right)^2 \frac{1}{c^2 X_i^2} = \Theta\left(\frac{\sum_{j=1}^d X_j^2}{X_i^2 \sqrt{d}}\right).$$

Combining with eq. (49), we get that

$$\frac{\left\|E^{(t)}\right\|_2}{u_i^2 \left\|\boldsymbol{v}^{(t)}\right\|_2^2} \leq \mathcal{O}\left(\frac{\sum_{j=1}^d X_j^2}{X_i^2} \cdot \sqrt{\epsilon_0}\right), \quad \frac{\left|E_k^{(t)}\right|}{\left(c^{(t)}\right)^2 u_i^2} \leq \mathcal{O}\left(\frac{\sum_{j=1}^d X_j^2}{X_i^2} \cdot \sqrt{\frac{\epsilon_0}{d}}\right). \tag{50}$$

Substituting eq. (48) and (50) into eq. (44) and (45) gives us

$$\forall 1 \leq i \leq d : \frac{\sqrt{\sum_{j \neq i} \left(H^{(t)}[i,j]\right)^2}}{\left|H^{(t)}[i,i]\right|} \leq \mathcal{O}\left(\frac{\sqrt{\sum_{j=1}^d X_j^2}}{|X_i|}\right) + q_{1i}^{(t)},$$

where the trend of $q_{1i}^{(t)}$ is to decrease over time and $q_{1i}^{(T_{\text{SGD},2})} \leq \mathcal{O}\left(\frac{\sum_{j=1}^d X_j^2}{X_i^2} \cdot \sqrt{\epsilon_0}\right)$.

We also have

$$\forall 1 \leq i \leq d, 1 \leq k \leq d : \frac{\sqrt{\sum_{j \neq id+k} \left(H^{(t)}[i,j]\right)^2}}{\left|H^{(t)}[id+k, id+k]\right|} \leq \mathcal{O}\left(\frac{\sqrt{\sum_{j=1}^d X_j^2}}{|X_i|}\right) + q_{2i}^{(t)},$$

where the trend of $q_{2i}^{(t)}$ is to decrease over time and $q_{2i}^{(T_{\text{SGD},2})} \leq \mathcal{O}\left(\frac{\sum_{j=1}^d X_j^2}{X_i^2} \cdot \sqrt{\frac{\epsilon_0}{d}}\right)$.

Hence

$$R_{\text{diag}}^{\text{SGDM}}(t) = \mathcal{O}\left(\frac{1}{d} \sum_{i=1}^d \frac{\sqrt{\sum_{j=1}^d X_j^2}}{|X_i|}\right) + \frac{1}{d^2 + d} \sum_{i=1}^d q_{1i}^{(t)} + \frac{d}{d^2 + d} \sum_{i=1}^d q_{2i}^{(t)}$$

$$:= \mathcal{O}\left(\frac{1}{d} \sum_{i=1}^d \frac{\sqrt{\sum_{j=1}^d X_j^2}}{|X_i|}\right) + q^{(t)},$$

where the trend of $q^{(t)}$ is to decrease over time and

$$q^{(T_{\text{SGD},2})} \le \frac{1}{d^2+d} \sum_{i=1}^{d} \mathcal{O}\left(\frac{\sum_{j=1}^{d} X_j^2}{X_i^2} \cdot \sqrt{\epsilon_0}\right) + \frac{d}{d^2+d} \sum_{i=1}^{d} \mathcal{O}\left(\frac{\sum_{j=1}^{d} X_j^2}{X_i^2} \cdot \sqrt{\frac{\epsilon_0}{d}}\right)$$

$$\le \mathcal{O}\left(\frac{1}{d^2+d} \sum_{i=1}^{d} \frac{\sum_{j=1}^{d} X_j^2}{X_i^2} \cdot \sqrt{\epsilon_0 d}\right) = \mathcal{O}\left(\frac{1}{d} \sum_{i=1}^{d} \frac{\sum_{j=1}^{d} X_j^2}{X_i^2} \cdot \sqrt{\frac{\epsilon_0}{d}}\right).$$

Denote $\sigma^2$ as the variance of $X_i$ for $i \in [d]$. By concentration of chi-squared distribution, we know that with probability at least $1 - \delta$ for $\delta > 0$,

$$\sum_{i=1}^{d} X_i^2 \le \sigma^2 d + \sigma^2 \mathcal{O}\left(\sqrt{d \log \frac{1}{\delta}}\right).$$

By Lemma 36 in Appendix H, we know that with constant probability $\frac{1}{d} \sum_{i=1}^{d} \frac{1}{|X_i|} = \mathcal{O}\left(\frac{1}{\sigma} \log d\right)$. Then with constant probability, $\frac{1}{d} \sum_{i=1}^{d} \frac{1}{X_i^2} \le \frac{1}{d} \left(\sum_{i=1}^{d} \frac{1}{|X_i|}\right)^2 = \mathcal{O}\left(\frac{d}{\sigma^2} \log^2 d\right)$. Hence

$$\frac{1}{d} \sum_{i=1}^{d} \frac{\sqrt{\sum_{j=1}^{d} X_j^2}}{|X_i|} = \tilde{\mathcal{O}}(\sqrt{d}), \quad \frac{1}{d} \sum_{i=1}^{d} \frac{\sum_{j=1}^{d} X_j^2}{X_i^2} = \tilde{\mathcal{O}}\left(d^2\right).$$

Therefore with constant probability,

$$R_{\text{diag}}^{\text{SGDM}}(t) = \tilde{\mathcal{O}}\left(\sqrt{d}\right) + q^{(t)},$$

where the trend of $q^{(t)}$ is to decrease over time and $q^{(T_{\text{SGD},2})} \le \tilde{\mathcal{O}}\left(d\sqrt{\epsilon_0 d}\right)$. For any $p > 0$, by picking the same hyperparameters as in Theorem 1, we have $\epsilon_0 d \le \tilde{\mathcal{O}}\left(\frac{1}{d^p}\right)$ and hence $q^{(T_{\text{SGD},2})} \le \tilde{\mathcal{O}}\left(\frac{1}{d^{p/2-1}}\right) = o(d)$.

### F.3.3 Proof of eq. (43)

By the analyses in Section E.1, we know that for $t \in [T_{\text{Adam},1}, T_{\text{Adam},2}]$, the weights obtained by Adam satisfy

$$W_1^{(t)} = \boldsymbol{u}\boldsymbol{v}^{(t)T} + R_1^{(t)},$$
$$W_2^{(t)} = c^{(t)}\boldsymbol{u}^T + R_2^{(t)T},$$

where $\forall i \in [d] : u_i = \text{sign}(w_{2i}^{(0)}) \in \{\pm 1\}$ and

$$\forall 1 \le i, j \le d: \quad \frac{\left|R_1^{(t)}[i,j]\right|}{\left|u_i v_j^{(t)}\right|} \le \delta := \tilde{\mathcal{O}}\left(\eta^{\frac{1}{4}} + \frac{1}{d^{\frac{\alpha}{2}-\frac{1}{4}}}\right), \quad \frac{\left|R_{2i}^{(t)}\right|}{\left|c^{(t)} u_i\right|} \le \delta.$$

Hence by Lemma 29, when $t \in [T_{\text{Adam},1}, T_{\text{Adam},2}]$, we have for $1 \le i \le d$,

$$\frac{\sqrt{\sum_{j\neq i} \left(H^{(t)}[i,j]\right)^2}}{\left|H^{(t)}[i,i]\right|} \le \frac{1+\delta}{1-\delta}\left(1 + \frac{\left|c^{(t)}\right|}{\|\boldsymbol{v}^{(t)}\|_2}\right) \sqrt{\frac{\sum_{j=1}^{d} u_j^2}{u_i^2} + \frac{\|E^{(t)}\|_2}{(1-\delta)^2 u_i^2 \|\boldsymbol{v}^{(t)}\|_2^2}}$$
$$= \mathcal{O}\left(1 + \frac{\left|c^{(t)}\right|}{\|\boldsymbol{v}^{(t)}\|_2}\right) \sqrt{d} + \mathcal{O}\left(\frac{\|E^{(t)}\|_2}{\|\boldsymbol{v}^{(t)}\|_2^2}\right), \tag{51}$$

and for $1 \le i \le d, 1 \le k \le d$,

$$\frac{\sqrt{\sum_{j\neq id+k} \left(H^{(t)}[i,j]\right)^2}}{\left|H^{(t)}[id+k, id+k]\right|} \le \frac{1+\delta}{1-\delta}\left(1 + \frac{\left|v_k^{(t)}\right|}{\left|c^{(t)}\right|}\right) \sqrt{\frac{\sum_{j=1}^{d} u_j^2}{u_i^2} + \frac{\left|E_k^{(t)}\right|}{(1-\delta)^2 \left(c^{(t)}\right)^2 u_i^2}}$$
$$= \mathcal{O}\left(1 + \frac{\left|v_k^{(t)}\right|}{\left|c^{(t)}\right|}\right) \sqrt{d} + \mathcal{O}\left(\frac{\left|E_k^{(t)}\right|}{\left(c^{(t)}\right)^2}\right). \tag{52}$$

Recall the following facts of Adam.

(A) By Lemma 15, we know that for $t \in [T_{\text{Adam},1}, T_1]$ (where $T_1$ is defined in Definition 4), w.h.p. $\forall k \in [d] : v_k^{(t)} = c^{(t)} = \eta(t - t_{\text{inc}})$. Specially, when $t = T_{\text{Adam},1}$, $\forall k \in [d] :$ $v_k^{(t)} = c^{(t)} = \frac{1}{d^{\frac{\alpha}{2}}}$. Lemma 25 and 27 tell us that for $t \in [T_1, T_{\text{Adam},2}]$ w.h.p. $\forall i, j \in [d] :$ $|W_1[i,j]| = \tilde{\Theta}\left(\frac{1}{\sqrt{d}}\right), |w_{2i}| = \tilde{\Theta}\left(\frac{1}{\sqrt{d}}\right)$, which gives us $\forall k \in [d] : \left|v_k^{(t)}\right| = \tilde{\Theta}\left(\frac{1}{\sqrt{d}}\right)$ and $\left|c^{(t)}\right| = \tilde{\Theta}\left(\frac{1}{\sqrt{d}}\right)$. That means when $t \in [T_{\text{Adam},1}, T_{\text{Adam},2}]$, $\forall k \in [d] : \left|v_k^{(t)}\right|$ and $\left|c^{(t)}\right|$ increase from $\frac{1}{d^{\frac{\alpha}{2}}}$ to $\tilde{\Theta}(\frac{1}{\sqrt{d}})$ and $\frac{\left|v_k^{(t)}\right|}{\left|c^{(t)}\right|} = \tilde{\Theta}(1)$, $\frac{\left|c^{(t)}\right|}{\left\|v^{(t)}\right\|_2} = \tilde{\Theta}\left(\frac{1}{\sqrt{d}}\right)$.

(B) Lemma 15 and 26 tell us that w.h.p. $\left\|E^{(t)}\right\|_2$ (resp. $\forall k \in [d], \left|E_k^{(t)}\right|$) decreases from $\Theta(d)$ (resp. $\Theta(1)$) when $t = T_{\text{Adam},1}$ to $\tilde{\mathcal{O}}\left(d^2\sqrt{\eta}\right)$ (resp. $\tilde{\mathcal{O}}\left(d\sqrt{\eta d}\right)$) when $t = T_{\text{Adam},2}$.

Combining (A) and (B), we get that the trend of $\frac{\left\|E^{(t)}\right\|_2}{\left\|v^{(t)}\right\|_2^2}$ and $\frac{\left|E_k^{(t)}\right|}{\left(c^{(t)}\right)^2}$ is to decrease over time, and when $t = T_{\text{Adam},2}$, we have w.h.p.

$$\frac{\left\|E^{(t)}\right\|_2}{\left\|v^{(t)}\right\|_2^2} \leq \tilde{\mathcal{O}}\left(d^2\sqrt{\eta}\right), \quad \frac{\left|E_k^{(t)}\right|}{\left(c^{(t)}\right)^2} \leq \tilde{\mathcal{O}}\left(d^2\sqrt{\eta d}\right). \tag{53}$$

Substituting (A) and eq. (53) into eq. (51) and (52) gives us w.h.p.,

$$\forall 1 \leq i \leq d : \frac{\sqrt{\sum_{j\neq i}\left(H^{(t)}[i,j]\right)^2}}{\left|H^{(t)}[i,i]\right|} \leq \mathcal{O}\left(\sqrt{d}\right) + r_{1i}^{(t)},$$

where the trend of $r_{1i}^{(t)}$ is to decrease over time and $r_{1i}^{(T_{\text{Adam},2})} \leq \tilde{\mathcal{O}}\left(d^2\sqrt{\eta}\right)$.
We also have

$$\forall 1 \leq i \leq d, 1 \leq k \leq d : \frac{\sqrt{\sum_{j\neq id+k}\left(H^{(t)}[i,j]\right)^2}}{\left|H^{(t)}[id+k, id+k]\right|} \leq \tilde{\mathcal{O}}\left(\sqrt{d}\right) + r_{2i}^{(t)},$$

where the trend of $r_{2i}^{(t)}$ is to decrease over time and $r_{2i}^{(T_{\text{Adam},2})} \leq \tilde{\mathcal{O}}\left(d^2\sqrt{\eta d}\right)$.

Hence $R_{\text{diag}}^{\text{Adam}}(t) = \tilde{\mathcal{O}}\left(\sqrt{d}\right) + \frac{1}{d^2+d}\sum_{i=1}^{d} r_{1i}^{(t)} + \frac{d}{d^2+d}\sum_{i=1}^{d} r_{2i}^{(t)} := \tilde{\mathcal{O}}\left(\sqrt{d}\right) + r^{(t)}$ where the trend of $r^{(t)}$ is to decrease over time and

$$r^{(T_{\text{Adam},2})} \leq \frac{1}{d^2+d}\sum_{i=1}^{d}\tilde{\mathcal{O}}\left(d^2\sqrt{\eta}\right) + \frac{d}{d^2+d}\sum_{i=1}^{d}\tilde{\mathcal{O}}\left(d^2\sqrt{\eta d}\right) \leq \tilde{\mathcal{O}}\left(d^2\sqrt{\eta d}\right).$$

For any $p > 0$, by picking the same hyperparameters as in Theorem 1, we have $\eta d^4 \leq \tilde{\mathcal{O}}\left(\frac{1}{d^p}\right)$ and hence $r^{(T_{\text{Adam},2})} \leq \tilde{\mathcal{O}}\left(\frac{1}{d^{\frac{p-1}{2}}}\right) = o\left(\sqrt{d}\right)$.

### F.4 Proof of Lemma 29

By the assumed weight structure, we get that

$$\forall i \in [d] : (1-\delta)^2(cu_i)^2 \leq (w_{2i})^2 \leq (1+\delta)^2(cu_i)^2,$$
$$(1-\delta)^2(u_i)^2\|v\|_2^2 \leq \|W_1[i,:]\|_2^2 \leq (1+\delta)^2(u_i)^2\|v\|_2^2.$$

For the $i$-th row where $1 \le i \le d$, i.e. the $i$-th row of the submatrix $[H_{22} \quad H_{21}^T]$, by triangle inequality, we have

$$\sqrt{\sum_{j \ne i} H^2[i,j]} \le \sqrt{\sum_{j \ne i} H_{22}^2[i,j]} + \sqrt{\sum_{j=1}^{d^2} H_{21}^2[j,i]}$$

$$\le \sqrt{\sum_{j \ne i} \langle W_1[i,:], W_1[j,:] \rangle^2} + \sqrt{\sum_{j=1}^{d} w_{2j}^2 \sum_{k=1}^{d} W_1^2[i,k]} + \|E\|_2$$

$$\le \|W_1[i,:]\|_2 \left( \sqrt{\sum_{j \ne i} \|W_1[j,:]\|_2^2} + \sqrt{\sum_{j=1}^{d} w_{2j}^2} \right) + \|E\|_2.$$

Then we have that for $1 \le i \le d$,

$$\frac{\sqrt{\sum_{j \ne i} H^2[i,j]}}{|H[i,i]|} \le \frac{\|W_1[i,:]\|_2 \sqrt{\sum_{j \ne i} \|W_1[j,:]\|_2^2} + \sqrt{\sum_{j=1}^{d} w_{2j}^2}}{\|W_1[i,:]\|_2^2} + \frac{\|E\|_2}{\|W_1[i,:]\|_2^2}$$

$$= \sqrt{\frac{\sum_{j \ne i} \|W_1[j,:]\|_2^2}{\|W_1[i,:]\|_2^2}} + \sqrt{\frac{\sum_{j=1}^{d} w_{2j}^2}{\|W_1[i,:]\|_2^2}} + \frac{\|E\|_2}{\|W_1[i,:]\|_2^2}$$

$$\le \sqrt{\frac{(1+\delta)^2}{(1-\delta)^2} \cdot \frac{\sum_{j \ne i} u_j^2 \|\boldsymbol{v}\|_2^2}{u_i^2 \|\boldsymbol{v}\|_2^2}} + \sqrt{\frac{(1+\delta)^2}{(1-\delta)^2} \cdot \frac{c^2 \sum_{j=1}^{d} u_j^2}{u_i^2 \|\boldsymbol{v}\|_2^2}} + \frac{\|E\|_2}{(1-\delta)^2 u_i^2 \|v\|_2^2}$$

$$\le \frac{1+\delta}{1-\delta} \left( 1 + \frac{|c|}{\|\boldsymbol{v}\|_2} \right) \sqrt{\frac{\sum_{j=1}^{d} u_j^2}{u_i^2}} + \frac{\|E\|_2}{(1-\delta)^2 u_i^2 \|v\|_2^2}.$$

For the $(id+k)$-th row where $1 \le i \le d, 1 \le k \le d$, i.e. the $((i-1)d+k)$-th row of the submatrix $[H_{21} \quad H_{11}]$, by triangle inequality again, we have

$$\sqrt{\sum_{j \ne id+k} H^2[i,j]} \le \sqrt{\sum_{j \ne (i-1)d+k} H_{11}^2[(i-1)d+k,j]} + \sqrt{\sum_{j=1}^{d} H_{21}^2[(i-1)d+k,j]}$$

$$\le \sqrt{\sum_{j \ne i} w_{2i}^2 w_{2j}^2} + \sqrt{\sum_{j=1}^{d} w_{2i}^2 W_1^2[j,k]} + |E_k|$$

$$= |w_{2i}| \left( \sqrt{\sum_{j \ne i} w_{2j}^2} + \sqrt{\sum_{j=1}^{d} W_1^2[j,k]} \right) + |E_k|.$$

Then we have that for $1 \le i \le d, 1 \le k \le d$,

$$\frac{\sqrt{\sum_{j \ne id+k} H^2[i,j]}}{|H[id+k, id+k]|} \le \frac{|w_{2i}| \sqrt{\sum_{j \ne i} w_{2j}^2} + \sqrt{\sum_{j=1}^{d} W_1^2[j,k]}}{w_{2i}^2} + \frac{|E_k|}{w_{2i}^2}$$

$$= \sqrt{\frac{\sum_{j \ne i} w_{2j}^2}{w_{2i}^2}} + \sqrt{\frac{\sum_{j=1}^{d} W_1^2[j,k]}{w_{2i}^2}} + \frac{|E_k|}{w_{2i}^2}$$

$$\le \sqrt{\frac{(1+\delta)^2}{(1-\delta)^2} \cdot \frac{\sum_{j \ne i} c^2 u_j^2}{c^2 u_i^2}} + \sqrt{\frac{(1+\delta)^2}{(1-\delta)^2} \cdot \frac{v_k^2 \sum_{j=1}^{d} u_j^2}{c^2 u_i^2}} + \frac{|E_k|}{(1-\delta)^2 c^2 u_i^2}$$

$$\le \frac{1+\delta}{1-\delta} \left( 1 + \frac{|v_k|}{|c|} \right) \sqrt{\frac{\sum_{j=1}^{d} u_j^2}{u_i^2}} + \frac{|E_k|}{(1-\delta)^2 c^2 u_i^2}.$$

# G   Connection between diagonal of loss Hessian and weights

The partial derivative at $W_i$ of the cost function for each $i$ is given by:

$$\nabla_{W_i} L(W) = W_{i+1}^T \ldots W_{H+1}^T (W_{H+1} W_H \ldots W_1 - A) W_1^T \ldots W_{i-1}^T \quad (54)$$

In our experiments, we were interested in the diagonal elements of the hessian. These are given by:

$$\nabla_{(W_i)_{a,b}} (\nabla_{W_i} L(W))_{a,b} = \nabla_{(W_i)_{a,b}} (W_{i+1}^T \ldots W_{H+1}^T (W_{H+1} W_H \ldots W_1 - A) W_1^T \ldots W_{i-1}^T)_{a,b}$$

for each possible $i, a, b$. For ease in notation, define for each $i$, the quantities $M_i := W_{i+1}^T \ldots W_{H+1}^T$ and $N_i := W_1^T \ldots W_{i-1}^T$. Then we have the following lemma.

**Lemma 30.** *The diagonal elements of the hessian of the cost function are given by:*

$$\nabla_{(W_i)_{a,b}} (\nabla_{W_i} L(W))_{a,b} = (M_i M_i^T)_{a,a} (N_i^T N_i)_{b,b}$$

*for each possible $i, a, b$.*

*Proof.* We have:

$$\begin{aligned}
\nabla_{W_i} L(W) &= W_{i+1}^T \ldots W_{H+1}^T (W_{H+1} W_H \ldots W_1 - A) W_1^T \ldots W_{i-1}^T \\
&= M_i (W_{H+1} W_H \ldots W_1 - A) N_i \\
&= M_i W_{H+1} W_H \ldots W_1 N_i - M_i Y N_i.
\end{aligned}$$

This implies that:

$$\begin{aligned}
\nabla_{(W_i)_{a,b}} (\nabla_{W_i} L(W))_{a,b} &= \nabla_{(W_i)_{a,b}} (M_i W_{H+1} W_H \ldots W_1 N_i - M_i Y N_i)_{a,b} \\
&= \nabla_{(W_i)_{a,b}} (M_i W_{H+1} W_H \ldots W_1 N_i)_{a,b},
\end{aligned}$$

where the last step follows since $M_i$ and $N_i$ are not functions of $W_i$.

Since $M_i := W_{i+1}^T \ldots W_{H+1}^T$, $N_i := W_1^T \ldots W_{i-1}^T$, by defining $C_i := M_i W_{H+1} W_H \ldots W_{i+1} = M_i M_i^T$ and $D_i := W_{i-1} \ldots W_2 W_1 N_i = N_i^T N_i$ we have that:

$$\nabla_{(W_i)_{a,b}} (\nabla_{W_i} L(W))_{a,b} = \nabla_{(W_i)_{a,b}} (C_i W_i D_i)_{a,b},$$

where $C_i$ and $D_i$ are not functions of $W_i$. Now, Equation 74 in the Matrix Cookbook[13] shows us that for any matrices $A$ and $X$ we have:

$$\nabla_{X_{mn}} (XA)_{ij} = \delta_{im} A_{nj}.$$

Note that $W_i \in \mathbb{R}^{d_i \times d_{i-1}}$, then we can apply this to obtain that:

$$\begin{aligned}
\nabla_{(W_i)_{a,b}} (\nabla_{W_i} L(W))_{a,b} &= \nabla_{(W_i)_{a,b}} (C_i W_i D_i)_{a,b} \\
&= \nabla_{(W_i)_{a,b}} \left[ \sum_{k=1}^{d_i} (C_i)_{a,k} (W_i D_i)_{k,b} \right] \\
&= \sum_{k=1}^{d_i} (C_i)_{a,k} \nabla_{(W_i)_{a,b}} (W_i D_i)_{k,b} \\
&= \sum_{k=1}^{d_i} (C_i)_{a,k} \delta_{ak} (D_i)_{b,b} \\
&= (C_i)_{a,a} (D_i)_{b,b} \\
&= (M_i M_i^T)_{a,a} (N_i^T N_i)_{b,b}.
\end{aligned}$$

This completes the proof. □

---

[13] https://www.math.uwaterloo.ca/~hwolkowi/matrixcookbook.pdf

For ease of notation, let's now drop the superscript OPT and $(t)$ and write $R_{\text{med},1}^{\text{OPT}}(t)$ as $R_{\text{med},1}$ and $R_{\text{med},2}^{\text{OPT}}(t)$ as $R_{\text{med},2}$. For a 2-layer linear network, $H = 1$. Consider the Hessian w.r.t $W_1$, we have $M_1 M_1^T = W_2^T W_2$ and $N_1^T N_1$ is an identity matrix. Under Assumption 1, we know that $W_2$ is a row vector, which can be denoted as $W_2 = [w_{21}, w_{22}, ..., w_{2d_1}]$. Then we have

$$(M_1 M_1^T)_{a,a} = w_{2a}^2, (N_1^T N_1)_{b,b} = 1, \quad \Rightarrow \quad R_{\text{med},1} = \frac{\max_i (w_{2i})^2}{\text{median}(w_{2i})^2}.$$

Similarly, consider the Hessian w.r.t. $W_2$, we have that $M_1 M_1^T$ is an identity matrix and $N_1^T N_1 = W_1 W_1^T$. Therefore,

$$(M_1 M_1^T)_{a,a} = 1, (N_1^T N_1)_{b,b} = \|W_1[b,:]\|_2^2, \quad \Rightarrow \quad R_{\text{med},2} = \frac{\max_i \|W_1[i,:]\|_2^2}{\text{median}\|W_1[i,:]\|_2^2}.$$

Hence we have related the uniformity of diagonal Hessian to that of weight matrices. In the detailed analysis, for both GD and Adam, we can prove that $W_1$ converges to an approximately rank 1 matrix. The following lemma allows us to use this rank 1 structure to compute $R_{\text{med},1}$ and $R_{\text{med},2}$.

**Lemma 31.** *Suppose $W_1 \in \mathbb{R}^{d \times d}$ and $W_2 \in \mathbb{R}^{1 \times d}$ have the following structure:*

$$W_1 = \boldsymbol{u}\boldsymbol{v}^T + R_1,$$
$$W_2 = c\boldsymbol{u}^T + R_2,$$

*where $\boldsymbol{u} \in \mathbb{R}^d, \boldsymbol{v} \in \mathbb{R}^d, R_1 \in \mathbb{R}^{d \times d}, R_2 \in \mathbb{R}^{1 \times d}$ and that*

$$\forall 1 \le i, j \le d: \quad \frac{|R_1[i,j]|}{|u_i v_j|} \le \delta, \quad \frac{|R_{2i}|}{|cu_i|} \le \delta, \quad \delta \in (0,1).$$

*Then we have*

$$R_{\text{med},1}, R_{\text{med},2} \in \left[\frac{(1-\delta)^2}{(1+\delta)^2} \cdot \frac{\max_i u_i^2}{\text{median } u_i^2}, \frac{(1+\delta)^2}{(1-\delta)^2} \cdot \frac{\max_i u_i^2}{\text{median } u_i^2}\right].$$

*Proof.* Let's first consider $R_{\text{med},1}$. we have

$$\forall i \in [d]: (1-\delta)^2 (cu_i)^2 \le w_{2i}^2 \le (1+\delta)^2 (cu_i)^2$$
$$\Rightarrow \quad (1-\delta)^2 \max_i (cu_i)^2 \le \max_i w_{2i}^2 \le (1+\delta)^2 \max_i (cu_i)^2$$
$$(1-\delta)^2 \text{median } (cu_i)^2 \le \text{median } w_{2i}^2 \le (1+\delta)^2 \text{median } (cu_i)^2,$$

which yields

$$\frac{(1-\delta)^2}{(1+\delta)^2} \cdot \frac{\max_i u_i^2}{\text{median } u_i^2} \le R_{\text{med},1} = \frac{\max_i w_{2i}^2}{\text{median } w_{2i}^2} \le \frac{(1+\delta)^2}{(1-\delta)^2} \cdot \frac{\max_i u_i^2}{\text{median } u_i^2}.$$

Similarly, for $R_{\text{med},2}$. We have that

$$\forall i, j \in [d]: (1-\delta)^2 (u_i v_j)^2 \le W_1^2[i,j] \le (1+\delta)^2 (u_i v_j)^2$$
$$\Rightarrow \quad (1-\delta)^2 u_i^2 \|\boldsymbol{v}\|_2^2 \le \|W_1[i,:]\|_2^2 \le (1+\delta)^2 u_i^2 \|\boldsymbol{v}\|_2^2$$
$$\Rightarrow \quad (1-\delta)^2 \max_i u_i^2 \|\boldsymbol{v}\|_2^2 \le \max_i \|W_1[i,:]\|_2^2 \le (1+\delta)^2 \max_i u_i^2 \|\boldsymbol{v}\|_2^2$$
$$(1-\delta)^2 \text{median } u_i^2 \|\boldsymbol{v}\|_2^2 \le \text{median } \|W_1[i,:]\|_2^2 \le (1+\delta)^2 \text{median } u_i^2 \|\boldsymbol{v}\|_2^2,$$

which yields

$$\frac{(1-\delta)^2}{(1+\delta)^2} \cdot \frac{\max_i u_i^2 \|\boldsymbol{v}\|_2^2}{\text{median } u_i^2 \|\boldsymbol{v}\|_2^2} \le R_{\text{med},2} = \frac{\max_i \|W_1[i,:]\|_2^2}{\text{median}\|W_1[i,:]\|_2^2} \le \frac{(1+\delta)^2}{(1-\delta)^2} \cdot \frac{\max_i u_i^2 \|\boldsymbol{v}\|_2^2}{\text{median } u_i^2 \|\boldsymbol{v}\|_2^2}.$$

That means

$$\frac{(1-\delta)^2}{(1+\delta)^2} \cdot \frac{\max_i u_i^2}{\text{median } u_i^2} \le R_{\text{med},2} \le \frac{(1+\delta)^2}{(1-\delta)^2} \cdot \frac{\max_i u_i^2}{\text{median } u_i^2}.$$

$\square$

# H  Auxiliary lemmas

**Lemma 32.** *Let $A = \frac{1}{m}YX^T$, $\Lambda_{xx} := \frac{1}{m}XX^T$, $g_k^{(t)} = \nabla_{W_k}L(W^{(t)})$, $k = 1, 2$. Denote $\tilde{A}^{(t)}$, $\tilde{\Lambda}_{xx}^{(t)}$ and $\tilde{g}_k^{(t)}$, $k = 1, 2$ as the corresponding batch versions at time $t$. Let $M_1^{(t)} = \max_{i,j} \left| W_1^{(t)}[i,j] \right|$ and $M_2^{(t)} = \max_i \left| w_{2i}^{(t)} \right|$. Under Assumption 3, we have with probability at least $1 - \frac{1}{d}$, for $\forall t \leq T$ and $\forall i, j \in [d]$,*

$$\left| \tilde{g}_1^{(t)}[i,j] - g_1^{(t)}[i,j] \right| \leq d^3 M_1^{(t)} \left( M_2^{(t)} \right)^2 \sigma\sqrt{dT} + M_2^{(t)}\sigma\sqrt{d^2T},$$

$$\left| g_{2i}^{(t)} - g_{2i}^{(t)} \right| \leq d^4 \left( M_1^{(t)} \right)^2 M_2^{(t)}\sigma\sqrt{dT} + dM_1^{(t)}\sigma\sqrt{d^2T}.$$

*Proof.* By Assumption 3 and Chebyshev's inequality, we have for fixed $i, j \in [d]$ and $t \leq T$,

$$\mathbb{P}\left( \left| \tilde{A}_i^{(t)} - A_i \right| > \lambda \right) \leq \frac{\sigma^2}{\lambda^2}, \quad \mathbb{P}\left( \left| \tilde{\Lambda}_{xx}^{(t)}[i,j] - \Lambda_{xx}[i,j] \right| > \lambda \right) \leq \frac{\sigma^2}{\lambda^2}.$$

Applying the union bound gives us

$$\mathbb{P}\left( \exists i \in [d], \exists t \leq T : \quad \left| \tilde{A}_i^{(t)} - A_i \right| > \lambda \right) \leq \frac{Td\sigma^2}{\lambda^2},$$

$$\mathbb{P}\left( \exists i, j \in [d], \exists t \leq T : \quad \left| \tilde{\Lambda}_{xx}^{(t)}[i,j] - \Lambda_{xx}[i,j] \right| > \lambda \right) \leq \frac{Td^2\sigma^2}{\lambda^2},$$

which gives us with probability at least $1 - \frac{1}{d}$, for $\forall t \leq T, \forall i, j \in [d]$,

$$\left| \tilde{A}_i^{(t)} - A_i \right| \leq \sigma\sqrt{d^2T}, \quad \left| \tilde{\Lambda}_{xx}^{(t)}[i,j] - \Lambda_{xx}[i,j] \right| \leq \sigma d\sqrt{dT}.$$

Now we are ready to bound $\tilde{g}_k^{(t)} - g_k^{(t)}$ for $k = 1, 2$ and $t \leq T$.

Note that for all $t \leq T$ and $\forall i \in [d]$,

$$\left| \left( W_2^{(t)} W_1^{(t)} \right)_i \right| = \left| \sum_{j=1}^d w_{2j}^{(t)} W_1^{(t)}[j,i] \right| \leq \sum_{j=1}^d \left| w_{2j}^{(t)} \right| \left| W_1^{(t)}[j,i] \right| \leq dM_1^{(t)}M_2^{(t)}.$$

Then we have with probability at least $1 - \frac{1}{d}$, for all $t \leq T$ and $\forall i \in [d]$,

$$\left| \left( W_2^{(t)} W_1^{(t)} \left( \tilde{\Lambda}_{xx}^{(t)} - \Lambda_{xx} \right) \right)_i \right| \leq \sum_{j=1}^d \left| \left( W_2^{(t)} W_1^{(t)} \right)_j \right| \left| \tilde{\Lambda}_{xx}^{(t)}[j,i] - \Lambda_{xx}[j,i] \right|$$

$$\leq d^3 M_1^{(t)} M_2^{(t)}\sigma\sqrt{dT}.$$

Combining with $\tilde{g}_1^{(t)} - g_1^{(t)} = W_2^{(t)T} \left( W_2^{(t)} W_1^{(t)} \left( \tilde{\Lambda}_{xx}^{(t)} - \Lambda_{xx} \right) - \left( \tilde{A}^{(t)} - A \right) \right)$, we get that with probability at least $1 - \frac{1}{d}$, for all $t \leq T$ and $\forall i, j \in [d]$,

$$\left| \tilde{g}_1^{(t)}[i,j] - g_1^{(t)}[i,j] \right| \leq \left| w_{2i}^{(t)} \right| \left| \left( W_2^{(t)} W_1^{(t)} \left( \tilde{\Lambda}_{xx}^{(t)} - \Lambda_{xx} \right) \right)_j \right| + \left| w_{2i}^{(t)} \right| \left| \tilde{A}_j^{(t)} - A_j \right|$$

$$\leq d^3 M_1^{(t)} \left( M_2^{(t)} \right)^2 \sigma\sqrt{dT} + M_2^{(t)}\sigma\sqrt{d^2T}.$$

Similarly, note that $\tilde{g}_{2i}^{(t)} - g_{2i}^{(t)} = \left( W_2^{(t)} W_1^{(t)} \left( \tilde{\Lambda}_{xx}^{(t)} - \Lambda_{xx} \right) - \left( \tilde{A}^{(t)} - A \right) \right) W_1^{(t)T}$, we then have that with probability at least $1 - \frac{1}{d}$, for all $t \leq T$ and $\forall i, j \in [d]$,

$$\left| \tilde{g}_{2i}^{(t)} - g_{2i}^{(t)} \right| \leq \sum_{j=1}^d \left| \left( W_2^{(t)} W_1^{(t)} \left( \tilde{\Lambda}_{xx}^{(t)} - \Lambda_{xx} \right) \right)_j \right| \left| W_1^{(t)}[i,j] \right| + \sum_{j=1}^d \left| \tilde{A}_j^{(t)} - A_j \right| \left| W_1^{(t)}[i,j] \right|$$

$$\leq d^4 \left( M_1^{(t)} \right)^2 M_2^{(t)}\sigma\sqrt{dT} + dM_1^{(t)}\sigma\sqrt{d^2T}.$$

$\square$

**Lemma 33.** *Consider two sequences $\{a^{(t)}\}_{t\geq 0}, \{b^{(t)}\}_{t\geq 0}$, which satisfy*

$$a^{(t)} = (1-\beta) \sum_{\tau=0}^{t} \beta^\tau b^{(t-\tau)}, \beta \in (0,1).$$

*Suppose $\forall \tau \leq t : |b^{(t)}| \leq B$, then for any $\epsilon > 0$, the following truncated version*

$$\tilde{a}^{(t)} = (1-\beta) \sum_{\tau=0}^{H} \beta^\tau b^{(t-\tau)}$$

*with $H \geq \frac{1}{1-\beta} \log \frac{B}{\epsilon} = \tilde{\Omega}\left(\frac{1}{1-\beta}\right)$ satisfies*

$$\left| a^{(t)} - \tilde{a}^{(t)} \right| \leq \epsilon.$$

*Proof.* We have that

$$\left| a^{(t)} - \tilde{a}^{(t)} \right| \leq \left| (1-\beta) \sum_{\tau=H+1}^{t} \beta^\tau b^{(t-\tau)} \right| \leq (1-\beta) \sum_{\tau=H+1}^{t} \beta^\tau B \leq B\beta^{H+1}.$$

To make it less than $\epsilon$, it suffices to choose $H \geq \log(\frac{\epsilon}{B})/\log \beta$.

Since $\beta \in (0,1)$, we know that $\log \beta \leq \beta - 1 < 0$. We also have $\log \frac{\epsilon}{B} < 0$. Then it suffices to choose

$$H \geq \frac{\log(\epsilon/B)}{\beta - 1} \geq \frac{\log(\epsilon/B)}{\log \beta} \quad \Rightarrow \quad H \geq \frac{1}{1-\beta} \log \frac{B}{\epsilon} = \tilde{\Omega}\left(\frac{1}{1-\beta}\right).$$

$\square$

**Lemma 34.** *Suppose $a, b, c, e_a, e_b, e_c \in \mathbb{R}, b > 0, c > 0$ satisfy $b + e_b + e_c > 0, |e_a| \leq \delta|a|, |e_b| \leq \delta b, |e_c| \leq \delta^2 c^2$ with $0 < \delta \ll 1$, then we have*

$$\frac{a + e_a}{\sqrt{b + e_b + e_c} + c} = \frac{a}{\sqrt{b} + c}(1 + R), \quad \text{where } |R| = \mathcal{O}(\delta).$$

*Proof.* We have

$$\frac{a + e_a}{\sqrt{b + e_b + e_c} + c} = \frac{a}{\sqrt{b} + c} + \frac{a}{\sqrt{b + e_b + e_c} + c} - \frac{a}{\sqrt{b} + c} + \frac{e_a}{\sqrt{b + e_b + e_c} + c}$$

$$= \frac{a}{\sqrt{b} + c}\left( 1 + \underbrace{\frac{\sqrt{b} + c}{\sqrt{b + e_b + e_c} + c} - 1}_{q_1} + \underbrace{\frac{e_a}{a} \cdot \frac{\sqrt{b} + c}{\sqrt{b + e_b + e_c} + c}}_{q_2} \right).$$

Define $R := q_1 + q_2$. The term $|q_1|$ can be bounded by

$$|q_1| = \frac{\left| \sqrt{b} - \sqrt{b + e_b + e_c} \right|}{\sqrt{b + e_b + e_c} + c}$$

$$= \frac{|e_b + e_c|}{(\sqrt{b + e_b + e_c} + c)\left( \sqrt{b} + \sqrt{b + e_b + e_c} \right)}$$

$$\leq \frac{|e_b|}{(\sqrt{b + e_b + e_c} + c)\left( \sqrt{b} + \sqrt{b + e_b + e_c} \right)} + \frac{|e_c|}{(\sqrt{b + e_b + e_c} + c)\left( \sqrt{b} + \sqrt{b + e_b + e_c} \right)}$$

$$\leq \frac{|e_b|}{(\sqrt{b + e_b + e_c} + c)\sqrt{b}} + \frac{\sqrt{|e_c|}}{c} \cdot \frac{\sqrt{|e_c|}}{\sqrt{b} + \sqrt{b + e_b + e_c}}$$

$$\leq \underbrace{\frac{|e_b|}{(\sqrt{b + e_b + e_c} + c)\sqrt{b}}}_{q_3} + \delta \underbrace{\frac{\sqrt{|e_c|}}{\sqrt{b} + \sqrt{b + e_b + e_c}}}_{q_4},$$

where $|q_3|$ can be bounded by

$$|q_3| \overset{(i)}{\leq} \frac{\delta b}{(\sqrt{b+e_b}-\sqrt{|e_c|}+c)\sqrt{b}} \leq \frac{\delta\sqrt{b}}{\sqrt{b(1-\delta)}+c(1-\delta)} \leq \frac{\delta\sqrt{b}}{\sqrt{b(1-\delta)}} = \mathcal{O}(\delta).$$

Here the denominator of $(i)$ uses $b + e_b \geq b(1-\delta) > 0$ and $\sqrt{x+y} \geq \sqrt{x} - \sqrt{|y|}$ when $x \geq 0, x + y \geq 0$.

Now let's bound $|q_4|$. If $e_c > 0$, we have $e_c = |e_c|$ and $|q_4| \leq \frac{\sqrt{e_c}}{\sqrt{e_c}} = 1$ since $b + e_b \geq b(1-\delta) > 0$.

If $e_c \leq 0$, note that $b + e_b + e_c > 0$, we have $|e_c| < b + b_e \leq b(1+\delta)$, which yields $|q_4| \leq \frac{\sqrt{|e_c|}}{\sqrt{b}} = \mathcal{O}(1)$. Combining the above bounds give us $|q_1| \leq |q_3| + \delta|q_4| = \mathcal{O}(\delta)$.

On the other hand, $|q_2|$ can be bounded by

$$|q_2| \leq \delta \frac{\sqrt{b}+c}{\sqrt{b+e_b}-\sqrt{|e_c|}+c} \leq \delta \frac{\sqrt{b}+c}{\sqrt{b(1-\delta)}+c(1-\delta)} = \mathcal{O}(\delta).$$

Then $|R| \leq |q_1| + |q_2| = \mathcal{O}(\delta)$ $\qquad\square$

**Lemma 35.** *Suppose $X_1, X_2, ..., X_d$ are i.i.d Gaussian with mean 0 and variance $\sigma^2$, then for $0 < \delta < \frac{1}{e}$, we have with probability at least $1 - \delta$,*

$$\max_{1\leq i\leq d} X_i^2 \geq \sigma^2 \left( C_1 \log d - C_2 \log\log\frac{1}{\delta} \right)$$

*for some $C_1, C_2 > 0$.*

*Proof.* It suffices to assume that $\sigma^2 = 1$ and prove that w.p. at least $1 - \delta$, $\max_{1\leq i\leq d} X_i^2 \geq C_1 \log d - C_2 \log\log\frac{1}{\delta}$.

First, by the lower bound of Gaussian tail, there exists $\alpha, \beta > 0$ such that $\mathbb{P}(|X_i| > x) = 2\mathbb{P}(X_i > x) \geq \alpha e^{-\beta x^2}$ for $x \geq 0$. Then by i.i.d., we have

$$\mathbb{P}(\max_i |X_i| \leq x) = \mathbb{P}\left( \bigcap_{i=1}^d \{|X_i| \leq x\} \right)$$

$$= \prod_{i=1}^d \mathbb{P}(|X_i| \leq x) = (1 - \mathbb{P}(|X_i| > x))^d$$

$$\leq (1 - \alpha e^{-\beta x^2})^d$$

$$\leq \exp(-d\alpha e^{-\beta x^2}),$$

where the last inequality uses $1 - x \leq e^{-x}$ for $x \in [0, 1]$. Let $\exp(-d\alpha e^{-\beta x^2}) = \delta$, we get that w.p. at least $1 - \delta$,

$$\max_{1\leq i\leq d} |X_i| \geq \sqrt{\frac{1}{\beta}\left( \log(\alpha d) - \log\log\frac{1}{\delta} \right)}.$$

Then we have w.p. at least $1 - \delta$,

$$\max_{1\leq i\leq d} X_i^2 = \left( \max_{1\leq i\leq d} |X_i| \right)^2 \geq \frac{1}{\beta}\left( \log(\alpha d) - \log\log\frac{1}{\delta} \right).$$

$\qquad\square$

**Lemma 36.** *Suppose $X_1, X_2, ..., X_d$ are i.i.d Gaussian with mean 0 and variance $\sigma^2$, then we have with constant probability,*

$$\frac{1}{d}\sum_{i=1}^d \frac{1}{|X_i|} \leq \mathcal{O}\left( \frac{1}{\sigma}\log d \right).$$

*Proof.* It suffices to assume that $\sigma^2 = 1$ and prove that with constant probability, $\frac{1}{d}\sum_{i=1}^{d}\frac{1}{|X_i|} \leq \mathcal{O}(\log d)$.

Consider $X_i$ for some fixed $i$. Since $X_i \sim \mathcal{N}(0,1)$, we have $\mathbb{P}(|X_i| \leq t) \leq \frac{2t}{\sqrt{2\pi}}$. Then we know that with probability at least $1 - \Theta\left(d^{-1}\right)$, $|X_i| \geq \frac{C}{d}$ for some $C > 0$. Then by union bound, with constant probability, $\forall i \in [d] : |X_i| \geq \frac{C}{d}$.

Now we split the interval $[\frac{C}{d}, 1]$ into several subintervals $\mathcal{I}_k = \{i : |X_i| \in [2^{-k-1}, 2^{-k}]\}$ for $k = 0, 1, ..., \lceil \log_2 \frac{d}{C} \rceil - 1$. Let $p_k = \mathbb{P}(|X_i| \in [2^{-k-1}, 2^{-k}])$, we know that $|\mathcal{I}_k| \sim \text{Binomial}(d, p_k)$ and $p_k \leq C_1 \cdot 2^{-k-1}$. Then by the concentration of binomial variables, we have w.p. at least $1 - d^{-p}$ for $p > 0$, $|\mathcal{I}_k| = \mathcal{O}\left(dp_k + \sqrt{dp_k \log d} + \log d\right) = \mathcal{O}\left(d \cdot 2^{-k-1} + \sqrt{d \cdot 2^{-k-1} \log d} + \log d\right)$. Then we have

$$\sum_{i \in \mathcal{I}_k} \frac{1}{|X_i|} \leq |\mathcal{I}_k| 2^{k+1} = \mathcal{O}\left(d + \sqrt{d \cdot 2^{k+1} \log d} + 2^{k+1} \log d\right), \quad k = 0, 1, ..., \lceil \log_2 \frac{d}{C} \rceil - 1.$$

Therefore, with constant probability,

$$\begin{aligned}
\sum_{i=1}^{d} \frac{1}{|X_i|} &= \sum_{k=0}^{\lceil \log_2 \frac{d}{C} \rceil - 1} \sum_{i \in \mathcal{I}_k} \frac{1}{|X_i|} + \sum_{|X_i| > 1} \frac{1}{|X_i|} \\
&\leq \sum_{k=0}^{\lceil \log_2 \frac{d}{C} \rceil - 1} \mathcal{O}\left(d + \sqrt{d \cdot 2^{k+1} \log d} + 2^{k+1} \log d\right) + d \\
&= \mathcal{O}\left(d \log_2 \frac{d}{C}\right) + \mathcal{O}\left(\sqrt{d \log d} \cdot (\sqrt{2})^{\lceil \log_2 \frac{d}{C} \rceil + C_2}\right) + 2^{\lceil \log_2 \frac{d}{C} \rceil + 1} \log d + d \\
&= \mathcal{O}(d \log d),
\end{aligned}$$

which means with constant probability, $\frac{1}{d}\sum_{i=1}^{d}\frac{1}{|X_i|} = \mathcal{O}(\log d)$. $\qquad \square$

