# OpenReview forum: "How Does Adaptive Optimization Impact Local Neural Network Geometry?"
_NeurIPS.cc/2023/Conference — NeurIPS 2023 poster_

### Official Review · Reviewer_qhhF · 2023-07-06

**Soundness:** 3 good
**Presentation:** 2 fair
**Contribution:** 3 good
**Rating:** 6
**Confidence:** 4

**Summary:**

This paper aims to understand why Adam works better than SGD for language model training from the perspective of local geometry of the training loss around the algorithm iterates.

**Strengths:**

The problem that this paper considers is indeed a very important question, and this work sets out to answer the question from first principles.
This paper has many good intuitions. This paper also has some partial theory that backs up their empirical observation. Given their empirical observation is quite hard to theoretically analyze, I think their theory is quite valuable although it only shows a weak result for a very specific model.

**Weaknesses:**

- Many plots have too small fonts. Please adjust the font sizes for better readability.
- In general, the plots have legends and axes that are quite hard to read, please adjust them.

- In Figure 8, how can the spectrum on the right plot approximately low-rank? All singular values are greater than 0.5 and compared to the left plot, I wouldn't consider this approximate low rank. Please explain

- Regarding Section A.6, the plots seem a bit inconclusive. Given how noisy the plots are, I don't see this as alignment. Do you have any intuitions behind why the layer gradients align with diagonal of Hessian?

- Are the results established in Theorem 1 empirically tight? I think the experiments on this very simple setting to see if your empirical observations hold true for this very simple setting is required. In particular, for this two layer MLP, do you see lower R-value for Adam than SGD-M?

- In Figure 9, I see that the observation doesn't hold for the right singular vectors.. This makes the observation quite brittle in my opinion. Could you explain why it doesn't hold for the right singular vectors?

- In general, this paper lacks empirical investigation on smaller/toy models. I think to what extent this observation holds for simpler models would help readers comprehend how universal this observed phenomenon is.

- Given that the language model training is done mostly with the cross entropy loss, I'm curious whether the conclusion you made in Theorem 1 similarly holds true for the logistic loss?

- In the Gaussian initialization, I see the variances of the different layers are chosen quite carefully. Also the variance looks quite small. Could you justify this assumption as to why making these assumptions is practical. Also, could you explain intuition behind why these assumptions are required for the proof?

- In the Theorem statement, how big are $T_1^{sgd}, T_2^{sgd}$ compared to $T_1^{adam}, T_2^{adam}$? Are they comparable? Or as you observed in the empirical results, are $T_1^{sgd}, T_2^{sgd}$ provably larger than $T_1^{adam}, T_2^{adam}$?

- Could you share some intuitions as to why training behavior of vision models (CIFAR 10, IMAGENET) is so different than that of language models? I don't find the explanation that it has to do with transformer architecture quite conclusive...

- Minor comment: It would be helpful if the proof sketch of Theorem 1 appears in the main text. I think it has many good intuitions.

**Questions:**

See above

---

> ### Author Rebuttal · Authors · 2023-08-10
>
> We thank the reviewer for their comments and answer their questions below.
>
> 1. **Fonts of plots**. Thanks for your advice. We will increase the fonts and adjust the legends and axes of the plots.
>
> 2. **Low-rank structure.** Figure 8 is to depict a low-rank trend of the spectrum instead of a mathematically rigorous low-rank property. Actually for practical deep models, due to various noises, we don't expect it to have a very nice low-rank property.
>
> 3. **Alignment in Section A.6.** Though not very significant, if we focus on *large* gradients, we can find that they align with *large* diagonals of Hessian. Intuitively, when the diagonal of Hessian is large (sharp region), we should use a small update. However, for SGD, the corresponding updates are usually large due to large gradients. This gives us some intuitive evidence of why a non-uniform diagonal Hessian will harm fast optimization for SGD.
>
> However, things are different for adaptive methods such as Adam. As is shown in the right columns of Figure 10-12, their *adaptive updates* do not align with the diagonal of Hessian and are more uniform. Actually, we don't want to demonstrate the alignment between *true gradients* and diagonals of Hessian. Instead, we want to emphasize the uniformity of Adam's *adaptive updates* and the misalignment between them and the diagonals of Hessian, as it provides intuition on the optimization advantage of adaptive methods. Finally, the figures in Section A.6 are just intuitive and we don't want to draw a conclusion. That's why we put them into an appendix.
>
> 4. **Empirical results on the setting of Theorem 1 and other smaller/toy models.** Below are the empirical results on a 2-layer linear network. We trained the models for 20 epochs to convergence and chose the best learning rates of SGD+M and Adam.  As we can see, our empirical observations hold true for this very simple setting, i.e. $R^{\text{Adam}}\_{\text{med}}(t)<R^{\text{SGDM}}\_{\text{med}}(t)$.
> |   |  Epoch0|Epoch0|Epoch5|Epoch5|Epoch10|Epoch10| Epoch20|Epoch20|
> | :---: | :---: | :---: | :---: | :---: | :---: | :---: | :---: | :---: |
> | Layer # | $R^{\text{SGDM}}_{\text{med}}(t)$ | $R^{\text{Adam}}_{\text{med}}(t)$ | $R^{\text{SGDM}}_{\text{med}}(t)$ | $R^{\text{Adam}}_{\text{med}}(t)$ |$R^{\text{SGDM}}_{\text{med}}(t)$ | $R^{\text{Adam}}_{\text{med}}(t)$ | $R^{\text{SGDM}}_{\text{med}}(t)$ | $R^{\text{Adam}}_{\text{med}}(t)$|
> |1|57.18 |57.18|59.85|35.37 |61.20 |38.16|75.10|59.13|
> |2|1.54|1.54|3.68|1.58|3.33|1.60|3.97|1.86|
>
> Moreover, Appendix A.1 presents empirical results on a shallow transformer (8 layers), which is a simpler model than those in the main text. Our observation still holds for this shallow transformer. The above results reveal the universality of our observation on simple models.
>
> 5. **Right singular vectors.** We planned to use a paragraph to present the behavior of right singular vectors and discuss possible explanations. However, due to an editing error, we forgot to put that paragraph into the appendix. Roughly speaking, we noticed that the right singular vectors do not have uniformity for Adam. We are not sure of the reason and one possible explanation is that for a weight matrix, its right singular vectors are closer to the input data than left singular vectors and more easily influenced by the data, therefore may not show uniformity.
>
> 6. **Logistic loss.** The answer is yes. Below are the empirical results on the setting of Theorem 1 but using the logistic loss. We trained the models for 25 epochs to convergence and chose the best learning rates of SGD+M and Adam. As we can see, the conclusion similarly holds, i.e. $R^{\text{Adam}}\_{\text{med}}(t)<R^{\text{SGDM}}\_{\text{med}}(t)$.
> | |Epoch0|Epoch0 | Epoch5 | Epoch5 |Epoch15| Epoch15|Epoch25|Epoch25|
> | :---: | :---: | :---: | :---: | :---: | :---: | :---: | :---: | :---: |
> | Layer # | $R^{\text{SGDM}}_{\text{med}}(t)$ | $R^{\text{Adam}}_{\text{med}}(t)$ | $R^{\text{SGDM}}_{\text{med}}(t)$ | $R^{\text{Adam}}_{\text{med}}(t)$ | $R^{\text{SGDM}}_{\text{med}}(t)$ | $R^{\text{Adam}}_{\text{med}}(t)$ | $R^{\text{SGDM}}_{\text{med}}(t)$ | $R^{\text{Adam}}_{\text{med}}(t)$|
> |1|47.75| 47.75| 18.5|15.55|32.08|14.54|56.42|12.58|
> |2|2.06 |2.06| 3.57|1.74| 2.28|1.81|4.34|1.60|
>
> 7. **About initialization.** Empirically, our observation still holds under standard initialization. The empirical results in the fourth point are conducted under standard initialization. That means the small initialization assumption in our theoretical analysis is not essential. Actually it is only for technical reasons to make the proof of low-rank structures easier. Besides, it is a common choice for theoretical work to make the initialization close to 0.
>
> 8. **Comparison between $T\_{\text{Adam,2}}, T_{\text{Adam,1}}$ and $T_{\text{SGDM,2}}, T_{\text{SGDM,1}}$.** In the detailed proofs in Appendix D and E, we prove that $T\_{\text{Adam,2}}-T_{\text{Adam,1}}=\Theta(\frac{1}{\eta\sqrt{d}})$ and provide an upper bound for SGD+M: $T_{\text{SGDM,2}}-T_{\text{SGDM,1}}\le\tilde{\mathcal{O}}(\frac{d^\alpha}{\eta})$. We can see that $T_{\text{SGDM,2}}-T_{\text{SGDM,1}}$ is larger than $T\_{\text{Adam,2}}-T_{\text{Adam,1}}$, but actually, they are not comparable because the bound for SGD+M is just an upper bound.
>
> 9. **Results on image tasks.** As is discussed in A.8, the behavior on image tasks seems different, but the underlying correlation between $R^{\text{OPT}}\_{\text{med}}$ and optimization speed is actually consistent with that on language tasks. As is shown in A.8, on image tasks, Adam does not converge faster than SGD+M and in the meantime, $R^{\text{Adam}}_{\text{med}}$ values are no longer smaller than $R^{\text{SGDM}}\_{\text{med}}$ during training. This reveals the connection between the local diagonal geometry and the convergence speed from another perspective. That is, when the diagonal of Hessian of Adam is not more uniform than SGD+M, its convergence speed is not better, either.

---

> > ### Comment · Reviewer_qhhF · 2023-08-13
> >
> > I read the response and it's satisfactory.
> > I raise my score to 6.

---

> > > ### Author Response · Authors · 2023-08-21
> > >
> > > Thanks a lot for raising the score!

---

### Official Review · Reviewer_8Quz · 2023-07-07

**Soundness:** 2 fair
**Presentation:** 3 good
**Contribution:** 2 fair
**Rating:** 4
**Confidence:** 4

**Summary:**

This paper aims to study the connections between an optimization algorithm and the geometric properties observed during the training of a neural network with that algorithm. The authors argue that when analyzing these connections, it is important to consider the local iterates. To address this, they propose the utilization of a metric called $R^{OPT}_{med}$, which involves the local condition number of the Hessian calculated at the iterates. The authors conduct experiments to illustrate their findings that fast convergence is often associated with low statistics, and adaptive methods such as Adam often has low statistics. They proves that for some small theoretical setting, Adam has better statistic than SGD with some probability.

**Strengths:**

Studying the influence of algorithms on the loss geometry is interesting. This paper provides some insights on the uniformity of diagonal geometry by the $R^{OPT}_{med}$ metric where the values found by Adam are smaller than those found by SGDM in the empirical results.

The theoretical setting shows that the diagonal of loss Hessian for Adam has good uniformity while the diagonal of loss Hessian for SGD+M is less uniform.



**Weaknesses:**

While there may exist a correlation between the proposed statistic and the optimization algorithm, the reasons behind the favorable behavior of these algorithms remain uncertain. Is it better due to the uniformity or is the diagonal uniformity a byproduct of Adam's inherent algorithmic properties that are unrelated to its overall effectiveness? As a result, it is more important to demonstrate the contribution of the proposed statistic to achieving successful convergence rather than focusing on the relationship between Adam and the the diagonal uniformity.

Why the large batch setting are needed to prove the correlation of Adam and diagonal uniformity effectiveness? It is not clear how the algorithms behave in other settings and how high the probability can be in the theoretical analysis.

**Questions:**

Please see above.

---
I thank the authors for your rebuttal. My recommendation remains the same.

---

> ### Author Rebuttal · Authors · 2023-08-09
>
> We thank the reviewer for their comments and answer their questions below.
>
> 1. **Contribution of our statistic to fast optimization.** We actually discuss and demonstrate the contribution of small $R^{\text{OPT}}_{\text{med}}$ to fast optimization. Please see the first paragraph on Page 7 and Appendix B for more details.
>
> Roughly speaking, to rule out the possibility that small $R^{\text{OPT}}\_{\text{med}}$ is just a byproduct of adaptive methods and is unrelated to fast optimization, in Appendix B.1, we add a supplementary experiment similar to that in Figure 1. We select two iterates $x_1$ and $x_2$ from two trajectories that both come from SGD+M (instead of one from Adam and one from SGD+M in Figure 1), such that the loss $f(x_1)=f(x_2)$ but $x_2$ has a smaller $R^{\text{OPT}}\_{\text{med}}$ than $x_1$. We then run SGD+M with the same configuration twice, once from $x_1$ and once from $x_2$. Under this setting, we get a similar observation: running SGD+M from $x_2$ (with smaller $R^{\text{OPT}}\_{\text{med}}$) achieves faster convergence than from $x_1$. This suggests that the uniformity of the diagonal of loss Hessian (measured by $R^{\text{OPT}}\_{\text{med}}$) reveals some intrinsic trajectory property beyond the algorithm choice and is indeed a contributing factor to fast optimization. In Appendix B.2, we theoretically prove the contribution of small $R^{\text{OPT}}_{\text{med}}$ to fast optimization in a simplified setting.
>
> 2. **The large batch assumption.** Our proof relies on a very elegant analysis and tracks detailed dynamical properties of weight matrices during training, which is a big technical challenge. To overcome the difficulty and simplify the proof, we require large batches to reduce the stochastic variances.
> Moreover, we want to emphasize that people in practice use very large batches when training language models, especially when using multiple GPUs.

---

> ### Author Response · Authors · 2023-08-18
>
> Thanks for your feedback on our rebuttal. We would like to know which part of our rebuttal is still unclear and are willing to elaborate more on it. As for the demonstration of the contribution of our metric to fast optimization, please see the first paragraph on Page 7 and Appendix B for more details, in case of any unclear part in our rebuttal.

---

### Official Review · Reviewer_cQDf · 2023-07-07

**Soundness:** 3 good
**Presentation:** 3 good
**Contribution:** 3 good
**Rating:** 6
**Confidence:** 4

**Summary:**

This study makes the comparison between the adaptive optimization method (Adam) and the non-adaptive one (stochastic gradient method with momentum) through a lens of ``uniformity'' of diagonal components of the Hessian denoted by $R_{\rm med}^{\rm OPT}(t)$. The comparison is conducted experimentally for various deep learning models such as BERT-small and theoretically for the two-layer linear neural networks under several assumptions. Both comparisons conclude that the uniformity will be smaller when using the Adam, which results in faster convergence.

**Strengths:**

- The experimental comparison is conducted carefully.
- The theoretical results have a novelty, which gives new insight into analyzing the implicit bias of various optimization methods.

**Weaknesses:**

- I couldn't fully understand the motivation for employing $R_{\rm med}^{\rm OPT}(t)=\frac{{\rm max}\{|H_{ii}^{(t)}|\}}{{\rm median}\{|H_{ii}^{(t)}}|\}$ as for the notion of uniformity. I realize this value is more stable than the condition number; as explained in footnote 1, many types of variants could be considered. Indeed, in the experiment, there are some cases where $\frac{R_{\rm med}^{\rm SGD}(t)}{R_{\rm med}^{\rm Adam}(t)}$ is smaller than 1.
- In the theoretical comparison (Theorem 1), there are several unclear points to me. First, the training dynamics will change by the selection of hyperparameter and affects to $R_{\rm med}^{\rm OPT}(t)$, but some parameters (such as $\alpha$, $\sigma$) can be taken in a different order. Will it be a fair comparison? Moreover, I think it will be more helpful for readers if the authors clarify the dependence of $T_{\rm OPT,(1,2)}$ on other parameters.
- (minor) The reference to Figure 4. seems wrong. The authors write ``see Section 4.1'' in the caption, but it is explained in Section 6.

**Questions:**

- Although the initialization order required in Theorem 1 seems to be very small, can the authors import the theoretical result or its insights into the practical settings? If not, what is the difficulty?
- The setting in Section 5 requires $d_1=d_0=d$, but can the overparameterized settings (i.e., $d_1\ge d_0$ holds) be treated in the same way?

---

> ### Author Rebuttal · Authors · 2023-08-10
>
> We thank the reviewer for their comments and answer their questions below.
>
> 1. **About our metric $R^{\text{OPT}}_{\text{med}}(t)$.** First, we did consider another type of variant, which is a singular value-based metric. More discussions and experiments can be found in the paragraph starting from Line 79 and Appendix A.9. The short conclusion is that when measured by that singular value-based metric, our observation still holds: the local geometry obtained by Adam is more uniform than that obtained by SGD+M.
>
> Second, we want to emphasize that although in some cases, $\frac{R^{\text{SGDM}}\_{\text{med}}(t)}{R^{\text{Adam}}\_{\text{med}}(t)}$ are smaller than 1, this only happens on a small number of layers. For most layers, $\frac{R^{\text{SGDM}}\_{\text{med}}(t)}{R^{\text{Adam}}\_{\text{med}}(t)}$ are larger than 1.
>
> 2. **Selection of hyperparameters.** Theorem 1 holds for hyperparameters (such as $\alpha,\sigma$) in certain ranges instead of just particular values. The ranges of SGD+M and Adam overlap with each other. That means we can choose the same hyperparameters for SGD+M and Adam in the overlapped region to make a fair comparison, for example, the same $\alpha,\sigma$ such that $\alpha\ge 4(p+2)$ and $\sigma\le\min\left(\frac{\eta^{3/2}}{d^{\alpha/2+1}},\frac{\eta^{3/2}\xi^2}{d^{13/4}}\right)$.
>
> We provide the dependence of $T_{\text{OPT},(1,2)}$ on other parameters in the detailed proofs in Appendix D and E. For example, we prove that $T_{\text{Adam,2}}-T_{\text{Adam,1}}=\Theta(\frac{1}{\eta\sqrt{d}})$ and provide an upper bound for SGD+M: $T_{\text{SGDM,2}}-T_{\text{SGDM,1}}\le\mathcal{O}(\frac{d^\alpha\log(d/\epsilon)}{\eta})$. However, due to the page limit, we didn't write these details in the main text.
>
> 3. **About initialization.** Empirically, our observation still holds under standard initialization. That means the small initialization assumption in our theoretical analysis is not essential. Below are the empirical results on a 2-layer linear network under standard initialization. We trained the models for 20 epochs to convergence and tuned and chose the best learning rates of SGD+M and Adam.  As we can see, our empirical observation still holds, i.e. $R^{\text{Adam}}\_{\text{med}}(t)<R^{\text{SGDM}}_{\text{med}}(t)$.
> |   |  Epoch0|Epoch0|Epoch5|Epoch5|Epoch10|Epoch10| Epoch20|Epoch20|
> | :---: | :---: | :---: | :---: | :---: | :---: | :---: | :---: | :---: |
> | Layer # | $R^{\text{SGDM}}_{\text{med}}(t)$ | $R^{\text{Adam}}_{\text{med}}(t)$ | $R^{\text{SGDM}}_{\text{med}}(t)$ | $R^{\text{Adam}}_{\text{med}}(t)$ |$R^{\text{SGDM}}_{\text{med}}(t)$ | $R^{\text{Adam}}_{\text{med}}(t)$ | $R^{\text{SGDM}}_{\text{med}}(t)$ | $R^{\text{Adam}}_{\text{med}}(t)$|
> |1|57.18 |57.18|59.85|35.37 |61.20 |38.16|75.10|59.13|
> |2|1.54|1.54|3.68|1.58|3.33|1.60|3.97|1.86|
>
> Actually, the small initialization assumption is only for technical reasons to make the proof of low-rank structures easier. Besides, it is a common choice for theoretical work to make the initialization close to 0.
>
> 4. **Overparameterized case.** The answer is yes. The assumption $d_0=d_1=d$ is to make the notation simple and is not essential. For the overparameterized case $d_1\ge d_0$, the bounds in our theorem will have a more complicated dependence on $d_1,d_0$ (for example, $\frac{1}{poly(d)}$ becomes $\frac{1}{poly(d_1,d_0)}$) but the main message will not change, i.e. $R^{\text{Adam}}\_{\text{med}}(t)<R^{\text{SGDM}}_{\text{med}}(t)$.
>
> 5. **Reference to Figure 4.** Sorry for the confusion. In this caption, we write "Singular values and $R_u$ of the weight matrix in the 27-th layer on the translation task (see Section 4.1)". Although Figure 4 is explained in Section 6, actually the note "see Section 4.1" is in terms of the translation task because the setup of the translation task is in Section 4.1. Thanks for pointing out this confusion. We will edit the caption to clarify this.

---

> > ### Comment · Reviewer_cQDf · 2023-08-18
> >
> > Thanks for the author's reply. My concerns are adequately addressed.

---

### Official Review · Reviewer_pmE6 · 2023-07-11

**Soundness:** 3 good
**Presentation:** 3 good
**Contribution:** 2 fair
**Rating:** 6
**Confidence:** 3

**Summary:**

This paper aims to study the interaction between optimizers and the local loss landscape. The authors introduce the notion of $R_{\mathrm{med}}^{\mathrm{OPT}}$ to characterize the uniformity of the Hessian diagonal. Through extensive experiments, they show that, compared with SGD,  Adam biases the trajectory towards regions with higher Hessian diagonal uniformity, which, they argue, contributes to faster optimization. As a theoretical support, the authors demonstrate that Adam indeed possesses this property in a two-layer linear network setting.

**Strengths:**

1. This paper is well-written, with extensive experiments and solid theoretical analysis.

2. The interaction between optimizers and local loss landscape is an important and interesting topic. The findings of this paper may inspire future studies on developing a deeper understanding of the success of adaptive methods.

**Weaknesses:**

I was wondering whether $R_{\mathrm{med}}^{\mathrm{OPT}}$ can characterize the degree of the loss landscape being ill-conditioned. While it is true that $R_{\textrm{med}}^{\textrm{OPT}}$ conveys the similar message as the condition number when the Hessian is dominated by the diagonal ($\nabla^2 \mathcal{L}(\theta)\approx\mathrm{diag}(\nabla^2 \mathcal{L}(\theta))$,  it remains unclear whether $\nabla^2 \mathcal{L}(\theta)\approx\mathrm{diag}(\nabla^2 \mathcal{L}(\theta))$ holds throughout the training trajectory.  Can the authors provide some empircial evidence?

**Questions:**

Regarding the comparison of different training runs, specifically run 1 (using Adam throughout), run 2 (using Adam initially and then switching to SGDM halfway), and run 3 (using SGDM throughout), Figure 1 illustrates that run 2 achieves a considerably lower loss level compared to run 3. However, I am interested in the gap in the final loss between run 1 and run 2.

 If this gap is acceptable, it raises a possibility: Could we design an algorithm that employs Adam during the initial phase and then seamlessly switches to SGDM halfway through the training process? Such an approach may offer the potential benefit of memory savings.

**Limitations:**

Yes

---

> ### Author Rebuttal · Authors · 2023-08-09
>
> We thank the reviewer for their comments and answer their questions below.
>
> 1. **Diagonal approximation.** We actually add empirical evidence on the diagonal approximation in Appendix F. In Appendix F.1, we conduct experiments on a language transformer model and demonstrate that the trend of loss Hessian is to become more and more diagonal during training. In Appendix F.2, we give a rigorous theoretical analysis on a two-layer linear network for this trend.
>
> 2. **Gap between run1 and run 2**. We add a figure in the global rebuttal to demonstrate that the gap between run 1 and run 2 is not acceptable. The convergence speed of run 2 (using Adam initially and then switching to SGDM halfway) is slower than that of run 1 (using Adam throughout). A possible reason is that after switching to SGDM halfway, the trajectory geometry becomes worse and worse, which harms fast optimization.

---

> > ### Comment · Reviewer_pmE6 · 2023-08-13
> >
> > The authors' rebuttal has well addressed my concerns. I will keep my positive score.

---

### Official Review · Reviewer_PbqS · 2023-07-20

**Soundness:** 3 good
**Presentation:** 3 good
**Contribution:** 3 good
**Rating:** 6
**Confidence:** 3

**Summary:**

The paper provides a new explanation of why adaptive gradient methods perform better than SGD with momentum though extensive experiments and theoretical analysis. The key insight is that adaptive gradient methods especially Adam bias toward solutions with uniform Hessian diagonal values, and this property may contribute to faster convergence. Theoretical analysis is conducted on a simple setting with 2 layer neural network, it is proven that Adam guarantees to converge to solution with more uniform Hessian diagonal values than SGD.

**Strengths:**

The insight that Adam bias toward solutions with uniform Hessian diagonal values is an interesting and novel observation and this measure is easy to compute compared with Hessian singular value-based metrics. The observation is supplemented with comprehensive experiment results and an analysis in simplified setting of neural network, which make the work pretty solid in both theory and empirical sides.

**Weaknesses:**

To my understanding, the weaknesses have two folds:

The experiments in the paper mainly focus on settings where Adam outperforms SGD, in which more uniformity in Hessian diagonals is observed with Adam. However, it lacks a comparison in settings where SGD outperforms Adam. It remains a question whether uniformity is still a good measure of performance in this case. In other words, would SGD iterates observe more uniformity in Hessian diagonals instead?

In the explanation of the paper, Adam biases towards solutions with more uniformity in Hessian diagonals, and this contributes to faster convergence. Thus, Adam converges faster than SGD. The bias in Adam has been well experimented with and studied. However, it is not very clear why uniformity in Hessian diagonals implies faster convergence. The paper does not provide a theory on it, and the experimental support of this claim is not strong. Additionally, intuitively, it is hard to connect uniformity with faster convergence since regions with more uniformity are allowed to have either large curvature or small curvature. The folklore intuition on a flat region encourages better optimization and generalization does not seem to hold for this measure.

**Questions:**

Could the authors provide more intuition on how uniformity in Hessian diagonals can induce faster optimization?

**Limitations:**

Limitations are adequately discussed in the paper.

---

> ### Author Rebuttal · Authors · 2023-08-10
>
> We thank the reviewer for their comments and answer their questions below.
>
> 1. **Comparison in the opposite setting.** In Appendix A.8, we have empirical results on image tasks, which provide the comparison in the opposite setting. As is shown in A.8, on image tasks, Adam does not converge faster than SGD+M and in the meantime, $R^{\text{Adam}}\_{\text{med}}$ values are no longer smaller than $R^{\text{SGDM}}\_{\text{med}}$ during training. This reveals the connection between the local diagonal geometry and the convergence speed from another perspective. That is, when the diagonal of Hessian of Adam is not more uniform than SGD+M, its convergence speed is not better, either.
>
> 2. **Contribution of uniformity in Hessian diagonals to fast optimization.** We actually discuss and demonstrate the contribution of small $R^{\text{OPT}}\_{\text{med}}$ (more uniform diagonal of Hessian) to fast optimization. Please see the second and third paragraphs of Section 4.3 and Appendix B for more details.
>
> Roughly speaking, to rule out the possibility that small $R^{\text{OPT}}\_{\text{med}}$ is just a byproduct of adaptive methods and is unrelated to fast optimization, in Appendix B.1, we add a supplementary experiment similar to that in Figure 1. We select two iterates $x_1$ and $x_2$ from two trajectories that both come from SGD+M (instead of one from Adam and one from SGD+M in Figure 1), such that the loss $f(x_1)=f(x_2)$ but $x_2$ has a smaller $R^{\text{OPT}}\_{\text{med}}$ than $x_1$. We then run SGD+M with the same configuration twice, once from $x_1$ and once from $x_2$. Under this setting, we get a similar observation: running SGD+M from $x_2$ (with smaller $R^{\text{OPT}}\_{\text{med}}$) achieves faster convergence than from $x_1$. This suggests that the uniformity of the diagonal of loss Hessian (measured by $R^{\text{OPT}}\_{\text{med}}$) reveals some intrinsic trajectory property beyond the algorithm choice and is indeed a contributing factor to fast optimization. In Appendix B.2, we theoretically prove the contribution of small $R^{\text{OPT}}_{\text{med}}$ to fast optimization in a simplified setting.
>
> 3. **More about uniformity and curvature.** The uniformity measured by our statistic $R^{\text{OPT}}_{\text{med}}$ can be viewed as some variant of the diagonal condition number. We want to emphasize that the condition number is also a relative measure instead of an absolute measure of large or small curvature and is well-known to have a direct connection to the speed of optimization. As an analogy of the condition number, our statistic $R^{\text{OPT}}\_{\text{med}}$ also has an intuitive connection with the optimization speed. In the theoretical analysis in Appendix B.2 on a simplified setting, we can see this connection more clearly.

---

### Author Rebuttal · Authors · 2023-08-10

The attached figure is to address a question raised by Reviewer pmE6.

---

### Decision · Program_Chairs · 2023-09-21

**Decision:**

Accept (poster)

**Comment:**

This paper analyzes how adaptive gradient algorithms affect the local geometry of the neural network learning landscape. The key contribution of this paper is to introduce a statistic that characterizes the "condition number" of the Hessian (which is defined as the ratio of the maximum diagonal value and the median diagonal value of the Hessian matrix) and shows empirically that this new statistic is related to the optimization speed. The authors further prove that in a two-layer linear network, Adam and SGD training trajectories have different hessian uniformity behaviors during the training, which may explain why Adam is faster than SGD in neural network training on NLP tasks.

One of the reviewers criticized that it is unclear whether the proposed statistic is really related to fast optimization. Although the authors do not provide a theoretical answer for this, they have provided experiments in the Appendix to show that they seem to be related. The large batch assumption is indeed a bit unrealistic in practice, but I believe the new trajectory-based analysis will be appreciated by the NeurIPS community and it might inspire future work studying this.

Therefore I recommend accepting this paper.